# GlotLID: Language Identification for Low-Resource Languages

**Amir Hossein Kargaran**[*◇], **Ayyoob Imani**[*◇], **François Yvon**[†] **and Hinrich Schütze**[*◇]
[*]Center for Information and Language Processing, LMU Munich, Germany
[◇]Munich Center for Machine Learning (MCML), Germany
[†]Sorbonne Université, CNRS, ISIR, France
`amir@cis.lmu.de`

## Abstract

Several recent papers have published good solutions for language identification (LID) for about 300 high-resource and medium-resource languages. However, there is no LID available that (i) covers a wide range of low-resource languages, (ii) is rigorously evaluated and reliable and (iii) efficient and easy to use. Here, we publish GlotLID-M, an LID model that satisfies the desiderata of wide coverage, reliability and efficiency. It identifies 1665 languages, a large increase in coverage compared to prior work. In our experiments, GlotLID-M outperforms four baselines (CLD3, FT176, OpenLID and NLLB) when balancing F1 and false positive rate (FPR). We analyze the unique challenges that low-resource LID poses: incorrect corpus metadata, leakage from high-resource languages, difficulty separating closely related languages, handling of macrolanguage vs varieties and in general noisy data. We hope that integrating GlotLID-M into dataset creation pipelines will improve quality and enhance accessibility of NLP technology for low-resource languages and cultures. GlotLID-M model, code, and list of data sources are available: https://github.com/cisnlp/GlotLID.

## 1 Introduction

The NLP community should create technology that covers as many languages as possible, not only medium-resource and high-resource languages. This goal can only be achieved if corpora for low-resource languages are available. Web-mined datasets – including CC100 (Wenzek et al., 2020), mC4 (Xue et al., 2021) and OSCAR (Abadji et al., 2021; Ortiz Suárez et al., 2019) – have made important contributions to low-resource NLP. In particular, they lay the ground for multilingual neural models like XLM-R (Conneau et al., 2020), mT5 (Xue et al., 2021) and Glot500 (ImaniGooghari et al., 2023). However, existing web-mined datasets have systematic quality issues (Kreutzer et al., 2022) and insufficient coverage of low-resource languages.

Low-quality datasets cause poor performance for downstream applications. They can also give rise to a misleading perception of progress when coverage of a low-resource language is claimed based on noisy data. NLP for low-resource languages requires high-quality datasets and high-quality datasets require high-quality LID (language identification). For this reason, high-quality LID for low-resource languages is paramount. To address this need, in this paper we present GlotLID-M, a high-quality LID that covers 1665 languages. We use ISO 639-3 to individuate languages.

When expanding the scope of LID from a few hundred to 1665 languages, the problem of *granularity* becomes severe. In real-world settings, LID needs to support both macrolanguages and their varieties; it also needs to be robust against out-of-model cousins (Caswell et al., 2020; Kreutzer et al., 2022). We pay particular attention to this issue.

While low-resource is our main focus, Blevins and Zettlemoyer (2022) point out that low-quality LID also affects high-resource corpora through contamination, resulting in claims of successful crosslingual transfer that are due to unrecognized coverage of low-resource languages. We also address this issue, e.g., we improve English F1 on the "Universal Declaration of Human Rights" corpus (UDHR) to .85 compared to .43 for OpenLID.

**Contributions.** (i) We curate GlotLID-C, a comprehensive dataset covering 1665 languages, most of them low-resource, from a diverse set of domains. (ii) We train GlotLID-M on GlotLID-C, an open-source LID covering these 1665 languages. (iii) In our experiments, GlotLID-M outperforms several baselines by more than 12% absolute F1 on UDHR, which we take as the best benchmark for our focus on low-resource languages. (iv) When balancing F1 and false positive rate (FPR), GlotLID-M also outperforms baselines on FLORES-200, which is dominated by high-/medium-resource languages.

## 2 Requirements for low-resource LID

**Main use case: Corpus creation.** Corpus creation and cleaning is the main use case for our low-resource LID because we want to address the need for high-quality corpora for low-resource languages. Line-by-line LID filtering is an effective method for achieving high corpus quality. Reliable LID can eliminate various types of noise (see (Caswell et al., 2020; Kreutzer et al., 2022)) – including data from other languages and non-linguistic data – that is frequent, especially in web-crawled content. By adjusting the confidence threshold, users will have control over the level of quality of the corpora they create.

**Broad coverage of languages, minimize out-of-model cousin errors.** We strive for as broad a coverage as is possible given available datasets. This has two benefits. First, it reduces "out-of-model cousin" errors (Caswell et al., 2020; Kreutzer et al., 2022), i.e., it reduces the risk that a language not covered is misclassified as a closely related covered language. Second, having LIDs that discriminate many low-resource languages is a pre-requisite for developing NLP technologies for the largest possible number of languages. Yet many existing LIDs only cover a few hundred languages. In this study, we therefore focus on LIDs having a broad coverage, excluding CLD2 (McCandless, 2010), Equilid (Jurgens et al., 2017), Langdetect (Shuyo, 2010) and langid.py (Lui and Baldwin, 2012). These LIDs cover less than 100 languages or are outperformed by the models we compare with.

**Open-source.** LIDs should be open-source to encourage open collaboration and conform to best research practices. Some LIDs that meet our other requirements are not open-source, e.g., those published by Caswell et al. (2020), Bapna et al. (2022) and Kudugunta et al. (2023). CLD3 (Botha et al., 2017; Salcianu et al., 2018) is freely available, but its training code is not open-source.

**Ease of use.** LIDs should be easily deployable across platforms and programming environments without having to worry about dependencies, compatibility and lack of maintenance.

Because of this ease-of-use requirement, we do not consider whatlang (Brown, 2014b,a) nor idNet (Dunn, 2020), two broad-coverage LIDs that meet many other requirements, but are hard to use in many practical scenarios due to software issues and lack of maintenance.

**Uncertainty assessment.** In our use cases, we would like to rely on uncertainty measures to distinguish cases where the highest-probability language is certain from those where it is not. This would allow us to choose a level of confidence for the resulting corpus. For example, we may want to retain only sentences identified with a high confidence (say, 70%). This is essential to produce high-quality low-resource corpora.

Because of this requirement, we do not consider Franc (Wormer, 2014) as a baseline. While it has many desirable properties, it generally does not provide well-calibrated probabilities. It usually returns several classes, giving 1.0 to the top class and values close to 1.0 to several others.

**Efficiency.** LID is easy to run in parallel, but we still need an efficient solution to make it applicable to large corpora, not least for ecological reasons.

Lack of efficiency is the reason why we do not use AfroLID (Adebara et al., 2022) as a baseline, despite its excellent coverage of African languages.[1] AfroLID is a transformer architecture and less efficient than its competitors.

**Granularity flexibility.** When scaling LID from a few hundred languages to more than 1500, it is hardly practical to restrict the set of labels to a single level of the language hierarchy (e.g., using resources like `https://iso639-3.sil.org`). This is due to the complexity of defining and delimiting languages, including the coexistence of macrolanguages and their varieties. In many cases, we want to keep both the macrolanguage and the varieties in our label set because the varieties we have data for are important languages in their own right. But for other varieties, we do not have variety-labeled data, so the only way to include them is through the macrolanguage. For example, FLORES covers the macrolanguage aka (Akan) and its variety twi (Twi), but not its variety fat (Fanti). Keeping both aka and twi gives flexibility to LID users: they can either differentiate aka and twi or they can consolidate the two labels to the single label aka, depending on what makes more sense in their setting.

## 3 Dataset curation

We now describe GlotLID-C, a corpus for LID training that covers 1665 languages.

**Source selection.** We choose sources that we deem trustworthy (i.e., high chance of correct language label). To address the domain sensitivity of

---

[1]It has no coverage of other low-resource languages.

LID and broaden language coverage, we curate a diverse set of text domains.

We review sources referenced by ImaniGooghari et al. (2023); Burchell et al. (2023); Blaschke et al. (2023); Adebara et al. (2022); Adebara and Abdul-Mageed (2022). In each case, we consider the collection methodology, selecting sources whose language labels are trustworthy. We generally do not use web-crawled sources to avoid the associated problems (Kreutzer et al., 2022). Most selected sources are derived from Wikipedia, religious texts, collaborative translations, storybooks, and news sites. This gives us a coverage of 1832 languages, more than any other public LID. For a list of data sources, see §A.

**Preprocessing.** We ensure that each sentence is written in the correct script, based on the writing system databases of Kargaran et al. (2023) and van Esch et al. (2022). We use the GlotScript (Kargaran et al., 2023) Python library to determine scripts.[2] We also eliminate duplicate sentences.

**Statistics.** Our final corpus, GlotLID-C, comprises 289 million sentences (i.e., lines of data) totaling 40GB and spans 1832 languages (identified by their ISO 639-3 code). 1677 languages have more than 1000 sentences. Refer to §D for the total number of sentences per language.

**Train/test split.** We designate 85% of the data as **GlotLID-C train**. Let $n_l$ be the number of sentences from language $l$ in the remaining 15%. Then we sample $\min(1000, n_l)$ sentences from it. We refer to the resulting dataset as **GlotLID-C test**.

**Contamination.** To make sure our evaluation data (especially UDHR, refer to §5.1) do not overlap with our sources, we compute contamination of UDHR in GlotLID-C train.

We count a UDHR test sentence as occurring in the training set if all of its word four-grams occur in one sentence of GlotLID-C. Most of these contaminations are due to two resources: Wikipedia and Tatoeba.[3] GlotLID-C train shares 374 languages with UDHR.

For 292 languages, we find that none of the UDHR test sentences occurs in the training data. For 57 languages, less than 10% of UDHR test sentences occur in the training data. The remaining 25 languages with a contamination rate over 10% are all high/medium resource languages.

In our experiments, we decided against remov-

ing any sentences from GlotLID-C, as there is little contamination of UDHR for low-resource languages. We follow here most prior work which has the problem of contamination of UDHR for high-resource languages. We will however remove from GlotLID-C train the sentences causing contamination as part of our next release.

## 4 GlotLID-M

We select FastText (Joulin et al., 2017) as the architecture for GlotLID-M, because it satisfies all requirements outlined in §2 as we will explain now.

We train our FastText model GlotLID-M on GlotLID-C train with 1832 languages. FastText can easily handle the large number of languages in the corpus. Because of this **broad coverage**, **out-of-model cousin errors are reduced**. Although we restrict the number of classes to 1665 for some experiments (e.g., in Table 1), GlotLID-M's classification always uses all 1832 languages to mitigate out-of-model cousin errors. This satisfies the first requirement from §2: GlotLID-M is a useful tool for corpora creation because it has a broad coverage of languages that can occur in raw data.

FastText provides an **open-source** codebase for training, which supports customization and extension of GlotLID-M.

FastText is **easy to use**: It offers a number of language bindings, making it compatible with multiple programming languages (including C++, Python, Java, Node.js, Rust, Ruby, R) and reducing dependency, incompatibility and other software issues.

FastText meets the requirement of **uncertainty assessment** because it provides confidence scores that can serve as thresholds to effectively mitigate noise in the data. For the same reason, FastText also supports **granularity flexibility**: we can accumulate probabilities over language varieties to get a good estimate of the probability of the macrolanguage. To this end, we simply add to the macrolanguage probability the probabilities of its varieties. This way, the system can return appropriate estimates at various levels of granularity.

As a professionally designed and implemented linear classifier, FastText is **efficient**: it had the best throughput of the candidate solutions we tested and can process large corpora with high speed. As a linear model, FastText has the additional advantage of delivering explainable classification decisions. FastText is a multinomial logistic classifier. The input sentence is represented as an average of n-gram

---

[2] https://github.com/cisnlp/GlotScript
[3] https://tatoeba.org/en/downloads

embeddings. This allows us to visualize how much each n-gram contributed to the final prediction. See NLLB Team et al. (2022), Fig. 8, for details.

Taking all these requirements together (and its good LID performance demonstrated in §6 and acceptable calibration in §F), GlotLID-M, based on FastText, is, in our opinion, an excellent tool for supporting our use case, the **creation of high-quality low-resource corpora**.

# 5 Experimental setup

We train GlotLID-M on GlotLID-C train using the hyperparameters in (NLLB Team et al., 2022; Burchell et al., 2023) and otherwise FastText defaults (see §B). Following Arivazhagan et al. (2019), NLLB Team et al. (2022) and Burchell et al. (2023), we perform up-sampling for low resource languages. Sentences from a language $l$ representing $p_l$ of the dataset are sampled proportionally to $p_l^{\frac{1}{T}}$ where $T$ is the temperature. Following NLLB Team et al. (2022) and Burchell et al. (2023), we set $\frac{1}{T} = .3$.

## 5.1 Evaluation data

We evaluate GlotLID-M on GlotLID-C test, FLORES-200 (NLLB Team et al., 2022) and UDHR[4] (Universal Declaration of Human Rights).

While testing on data unseen in training is standard in NLP, the results have to be taken with a grain of salt because there is often a domain mismatch in real-world applications of LID (Caswell et al., 2020; Dunn, 2020). FLORES-200 and UDHR address this concern: they are not part of our training set (however, see discussion in §3) and do not draw on our sources. Many other benchmarks share sources like Wikipedia with us (Thoma, 2018; Haas and Derczynski, 2021; Ahmadi et al., 2023). FLORES-200 and UDHR are also the benchmarks with the broadest available language coverage.

**FLORES-200** is a collection of 842 articles obtained from English-language Wikimedia projects. Each sentence in the articles was translated into 204 distinct language-script combinations, corresponding to 196 distinct languages, and human-verified. It provides 997 sentences for development, 1012 for dev-test and 992 for test. FLORES-200 test is not publicly available. Following prior work, we use dev-test as our FLORES test set.

The level of granularity across language (sub)families varies in FLORES; e.g., it includes nine varieties of Arabic. On the other hand, some languages (e.g., est:Estonian) are only available as macrolanguage. In some cases, FLORES includes both a macrolanguage and varieties, e.g., aka (Akan) and its variety twi (Twi), and zho (Chinese) and its variety yue (Yue Chinese). Although some issues have been reported (see §C.1) with FLORES, we do not have the resources to investigate them, so we use it as is.

**UDHR** consists of more than 500 translations of the "Universal Declaration of Human Rights". 419 translations available from the "UDHR in Unicode" project have a iso-639-3 code that is not "und" (undetermined). We discard short sentences (e.g., consisting of just an article number or the single English word 'missing') by discarding the 35% shortest sentences for each language.

In some cases (e.g., Zulu and Quechua), UDHR contains both a macrolanguage and one of its varieties. We have also seen some issues in UDHR (see §C.2), but we have not extensively investigated these potential problems.

## 5.2 Baselines

Our baselines are FT176,[5] CLD3, NLLB (NLLB Team et al., 2022) and OpenLID (Burchell et al., 2023). The first two were used for filtering the resources OSCAR and mC4 (Kreutzer et al., 2022).

**CLD3.** CLD3 uses an n-gram ($1 \leq n \leq 3$) based neural network model. CLD3 sometimes deviates from established metadata conventions. For example, ISO-639-1 ku refers to kur (Kurdish), but in CLD3 ku refers to its variety kmr (Northern Kurdish). It refers to Hebrew as iw, but the ISO code for Hebrew has changed to he and heb.

**FT176.** FT176 is a FastText model that uses Wikipedia (WP) codes as labels. The documentation of language metadata is sometimes unclear; e.g., FT176 refers to Alemannic German as als although ISO-639-3 als is Tosk Albanian. It refers to the Malay macrolanguage as ms, but unlike ISO-639-3, this does not include ind (Indonesian).

**NLLB and OpenLID.** NLLB and OpenLID are FastText models. Their language label sets are mostly taken from FLORES, so granularity and coverage are similar to FLORES.

**Language metadata matching.** Matching the

---

[4]http://www.unicode.org/udhr/d/

[5]https://fasttext.cc/docs/en/language-identification.html

**Decision rule**

Given an LID classifier $m$, a base set $B$ of languages and a threshold $\theta$, we assign label $\phi(s, m, B, \theta)$ to sentence $s$ as follows:

$$\phi(s, m, B, \theta) = \begin{cases} \text{undetermined} & \text{if } \max_{l \in B} P_m(l|s) < \theta \\ \text{argmax}_{l \in B} P_m(l|s) & \text{otherwise} \end{cases}$$

We distinguish two scenarios: SET! and SET?.

In scenario SET!, the set of languages covered by the evaluation benchmark is known. We restrict a model's predictions to those languages that occur in the benchmark. This means that $B$ is a (proper or improper, see table captions for details) subset of the languages occurring in the benchmark.

In scenario SET?, the set of languages covered by the evaluation benchmark is not known. We do not restrict a model $m$'s predictions: the model considers the entire set of languages it was trained on. This means that $B$ is the set of languages that $m$ was trained on.

Figure 1: Decision rule for assigning classes (i.e., languages) in language identification

metadata of the models to the metadata of the benchmarks (FLORES, UDHR, GlotLID-C) is not easy. First, models do not consistently adhere to standard language codes. In addition, differences in granularity require matching rules. For example, if a benchmark only covers a macrolanguage and none of its varieties, then we consolidate classification decisions for the macrolanguage and its variations into the macrolanguage label. See §E for details on metadata matching.

**Confidence thresholds.** For CLD3, we use .5 and .7, the two preset thresholds in Google's CLD3 repository. For the other three baselines and GlotLID-M, we also use .5, but we use .3 as the second threshold value because .7 severely reduces the number of positive predictions for the FastText models, resulting in low F1.

Prior work has not systematically investigated the effect of confidence thresholding. However, it is of key importance for our use case of creating high-quality corpora for low-resource languages. See §5.3 and §6 for discussion of this point.

### 5.3 Decision rule

Figure 1 defines our decision rule.

**SET! scenario.** When comparing LIDs $m_1$ and $m_2$ (trained on the set of languages $M_1$ and $M_2$) on a benchmark $T$ (supporting the set of languages $B(T)$), many evaluations create a subset $M_1 \cap M_2 \cap B(T)$ and remove all sentences in the benchmark that are labeled with languages outside of $M_1 \cap M_2 \cap B(T)$. SET! evaluation replicates this standard way of evaluating LIDs.

**SET? scenario.** We believe that the SET! scenario makes the LID task unrealistically easy: a portion of the data that could give rise to false positives (data not in $M_1 \cap M_2 \cap B(T)$) is removed. It is particularly unrealistic for our low-resource

scenario. Instead of hundreds of languages that are not supported by all models, we have more than a thousand. *We therefore run evaluations on the data for all languages* – not just for $M_1 \cap M_2 \cap B(T)$. That is, we run evaluations on the entire benchmark $T$, not on the subset in $M_1 \cap M_2 \cap B(T)$. This is the SET? setting in Table 2 where SET? signifies that the LID is not given prior knowledge about which languages occur in $T$. For example, for the comparison of CLD3 and GlotLID-M on FLORES in the top part (SET?) of Table 2, both CLD3 and GlotLID-M are run on the entire FLORES test set. We do not exclude the languages that are present in $T$, but are not part of $M_{\text{CLD3}} \cap M_{\text{GlotLID}}$, i.e., the languages outside of the set of 95 languages common to CLD3 and GlotLID-M.

**Macro average.** For a fair comparison to prior work, we restrict the macro average over languages to a subset of languages in order to replicate the experimental setup of this prior work. This subset is indicated in the tables.

**Realistic evaluation for low-resource scenarios.** We believe that our new evaluation setup SET? better approximates real world situations. In cleaning pipelines, LID models are often presented with an unknown set of languages without prior knowledge. Therefore, it is crucial for an LID to have the capacity to handle unknown languages. This can be achieved by setting a threshold $\theta$ on the confidence scores. If the confidence score for a predicted label falls below the threshold, the model should label the input text as "undetermined". This reduces the risk of languages unknown to the model being incorrectly categorized as a known language (the out-of-model problem). Consequently, when comparing LIDs, it is necessary to apply each model to the entire benchmark.

| Benchmark | | $|L|$ | GlotLID-M, $\theta$=.0 | | GlotLID-M, $\theta$=.5 | |
|---|---|---|---|---|---|---|
| | | | F1↑ | FPR↓ | F1↑ | FPR↓ |
| GlotLID-C | all | 1832 | .940 | .0005 | .938 | .0003 |
| GlotLID-C | subset | 1665 | .977 | .0003 | .973 | .0002 |
| UDHR | all | 374 | .750 | .0015 | .734 | .0007 |
| UDHR | subset | 342 | .784 | .0014 | .770 | .0006 |
| FLORES-200 | all | 196 | .917 | .0042 | .887 | .0013 |
| FLORES-200 | subset | 177 | .957 | .0029 | .924 | .0010 |

Table 1: Performance of GlotLID-M on GlotLID-C, UDHR and FLORES-200 test sets. Subset: restriction to an "operational" subset of languages that are either high-resource or for which GlotLID-M achieves F1≠0 and FPR≤.0005 on GlotLID-C test. $L$: intersection of GlotLID-M languages (all: 1832 or subset: 1665) and languages present in benchmark. Referring to Figure 1, the size of the base set $B$ is either 1832 (all) or 1665 (subset). $L$ is the set of languages over which the macro average is computed. For example, for the last line (FLORES-200 subset), $B$ consists of 1665 languages and the reported macro averages are computed over 177 languages.

## 5.4 Evaluation measures

Unlike some older prior work (Jauhiainen et al., 2019b), we do not use accuracy because classes are highly imbalanced. Instead, we follow recent prior work (NLLB Team et al., 2022; Burchell et al., 2023) and use F1 and false positive rate (FPR). F1 is an aggregate measure of precision and recall, both of which are important: we want accurate classifications decisions (precision) and we do not want to lose too much data (recall). FPR is defined as $\text{FPR} = \frac{\text{FP}}{\text{FP}+\text{TN}}$, where FP is the number of false positives, and TN is the number of true negatives. FPR helps us assess the potentially fatal effect of an even low false positive rate when the negative class is huge – which is the case in our scenario. For example, an FPR of .01 (which *prima facie* may seem ok) for a language $l$ with base frequency .01 can result in a corpus for $l$ that contains 50% noise, an unacceptably high level.

## 6 Results

Table 1 gives results on GlotLID-C test, UDHR and FLORES-200. GlotLID-M does not perform well on some languages. In particular, there are 167 (1832-1665) low-resource languages for which either F1<.01 or FPR>.0005, often due to very small GlotLID-C training sets. The table gives results for "all" 1832 languages as well as for the "subset" of 1665 well-performing languages. We run GlotLID-M in two settings: $\theta$=.0 (i.e., we choose the highest probability class no matter how low its

probability is) and $\theta = .5$ (i.e., we only assign a language label if its probability exceeds .5). See Figure 1 for the definition of our decision rule.

Focusing on the "subset" results for $\theta = .5$, F1 is .973 on GlotLID-C and .924 on FLORES; and FPR is .0002 on GlotLID-C and .0010 on FLORES. This is a very good performance, in particular for the use case of low-resource corpus creation because low FPR means that the resulting corpora will be less contaminated. On UDHR, again for the "subset" results for $\theta = .5$, F1 is .770 and FPR .0006. This is again an encouragingly low FPR, but F1 is quite a bit lower than for GlotLID-C and FLORES. The reason is that we have a domain shift (compared to GlotLID-C) and many more languages (compared to FLORES), resulting in lower F1. Although the UDHR results should be improved further, we will now show that they outperform the state of the art.

Table 2 compares GlotLID-M with four baselines. We consider two evaluation settings (SET? and SET!) and three thresholds $\theta$. The top part of the table (SET?) corresponds to the case where the set of languages in the benchmark is not known, i.e., the LID makes predictions for all languages it was trained on. In contrast, in the SET! setting (bottom part), the set of languages in the benchmark is known, and each LID only makes predictions for those languages. SET? is a more realistic setting, as we usually do not know which languages occur in a corpus that needs to be cleaned.

For the SET? setting, GlotLID-M consistently outperforms CLD3 by a large margin. Taking into account that F1 and FPR should be balanced, we also take it to outperform FT176. Even though GlotLID-M's FPR is slightly higher in some cases, its F1 is better by a large margin, so that it is clearly the better performing system.

On UDHR, GlotLID-M also clearly outperforms OpenLID and NLLB for F1 and FPR by large margins. On FLORES, F1 is slightly worse and FPR slightly better compared with OpenLID and NLLB. We point out that this comparison is not entirely fair since OpenLID and NLLB were designed with FLORES in mind. More importantly, our use case is the creation of low-resource corpora for which UDHR is the more appropriate benchmark.

Comparing results for different thresholds, we observe that increasing $\theta$ lowers F1 (because recall is hurt) and lowers FPR (because precision is increased). This suggests that a higher thresh-

| | | FLORES-200 | | | | | | | | UDHR | | | | | | | |
|---|---|---|---|---|---|---|---|---|---|---|---|---|---|---|---|---|---|
| | | CLD3 | | FT176 | | OpenLID | | NLLB | | CLD3 | | FT176 | | OpenLID | | NLLB | |
| | | $|L|=96$ | | $|L|=108$ | | $|L|=195$ | | $|L|=188$ | | $|L|=100$ | | $|L|=124$ | | $|L|=159$ | | $|L|=172$ | |
| LID Model | $\theta$ | F1↑ | FPR↓ | F1↑ | FPR↓ | F1↑ | FPR↓ | F1↑ | FPR↓ | F1↑ | FPR↓ | F1↑ | FPR↓ | F1↑ | FPR↓ | F1↑ | FPR↓ |
| **SET?** | | | | | | | | | | | | | | | | | |
| baselines | .0 | .753 | .0098 | .775 | .0090 | .923 | .0051 | .947 | .0053 | .544 | .0099 | .566 | .0079 | .645 | .0056 | .641 | .0051 |
| baselines | $\theta_1$ | .779 | .0081 | .816 | .0033 | .923 | .0050 | .948 | .0051 | .576 | .0081 | .644 | .0025 | .676 | .0046 | .677 | .0040 |
| baselines | $\theta_2$ | .799 | .0060 | .796 | .0021 | .923 | .0044 | .947 | .0047 | .618 | .0060 | .647 | .0014 | .718 | .0034 | .717 | .0030 |
| GlotLID-M | .0 | .978 | .0051 | .987 | .0042 | .916 | .0043 | .947 | .0035 | .868 | .0033 | .868 | .0030 | .848 | .0020 | .847 | .0019 |
| GlotLID-M | .3 | .980 | .0042 | .987 | .0037 | .898 | .0020 | .927 | .0019 | .881 | .0028 | .879 | .0026 | .846 | .0015 | .844 | .0015 |
| GlotLID-M | .5 | .980 | .0031 | .987 | .0029 | .886 | .0014 | .916 | .0013 | .903 | .0023 | .890 | .0021 | .847 | .0012 | .846 | .0011 |
| **SET!** | | | | | | | | | | | | | | | | | |
| baselines | .0 | .952 | .0104 | .881 | .0093 | .923 | .0051 | .950 | .0053 | .922 | .0101 | .739 | .0081 | .881 | .0063 | .854 | .0058 |
| GlotLID-M | .0 | .983 | .0104 | .991 | .0093 | .922 | .0051 | .954 | .0053 | .952 | .0100 | .927 | .0081 | .926 | .0064 | .925 | .0060 |

Table 2: Evaluation of LID performance. Top ("SET?"): The set of languages is not known, i.e., each LID makes predictions for all languages it was trained on. Bottom ("SET!"): The set of languages is known: each LID only makes predictions for languages that occur in the benchmark. For the more realistic "SET?" setting, GlotLID-M outperforms the baselines on UDHR (which we take to be the best benchmark for the low-resource case) assuming a good tradeoff between FPR and F1 is desired; it either matches or outperforms them on FLORES. Let $M_i$ be the set of languages model $m_i$ was trained on and $B(T)$ the set of languages covered by benchmark $T$. Then F1 and FPR are averages over $L = M_1 \cap M_2 \cap B(T)$ when comparing models $m_1$ and $m_2$; this is indicated in the third row of table, e.g., $|L| = 96$ for $m_1 =$ CLD3, $m_2 =$ GlotLID. $\theta_1$=.5 for CLD3, $\theta_1$=.3 for FT176, OpenLID and NLLB. $\theta_2$=.7 for CLD3, $\theta_2$=.5 for FT176, OpenLID and NLLB. Referring to Figure 1, the base set $B$ in SET? has size 103 for CLD3, 176 for FT176, 195 for OpenLID, 211 for NLLB and 1832 for GlotLID-M (i.e., the languages the LID was trained on). For scenario SET!, $B = L$, i.e., $B = M_1 \cap M_2 \cap B(T)$. For example, $|B| = 96$ (for both CLD3 and GlotLID) for the four cells in the the SET! rows and the CLD3 columns in the lower left corner of the table. The best result in each column is **bolded**, and the second-best result is underlined.

old should be used since lower FPR will result in low-resource corpora with less contamination from high-resource languages.

For the less realistic SET! setting, GlotLID-M performs better than CLD3 and FT176 and comparably to OpenLID and NLLB. Overall, GlotLID-M clearly outperforms all baselines for the low-resource corpus creation use case.

To analyze variance of results, we ran three GlotLID experiments with different initial seeds on the 200 languages with the most data, splitting the data into 80% train and 20% test. The F1 score was .991 each time. This indicates that the variance of FastText in this task (and by extension GlotLID) is negligible.

# 7 Analysis

In this section, we analyze the GlotLID-M results summarized in Table 1 ($\theta$=.0, "all") for our main use case, the creation of high-quality corpora. We address four questions. (i) For which languages do we get a high number of false positives? (ii) For which languages do we produce a corpus with a high contamination rate? (iii) For which languages does learning completely fail? (iv) Is it more realistic to evaluate LID on a balanced test set (as in prior work) or on one that is skewed in favor of high-resource languages?

**Most errors.** We first analyze languages with a high number of errors. Table 3 (top, "most errors") gives for each of the three benchmarks the five languages that have the highest number of errors (column "language"). "FP" is the number of false positives, "cl" the ratio of true positives to all positives (that is the "cleanness" of the corpus), "top FP source" the language that contributed most of the errors and "%" is the portion of these false positives as a percentage of all false positives. We use the cl measure in our analysis because it is ultimately the measure we want to optimize to produce high-quality low-resource corpora. Note that cl (the denominator is the total number of positive sentences) is not directly related to FPR (the denominator is the number of sentences that do not belong to the language). cl is a more direct measure of the utility of the resulting corpus of a low-resource language (e.g., for training a language model) than FPR.

Most of the fifteen pairs of "conflated" languages shown in the table are closely related languages: varieties of Arabic (Standard, Najdi, Egyptian and Levantine), Persian (Iranian, Dari), Chinese (Mandarin, Yue, Wu, Hakka), English (Standard, Liberian), Quechua (Huallaga Huánuco, Huamalíes-Dos de Mayo Huánuco), Finnic (Finnish, Karelian), Slavic (Russian, Church Slavic), Bihari (Bihari, Bhojpuri) and Hindi (Standard, Awadhi). In many of these cases, speakers of

Table 3 with three benchmark column groups (FLORES-200, UDHR, GlotLID-C):

| | language (FLORES-200) | FP | cl | top FP source | #FP | % | language (UDHR) | FP | cl | top FP source | #FP | % | language (GlotLID-C) | FP | cl | top FP source | #FP | % |
|---|---|---|---|---|---|---|---|---|---|---|---|---|---|---|---|---|---|---|
| **most errors** | arb:St Arabic | 3787 | .18 | ars:Najdi Arabi | 829 | .22 | cmn:Mandarin Ch | 596 | .38 | chr:Cherokee | 81 | .14 | spa:Spanish | 1952 | .34 | pid:Piaroa | 156 | .08 |
| | arz:Egyptian Ar | 1726 | .32 | apc:Levantine A | 440 | .25 | qub:Huallaga Hu | 247 | .00 | qvh:Huamalíes-D | 55 | .22 | eng:English | 1168 | .46 | lir:Liberian En | 254 | .22 |
| | pes:Ir. Persian | 1495 | .40 | prs:Dari | 905 | .61 | fin:Finnish | 224 | .22 | krl:Karelian | 138 | .62 | rus:Russian | 1057 | .49 | chu:Church Slav | 661 | .63 |
| | cmn:Mandarin Ch | 1008 | .00 | yue:Yue Chinese | 1008 | .99 | wuu:Wu Chinese | 172 | .24 | hak:Hakka Chine | 44 | .26 | bho:Bhojpuri | 882 | .50 | bih:Bihari Lgs | 854 | .97 |
| | hin:Hindi | 977 | .51 | awa:Awadhi | 693 | .71 | rus:Russian | 157 | .28 | niv:Gilyak | 44 | .28 | lir:Liberian En | 712 | .47 | din:Dinka | 174 | .24 |
| **most noisy** | arb:St Arabic | 3787 | .18 | ars:Najdi Arabi | 829 | .22 | evn:Evenki | 36 | .23 | oaa:Orok | 19 | .53 | rus:Russian | 1057 | .49 | chu:Church Slav | 661 | .63 |
| | arz:Egyptian Ar | 1726 | .32 | apc:Levantine A | 440 | .25 | quz:Cusco Quech | 82 | .40 | qxu:Arequipa-La | 61 | .74 | eng:English | 1168 | .46 | lir:Liberian En | 254 | .22 |
| | prs:Dari | 338 | .24 | pbt:S Pashto | 310 | .92 | hrv:Croatian | 84 | .42 | bos:Bosnian | 39 | .46 | spa:Spanish | 1952 | .34 | pid:Piaroa | 156 | .08 |
| | dyu:Dyula | 255 | .25 | bam:Bambara | 255 | .99 | tzm:C Atlas Tam | 52 | .02 | zgh:St Moroccan | 52 | .99 | crq:Iyo'wujwa C | 347 | .47 | crt:Iyojwa'ja C | 347 | .99 |
| | apc:Levantine A | 161 | .42 | ajp:S Levantine | 70 | .43 | uzn:N Uzbek | 72 | .46 | cbu:Candoshi-Sh | 16 | .22 | crt:Iyojwa'ja C | 698 | .48 | crq:Iyo'wujwa C | 697 | .99 |
| **no positives** | | | | | | | tet:Tetum | 0 | .00 | | | | sck:Sadri | 0 | .00 | | | |
| | | | | | | | hsn:Xiang Chine | 0 | .00 | | | | chg:Chagatai | 0 | .00 | | | |
| | | | | | | | abk:Abkhazian | 0 | .00 | | | | liv:Liv | 0 | .00 | | | |
| | | | | | | | vep:Veps | 0 | .00 | | | | gbm:Garhwali | 0 | .00 | | | |
| | | | | | | | niv:Gilyak | 0 | .00 | | | | tmw:Temuan | 0 | .00 | | | |
| **hi resource** | arb:St Arabic | 3787 | .99 | ars:Najdi Arabi | 829 | .22 | cmn:Mandarin Ch | 596 | .99 | chr:Cherokee | 81 | .14 | cmn:Mandarin Ch | 367 | .99 | wuu:Wu Chinese | 208 | .57 |
| | dzo:Dzongkha | 10300 | .09 | bod:Tibetan | 10300 | .99 | fin:Finnish | 224 | .99 | krl:Karelian | 138 | .62 | eng:English | 1267 | .99 | lir:Liberian En | 254 | .20 |
| | hin:Hindi | 977 | .99 | awa:Awadhi | 693 | .71 | hin:Hindi | 76 | .99 | mai:Maithili | 24 | .32 | hin:Hindi | 488 | .99 | bho:Bhojpuri | 98 | .20 |
| | rus:Russian | 1 | .99 | bul:Bulgarian | 1 | .99 | rus:Russian | 256 | .99 | eng:English | 100 | .39 | rus:Russian | 1156 | .99 | chu:Church Slav | 661 | .57 |
| | spa:Spanish | 10 | .99 | ast:Asturian | 7 | .70 | spa:Spanish | 62 | .99 | agr:Aguaruna | 20 | .32 | spa:Spanish | 1952 | .99 | pid:Piaroa | 156 | .08 |

Table 3: Analysis of the GlotLID-M runs with settings $\theta$=.0, SET? from Table 1 and Table 2. "most errors": languages with the most false positives. "most noisy": a sample of languages with cleanness between 0 and .5. "no positives": a sample of languages without positives. "hi resource": a more realistic setting in which the distribution is skewed in favor of high-resource languages. For each "language", we give the number of false positives ("FP"), the cleanness of the resulting corpus ("cl": ratio true positives to all positives), its most conflated language ("top FP source"), FP contributed by that language and the ratio of the two FP numbers ("%"). To save space, we write .99 for 1.00.

one variety of the pair also have good knowledge of the other; e.g., many speakers of Arabic varieties know Standard Arabic. The two Quechua varieties are spoken in neighboring areas of Peru. The quantitatively largest use of Church Slavic (which may be reflected in the size of our corpora) is in Russia by Russian speakers.

Arabic, Chinese and English (and perhaps also Hindi, Persian and Bihari) are diglossic linguistic communities. There may be a lack of clear separation between the two conflated varieties in the available corpora because speakers switch back and forth between more formal and less formal ways of speaking depending on factors like context, audience and subject. This type of fluid switching between languages often occurs in a single sentence or conversation, i.e., it manifests as code switching. As a result, much of the text (and speech) produced in one language may be mixed with the other language. New methods will have to be developed to deal with these quite complex challenges of creating training corpora for language identification; see also (Aguilar et al., 2020).

Apart from these related languages, at least four conflated language pairs in Table 3 are clear errors: Mandarin/Cherokee, Russian/Gilyak, Spanish/Piaroa and Liberian English/Dinka. Similar to the situation we described for the closely related languages, Gilyak (resp. Piaroa) is spoken in an area where Russian (resp. Spanish) is the dominant official language. This means that our training cor-

pora will need to be improved: they most likely contain many sentences labeled as Gilyak/Piaroa that are partially or completely Russian/Spanish. We leave it to future work to revisit and improve our corpus selection and preprocessing methodology to address this data quality problem.

GlotLID-M confuses Mandarin and Cherokee because our Cherokee training data do not cover the Cherokee syllabary script. Sentences written in this script are returned with a close to uniform distribution over several other scripts, including Chinese, Japanese and Thai, which explains the confusion. The Dinka test set is noisy. In a manual inspection, we found 377 sentences that are clearly English, not Dinka. Because GlotLID-M did not learn very well to discriminate English and Liberian English, 174 of these 377 sentence were classified as Liberian English.

**Most noisy corpora.** The second part of Table 3 ("most noisy") gives, for each benchmark, a random selection of five languages whose cleanness score cl (ratio of true positive to all positives) is in the range 0<cl<.5. The total number of languages in this range is 9 for FLORES, 27 for UDHR and 6 for GlotLID-C. Again, most of the conflated pairs are closely related languages as in the last section. Additional pairs that occur here are Dyula/Bambara, Evenki/Orok, Croation/Bosnian, Berber languages (Standard Moroccan Tamazight, Atlas Tamazight) and two varieties of Chorote (Iyo'wujwa, Iyojwa'ja). The resulting

corpora are noisy, an issue that we will have to address in future work.

**No positives.** Part 3 of Table 3 ("no positives") gives five random examples from languages for which there was not a single positive classification. There were no such languages for FLORES.

For UDHR, we identified two reasons. (i) Performance on GlotLID-C is good, but poor on UDHR. Tetum is an example. The most likely cause is a domain shift or some other big train/test difference. (ii) The training set is too small (less than 30 sentences): hsn (Xiang Chinese), abk (Abkhazian), vep (Veps) and niv (Gilyak) are in this class.

For the five GlotLID-C random examples with no positives, the reason is also that the training sets were too small (less than 40 sentences): sck (Sadri), chg (Chagatai), liv (Liv), gbm (Garhwali) and tmw (Temuan). We should have set a higher threshold for minimum size of the training corpus. Note that the number of 1665 languages that we use throughout the paper already reflects this insight. Even though we train on 1832 languages, we claim reasonable performance for only 1665 (Table 1).

**Test set skewed in favor of high-resource.** FLORES and UDHR test sets are balanced: high-resource and low-resource languages have about the same size. Following this model, we constructed the test set of GlotLID-C in the same way. F1 is independent of this distribution, but FPR and cleanness ("cl") are strongly dependent on it. The Spanish corpus generated by GlotLID-M on GlotLID-C test has a dismal cleanness of only .34. Is this a problem for GlotLID-M?

We believe the answer is no, as the corpora we run LID on will have a distribution skewed in favor of high-resource languages. To simulate this more realistic scenario, the last part of Table 3 ("hi resource") gives five selected languages for each benchmark where we have inflated the subsets for high-resource languages by a factor of 100. For example, instead of a single copy of the English part of FLORES, the test set now contains 100 copies.

We see in Table 3 that this results in clean corpora (cl=.99) for each of the fourteen high-resource languages shown: Standard Arabic, Hindi, Russian, Spanish (FLORES); Mandarin, Finnish, Hindi, Russian, Spanish (UDHR); Mandarin, English, Hindi, Russian, Spanish (GlotLID-C). As an example, looking at Spanish for GlotLID-C (the first and last lines in the table), the number of false positives (1952) and the number of false positives

contributed by the low-resource language Piaroa (156) are the same. But since the size of Spanish is increased 100x, its cleanness improves from .34 for the unrealistic uniform distribution to .99 for the realistic skewed distribution. Thus, as we would expect, LID for high-resource languages is a relatively easy problem and this does not change much if we run a broad-coverage LID like GlotLID-M.

Conversely, LID numbers for low-resource languages can be *negatively* affected. The Dzongkha corpus generated from FLORES in the uniform setting has 103 false positives and a cleanness of .91 (not shown). In the skewed setting, making Tibetan a high-resource language causes 10,300 false positives from Tibetan to leak into Dzongkha, reducing its cleanness to an unacceptable .09.

This discussion suggests that the established evaluation methodology for LID is unsatisfactory. We recommend that future work considers both unifom and skewed test sets to better assess how LID is expected to perform in the real world.

This analysis demonstrates how much harder LID becomes when we represent as large and diverse sets of languages as we do. What we have shown is that there is a real danger of creating corpora that are badly contaminated. To address this, we need to develop methodologies and resources that better handle low-resource languages.

Based on the analysis described in this section we created and open-sourced a much improved version of the UDHR test set for evaluation of LID.[6] All UDHR results in this paper are based on the version of the UDHR test set descibed in §5.1.

## 8 Conclusion

We create GlotLID-C, an LID resource that covers 1832 languages, several times more than prior work. We introduce GlotLID-M, an open-source LID that covers 1665 languages with good results. The comparison of GlotLID-M against four LID baselines shows superior performance for the low-resource use case. In future research, we would like to improve quality of our training corpora and add more low-resource languages in to GlotLID. We hope GlotLID will be a valuable resource in creating higher-quality corpora for low-resource languages.

---

[6]https://huggingface.co/datasets/cis-lmu/udhr-lid

## Limitations

(1) We publish list of GlotLID-C data sources as part of this work. There is no other LID benchmark available that covers as many languages as GlotLID-C does. GlotLID-C, FLORES and UDHR all have drawbacks as evaluation datasets for LID. An LID trained on GlotLID-C train and tested on GlotLID-C test will often find the same domain in the test set as in the training set. It is well known that this results in overly optimistic evaluation numbers. FLORES and UDHR consist of data that were not originallly produced in each language. Rather, they were translated from high-resource languages. The same is true to a lesser extent for GlotLID-C. Translated language is only an imperfect evaluation benchmark because it can differ greatly from natural language data, i.e., translationese is often not a good model of natural language data.

(2) Many corpora for the lowest resource languages are derived from religious sources. It should be noted that many Bible translations do not reflect actual language use.

(3) We do not conduct hyperparameter search and instead use the hyperparameters employed by previous studies. However, conducting such a search can make our findings more robust, considering the difference in the number of languages included in our study compared to the prior work.

(4) Although we tried our best to select the most suitable LIDs as the baseline. We could not compare against all of the LID models. This includes CLD2 (McCandless, 2010), Equilid (Jurgens et al., 2017), Langdetect (Shuyo, 2010), langid.py (Lui and Baldwin, 2012), whatlang (Brown, 2014b,a), idNet (Dunn, 2020), Franc (Wormer, 2014), AfroLID (Adebara et al., 2022), HeLI-OTS (Jauhiainen et al., 2022), transliterate[7], whatthelang[8], whatlang-rs[9], lingua[10], Google/Bing Online, LanideNN (Kocmi and Bojar, 2017), Paasaa[11], Q-LID (Ren et al., 2022), UDLDI (Goswami et al., 2020), PALI (Ahmadi et al., 2023), SS-LID (Caswell et al., 2020; Bapna et al., 2022; Kudugunta et al., 2023) and TextCat (Cavnar et al., 1994).

---

[7] https://github.com/barseghyanartur/transliterate
[8] https://github.com/indix/whatthelang
[9] https://github.com/greyblake/whatlang-rs
[10] https://github.com/pemistahl/lingua
[11] https://github.com/minibikini/paasaa

## Ethics Statement

We here highlight key ethical considerations for GlotLID.

**Data.** The data used in our study comes from openly available (but not necessarily freely redistributable) datasets, including resources previously published by researchers, publishers, and translators. We ensured that the data collection process complied with licensing of each dataset.

**Bias.** We recognize potential biases towards higher resource languages. We conducted a comprehensive analysis of errors and evaluated their impact on our results.

**Inclusivity.** We acknowledge the challenges associated with low-resource languages and have taken steps to include a diverse range of languages in our study.

**Ethical Use.** We have demonstrated both positive and negative outcomes of applying GlotLID-M as an LID tool. We acknowledge that GlotLID-M has a high error rate for some low-resource languages. This means that there is a potential risk of excluding low-resource languages during the collection and processing of NLP corpora.

**Transparency.** We provide detailed descriptions of our methodology, model architecture, and evaluation process. Additionally, we make our research artifacts, including model, code, and list of data sources openly available to foster collaboration and reproducibility.

## 9 Acknowledgements

We would like to thank anonymous reviewers. This work was funded by the European Research Council (grant #740516).

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

## A List of data sources

- **Wikipedia articles:** Wikipedia dumps,[12] WiLI-2018 (Thoma, 2018), Leipzig corpora-wikipedia split (Goldhahn et al., 2012)
- **News:** BBC News (Hasan et al., 2021), Global Voices (Tiedemann, 2012), Leipzig corpora-news split (Goldhahn et al., 2012), SETIMES (Tiedemann, 2012)
- **Translation:** NLLB Seed (NLLB Team et al., 2022)
- **Religious:** PBC (Mayer and Cysouw, 2014), Jehovah's Witnesses,[13] 1000Langs[14]
- **Crowdsourcing:** Tatoeba[15]
- **Multiple domain:** MT-560 (Gowda et al., 2021; Tiedemann, 2012; Burchell et al., 2023; Post et al., 2012; Ziemski et al., 2016; Rozis and Skadiņš, 2017; Kunchukuttan et al., 2018; Qi et al., 2018; Zhang et al., 2020; Bojar et al., 2013, 2014, 2015, 2016, 2017, 2018; Barrault et al., 2019, 2020), LTI (Brown, 2012), Arabic (Zahir, 2022; Alsarsour et al., 2018; Abu Kwaik et al., 2018; Medhaffar et al., 2017; Meftouh et al., 2015; Zaidan and Callison-Burch, 2011; El-Haj et al., 2018; Bouamor et al., 2019), Persian (Pilevar et al., 2011; Kashefi, 2018), Turkic (Mirzakhalov et al., 2021), Bhojpuri (Ojha, 2019), Cantonese (Luke and Wong, 2015), Guaraní (Góngora et al., 2022), Manipuri (Huidrom et al., 2021)
- **Government domain:** Autshumato (Groenewald and du Plooy, 2010)

We also introduce additional data sources suited for LID, but they are not included in the training of the version of GlotLID-M discussed in the paper:

- **Crowdsourcing:** CommonVoice v11 (Ardila et al., 2020)
- **Web:** Wanca 2016 (Jauhiainen et al., 2019a)
- **News:** GlotSparse[16] which is a collection of news websites in low-resource languages, MasakhaNEWS (Adelani et al., 2023), Goud.ma (Issam and Mrini, 2022), AI4D Siminyu et al. (2021), Radio Ramogi,[17]

smugri (Yankovskaya et al., 2023), finno-ugric (Yankovskaya et al., 2023)
- **Trasnlation:** GlotStoryBook[18] which is a collection of children storybooks in 174 languages from Global Storybooks[19], AfriQA Ogundepo et al. (2023), smugri-flores (Yankovskaya et al., 2023)
- **Multiple domain:** Universal Dependencies v2.12 (Nivre et al., 2020), Abkhaz National Corpus[20]
- **Lyrics:** lyricstranslate[21]
- **Government domain:** Vuk'uzenzele (Lastrucci et al., 2023)

Specifically, GlotSparse[16] and GlotStoryBook[18] are two corpora that compiled as a side of this project to include more languages and domains for LID.

## B GlotLID-M hyperparameters

We provide the hyperparameters used to train the GlotLID-M in Table 4.

| argument | description | value |
|---|---|---|
| -minCount | minimal number of word occurrences | 1000 |
| -minCountLabel | minimal number of label occurrences | 0 |
| -wordNgrams | max length of word ngram | 1 |
| -bucket | number of buckets | 1e6 |
| -minn | min length of char ngram | 2 |
| -maxn | max length of char ngram | 5 |
| -loss | loss function | softmax |
| -dim | size of word vectors | 256 |
| -epoch | number of epochs | 2 |
| -lr | learning rate | .8 |

Table 4: GlotLID-M training hyperparameters

## C Evaluation data issues

### C.1 FLORES-200

There are some mistakes in the FLORES-200 dataset which have been raised by the community.

For example, in a GitHub issue,[22] it is pointed out that yue_Hant and zho_Hant should actually be very easy to distinguish from each other, and the Cantonese (Yue Chinese, yue_Hant) data in FLORES-200 is completely wrong.

---

[12] https://dumps.wikimedia.org/
[13] https://www.jw.org/
[14] https://github.com/ehsanasgari/1000Langs
[15] https://tatoeba.org/en/downloads
[16] https://github.com/cisnlp/GlotSparse
[17] https://github.com/Pogayo/Luo-News-Dataset

[18] https://github.com/cisnlp/GlotStoryBook
[19] https://github.com/global-asp/
[20] https://clarino.uib.no/abnc/page
[21] https://lyricstranslate.com/
[22] https://github.com/facebookresearch/flores/issues/61

In another issue[23], it is mentioned that the Central Atlas Tamazight (tzm) is actually in Standard Moroccan Tamazight (zgh), as confirmed by a native speaker of Central Atlas Tamazight.

## C.2 UDHR

There are some mistakes with UDHR. For example, both ckb and kmr files are the same. ckb is known for the Arabic script, although it can also be written in Latin. There are also some files that the writing system is not in popular use (based on Kargaran et al. (2023) metadata):

- ckb_Latn (Arabic script is in use.)
- azb_Latn (Arabic script is in use.)
- khk_Mong (Cyrillic script is in use.)
- vie_Hani (Latin script is in use.)

## D Performance of GlotLID-M per language

The list of languages used to train GlotLID-M, along with the corresponding amount of available data and detailed results for each language, can be found in Tables 5-29

## E Language metadata matching

The per-language comparison between GlotLID-M and the baselines (CLD3, FT176, OpenLID, and NLLB) for each benchmark in scenario is as follows:

**FLORES-200.** (i) GlotLID-M vs CLD3: Tables 30-31 (ii) GlotLID-M vs FT176: Tables 32-33 (iii) GlotLID-M vs OpenLID: Tables 34-35 (iv) GlotLID-M vs NLLB: Tables 36-37

**UDHR.** (i) GlotLID-M vs CLD3: Tables 38-41 (ii) GlotLID-M vs FT176: Tables 42-45 (iii) GlotLID-M vs OpenLID: Tables 46-47 (iv) GlotLID-M vs NLLB: Tables 48-49

The underlined results in each table show the best result for each model, and the **bold** result indicates the overall best result.

The tables also contain the metadata matching rules we define. Column "isocode639-3" contains the ISO 639-3 code of each language. This corresponds to the class used by GlotLID-M (since all our classes are ISO 639-3 codes). The following columns contain the codes that we mapped the ISO

---

[23]https://github.com/facebookresearch/flores/issues/63

---

639-3 codes to. For example, Table 30 indicates that we map ISO 639-3 code fas to pes/prs in FLORES. In other words, to evaluate our performance for the language fas in FLORES, we (only conceptually) create a new test set in which all sentences labeled as pes or prs in FLORES, are relabeled as fas.

## F Calibration

As stated in §2, an LID model should provide a calibrated confidence measure in addition to its prediction. Reliability diagrams illustrate model calibration (DeGroot and Fienberg, 1983; Niculescu-Mizil and Caruana, 2005). These diagrams use expected sample accuracy as a function of confidence. If the model is perfectly calibrated, then the diagram plots the identity function.

We provide the reliability diagram for GlotLID-M on GlotLID-C test in Figure 2. For GlotLID-C test, the plot is nearly close to the identity function. However, for some of the low confidence scores, it's not calibrated. This mostly happens because we included so many languages in our models, and some of these languages are very similar to each other or have small training sizes.

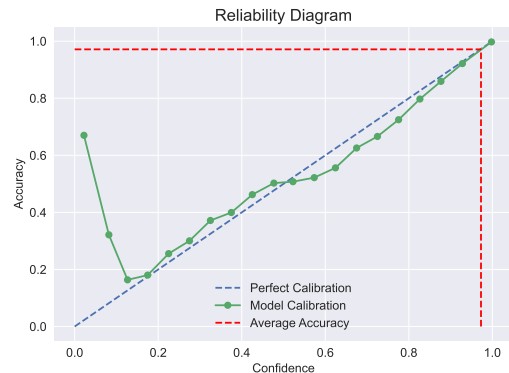

Figure 2: Reliability diagram for GlotLID-M on GlotLID-C test

## G Analysis of Baseline Results

In this section, we analyze baseline results summarized in Table 2. For a detailed breakdown of results for each language in scenario SET?, see §E.

**CLD3**: The largest performance gap between SET? and SET! is for CLD3. This could be attributed to the different architecture of CLD3, which performs better when the set is known. Also, since CLD3 supports fewer languages compared

to other models, it has an advantage in terms of supporting a higher percentage of high-resource languages within its base set.

CLD3 achieves a lower F1 score than the other FastText-based models in the UDHR benchmark in scenario SET?. However, in scenario SET! for the UDHR, it outperforms all other FastText-based models on F1. Additionally, the good performance achieved in both benchmarks (.952 and .922) in scenario SET! illustrates the robustness of this LID.

**FT176.** When comparing FT176 and GlotLID-M in FLORES-200, GlotLID-M achieves the highest F1 scores in FLORES. This may be attributed to the fact that all languages in FT176 are supported by Wikipedia, and GlotLID-M has strong support for these languages. On the other hand, FT176 has the worst overall performance among the FastText models.

**NLLB and OpenLID.** In FLORES-200, OpenLID and NLLB have an advantage in scenario SET?, as this scenario closely aligns with SET!. Both models provide near-complete support for the languages available in FLORES-200. For the rest of the models and benchmarks, GlotLID-M displays a marked difference in performance and takes the lead. Among the baseline models in scenario SET?, OpenLID shows the best performance in both benchmarks. We will now investigate which languages OpenLID performs better or worse in compared to GlotLID-M.

In scenario SET?, when comparing OpenLID and GlotLID-M on FLORES-200, most of the time the per language scores are very close to each other (see Tables 34- 35). However, there are cases where OpenLID performs noticeably better, for example, with a .39 improvement for azb and .29 for awa. On the other hand, GlotLID-M performs better by .2 for zho, which can be attributed to the poorer performance of OpenLID in zho_Hant. Additionally, both models perform poor on languages such as Yue Chinese, which could be attributed to an issue with FLORES-200 (see §C.1). However, this situation is quite different for UDHR, as GlotLID-M supports more languages than OpenLID, GlotLID-M performs much better in handling languages that are outside the intersection of both models' base sets.

| iso639-3 | Language | \|Sentences\| | GlotLID-C F1↑ | GlotLID-C FPR↓ | FLORES-200 F1↑ | FLORES-200 FPR↓ | UDHR F1↑ | UDHR FPR↓ |
|---|---|---|---|---|---|---|---|---|
| aai | Arifama-Miniafia | 7954 | 0.9995 | 0.0 | | | | |
| aak | Ankave | 7784 | 1.0 | 0.0 | | | | |
| aau | Abau | 7888 | 1.0 | 0.0 | | | | |
| aaz | Amarasi | 9071 | 0.999 | 0.0 | | | | |
| aba | Abé | 28725 | 0.81648 | 0.00371 | | | | 0.00416 |
| abk | Abkhazian | 28 | | | | | | |
| abn | Abua | 61385 | 0.94699 | 0.00091 | | | | |
| abq | Abaza | 48 | 1.0 | 0.0 | | | | |
| abt | Ambulas | 15129 | 0.999 | 6e-05 | | | | |
| abx | Inabaknon | 7855 | 0.999 | 3e-05 | | | | |
| aby | Aneme Wake | 7070 | 0.998 | 3e-05 | | | | 0.00011 |
| abz | Abui | 16642 | 0.98171 | 6e-05 | | | | |
| aca | Achagua | 4408 | 0.99929 | 0.0 | | | | |
| acd | Gikyode | 7884 | 1.0 | 0.0 | | | | |
| ace | Achinese | 31228 | 0.98176 | 0.00014 | 0.95503 | 0.00579 | 0.90909 | 0.0012 |
| acf | Saint Lucian Creole French | 21811 | 0.97035 | 0.00019 | | | | |
| ach | Acoli | 42735 | 0.995 | 0.00017 | | | | 0.00044 |
| acm | Mesopotamian Arabic | 4902 | 0.79445 | 0.00091 | 0.01562 | 0.00023 | | |
| acn | Achang | 7919 | 0.9995 | 0.0 | | | | 0.00099 |
| acq | Ta'izzi-Adeni Arabic | 1598 | 0.60947 | 0.00058 | 0.00197 | 6e-05 | | |
| acr | Achi | 27176 | 0.996 | 0.00011 | | | | 0.00044 |
| acu | Achuar-Shiwiar | 11509 | 0.999 | 0.0 | | | 0.584 | 0.00011 |
| ada | Adangme | 217601 | 0.998 | 8e-05 | | | 0.91045 | 0.00131 |
| ade | Adele | 7924 | 0.9985 | 6e-05 | | | | |
| adh | Adhola | 8975 | 0.99549 | 6e-05 | | | | 0.00055 |
| adi | Adi | 30717 | 0.9945 | 0.00017 | | | | 0.00022 |
| adj | Adioukrou | 7883 | 0.998 | 0.0 | | | | |
| adl | Galo | 7956 | 0.998 | 3e-05 | | | | |
| ady | Adyghe | 4885 | 0.99596 | 0.0 | | | 0.83495 | 0.0 |
| adz | Adzera | 1489 | 0.99541 | 0.0 | | | | |
| aeb | Tunisian Arabic | 26935 | 0.86433 | 0.00231 | 0.28501 | 0.00199 | | |
| aer | Eastern Arrernte | 9577 | 0.9995 | 0.0 | | | | 0.00011 |
| aeu | Akeu | 7853 | 1.0 | 0.0 | | | | 0.00504 |
| aey | Amele | 9061 | 0.9995 | 0.0 | | | | |
| afb | Gulf Arabic | 136 | 0.2 | 3e-05 | | | | |
| afh | Afrihili | 79 | 0.75862 | 3e-05 | | | | |
| afr | Afrikaans | 1436086 | 0.98216 | 0.00074 | 1.0 | 0.0 | 0.95238 | 0.00066 |
| agd | Agarabi | 7917 | 1.0 | 0.0 | | | | |
| agg | Angor | 7788 | 1.0 | 0.0 | | | | |
| agm | Angaataha | 7889 | 0.9995 | 0.0 | | | | |
| agn | Agutaynen | 7844 | 0.9985 | 6e-05 | | | | |
| agr | Aguaruna | 23895 | 0.93626 | 0.0005 | | | 0.81429 | 0.00033 |
| agt | Central Cagayan Agta | 7554 | 0.9985 | 3e-05 | | | | |
| agu | Aguacateco | 7928 | 0.9985 | 0.0 | | | | |
| agw | Kahua | 35771 | 0.998 | 0.00011 | | | | |
| agx | Aghul | 1150 | 0.97898 | 0.00011 | | | | |
| aha | Ahanta | 18467 | 0.999 | 0.0 | | | | |
| ahk | Akha | 134957 | 1.0 | 0.0 | | | | |
| aia | Arosi | 7804 | 0.9995 | 0.0 | | | | |
| aii | Assyrian Neo-Aramaic | 10736 | 1.0 | 0.0 | | | | |
| aim | Aimol | 7949 | 0.9975 | 3e-05 | | | | 0.00022 |
| ain | Ainu (Japan) | 324 | 0.91111 | 3e-05 | | | | 0.00022 |
| ajg | Aja (Benin) | 35237 | 0.99245 | 3e-05 | | 0.00028 | 0.64516 | 0.00722 |
| aji | Ajië | 9916 | 0.998 | 3e-05 | | | | 0.00175 |
| ajp | South Levantine Arabic | 28203 | 0.75111 | 0.00341 | 0.10836 | 0.00102 | | |
| ajz | Amri Karbi | 7956 | 0.999 | 3e-05 | | | | 0.00033 |
| aka | Akan | 1174 | 0.95114 | 0.0 | 0.99852 | 0.0 | | |
| akb | Batak Angkola | 7940 | 0.98993 | 8e-05 | | | | |
| ake | Akawaio | 7933 | 1.0 | 0.0 | | | | |
| akh | Angal Heneng | 7756 | 0.9995 | 3e-05 | | | | |
| akl | Aklanon | 28 | | | | | | |
| akp | Siwu | 7919 | 0.9985 | 3e-05 | | | | |
| ald | Alladian | 7939 | 1.0 | 0.0 | | | | |
| alj | Alangan | 7877 | 0.999 | 6e-05 | | | | 0.00011 |
| aln | Gheg Albanian | 71977 | 0.98587 | 0.00014 | | 6e-05 | | |
| alp | Alune | 7829 | 0.999 | 0.0 | | | | 0.00011 |
| alq | Algonquin | 8025 | 0.9995 | 0.0 | | | | |
| als | Tosk Albanian | 403221 | 0.97813 | 0.00077 | 0.99852 | 0.00011 | 0.85507 | 0.00208 |
| alt | Southern Altai | 89192 | 0.99701 | 0.00017 | | | 0.95495 | 0.00033 |
| aly | Alyawarr | 7411 | 1.0 | 0.0 | | | | |
| alz | Alur | 103655 | 0.99651 | 0.00014 | | | | |
| ame | Yanesha' | 7697 | 0.9995 | 0.0 | | | 1.0 | 0.0 |
| amf | Hamer-Banna | 7808 | 0.998 | 3e-05 | | | | 0.0012 |
| amh | Amharic | 682875 | 0.99206 | 0.00044 | 0.99951 | 6e-05 | 1.0 | 0.0 |
| ami | Amis | 104294 | 0.999 | 3e-05 | | | 0.26087 | 0.0 |

Table 5: Performance of GlotLID-M on GlotLID-C test, FLORES-200 and UDHR benchmarks (part 1)

| | | | GlotLID-C | | FLORES-200 | | UDHR | |
|---|---|---|---|---|---|---|---|---|
| **iso639-3** | **Language** | **\|Sentences\|** | **F1↑** | **FPR↓** | **F1↑** | **FPR↓** | **F1↑** | **FPR↓** |
| amk | Ambai | 7850 | 0.9995 | 3e-05 | | | | |
| amm | Ama (Papua New Guinea) | 7826 | 1.0 | 0.0 | | | | |
| amn | Amanab | 14975 | 0.9995 | 3e-05 | | | | 0.00011 |
| amp | Alamblak | 7720 | 0.9985 | 3e-05 | | | | |
| amr | Amarakaeri | 7671 | 1.0 | 0.0 | | | 1.0 | 0.0 |
| amu | Guerrero Amuzgo | 8056 | 0.999 | 0.0 | | | | 0.00022 |
| amx | Anmatyerre | 1949 | 1.0 | 0.0 | | | | |
| ang | Old English (ca. 450-1100) | 379 | 0.88889 | 3e-05 | | | | |
| anm | Anal | 30940 | 0.99499 | 8e-05 | | | | |
| ann | Obolo | 7948 | 0.9995 | 0.0 | | | | |
| anv | Denya | 7928 | 0.9995 | 3e-05 | | | | |
| any | Anyin | 7949 | 0.999 | 0.0 | | | | |
| aoc | Pemon | 98 | 1.0 | 0.0 | | | | 0.00044 |
| aoi | Anindilyakwa | 3813 | 0.99644 | 3e-05 | | 0.00028 | | |
| aoj | Mufian | 15638 | 0.9995 | 3e-05 | | | | |
| aom | Ömie | 7848 | 0.9995 | 3e-05 | | | | |
| aon | Bumbita Arapesh | 7832 | 1.0 | 0.0 | | | | 0.00011 |
| aoz | Uab Meto | 7962 | 0.99501 | 0.00019 | | | | |
| apb | Sa'a | 7764 | 1.0 | 0.0 | | | | |
| apc | Levantine Arabic | 68045 | 0.79142 | 0.00891 | 0.17857 | 0.00915 | | |
| ape | Bukiyip | 7832 | 0.9995 | 3e-05 | | | | 0.00033 |
| apn | Apinayé | 7353 | 1.0 | 0.0 | | | | |
| apr | Arop-Lokep | 7853 | 1.0 | 0.0 | | | | |
| apt | Apatani | 7941 | 1.0 | 0.0 | | | | 0.00011 |
| apu | Apurinã | 7937 | 1.0 | 0.0 | | | | |
| apw | Western Apache | 7930 | 0.9995 | 0.0 | | | | |
| apy | Apalaí | 30421 | 0.9995 | 0.0 | | | | |
| apz | Safeyoka | 7830 | 1.0 | 0.0 | | | | 0.00022 |
| ara | Arabic | 1044573 | 0.9985 | 8e-05 | | | | |
| arb | Standard Arabic | 7100859 | 0.81268 | 0.01268 | 0.25705 | 0.21515 | 0.98333 | 0.00011 |
| are | Western Arrarnta | 7810 | 1.0 | 0.0 | | | | |
| arg | Aragonese | 30103 | 0.97236 | 0.00011 | | | | |
| arh | Arhuaco | 3968 | 0.42298 | 0.00121 | | | | |
| arl | Arabela | 7914 | 1.0 | 0.0 | | | 0.99187 | 0.0 |
| arn | Mapudungun | 154241 | 0.81037 | 0.01037 | | | 0.93913 | 0.00011 |
| arp | Arapaho | 1151 | 0.9971 | 0.0 | | | | |
| arq | Algerian Arabic | 3826 | 0.89381 | 0.00036 | | 0.00114 | | 0.00011 |
| ars | Najdi Arabic | 23194 | 0.7232 | 0.00624 | 0.00894 | 0.00574 | | |
| ary | Moroccan Arabic | 31432 | 0.83395 | 0.00272 | 0.56588 | 0.05164 | | 0.00011 |
| arz | Egyptian Arabic | 183549 | 0.86439 | 0.00635 | 0.46309 | 0.09806 | | |
| asg | Cishingini | 7900 | 0.9975 | 0.00011 | | | | |
| asm | Assamese | 213937 | 0.99749 | 0.0 | 1.0 | 0.0 | | |
| aso | Dano | 7694 | 1.0 | 0.0 | | | | |
| ast | Asturian | 1030498 | 0.97023 | 0.00151 | 0.9916 | 0.00051 | 0.97521 | 0.00011 |
| ata | Pele-Ata | 9433 | 0.999 | 3e-05 | | | | |
| atb | Zaiwa | 7905 | 1.0 | 0.0 | | | | |
| atd | Ata Manobo | 7849 | 0.9985 | 6e-05 | | | | |
| atg | Ivbie North-Okpela-Arhe | 7947 | 0.9995 | 0.0 | | | | |
| ati | Attié | 17922 | 0.78761 | 0.00209 | | | | |
| att | Pamplona Atta | 7954 | 1.0 | 0.0 | | | | |
| auc | Waorani | 7930 | 1.0 | 0.0 | | | 0.01504 | 0.0 |
| aui | Anuki | 652 | 1.0 | 0.0 | | 6e-05 | | |
| auy | Awiyaana | 7710 | 0.9965 | 8e-05 | | | | 0.00044 |
| ava | Avaric | 7833 | 0.9975 | 0.00011 | | | | |
| avk | Kotava | 4103 | 0.95715 | 0.00019 | | 0.00017 | | 0.00022 |
| avt | Au | 7878 | 1.0 | 0.0 | | | | |
| avu | Avokaya | 7590 | 1.0 | 0.0 | | | | |
| awa | Awadhi | 13074 | 0.94648 | 0.00011 | 0.38951 | 6e-05 | | |
| awb | Awa (Papua New Guinea) | 7880 | 0.9995 | 3e-05 | | | | |
| awi | Aekyom | 7790 | 0.999 | 0.0 | | | | 0.00011 |
| awx | Awara | 1635 | 0.9958 | 0.0 | | | | |
| aym | Aymara | 368899 | 0.99402 | 0.00025 | | | | |
| ayo | Ayoreo | 7898 | 0.9995 | 3e-05 | | | | |
| ayr | Central Aymara | 173203 | 0.99449 | 8e-05 | 0.99557 | 0.00045 | 0.98361 | 0.00022 |
| azb | South Azerbaijani | 532 | 0.97778 | 3e-05 | 0.36583 | 0.00011 | | |
| aze | Azerbaijani | 1069419 | 0.9995 | 3e-05 | | | | |
| azg | San Pedro Amuzgos Amuzgo | 7939 | 1.0 | 0.0 | | | | |
| azj | North Azerbaijani | 472589 | 0.9935 | 0.00019 | 0.99901 | 0.0 | 0.74306 | 0.00668 |
| azz | Highland Puebla Nahuatl | 7943 | 1.0 | 0.0 | | | | 0.00011 |
| bak | Bashkir | 171555 | 0.98846 | 0.00022 | 1.0 | 0.0 | | 0.0012 |
| bal | Baluchi | 141 | 0.84211 | 0.0 | | | | |
| bam | Bambara | 20569 | 0.92308 | 0.00061 | 0.52563 | 0.05119 | 0.49682 | 0.00635 |
| ban | Balinese | 32978 | 0.99348 | 8e-05 | 0.97521 | 6e-05 | 0.97561 | 0.00011 |
| bao | Waimaha | 7940 | 0.9995 | 3e-05 | | | | |
| bar | Bavarian | 130954 | 0.98611 | 0.00061 | | | | 0.00011 |

Table 6: Performance of GlotLID-M on GlotLID-C test, FLORES-200 and UDHR benchmarks (part 2)

| iso639-3 | Language | \|Sentences\| | GlotLID-C | | FLORES-200 | | UDHR | |
|---|---|---|---|---|---|---|---|---|
| | | | F1↑ | FPR↓ | F1↑ | FPR↓ | F1↑ | FPR↓ |
| bas | Basa (Cameroon) | 132504 | 0.996 | 0.00014 | | 0.00017 | | 0.00022 |
| bav | Vengo | 7954 | 1.0 | 0.0 | | | | 0.00011 |
| bba | Baatonum | 33686 | 0.999 | 6e-05 | | | 0.92187 | 0.00099 |
| bbb | Barai | 10264 | 1.0 | 0.0 | | | | |
| bbc | Batak Toba | 279942 | 0.98618 | 0.00074 | | | | |
| bbj | Ghomálá' | 27499 | 0.79298 | 0.00465 | | | | 0.00109 |
| bbr | Girawa | 7675 | 0.999 | 3e-05 | | | | 0.00011 |
| bcc | Southern Balochi | 1753 | 0.99313 | 3e-05 | | | | |
| bch | Bariai | 10603 | 0.998 | 3e-05 | | | | |
| bci | Baoulé | 382411 | 0.998 | 8e-05 | | 0.00011 | 0.97521 | 0.00033 |
| bcl | Central Bikol | 1193855 | 0.99104 | 0.00039 | | | 1.0 | 0.0 |
| bco | Kaluli | 1544 | 1.0 | 0.0 | | | | |
| bcw | Bana | 7874 | 1.0 | 0.0 | | | | |
| bdd | Bunama | 7866 | 0.9985 | 6e-05 | | | | |
| bdh | Baka (South Sudan) | 7948 | 0.9995 | 0.0 | | | | 0.00011 |
| bea | Beaver | 677 | 1.0 | 0.0 | | | | |
| bef | Benabena | 7857 | 0.999 | 3e-05 | | | | 0.00044 |
| bel | Belarusian | 428690 | 0.998 | 6e-05 | 1.0 | 0.0 | 0.98333 | 0.0 |
| bem | Bemba (Zambia) | 1477399 | 0.98039 | 0.0011 | 0.9906 | 0.00045 | 0.98333 | 0.00022 |
| ben | Bengali | 1659455 | 0.97466 | 0.00143 | 0.99852 | 6e-05 | 1.0 | 0.0 |
| beq | Beembe | 7934 | 0.60249 | 0.01262 | | | | |
| ber | Berber languages | 628585 | 0.84139 | 0.00514 | | 0.01335 | | 0.00099 |
| bex | Jur Modo | 10586 | 0.9985 | 0.0 | | | | 0.00274 |
| bfd | Bafut | 7941 | 0.999 | 3e-05 | | | | |
| bfo | Malba Birifor | 7898 | 0.998 | 0.00011 | | | | |
| bfz | Mahasu Pahari | 57 | 0.83333 | 0.0 | | | | |
| bgr | Bawm Chin | 7859 | 0.99599 | 6e-05 | | | | |
| bgs | Tagabawa | 9968 | 0.999 | 0.0 | | | | |
| bgz | Banggai | 7819 | 0.9995 | 0.0 | | | | |
| bhg | Binandere | 2782 | 0.99632 | 0.0 | | | | |
| bhl | Bimin | 9573 | 0.999 | 3e-05 | | | | |
| bho | Bhojpuri | 62722 | 0.64017 | 0.02426 | 0.94329 | 0.00443 | 0.78519 | 0.00033 |
| bhp | Bima | 7876 | 0.9995 | 0.0 | | | | |
| bhw | Biak | 109097 | 0.99045 | 0.00011 | | | | |
| bib | Bissa | 7940 | 0.9995 | 0.0 | | | | |
| big | Biangai | 7814 | 0.9995 | 3e-05 | | | | 0.00022 |
| bih | Bihari languages | 10000 | 0.18688 | 0.00025 | | | | |
| bik | Bikol | 30000 | 0.98838 | 3e-05 | | | | |
| bim | Bimoba | 30166 | 0.998 | 0.00011 | | | | 0.00033 |
| bin | Bini | 132682 | 0.9975 | 3e-05 | | | 0.9927 | 0.00011 |
| bis | Bislama | 1062315 | 0.998 | 0.00011 | | | 1.0 | 0.0 |
| biu | Biete | 7929 | 0.99499 | 0.00011 | | | | |
| biv | Southern Birifor | 7936 | 0.999 | 0.0 | | | | 0.00022 |
| bjn | Banjar | 32196 | 0.95164 | 0.00022 | 0.79496 | 0.05329 | | 0.00536 |
| bjp | Fanamaket | 877 | 1.0 | 0.0 | | | | |
| bjr | Binumarien | 9867 | 1.0 | 0.0 | | | | |
| bjv | Bedjond | 7919 | 0.998 | 3e-05 | | | | |
| bkd | Binukid | 7773 | 0.998 | 0.0 | | | | |
| bkq | Bakairí | 7773 | 0.9995 | 0.0 | | | | |
| bku | Buhid | 7911 | 0.999 | 3e-05 | | | | |
| bkv | Bekwarra | 7860 | 1.0 | 0.0 | | | | |
| bla | Siksika | 204 | 0.98361 | 0.0 | | | | |
| blh | Kuwaa | 7902 | 0.999 | 0.0 | | | | 0.00011 |
| blw | Balangao | 7883 | 0.9995 | 0.0 | | | | 0.00033 |
| blz | Balantak | 7917 | 0.997 | 6e-05 | | | | |
| bmb | Bembe | 8008 | 0.56335 | 0.01023 | | | | |
| bmh | Kein | 7688 | 0.9995 | 3e-05 | | | | |
| bmk | Ghayavi | 650 | 0.95385 | 6e-05 | | | | |
| bmq | Bomu | 7930 | 0.9985 | 0.0 | | | | |
| bmr | Muinane | 7926 | 1.0 | 0.0 | | | | |
| bmu | Somba-Siawari | 9565 | 0.999 | 3e-05 | | | | 0.00011 |
| bnj | Eastern Tawbuid | 7881 | 0.9985 | 3e-05 | | | | |
| bnp | Bola | 14309 | 0.996 | 0.00011 | | | | |
| boa | Bora | 7650 | 1.0 | 0.0 | | | 0.99213 | 0.00011 |
| bod | Tibetan | 24952 | 0.98419 | 0.00077 | 0.94589 | 6e-05 | 0.89091 | 0.00011 |
| boj | Anjam | 14853 | 1.0 | 0.0 | | | | 0.00022 |
| bom | Berom | 7960 | 0.998 | 3e-05 | | | | |
| bon | Bine | 7901 | 1.0 | 0.0 | | | | |
| bos | Bosnian | 507207 | 0.76033 | 0.00138 | 0.58206 | 0.00312 | 0.18026 | 0.01007 |
| bov | Tuwuli | 7907 | 0.9985 | 0.0 | | | | |
| box | Buamu | 7830 | 1.0 | 0.0 | | 6e-05 | | 0.00011 |
| bpr | Koronadal Blaan | 7840 | 0.96566 | 0.00066 | | | | |
| bps | Sarangani Blaan | 7840 | 0.96682 | 0.00118 | | | | |
| bpy | Bishnupriya | 30000 | 0.98785 | 0.0 | | | | |
| bqc | Boko (Benin) | 30639 | 0.99551 | 0.00017 | | | | |

Table 7: Performance of GlotLID-M on GlotLID-C test, FLORES-200 and UDHR benchmarks (part 3)

| iso639-3 | Language | \|Sentences\| | GlotLID-C | | FLORES-200 | | UDHR | |
|---|---|---|---|---|---|---|---|---|
| | | | F1↑ | FPR↓ | F1↑ | FPR↓ | F1↑ | FPR↓ |
| bqj | Bandial | 7903 | 0.998 | 3e-05 | | | | |
| bqp | Busa | 7894 | 0.9955 | 0.00014 | | | | 0.00011 |
| bre | Breton | 16810 | 0.98688 | 0.00011 | | | 0.98361 | 0.0 |
| bru | Eastern Bru | 7888 | 0.999 | 0.0 | | | | |
| brx | Bodo (India) | 11 | | | | | | |
| bsc | Bassari | 7949 | 0.9995 | 3e-05 | | | | 0.00022 |
| bsn | Barasana-Eduria | 7547 | 0.999 | 3e-05 | | | | |
| bsp | Baga Sitemu | 4533 | 0.99089 | 0.00017 | | | | 0.00011 |
| bsq | Bassa | 4073 | 0.96104 | 3e-05 | | 0.00028 | | 0.00055 |
| bss | Akoose | 7905 | 0.999 | 0.0 | | | | |
| btd | Batak Dairi | 9423 | 0.99246 | 6e-05 | | | | 0.00011 |
| btg | Gagnoa Bété | 16784 | 0.77055 | 0.00396 | | | | 0.00055 |
| bth | Biatah Bidayuh | 7889 | 0.999 | 0.0 | | | | |
| bts | Batak Simalungun | 8277 | 0.99097 | 0.00017 | | | | |
| btt | Bete-Bendi | 7866 | 0.999 | 0.0 | | | | 0.00011 |
| btx | Batak Karo | 191461 | 0.97756 | 0.00069 | | | | |
| bua | Buriat | 10906 | 0.97807 | 6e-05 | | | | |
| bud | Ntcham | 7916 | 0.999 | 3e-05 | | | | 0.00022 |
| bug | Buginese | 15386 | 0.98379 | 8e-05 | 0.99802 | 6e-05 | 0.95312 | 0.00055 |
| buk | Bugawac | 7776 | 0.9995 | 0.0 | | | | |
| bul | Bulgarian | 1762751 | 0.98762 | 0.00061 | 0.99951 | 0.0 | 0.96 | 0.00055 |
| bum | Bulu (Cameroon) | 141188 | 0.99102 | 0.0003 | | 6e-05 | 0.54762 | 0.00022 |
| bus | Bokobaru | 7882 | 0.99399 | 0.00014 | | | | 0.00011 |
| bvr | Burarra | 7882 | 1.0 | 0.0 | | | | |
| bvy | Baybayanon | 76 | | | | | | |
| bvz | Bauzi | 7501 | 1.0 | 0.0 | | | | 0.00011 |
| bwd | Bwaidoka | 1522 | 0.99797 | 0.0 | | | | |
| bwi | Baniwa | 262 | 0.96104 | 0.0 | | | | |
| bwq | Southern Bobo Madaré | 7921 | 0.99699 | 3e-05 | | | | 0.00011 |
| bwu | Buli (Ghana) | 7824 | 0.998 | 3e-05 | | | | |
| bxh | Buhutu | 4196 | 0.99915 | 3e-05 | | | | 0.00011 |
| bxr | Russia Buriat | 8599 | 0.997 | 8e-05 | | | | |
| byr | Baruya | 8233 | 1.0 | 0.0 | | | | |
| byv | Medumba | 2171 | 0.97605 | 8e-05 | | | | 0.00394 |
| byx | Qaqet | 7783 | 0.9995 | 0.0 | | | | |
| bzd | Bribri | 8660 | 0.999 | 0.0 | | | | 0.00077 |
| bzh | Mapos Buang | 7937 | 0.9995 | 3e-05 | | | | 0.00011 |
| bzi | Bisu | 7830 | 0.999 | 0.0 | | | | |
| bzj | Belize Kriol English | 124087 | 0.94398 | 0.00179 | | | | |
| bzt | Brithenig | 357 | 0.92308 | 3e-05 | | | | 0.00011 |
| caa | Chortí | 7940 | 1.0 | 0.0 | | | | |
| cab | Garifuna | 228814 | 0.98498 | 0.00039 | | | 0.992 | 0.00011 |
| cac | Chuj | 38234 | 0.999 | 3e-05 | | | | 0.00055 |
| caf | Southern Carrier | 7943 | 0.99198 | 0.00017 | | | | |
| cag | Nivaclé | 9167 | 0.9995 | 0.0 | | | | 0.00033 |
| cak | Kaqchikel | 164900 | 0.99601 | 0.00019 | | | 1.0 | 0.0 |
| cao | Chácobo | 7902 | 0.999 | 3e-05 | | | | |
| cap | Chipaya | 15847 | 0.999 | 3e-05 | | | | 0.00011 |
| caq | Car Nicobarese | 32067 | 1.0 | 0.0 | | | | |
| car | Galibi Carib | 9359 | 0.9995 | 3e-05 | | | | 0.00055 |
| cas | Tsimané | 7870 | 0.998 | 3e-05 | | | | 0.00011 |
| cat | Catalan | 1137480 | 0.97554 | 0.00129 | 1.0 | 0.0 | 0.93023 | 0.00099 |
| cav | Cavineña | 7741 | 0.9995 | 0.0 | | | | 0.00011 |
| cax | Chiquitano | 15872 | 1.0 | 0.0 | | | | 0.00011 |
| cay | Cayuga | 31 | | | | | | |
| cbc | Carapana | 7791 | 0.9975 | 3e-05 | | | | |
| cbi | Chachi | 7863 | 1.0 | 0.0 | | | 0.9771 | 0.00022 |
| cbk | Chavacano | 111815 | 0.98029 | 0.00025 | | | | 0.00033 |
| cbr | Cashibo-Cacataibo | 7813 | 0.9995 | 0.0 | | | 0.7033 | 0.0 |
| cbs | Cashinahua | 7502 | 1.0 | 0.0 | | 6e-05 | 0.67308 | 0.00033 |
| cbt | Chayahuita | 7804 | 0.9995 | 0.0 | | | 0.97479 | 0.00011 |
| cbu | Candoshi-Shapra | 7588 | 1.0 | 0.0 | | | 0.33803 | 0.0 |
| cbv | Cacua | 7694 | 1.0 | 0.0 | | | | |
| cce | Chopi | 120353 | 0.99299 | 0.00014 | | 6e-05 | | 0.00011 |
| cco | Comaltepec Chinantec | 7890 | 1.0 | 0.0 | | | | 0.00131 |
| ceb | Cebuano | 2111383 | 0.97987 | 0.00107 | 0.99503 | 0.0 | 0.96721 | 0.00044 |
| ceg | Chamacoco | 7912 | 0.9995 | 0.0 | | | | |
| cek | Eastern Khumi Chin | 7873 | 1.0 | 0.0 | | | | |
| ces | Czech | 1639384 | 0.99156 | 0.00044 | 0.99951 | 6e-05 | 0.98387 | 0.0 |
| cfm | Falam Chin | 38517 | 0.98848 | 0.00028 | | | 0.85714 | 0.0 |
| cgc | Kagayanen | 7823 | 0.9995 | 3e-05 | | | | |
| cgg | Chiga | 40958 | 0.98943 | 0.00011 | | | | 0.00022 |
| cha | Chamorro | 16006 | 0.996 | 0.00014 | | 6e-05 | 0.8381 | 0.00011 |
| chd | Highland Oaxaca Chontal | 8393 | 0.998 | 0.0 | | 6e-05 | | 0.00011 |
| che | Chechen | 60837 | 0.995 | 0.00017 | | | | |

Table 8: Performance of GlotLID-M on GlotLID-C test, FLORES-200 and UDHR benchmarks (part 4)

| iso639-3 | Language | \|Sentences\| | GlotLID-C F1↑ | GlotLID-C FPR↓ | FLORES-200 F1↑ | FLORES-200 FPR↓ | UDHR F1↑ | UDHR FPR↓ |
|---|---|---|---|---|---|---|---|---|
| chf | Tabasco Chontal | 8213 | 0.998 | 6e-05 | | | | 0.00022 |
| chg | Chagatai | 8 | | | | | | |
| chj | Ojitlán Chinantec | 6824 | 0.72469 | 0.00091 | | | 0.12727 | 0.00471 |
| chk | Chuukese | 377198 | 0.99701 | 0.00017 | | | 0.97521 | 0.00011 |
| chn | Chinook jargon | 53 | 0.8 | 0.0 | | | | |
| cho | Choctaw | 124 | 0.78788 | 0.0 | | | | 0.00263 |
| chq | Quiotepec Chinantec | 7891 | 1.0 | 0.0 | | | | 0.00044 |
| chr | Cherokee | 7982 | 0.99699 | 3e-05 | | | 0.09677 | 0.0 |
| chu | Church Slavic | 7836 | 0.4985 | 0.0 | | | | |
| chv | Chuvash | 39476 | 0.99301 | 0.00025 | | | 0.86154 | 0.0 |
| chw | Chuwabu | 119383 | 0.997 | 0.00011 | | 6e-05 | | |
| chz | Ozumacín Chinantec | 7919 | 1.0 | 0.0 | | | | |
| cjk | Chokwe | 193952 | 0.9955 | 0.00011 | 0.84493 | 6e-05 | 0.92641 | 0.00033 |
| cjo | Ashéninka Pajonal | 7774 | 0.99354 | 0.00033 | | | | |
| cjp | Cabécar | 7921 | 0.9995 | 0.0 | | | | |
| cjs | Shor | 1553 | 0.98929 | 0.0 | | | 0.65217 | 0.0 |
| cjv | Chuave | 7852 | 0.9995 | 0.0 | | | | |
| cjy | Jinyu Chinese | 18 | | | | | | |
| ckb | Central Kurdish | 138398 | 0.99007 | 0.00047 | 0.99901 | 0.00011 | | |
| cko | Anufo | 7882 | 1.0 | 0.0 | | | | 0.00131 |
| ckt | Chukot | 1675 | 0.98619 | 6e-05 | | | | 0.00033 |
| cle | Lealao Chinantec | 7934 | 1.0 | 0.0 | | | | |
| clu | Caluyanun | 7850 | 0.99451 | 0.00019 | | | | |
| cly | Eastern Highland Chatino | 7928 | 0.77037 | 0.00217 | | | | 0.00066 |
| cme | Cerma | 7890 | 0.9995 | 0.0 | | | | 0.00011 |
| cmi | Emberá-Chamí | 16270 | 0.6085 | 0.01438 | | | | |
| cmn | Mandarin Chinese | 1073282 | 0.83808 | 0.01009 | | 0.05727 | 0.51064 | 0.07148 |
| cmo | Central Mnong | 16195 | 0.99699 | 0.0 | | | | 0.00011 |
| cnh | Hakha Chin | 435869 | 0.99057 | 0.00047 | | | 0.93023 | 0.00099 |
| cni | Asháninka | 11125 | 0.80793 | 0.00228 | | | 0.90226 | 0.00142 |
| cnl | Lalana Chinantec | 7911 | 1.0 | 0.0 | | | | |
| cnt | Tepetotutla Chinantec | 7924 | 1.0 | 0.0 | | | | |
| cnw | Ngawn Chin | 7929 | 0.99548 | 3e-05 | | | | 0.00055 |
| coe | Koreguaje | 7741 | 0.9995 | 3e-05 | | | | 0.00164 |
| cof | Colorado | 7555 | 1.0 | 0.0 | | | 0.74747 | 0.0 |
| cok | Santa Teresa Cora | 18162 | 0.99449 | 0.00011 | | | | |
| con | Cofán | 7876 | 0.9985 | 3e-05 | | | | 0.00011 |
| cop | Coptic | 23773 | 1.0 | 0.0 | | | | |
| cor | Cornish | 41272 | 0.99097 | 0.00017 | | | | |
| cos | Corsican | 11141 | 0.97444 | 8e-05 | | 0.00017 | 0.95082 | 0.00044 |
| cot | Caquinte | 7879 | 0.9975 | 3e-05 | | | 0.96774 | 0.00022 |
| cpa | Palantla Chinantec | 7946 | 0.9995 | 0.0 | | | | |
| cpb | Ucayali-Yurúa Ashéninka | 7947 | 0.98448 | 0.00039 | | | | 0.00022 |
| cpc | Ajyíninka Apurucayali | 7939 | 0.998 | 0.0 | | | | |
| cpi | Chinese Pidgin English | 7 | | | | | | |
| cpu | Pichis Ashéninka | 7945 | 0.98754 | 0.00044 | | | 0.89908 | 0.0 |
| cpy | South Ucayali Ashéninka | 6127 | 0.9937 | 6e-05 | | | | |
| crh | Crimean Tatar | 28216 | 0.97352 | 0.00022 | 0.98902 | 6e-05 | 0.97561 | 0.00033 |
| cri | Sãotomense | 2594 | 0.94133 | 8e-05 | | 6e-05 | 0.84404 | 0.00033 |
| crk | Plains Cree | 293 | 0.86486 | 0.00011 | | | | |
| crm | Moose Cree | 8076 | 0.9995 | 0.0 | | | | 0.00547 |
| crn | El Nayar Cora | 23287 | 0.99451 | 0.00019 | | | | |
| crq | Iyo'wujwa Chorote | 7731 | 0.36727 | 0.00954 | | | | 0.00055 |
| crs | Seselwa Creole French | 506502 | 0.99104 | 0.00039 | | | 1.0 | 0.0 |
| crt | Iyojwa'ja Chorote | 7826 | 0.55489 | 0.0192 | | | | 0.00066 |
| crx | Carrier | 7935 | 0.99151 | 0.00028 | | | | |
| csb | Kashubian | 10947 | 0.98221 | 3e-05 | | | | 0.00011 |
| csk | Jola-Kasa | 7918 | 0.999 | 0.0 | | 6e-05 | | |
| cso | Sochiapam Chinantec | 7936 | 1.0 | 0.0 | | | | |
| csw | Swampy Cree | 476 | 0.93671 | 3e-05 | | | 0.0 | 0.00186 |
| csy | Siyin Chin | 30034 | 0.98449 | 0.00041 | | | | |
| cta | Tataltepec Chatino | 7908 | 0.999 | 0.0 | | | | 0.00066 |
| ctd | Tedim Chin | 35149 | 0.88063 | 0.00564 | | | 0.78431 | 0.0 |
| cto | Emberá-Catío | 22638 | 0.86318 | 0.00358 | | 6e-05 | | 0.00011 |
| ctp | Western Highland Chatino | 7910 | 0.9995 | 3e-05 | | | | |
| ctu | Chol | 174449 | 0.9879 | 0.00011 | | | | 0.00044 |
| cub | Cubeo | 7943 | 0.9995 | 3e-05 | | | | 0.00033 |
| cuc | Usila Chinantec | 7901 | 1.0 | 0.0 | | | | |
| cui | Cuiba | 7882 | 0.9995 | 3e-05 | | | | |
| cuk | San Blas Kuna | 26510 | 0.998 | 3e-05 | | 6e-05 | | 0.00022 |
| cul | Culina | 7821 | 1.0 | 0.0 | | | | |
| cut | Teutila Cuicatec | 7835 | 0.999 | 0.0 | | | | |
| cux | Tepeuxila Cuicatec | 7954 | 1.0 | 0.0 | | | | |
| cwd | Woods Cree | 278 | 0.91304 | 0.00022 | | | | |
| cwe | Kwere | 7950 | 0.97405 | 0.00077 | | | | |

Table 9: Performance of GlotLID-M on GlotLID-C test, FLORES-200 and UDHR benchmarks (part 5)

| iso639-3 | Language | \|Sentences\| | GlotLID-C | | FLORES-200 | | UDHR | |
|---|---|---|---|---|---|---|---|---|
| | | | F1↑ | FPR↓ | F1↑ | FPR↓ | F1↑ | FPR↓ |
| cwt | Kuwaataay | 7828 | 1.0 | 0.0 | | | | |
| cya | Nopala Chatino | 7946 | 0.82102 | 0.00891 | | | | 0.00033 |
| cym | Welsh | 273301 | 0.99501 | 0.00019 | 0.99951 | 6e-05 | 1.0 | 0.0 |
| czt | Zotung Chin | 9478 | 0.999 | 3e-05 | | | | 0.00011 |
| daa | Dangaléat | 7839 | 0.999 | 3e-05 | | | | 0.00011 |
| dad | Marik | 7726 | 0.9995 | 3e-05 | | | | |
| daf | Dan | 12719 | 0.9995 | 0.0 | | | | |
| dah | Gwahatike | 7704 | 0.999 | 3e-05 | | | | |
| dak | Dakota | 2366 | 0.99856 | 0.0 | | | | |
| dan | Danish | 3921737 | 0.96039 | 0.00173 | 0.9931 | 0.00051 | 0.85135 | 0.00241 |
| dar | Dargwa | 2195 | 0.99543 | 3e-05 | | | | 0.00055 |
| ddg | Fataluku | 1603 | 0.98934 | 6e-05 | | | | 0.00077 |
| ddo | Dido | 20 | | | | | | |
| ded | Dedua | 10276 | 0.999 | 0.0 | | | | |
| des | Desano | 7740 | 1.0 | 0.0 | | | | |
| deu | German | 1800293 | 0.96193 | 0.00212 | 0.99901 | 0.0 | 0.98745 | 0.00011 |
| dga | Southern Dagaare | 12612 | 0.998 | 3e-05 | | | 0.71166 | |
| dgc | Casiguran Dumagat Agta | 7795 | 0.9995 | 0.0 | | | | |
| dgi | Northern Dagara | 7943 | 0.9975 | 3e-05 | | | | |
| dgr | Dogrib | 9376 | 0.9995 | 0.0 | | | | |
| dgz | Daga | 7802 | 0.9975 | 0.00011 | | | | 0.00011 |
| dhg | Dhangu-Djangu | 630 | 1.0 | 0.0 | | | | |
| dhm | Zemba | 7908 | 0.99649 | 6e-05 | | 0.00023 | | 0.00022 |
| dhv | Dehu | 403863 | 0.95252 | 0.00223 | | | | 0.00514 |
| dig | Digo | 7865 | 0.99449 | 0.00011 | | | | |
| dik | Southwestern Dinka | 33790 | 0.99551 | 0.00019 | 0.99653 | 6e-05 | | 0.00799 |
| din | Dinka | 3290 | 0.38141 | 0.0 | | | | |
| dip | Northeastern Dinka | 7933 | 0.98742 | 0.00017 | | | | |
| diq | Dimli (individual language) | 30006 | 0.997 | 6e-05 | | 0.00011 | | 0.00109 |
| dis | Dimasa | 7955 | 0.999 | 3e-05 | | 6e-05 | | |
| diu | Diriku | 198 | 0.88889 | 0.0 | | | | |
| div | Dhivehi | 30040 | 1.0 | 0.0 | | | 0.96774 | 0.0 |
| dje | Zarma | 7899 | 0.99699 | 3e-05 | | 6e-05 | | 0.00109 |
| djk | Eastern Maroon Creole | 98689 | 0.99548 | 3e-05 | | | | |
| djr | Djambarrpuyngu | 7771 | 0.9985 | 8e-05 | | 0.0004 | | |
| dks | Southeastern Dinka | 10409 | 0.98705 | 0.00047 | | 0.00011 | | |
| dng | Dungan | 944 | 0.98962 | 0.0 | | | | 0.00033 |
| dnj | Dan | 15219 | 1.0 | 0.0 | | | | |
| dob | Dobu | 7854 | 0.9985 | 0.0 | | | | |
| dop | Lukpa | 7934 | 1.0 | 0.0 | | | | |
| dow | Doyayo | 7919 | 0.9995 | 3e-05 | | | | 0.00022 |
| drg | Rungus | 244 | 0.82857 | 0.0 | | | | |
| drt | Drents | 40 | | | | | | |
| dru | Rukai | 30027 | 0.996 | 0.00011 | | | | |
| dsb | Lower Sorbian | 11095 | 0.97823 | 0.00025 | | | | |
| dtp | Kadazan Dusun | 19197 | 0.95972 | 0.00162 | | 6e-05 | | 0.00011 |
| dts | Toro So Dogon | 7822 | 0.9995 | 0.0 | | | | |
| dua | Duala | 52748 | 0.9975 | 3e-05 | | | | 0.00011 |
| due | Umiray Dumaget Agta | 7864 | 0.9985 | 0.0 | | | | |
| dug | Duruma | 7836 | 0.99499 | 0.00011 | | | | |
| duo | Dupaninan Agta | 7795 | 0.998 | 0.0 | | | | |
| dur | Dii | 7870 | 0.9985 | 3e-05 | | | | 0.00011 |
| dwr | Dawro | 7801 | 0.99044 | 8e-05 | | | | |
| dws | Dutton World Speedwords | 57 | 0.46154 | 0.0 | | | | |
| dww | Dawawa | 7876 | 0.9985 | 3e-05 | | | | |
| dyi | Djimini Senoufo | 7927 | 1.0 | 0.0 | | | | 0.00033 |
| dyo | Jola-Fonyi | 9559 | 0.9985 | 3e-05 | | | 0.97391 | 0.0 |
| dyu | Dyula | 218015 | 0.95472 | 0.00234 | 0.12435 | 0.01449 | 0.23188 | 0.0069 |
| dzo | Dzongkha | 6899 | 0.9843 | 8e-05 | 0.9496 | 0.00585 | 0.90769 | 0.0012 |
| ebk | Eastern Bontok | 7913 | 0.9975 | 3e-05 | | | | |
| efi | Efik | 1078995 | 0.99451 | 0.00022 | | | | 0.00755 |
| egl | Emilian | 144 | 0.38889 | 3e-05 | | 6e-05 | | |
| eka | Ekajuk | 7942 | 0.9985 | 3e-05 | | | | |
| ekk | Standard Estonian | 300000 | 0.98759 | 0.00055 | | | 0.90226 | 0.00142 |
| eko | Koti | 3176 | 0.99494 | 0.0 | | | | |
| ell | Modern Greek (1453-) | 4450890 | 0.98862 | 0.00061 | 1.0 | 0.0 | 0.97908 | 0.0 |
| emi | Mussau-Emira | 4352 | 0.99753 | 0.0 | | | | |
| eml | Emiliano-Romagnolo | 30000 | 0.98601 | 0.00041 | | 0.00045 | | 0.00952 |
| emp | Northern Emberá | 8463 | 0.997 | 6e-05 | | | | |
| emx | Erromintxela | 12 | | | | | | |
| enb | Markweeta | 7871 | 0.9995 | 3e-05 | | | | |
| eng | English | 10703345 | 0.63088 | 0.03212 | 0.98732 | 0.00148 | 0.85294 | 0.00197 |
| enl | Enlhet | 7903 | 1.0 | 0.0 | | | | |
| enm | Middle English (1100-1500) | 39815 | 0.9823 | 0.00017 | | | | 0.00022 |
| enx | Enxet | 7890 | 1.0 | 0.0 | | | | |

Table 10: Performance of GlotLID-M on GlotLID-C test, FLORES-200 and UDHR benchmarks (part 6)

| iso639-3 | Language | \|Sentences\| | GlotLID-C | | FLORES-200 | | UDHR | |
|---|---|---|---|---|---|---|---|---|
| | | | F1↑ | FPR↓ | F1↑ | FPR↓ | F1↑ | FPR↓ |
| epo | Esperanto | 1161088 | 0.98377 | 0.00091 | 0.99852 | 0.00017 | 0.96825 | 0.00044 |
| eri | Ogea | 7908 | 0.9995 | 3e-05 | | | | |
| ese | Ese Ejja | 7842 | 0.9995 | 0.0 | | | 0.75 | 0.00011 |
| esi | North Alaskan Inupiatun | 7915 | 0.992 | 0.00022 | | | | 0.00011 |
| esk | Northwest Alaska Inupiatun | 7921 | 0.99249 | 0.00017 | | | | 0.00011 |
| est | Estonian | 3664155 | 0.99399 | 0.00011 | 1.0 | 0.0 | | |
| esu | Central Yupik | 7942 | 0.9995 | 3e-05 | | | | |
| eto | Eton (Cameroon) | 208 | 0.95082 | 3e-05 | | 0.00011 | | |
| etr | Edolo | 3182 | 0.99898 | 0.0 | | | | |
| etu | Ejagham | 7900 | 1.0 | 0.0 | | | | |
| eus | Basque | 938424 | 0.97363 | 0.0014 | 0.99951 | 0.0 | 0.91045 | 0.00131 |
| eve | Even | 1149 | 0.98382 | 6e-05 | | | 0.27869 | 0.00372 |
| evn | Evenki | 116 | 0.73684 | 3e-05 | | | 0.20755 | 0.00394 |
| ewe | Ewe | 1698872 | 0.99206 | 0.00044 | 1.0 | 0.0 | 0.98361 | 0.00022 |
| ewo | Ewondo | 7942 | 0.9985 | 0.0 | | | | 0.00197 |
| ext | Extremaduran | 10066 | 0.96701 | 6e-05 | | 0.00017 | | 0.00011 |
| eza | Ezaa | 30880 | 0.99353 | 0.0003 | | | | |
| faa | Fasu | 7854 | 0.9995 | 3e-05 | | | | |
| fad | Wagi | 1147 | 1.0 | 0.0 | | | | |
| fai | Faiwol | 7941 | 0.999 | 3e-05 | | | | |
| fal | South Fali | 7916 | 0.999 | 3e-05 | | | | |
| fan | Fang (Equatorial Guinea) | 22472 | 0.99043 | 6e-05 | | | | 0.00142 |
| fao | Faroese | 90577 | 0.99399 | 0.00011 | 0.99951 | 0.0 | 0.98305 | 0.0 |
| fas | Persian | 1000000 | 0.88456 | 0.00718 | | | | |
| fat | Fanti | 54615 | 0.99448 | 3e-05 | | | 0.97521 | 0.00022 |
| ffm | Maasina Fulfulde | 7872 | 0.97054 | 0.00085 | | | | 0.00044 |
| fij | Fijian | 1232612 | 0.99106 | 0.00044 | 0.99951 | 0.0 | 1.0 | 0.0 |
| fil | Filipino | 38283 | 0.81967 | 0.00022 | | | | |
| fin | Finnish | 3909988 | 0.94857 | 0.00286 | 0.99901 | 0.00011 | 0.36 | 0.02452 |
| fkv | Kven Finnish | 539 | 0.67187 | 3e-05 | | | 0.28571 | 0.0 |
| fmp | Fe'fe' | 104 | 0.92308 | 0.0 | | | | 0.00066 |
| fon | Fon | 259192 | 0.9945 | 0.00017 | 0.99752 | 0.0 | 0.94118 | 0.00044 |
| for | Fore | 7892 | 1.0 | 0.0 | | | | 0.00263 |
| fra | French | 2570543 | 0.74664 | 0.01862 | 0.99951 | 6e-05 | 0.95238 | 0.00066 |
| frm | Middle French (ca. 1400-1600) | 32 | | | | | | |
| fro | Old French (842-ca. 1400) | 216 | 0.08333 | 0.0 | | | | |
| frr | Northern Frisian | 12856 | 0.98633 | 3e-05 | | 0.00017 | | 0.00022 |
| fry | Western Frisian | 131535 | 0.99651 | 0.00017 | | | 0.99174 | 0.00011 |
| fub | Adamawa Fulfulde | 34642 | 0.98303 | 0.00052 | | 0.00017 | | 0.00175 |
| fuc | Pulaar | 7 | | | | | | |
| fud | East Futuna | 11205 | 0.99246 | 6e-05 | | 0.00028 | | 0.00011 |
| fue | Borgu Fulfulde | 4664 | 0.9815 | 0.00022 | | 6e-05 | | |
| fuf | Pular | 12566 | 0.98898 | 0.00025 | | | 0.04762 | 0.00044 |
| fuh | Western Niger Fulfulde | 7899 | 0.97049 | 0.0008 | | | | 0.00022 |
| fuq | Central-Eastern Niger Fulfulde | 7908 | 0.96761 | 0.00099 | | | | |
| fur | Friulian | 54651 | 0.94985 | 0.00072 | 0.99951 | 6e-05 | 0.80272 | 0.00306 |
| fuv | Nigerian Fulfulde | 22406 | 0.97815 | 0.0008 | 0.96843 | 6e-05 | 0.76984 | 0.00241 |
| gaa | Ga | 1217812 | 0.99501 | 0.00019 | | | 0.93846 | 0.00088 |
| gag | Gagauz | 17347 | 0.99548 | 3e-05 | | | 0.94017 | 0.0 |
| gah | Alekano | 7879 | 0.999 | 3e-05 | | | | 0.00011 |
| gai | Borei | 7866 | 0.9995 | 3e-05 | | | | |
| gam | Kandawo | 7799 | 0.999 | 3e-05 | | | | 0.00011 |
| gaw | Nobonob | 7838 | 1.0 | 0.0 | | | | |
| gaz | West Central Oromo | 335746 | 0.99301 | 0.00022 | 0.99411 | 0.00068 | 0.83221 | 0.00274 |
| gba | Gbaya (Central African Republic) | 1010 | 0.99617 | 0.0 | | | | |
| gbi | Galela | 7722 | 0.9985 | 6e-05 | | | | |
| gbm | Garhwali | 36 | | | | | | |
| gbo | Northern Grebo | 7939 | 1.0 | 0.0 | | | | 0.00011 |
| gbr | Gbagyi | 7660 | 0.998 | 0.0 | | | | 0.00011 |
| gcf | Guadeloupean Creole French | 82017 | 0.98254 | 0.00055 | | 0.00023 | | 0.00471 |
| gcr | Guianese Creole French | 32425 | 0.98852 | 0.00036 | | | | 0.00022 |
| gde | Gude | 7901 | 1.0 | 0.0 | | | | |
| gdg | Ga'dang | 7827 | 0.9995 | 0.0 | | | | 0.00022 |
| gdn | Umanakaina | 7737 | 1.0 | 0.0 | | | | 0.00022 |
| gdr | Wipi | 7880 | 1.0 | 0.0 | | | | |
| geb | Kire | 7822 | 0.9995 | 3e-05 | | | | |
| gej | Gen | 7953 | 0.9985 | 6e-05 | | | | |
| gfk | Patpatar | 7657 | 0.9995 | 0.0 | | | | |
| ghe | Southern Ghale | 7953 | 0.9995 | 0.0 | | | | |
| ghs | Guhu-Samane | 7461 | 0.998 | 3e-05 | | | | |
| gid | Gidar | 7919 | 0.9995 | 0.0 | | | | |
| gil | Gilbertese | 428828 | 0.998 | 8e-05 | | | | |
| giz | South Giziga | 35118 | 0.9965 | 0.00011 | | | | |
| gjn | Gonja | 30289 | 0.997 | 6e-05 | | | 0.90476 | 0.00088 |
| gkn | Gokana | 96606 | 0.98947 | 0.00022 | | | | |

Table 11: Performance of GlotLID-M on GlotLID-C test, FLORES-200 and UDHR benchmarks (part 7)

| iso639-3 | Language | \|Sentences\| | GlotLID-C | | FLORES-200 | | UDHR | |
|---|---|---|---|---|---|---|---|---|
| | | | F1↑ | FPR↓ | F1↑ | FPR↓ | F1↑ | FPR↓ |
| gkp | Guinea Kpelle | 45287 | 0.54231 | 0.01537 | | | 0.92063 | 0.00109 |
| gla | Scottish Gaelic | 95320 | 0.99298 | 0.00011 | 0.99951 | 6e-05 | 0.94574 | 0.00022 |
| gle | Irish | 274216 | 0.9935 | 0.00017 | 1.0 | 0.0 | 0.95 | 0.00088 |
| glg | Galician | 343533 | 0.98495 | 0.00033 | 0.99703 | 0.00011 | 0.98305 | 0.00011 |
| glk | Gilaki | 10879 | 0.93362 | 0.00011 | | 0.00818 | | |
| glv | Manx | 45473 | 0.997 | 0.00011 | | | 1.0 | 0.0 |
| gmv | Gamo | 15674 | 0.99701 | 0.00014 | | 0.00011 | | |
| gnb | Gangte | 7955 | 0.99296 | 3e-05 | | | | 0.00022 |
| gnd | Zulgo-Gemzek | 7873 | 0.998 | 6e-05 | | | | |
| gng | Ngangam | 7833 | 0.999 | 0.0 | | | | |
| gnn | Gumatj | 13043 | 0.9985 | 0.0 | | | | 0.00022 |
| gnw | Western Bolivian Guaraní | 17302 | 0.996 | 0.00011 | | | | |
| goa | Guro | 1971 | 0.99286 | 3e-05 | | | | 0.00142 |
| gof | Gofa | 7908 | 0.98541 | 0.00022 | | | | |
| gog | Gogo | 8642 | 0.99348 | 8e-05 | | 0.00062 | | 0.00011 |
| gom | Goan Konkani | 89931 | 0.96933 | 0.00069 | | 0.00017 | | 0.00044 |
| gor | Gorontalo | 7895 | 0.999 | 3e-05 | | | | |
| gos | Gronings | 5670 | 0.94176 | 6e-05 | | | | |
| got | Gothic | 3857 | 0.99652 | 0.0 | | | | |
| gqr | Gor | 7931 | 0.9975 | 8e-05 | | | | |
| grc | Ancient Greek (to 1453) | 87146 | 0.98889 | 3e-05 | | | | 0.00055 |
| grn | Guarani | 62443 | 0.98069 | 8e-05 | 1.0 | 0.0 | | |
| grt | Garo | 8033 | 0.9985 | 0.0 | | 6e-05 | | |
| gso | Southwest Gbaya | 7902 | 0.9995 | 0.0 | | | | |
| gsw | Swiss German | 108513 | 0.95317 | 0.00201 | | 0.00011 | 0.98333 | 0.00011 |
| gub | Guajajára | 30075 | 0.51632 | 0.01438 | | | | |
| guc | Wayuu | 248890 | 0.999 | 6e-05 | | | 0.96063 | 0.00055 |
| gud | Yocoboué Dida | 7933 | 0.9985 | 6e-05 | | | | |
| gug | Paraguayan Guaraní | 384689 | 0.87191 | 0.00762 | | | 0.81967 | 0.0 |
| guh | Guahibo | 9114 | 0.95575 | 8e-05 | | | | 0.0012 |
| gui | Eastern Bolivian Guaraní | 15213 | 0.99499 | 8e-05 | | | | 0.00153 |
| guj | Gujarati | 1018488 | 1.0 | 0.0 | 1.0 | 0.0 | 1.0 | 0.0 |
| guk | Gumuz | 7906 | 0.9985 | 0.0 | | | | |
| gul | Sea Island Creole English | 7931 | 0.9985 | 6e-05 | | | | |
| gum | Guambiano | 8925 | 0.95248 | 0.00039 | | | | 0.00011 |
| gun | Mbyá Guaraní | 7929 | 0.9985 | 0.0 | | | | |
| guo | Guayabero | 7809 | 1.0 | 0.0 | | | | |
| guq | Aché | 7925 | 1.0 | 0.0 | | | | |
| gur | Farefare | 71300 | 0.996 | 0.00014 | | 6e-05 | | 0.00011 |
| guw | Gun | 765163 | 0.9975 | 0.00011 | | | | 0.00011 |
| gux | Gourmanchéma | 7927 | 0.9975 | 3e-05 | | | | 0.00022 |
| guz | Gusii | 7935 | 0.99599 | 6e-05 | | | | |
| gvc | Guanano | 7775 | 0.9995 | 0.0 | | | | |
| gvf | Golin | 7896 | 0.9985 | 0.0 | | | | |
| gvl | Gulay | 7956 | 0.9985 | 3e-05 | | | | 0.00022 |
| gvn | Kuku-Yalanji | 8219 | 0.9985 | 6e-05 | | | | |
| gwi | Gwichin | 7845 | 1.0 | 0.0 | | | | |
| gxx | Wè Southern | 24463 | 0.63256 | 0.01293 | | | | 0.00208 |
| gya | Northwest Gbaya | 35345 | 0.99499 | 0.00011 | | | | |
| gym | Ngäbere | 272246 | 0.99551 | 0.00017 | | | | 0.00022 |
| gyr | Guarayu | 15925 | 0.9985 | 8e-05 | | | 0.62745 | 0.0 |
| hae | Eastern Oromo | 7951 | 0.99649 | 6e-05 | | | | |
| hag | Hanga | 7659 | 0.9985 | 6e-05 | | | | 0.00044 |
| hak | Hakka Chinese | 38023 | 0.99598 | 0.0 | | | | |
| hat | Haitian | 380465 | 0.98213 | 0.00069 | 0.99852 | 0.00017 | 0.90706 | 0.00274 |
| hau | Hausa | 401986 | 0.98854 | 0.00041 | 0.95427 | 0.00551 | 0.94488 | 0.0023 |
| hav | Havu | 8780 | 0.98434 | 0.00014 | | 6e-05 | | |
| haw | Hawaiian | 7859 | 0.99497 | 0.0 | | 6e-05 | 1.0 | 0.0 |
| hay | Haya | 16371 | 0.92727 | 8e-05 | | 6e-05 | | |
| hbo | Ancient Hebrew | 102471 | 0.9985 | 8e-05 | | | | 0.00011 |
| hbs | Serbo-Croatian | 300000 | 0.99401 | 0.00019 | | | | |
| hch | Huichol | 13394 | 0.9985 | 0.0 | | | | |
| hdn | Northern Haida | 3 | | | | | | |
| heb | Hebrew | 2021869 | 0.98328 | 0.00094 | 0.99606 | 0.00045 | 0.99145 | 0.00011 |
| heg | Helong | 9068 | 0.999 | 3e-05 | | | | |
| heh | Hehe | 8605 | 0.9955 | 0.00011 | | | | 0.00022 |
| her | Herero | 154169 | 0.99549 | 6e-05 | | 6e-05 | | |
| hif | Fiji Hindi | 17894 | 0.98742 | 0.00017 | | 0.00034 | | 0.00077 |
| hig | Kamwe | 7900 | 0.999 | 0.0 | | | | |
| hil | Hiligaynon | 1762387 | 0.98665 | 0.00069 | | 0.00057 | 1.0 | 0.0 |
| hin | Hindi | 2227417 | 0.80097 | 0.01342 | 0.67444 | 0.05551 | 0.62 | 0.00832 |
| hix | Hixkaryána | 7756 | 1.0 | 0.0 | | | | |
| hla | Halia | 7706 | 0.999 | 3e-05 | | | | |
| hlt | Matu Chin | 7916 | 1.0 | 0.0 | | | 0.92982 | 0.0 |
| hmn | Hmong | 163469 | 1.0 | 0.0 | | | | |

Table 12: Performance of GlotLID-M on GlotLID-C test, FLORES-200 and UDHR benchmarks (part 8)

| iso639-3 | Language | \|Sentences\| | GlotLID-C F1↑ | FPR↓ | FLORES-200 F1↑ | FPR↓ | UDHR F1↑ | FPR↓ |
|---|---|---|---|---|---|---|---|---|
| hmo | Hiri Motu | 1029005 | 0.97791 | 0.00113 | | | | |
| hmr | Hmar | 50021 | 0.99449 | 8e-05 | | | | |
| hne | Chhattisgarhi | 60771 | 0.95589 | 0.00017 | 0.90296 | 0.00256 | | 0.00011 |
| hnj | Hmong Njua | 61134 | 0.99552 | 0.00022 | | | | |
| hnn | Hanunoo | 7931 | 0.9995 | 0.0 | | | | |
| hns | Caribbean Hindustani | 9791 | 0.99599 | 6e-05 | | | 0.88889 | 0.00033 |
| hoc | Ho | 1448 | 0.97035 | 8e-05 | | | | |
| hop | Hopi | 7910 | 1.0 | 0.0 | | | | |
| hot | Hote | 7646 | 1.0 | 0.0 | | | | |
| hra | Hrangkhol | 30908 | 0.9935 | 0.00017 | | | | |
| hrv | Croatian | 1901496 | 0.80685 | 0.01081 | 0.75157 | 0.03272 | 0.58824 | 0.00919 |
| hrx | Hunsrik | 46411 | 0.97319 | 0.00041 | | | | |
| hsb | Upper Sorbian | 31517 | 0.98205 | 0.00058 | | | 1.0 | 0.0 |
| hsn | Xiang Chinese | 4 | | | | | | |
| hto | Minica Huitoto | 7926 | 0.9995 | 3e-05 | | | | |
| hub | Huambisa | 15262 | 0.9985 | 6e-05 | | | | 0.00033 |
| hui | Huli | 7838 | 0.9995 | 0.0 | | | | 0.00011 |
| hun | Hungarian | 4361329 | 0.9726 | 0.00138 | 1.0 | 0.0 | 0.82192 | 0.00285 |
| hus | Huastec | 42824 | 0.98231 | 0.00019 | | | 0.97814 | 0.00044 |
| huu | Murui Huitoto | 7704 | 1.0 | 0.0 | | | 0.98305 | 0.0 |
| huv | San Mateo Del Mar Huave | 10600 | 0.998 | 0.0 | | | | 0.00011 |
| hvn | Sabu | 7902 | 0.9995 | 0.0 | | | | |
| hwc | Hawai'i Creole English | 17681 | 0.85699 | 0.00283 | | | | 0.00011 |
| hye | Armenian | 1425068 | 0.9985 | 6e-05 | 1.0 | 0.0 | 1.0 | 0.0 |
| hyw | Western Armenian | 679292 | 0.9995 | 3e-05 | | | | |
| ian | Iatmul | 7674 | 0.999 | 3e-05 | | | | |
| iba | Iban | 210100 | 0.9995 | 0.0 | | | | |
| ibg | Ibanag | 117204 | 0.997 | 8e-05 | | | | 0.00033 |
| ibo | Igbo | 537005 | 0.99402 | 0.00025 | 0.99951 | 6e-05 | 0.98718 | 0.00022 |
| icr | Islander Creole English | 7849 | 0.99699 | 0.0 | | | | |
| ido | Ido | 40359 | 0.97836 | 0.00041 | | 6e-05 | 0.95 | 0.00011 |
| idu | Idoma | 74684 | 0.97688 | 0.0005 | | | 0.0 | 0.00011 |
| ifa | Amganad Ifugao | 30672 | 0.98516 | 0.00072 | | | | |
| ifb | Batad Ifugao | 7913 | 0.98637 | 0.00011 | | 6e-05 | | |
| ife | Ifè | 7870 | 0.9995 | 0.0 | | 6e-05 | | 0.00011 |
| ifk | Tuwali Ifugao | 7699 | 0.99397 | 3e-05 | | | | |
| ifu | Mayoyao Ifugao | 7819 | 0.999 | 0.0 | | 6e-05 | | 0.00022 |
| ify | Keley-I Kallahan | 29643 | 0.99552 | 0.00022 | | | | |
| ige | Igede | 76711 | 0.9995 | 0.0 | | | | 0.00022 |
| ign | Ignaciano | 15364 | 0.9995 | 3e-05 | | | | |
| igs | Interglossa | 32 | | | | | | |
| iii | Sichuan Yi | 60 | | | | | | |
| ijc | Izon | 3785 | 0.60347 | 0.00069 | | | | 0.00022 |
| ike | Eastern Canadian Inuktitut | 30996 | 0.999 | 3e-05 | | | 0.96842 | 0.00033 |
| ikk | Ika | 7953 | 0.9995 | 0.0 | | | | |
| ikw | Ikwere | 7934 | 0.999 | 0.0 | | | | |
| ilb | Ila | 6805 | 0.99094 | 6e-05 | | | | |
| ile | Interlingue | 17750 | 0.96945 | 0.00033 | | 6e-05 | | 0.00011 |
| ilo | Iloko | 2917828 | 0.9828 | 0.00096 | 0.99951 | 6e-05 | 0.90625 | 0.00131 |
| imo | Imbongu | 7720 | 0.998 | 6e-05 | | | | |
| ina | Interlingua | 58339 | 0.97512 | 0.00083 | | 6e-05 | 0.84892 | 0.00219 |
| inb | Inga | 8000 | 0.9995 | 0.0 | | | | 0.00153 |
| ind | Indonesian | 2835053 | 0.8254 | 0.01117 | 0.91929 | 0.00983 | 0.72 | 0.00405 |
| ino | Inoke-Yate | 7474 | 0.9995 | 0.0 | | | | 0.00044 |
| iou | Tuma-Irumu | 12564 | 0.9985 | 0.0 | | | | |
| ipi | Ipili | 6181 | 0.99944 | 0.0 | | | | |
| iqw | Ikwo | 7943 | 0.98849 | 0.0003 | | | | |
| iri | Rigwe | 7948 | 1.0 | 0.0 | | | | |
| irk | Iraqw | 7936 | 0.9985 | 0.0 | | | | |
| iry | Iraya | 7904 | 0.999 | 3e-05 | | | | |
| isd | Isnag | 7892 | 0.999 | 0.0 | | | | |
| ish | Esan | 111255 | 0.97726 | 0.00033 | | | | 0.00011 |
| isl | Icelandic | 150748 | 0.9955 | 0.00011 | 0.99901 | 0.00011 | 0.9916 | 0.00011 |
| iso | Isoko | 656673 | 0.995 | 0.00017 | | | | 0.00022 |
| ita | Italian | 2485451 | 0.91225 | 0.00523 | 0.99803 | 0.00017 | 0.78947 | 0.00339 |
| its | Isekiri | 11791 | 0.70713 | 0.00267 | | | | |
| itv | Itawit | 7928 | 0.9975 | 3e-05 | | | | |
| ium | Iu Mien | 54967 | 1.0 | 0.0 | | | | 0.00099 |
| ivb | Ibatan | 7894 | 0.9975 | 0.00011 | | | | |
| ivv | Ivatan | 7895 | 0.9975 | 3e-05 | | | | |
| iws | Sepik Iwam | 7956 | 1.0 | 0.0 | | | | |
| ixl | Ixil | 24719 | 0.9985 | 3e-05 | | | | |
| izh | Ingrian | 21 | | | | | | |
| izr | Izere | 7945 | 0.998 | 8e-05 | | 6e-05 | | |
| izz | Izii | 7947 | 0.98445 | 0.00033 | | | | 0.00011 |

Table 13: Performance of GlotLID-M on GlotLID-C test, FLORES-200 and UDHR benchmarks (part 9)

| iso639-3 | Language | \|Sentences\| | GlotLID-C | | FLORES-200 | | UDHR | |
|---|---|---|---|---|---|---|---|---|
| | | | F1↑ | FPR↓ | F1↑ | FPR↓ | F1↑ | FPR↓ |
| jac | Popti' | 8027 | 0.998 | 0.0 | | | | 0.00011 |
| jae | Yabem | 7624 | 1.0 | 0.0 | | | | |
| jam | Jamaican Creole English | 23393 | 0.99299 | 0.00014 | | 6e-05 | | |
| jav | Javanese | 233182 | 0.98066 | 0.00077 | 0.98346 | 0.00187 | 0.97581 | 0.00044 |
| jbo | Lojban | 16508 | 0.99347 | 6e-05 | | | | 0.00099 |
| jbu | Jukun Takum | 7940 | 0.99599 | 6e-05 | | | | 0.00044 |
| jdt | Judeo-Tat | 9 | | | | | | |
| jic | Tol | 7907 | 0.9995 | 0.0 | | | | |
| jiv | Shuar | 17084 | 0.80832 | 0.00165 | | | 0.48387 | 0.01379 |
| jmc | Machame | 7957 | 0.9975 | 8e-05 | | 0.00017 | | |
| jmx | Western Juxtlahuaca Mixtec | 234 | 0.13636 | 3e-05 | | | | 0.00066 |
| jpa | Jewish Palestinian Aramaic | 17 | | | | | | |
| jpn | Japanese | 2236956 | 0.98375 | 0.00088 | 1.0 | 0.0 | 0.71498 | 0.01292 |
| jra | Jarai | 7944 | 0.9995 | 0.0 | | | | |
| jvn | Caribbean Javanese | 7750 | 0.99448 | 3e-05 | | | | |
| kaa | Kara-Kalpak | 53472 | 0.99298 | 0.00011 | | | 0.96667 | 0.00022 |
| kab | Kabyle | 704680 | 0.83958 | 0.00572 | 0.85967 | 0.01221 | | 0.00613 |
| kac | Kachin | 213841 | 0.99651 | 0.00014 | 1.0 | 0.0 | | 0.00011 |
| kal | Kalaallisut | 248414 | 0.996 | 0.00011 | | | 0.98305 | 0.0 |
| kam | Kamba (Kenya) | 359666 | 0.995 | 0.00017 | 0.92406 | 6e-05 | | |
| kan | Kannada | 733181 | 0.98377 | 0.00091 | 1.0 | 0.0 | 1.0 | 0.0 |
| kao | Xaasongaxango | 7909 | 0.9985 | 6e-05 | | | | 0.00033 |
| kap | Bezhta | 912 | 0.99259 | 0.0 | | | | |
| kaq | Capanahua | 7924 | 1.0 | 0.0 | | | | 0.00996 |
| kas | Kashmiri | 12946 | 0.98941 | 6e-05 | 0.97674 | 0.0 | | |
| kat | Georgian | 734914 | 0.9975 | 0.00011 | 1.0 | 0.0 | 1.0 | 0.0 |
| kaz | Kazakh | 386828 | 0.99452 | 0.00028 | 0.99951 | 0.0 | 0.96721 | 0.00033 |
| kbc | Kadiwéu | 7861 | 0.9995 | 3e-05 | | | | |
| kbd | Kabardian | 54247 | 0.99602 | 0.00022 | | | 0.87692 | 0.00175 |
| kbh | Camsá | 7884 | 1.0 | 0.0 | | | | |
| kbm | Iwal | 7796 | 0.9995 | 3e-05 | | | | |
| kbp | Kabiyè | 234402 | 0.9945 | 0.00017 | 0.99901 | 6e-05 | 0.85714 | 0.00219 |
| kbq | Kamano | 10526 | 0.9995 | 0.0 | | | | 0.00011 |
| kbr | Kafa | 7459 | 0.9995 | 3e-05 | | | 0.99174 | 0.00011 |
| kck | Kalanga | 23763 | 0.93843 | 0.0 | | | | 0.00033 |
| kdc | Kutu | 7929 | 0.97477 | 0.00044 | | | | 0.00011 |
| kde | Makonde | 18823 | 0.99649 | 6e-05 | | 0.00034 | 0.59813 | 0.0 |
| kdh | Tem | 1071 | 0.99408 | 0.0 | | | 0.77551 | 0.0 |
| kdi | Kumam | 7938 | 0.9965 | 0.00011 | | | | 0.00022 |
| kdj | Karamojong | 7928 | 0.99699 | 3e-05 | | | | 0.00011 |
| kdl | Tsikimba | 7901 | 0.996 | 8e-05 | | | | |
| kea | Kabuverdianu | 147918 | 0.9449 | 0.00226 | 0.95238 | 0.0 | 0.72727 | 0.00109 |
| kei | Kei | 297 | 0.95556 | 0.0 | | 6e-05 | | 0.00657 |
| kek | Kekchí | 203758 | 0.99701 | 0.00014 | | | 0.97521 | 0.00033 |
| ken | Kenyang | 7933 | 0.999 | 0.0 | | | | |
| ket | Ket | 20 | | | | | | |
| kew | West Kewa | 9393 | 0.99198 | 0.00017 | | | | |
| kex | Kukna | 873 | 0.97143 | 0.0 | | | | |
| kez | Kukele | 7935 | 1.0 | 0.0 | | | | |
| kff | Koya | 7934 | 0.999 | 0.0 | | | | |
| kgf | Kube | 10398 | 1.0 | 0.0 | | | | |
| kgk | Kaiwá | 7725 | 0.9985 | 8e-05 | | | | 0.00011 |
| kgp | Kaingang | 9799 | 0.9995 | 0.0 | | | | |
| kha | Khasi | 36201 | 0.98589 | 0.00017 | | 0.00011 | 0.97521 | 0.00022 |
| khk | Halh Mongolian | 176464 | 0.9965 | 8e-05 | 1.0 | 0.0 | 0.98462 | 0.00011 |
| khm | Khmer | 86506 | 0.999 | 6e-05 | 0.99951 | 0.0 | 1.0 | 0.0 |
| khs | Kasua | 7848 | 0.9985 | 8e-05 | | | | 0.00033 |
| khy | Kele (Democratic Republic of Congo) | 7911 | 0.9975 | 3e-05 | | 6e-05 | | 0.00011 |
| khz | Keapara | 7929 | 0.997 | 8e-05 | | | | |
| kia | Kim | 30712 | 0.99499 | 8e-05 | | | | |
| kik | Kikuyu | 519523 | 0.98863 | 0.00063 | 0.96562 | 0.00403 | | 0.00022 |
| kin | Kinyarwanda | 1575481 | 0.96376 | 0.0016 | 0.91471 | 0.00034 | 0.76336 | 0.0023 |
| kir | Kirghiz | 740492 | 0.98714 | 0.00066 | 1.0 | 0.0 | 0.94488 | 0.00077 |
| kiu | Kirmanjki (individual language) | 6 | | | | | | |
| kix | Khiamniungan Naga | 7935 | 1.0 | 0.0 | | | | |
| kjb | Q'anjob'al | 30505 | 0.996 | 0.00011 | | | | |
| kje | Kisar | 7827 | 0.998 | 3e-05 | | | | |
| kjh | Khakas | 43856 | 0.88244 | 0.00322 | | | 0.83099 | 0.00252 |
| kjs | East Kewa | 7917 | 0.99301 | 0.00025 | | | | 0.00011 |
| kkc | Odoodee | 5179 | 0.99933 | 0.0 | | | | |
| kki | Kagulu | 7957 | 0.99102 | 0.0003 | | | | 0.00077 |
| kkj | Kako | 7920 | 0.999 | 0.0 | | | | 0.00044 |
| kkl | Kosarek Yale | 3524 | 0.99907 | 0.0 | | | | |
| klj | Khalaj | 20 | | | | | | |
| kln | Kalenjin | 4550 | 0.99782 | 0.0 | | | | |

Table 14: Performance of GlotLID-M on GlotLID-C test, FLORES-200 and UDHR benchmarks (part 10)

| iso639-3 | Language | \|Sentences\| | GlotLID-C | | FLORES-200 | | UDHR | |
|---|---|---|---|---|---|---|---|---|
| | | | F1↑ | FPR↓ | F1↑ | FPR↓ | F1↑ | FPR↓ |
| klt | Nukna | 752 | 0.99567 | 3e-05 | | | | |
| klv | Maskelynes | 7892 | 0.999 | 3e-05 | | | | |
| kma | Konni | 7936 | 0.999 | 6e-05 | | | | |
| kmb | Kimbundu | 385054 | 0.98706 | 0.0005 | 0.96321 | 0.00415 | 0.99194 | 0.00011 |
| kmg | Kâte | 6156 | 1.0 | 0.0 | | | | |
| kmh | Kalam | 15160 | 0.999 | 3e-05 | | | | |
| kmk | Limos Kalinga | 7902 | 0.999 | 3e-05 | | | | |
| kmm | Kom (India) | 30098 | 0.99253 | 0.0003 | | | | |
| kmo | Kwoma | 7848 | 0.9995 | 3e-05 | | | | |
| kmr | Northern Kurdish | 262682 | 0.999 | 3e-05 | 0.99901 | 0.0 | 0.66667 | 0.00646 |
| kms | Kamasau | 7838 | 1.0 | 0.0 | | | | 0.00011 |
| kmu | Kanite | 7543 | 1.0 | 0.0 | | | | 0.00044 |
| knc | Central Kanuri | 16253 | 0.99498 | 3e-05 | 0.8634 | 0.00153 | 0.9635 | 0.00044 |
| kne | Kankanaey | 7675 | 0.997 | 8e-05 | | | | |
| knf | Mankanya | 9446 | 0.9985 | 0.0 | | | | |
| kng | Koongo | 7934 | 0.99497 | 0.0 | | | 0.0 | 0.00033 |
| knj | Western Kanjobal | 7919 | 0.99699 | 3e-05 | | | | 0.00022 |
| knk | Kuranko | 7863 | 0.998 | 3e-05 | | 6e-05 | | |
| kno | Kono (Sierra Leone) | 7827 | 0.9995 | 0.0 | | | | |
| knv | Tabo | 15492 | 0.9995 | 3e-05 | | | | 0.00131 |
| knx | Kendayan | 426 | 0.95238 | 6e-05 | | | | |
| kny | Kanyok | 4209 | 0.99619 | 3e-05 | | | | |
| kog | Cogui | 7874 | 0.9995 | 0.0 | | | | 0.00131 |
| koi | Komi-Permyak | 10043 | 0.98942 | 8e-05 | | | 0.95082 | 0.00055 |
| kom | Komi | 10000 | 0.98939 | 0.0 | | | | |
| kon | Kongo | 885632 | 0.999 | 0.0 | 0.99802 | 0.0 | | |
| koo | Konzo | 208525 | 0.99354 | 0.00033 | | 0.00017 | 0.79389 | 0.00011 |
| kor | Korean | 1736401 | 0.99701 | 0.00017 | 1.0 | 0.0 | 0.94488 | 0.00077 |
| kos | Kosraean | 87905 | 0.99501 | 0.00019 | | | | |
| kpf | Komba | 7855 | 0.998 | 3e-05 | | | | 0.00022 |
| kpg | Kapingamarangi | 30071 | 0.9955 | 0.00011 | | | | |
| kpj | Karajá | 7642 | 0.9995 | 0.0 | | | | |
| kpr | Korafe-Yegha | 7634 | 0.9995 | 3e-05 | | | | |
| kpv | Komi-Zyrian | 9672 | 0.99699 | 0.0 | | | | 0.00011 |
| kpw | Kobon | 7722 | 1.0 | 0.0 | | | | |
| kpx | Mountain Koiali | 7794 | 0.9995 | 0.0 | | | | |
| kpz | Kupsabiny | 7945 | 0.9975 | 8e-05 | | | | 0.00044 |
| kqc | Doromu-Koki | 5185 | 0.99873 | 0.0 | | | | |
| kqe | Kalagan | 7872 | 0.99298 | 0.00011 | | | | |
| kqf | Kakabai | 654 | 0.99487 | 0.0 | | | | |
| kql | Kyenele | 667 | 0.99 | 0.0 | | | | |
| kqn | Kaonde | 506556 | 0.99701 | 0.00014 | | 0.00011 | 1.0 | 0.0 |
| kqo | Eastern Krahn | 7891 | 1.0 | 0.0 | | | | |
| kqp | Kimré | 7916 | 1.0 | 0.0 | | | | 0.00022 |
| kqs | Northern Kissi | 7926 | 0.9985 | 3e-05 | | | 0.20896 | 0.0 |
| kqw | Kandas | 3093 | 0.99782 | 6e-05 | | | | |
| kqy | Koorete | 7827 | 0.9995 | 0.0 | | | | |
| krc | Karachay-Balkar | 19397 | 0.98844 | 0.00017 | | | | 0.00022 |
| kri | Krio | 200310 | 0.96654 | 0.00138 | | | 0.96875 | 0.0 |
| krj | Kinaray-A | 7942 | 0.98376 | 3e-05 | | | | |
| krl | Karelian | 194 | 0.57895 | 3e-05 | | | 0.0875 | 0.0 |
| kru | Kurukh | 7898 | 0.9995 | 0.0 | | | | |
| ksb | Shambala | 7920 | 0.99549 | 8e-05 | | | | |
| ksc | Southern Kalinga | 7833 | 0.999 | 3e-05 | | | | |
| ksd | Kuanua | 7947 | 0.99649 | 6e-05 | | | | |
| ksf | Bafia | 8272 | 0.99548 | 0.0 | | 6e-05 | | 0.00066 |
| ksh | Kölsch | 10080 | 0.96923 | 0.00014 | | 6e-05 | | 0.00022 |
| ksj | Uare | 4199 | 0.9992 | 0.0 | | | | |
| ksp | Kaba | 4842 | 0.99735 | 6e-05 | | | | |
| ksr | Borong | 9572 | 0.9995 | 3e-05 | | | | 0.00033 |
| kss | Southern Kisi | 180188 | 0.9985 | 6e-05 | | 6e-05 | | 0.00241 |
| ksw | S'gaw Karen | 149668 | 1.0 | 0.0 | | | | 0.00033 |
| ktb | Kambaata | 7788 | 0.9965 | 8e-05 | | | | |
| ktj | Plapo Krumen | 7841 | 0.9985 | 6e-05 | | | | |
| ktm | Kurti | 2425 | 0.98947 | 0.00014 | | | | |
| kto | Kuot | 7810 | 0.9995 | 3e-05 | | | | |
| ktu | Kituba (Democratic Republic of Congo) | 18217 | 0.99448 | 6e-05 | | 6e-05 | 0.79322 | 0.00635 |
| kua | Kuanyama | 473208 | 0.93245 | 0.00358 | | 0.0004 | | 0.00022 |
| kub | Kutep | 7926 | 0.999 | 3e-05 | | | | |
| kud | 'Auhelawa | 7895 | 0.997 | 8e-05 | | | | 0.00011 |
| kue | Kuman (Papua New Guinea) | 7942 | 0.9995 | 3e-05 | | | | |
| kuj | Kuria | 7933 | 0.9995 | 0.0 | | | | 0.00022 |
| kum | Kumyk | 10568 | 0.99548 | 3e-05 | | | | 0.00022 |
| kup | Kunimaipa | 7530 | 1.0 | 0.0 | | | | |
| kus | Kusaal | 7926 | 0.998 | 3e-05 | | | | 0.00109 |

Table 15: Performance of GlotLID-M on GlotLID-C test, FLORES-200 and UDHR benchmarks (part 11)

| iso639-3 | Language | \|Sentences\| | GlotLID-C | | FLORES-200 | | UDHR | |
|---|---|---|---|---|---|---|---|---|
| | | | F1↑ | FPR↓ | F1↑ | FPR↓ | F1↑ | FPR↓ |
| kvj | Psikye | 7908 | 1.0 | 0.0 | | | | |
| kvn | Border Kuna | 7926 | 1.0 | 0.0 | | | | |
| kwd | Kwaio | 7876 | 0.9995 | 3e-05 | | | | |
| kwf | Kwara'ae | 9038 | 0.9975 | 3e-05 | | | | |
| kwi | Awa-Cuaiquer | 7920 | 0.999 | 0.0 | | 6e-05 | 0.84716 | 0.0 |
| kwj | Kwanga | 7936 | 1.0 | 0.0 | | | | |
| kwn | Kwangali | 304624 | 0.99552 | 0.00022 | | 6e-05 | | 0.00022 |
| kwy | San Salvador Kongo | 471082 | 0.99352 | 0.00028 | | 0.0004 | | 0.00022 |
| kxc | Konso | 7905 | 0.9985 | 0.0 | | | | |
| kxm | Northern Khmer | 7912 | 1.0 | 0.0 | | | | |
| kxw | Konai | 7887 | 1.0 | 0.0 | | | | |
| kyc | Kyaka | 7891 | 0.998 | 8e-05 | | | | |
| kyf | Kouya | 7797 | 0.9995 | 3e-05 | | | | |
| kyg | Keyagana | 9517 | 0.999 | 6e-05 | | | | |
| kyq | Kenga | 7902 | 1.0 | 0.0 | | | | 0.00022 |
| kyu | Western Kayah | 7907 | 0.9995 | 0.0 | | | | |
| kyz | Kayabí | 7550 | 1.0 | 0.0 | | | | |
| kze | Kosena | 7704 | 0.9965 | 0.00011 | | | | |
| kzf | Da'a Kaili | 6792 | 0.99649 | 6e-05 | | | | |
| kzj | Coastal Kadazan | 6159 | 0.95996 | 0.00047 | | | | 0.00033 |
| kzn | Kokola | 2619 | 0.82961 | 3e-05 | | | | 0.00011 |
| laa | Southern Subanen | 10 | | | | | | |
| lac | Lacandon | 7923 | 0.999 | 3e-05 | | | | |
| lad | Ladino | 11721 | 0.96319 | 0.0 | | | 0.90598 | 0.00044 |
| lai | Lambya | 7869 | 0.99548 | 3e-05 | | | | |
| laj | Lango (Uganda) | 7941 | 0.99549 | 6e-05 | | | | 0.00011 |
| lam | Lamba | 38968 | 0.98941 | 6e-05 | | 0.00023 | | |
| lao | Lao | 26289 | 0.9995 | 0.0 | 1.0 | 0.0 | 1.0 | 0.0 |
| las | Lama (Togo) | 7903 | 0.9995 | 0.0 | | | | |
| lat | Latin | 217775 | 0.97705 | 0.00069 | | | 0.975 | 0.00011 |
| lav | Latvian | 345532 | 0.99551 | 0.00019 | | | | |
| lbb | Label | 667 | 1.0 | 0.0 | | | | |
| lbe | Lak | 218 | 0.98462 | 0.0 | | | | 0.00044 |
| lbj | Ladakhi | 7163 | 1.0 | 0.0 | | 0.00017 | | |
| lbk | Central Bontok | 9400 | 0.996 | 0.00014 | | | | |
| lch | Luchazi | 236 | 0.64286 | 0.0 | | 0.00114 | | 0.00011 |
| lcm | Tungag | 9544 | 0.99599 | 6e-05 | | | | |
| ldi | Laari | 61299 | 0.9935 | 0.00017 | | 6e-05 | | |
| ldn | Láadan | 144 | 0.97872 | | | | | |
| lea | Lega-Shabunda | 3823 | 0.9907 | 6e-05 | | 0.00011 | | 0.00011 |
| led | Lendu | 4500 | 0.99485 | 8e-05 | | | | 0.00022 |
| lee | Lyélé | 7873 | 0.999 | 3e-05 | | 6e-05 | | |
| lef | Lelemi | 7901 | 0.998 | | | | | |
| leh | Lenje | 60420 | 0.99249 | 0.00017 | | 0.00011 | | 0.00011 |
| lem | Nomaande | 7917 | 0.9995 | 3e-05 | | | | |
| leu | Kara (Papua New Guinea) | 7869 | 0.9995 | 0.0 | | | | |
| lew | Ledo Kaili | 7859 | 0.997 | 8e-05 | | | | |
| lex | Luang | 9630 | 0.9995 | 0.0 | | | | |
| lez | Lezghian | 421 | 0.98305 | 3e-05 | | | | |
| lfn | Lingua Franca Nova | 22272 | 0.97358 | 0.00028 | | | | 0.00011 |
| lgm | Lega-Mwenga | 7945 | 0.99551 | 0.00017 | | | | |
| lhi | Lahu Shi | 7872 | 1.0 | 0.0 | | | | 0.00022 |
| lhm | Lhomi | 7812 | 0.9995 | 0.0 | | | | |
| lhu | Lahu | 73972 | 1.0 | 0.0 | | | | |
| lia | West-Central Limba | 7904 | 1.0 | 0.0 | | | 0.94643 | 0.0 |
| lid | Nyindrou | 9513 | 1.0 | 0.0 | | | | 0.00011 |
| lif | Limbu | 15688 | 1.0 | 0.0 | | | | |
| lij | Ligurian | 36632 | 0.97681 | 0.00041 | 0.99901 | 6e-05 | 0.49785 | 0.0127 |
| lim | Limburgan | 141486 | 0.96436 | 0.00127 | 0.99253 | 0.0 | | 0.00011 |
| lin | Lingala | 1856585 | 0.98325 | 0.00088 | 0.99901 | 0.00011 | 0.99145 | 0.00022 |
| lip | Sekpele | 7899 | 0.9995 | 0.0 | | | | |
| lir | Liberian English | 24782 | 0.53173 | 0.01958 | | | | |
| lit | Lithuanian | 2813062 | 0.97971 | 0.00085 | 0.99951 | 0.0 | 0.9375 | 0.00088 |
| liv | Liv | 33 | | | | | | |
| ljp | Lampung Api | 7900 | 0.99448 | 3e-05 | | | | |
| lkt | Lakota | 22 | | | | | | |
| llb | Lolo | 30215 | 0.98327 | 0.00091 | | 0.00023 | | 0.00011 |
| lld | Ladin | 1049 | 0.9589 | 0.0 | | | 0.53465 | 0.00153 |
| lln | Lele (Chad) | 2151 | 0.90074 | 6e-05 | | 6e-05 | | 0.0012 |
| lmk | Lamkang | 7953 | 0.998 | 3e-05 | | | | 0.00011 |
| lmo | Lombard | 62982 | 0.97 | 0.00083 | 0.99554 | 0.00011 | | 0.00919 |
| lmp | Limbum | 7927 | 0.999 | 3e-05 | | | | |
| lob | Lobi | 7937 | 0.999 | 0.0 | | | 0.92857 | 0.00022 |
| loe | Saluan | 137 | 0.86667 | 0.0 | | | | |
| log | Logo | 7844 | 1.0 | 0.0 | | | | |

Table 16: Performance of GlotLID-M on GlotLID-C test, FLORES-200 and UDHR benchmarks (part 12)

| iso639-3 | Language | \|Sentences\| | GlotLID-C | | FLORES-200 | | UDHR | |
|---|---|---|---|---|---|---|---|---|
| | | | F1↑ | FPR↓ | F1↑ | FPR↓ | F1↑ | FPR↓ |
| lol | Mongo | 7934 | 0.997 | 0.00011 | | | | |
| lom | Loma (Liberia) | 8918 | 0.96124 | 0.00014 | | | | |
| loq | Lobala | 3745 | 0.99573 | 3e-05 | | | | |
| lou | Louisiana Creole | 22 | | | | | | |
| loz | Lozi | 961751 | 0.99303 | 0.00033 | | 0.00011 | 0.84892 | 0.0 |
| lsi | Lashi | 7897 | 0.9995 | 0.0 | | | | |
| lsm | Saamia | 7940 | 0.9985 | 3e-05 | | 6e-05 | | 0.00077 |
| ltg | Latgalian | 14697 | 0.97178 | 6e-05 | 0.99653 | 0.0 | | |
| ltz | Luxembourgish | 133586 | 0.98798 | 0.00028 | 0.99951 | 0.0 | 0.98305 | 0.0 |
| lua | Luba-Lulua | 1138848 | 0.99353 | 0.0003 | 0.99653 | 6e-05 | 0.71186 | 0.00175 |
| lub | Luba-Katanga | 651814 | 0.99552 | 0.00025 | | | | 0.00285 |
| lue | Luvale | 598110 | 0.998 | 8e-05 | | 0.00074 | 0.99145 | 0.0 |
| lug | Ganda | 297207 | 0.98406 | 0.00055 | 0.99653 | 0.0 | 0.9589 | 0.00066 |
| lun | Lunda | 394555 | 0.9985 | 3e-05 | | 0.00023 | 0.81752 | 0.00033 |
| luo | Luo (Kenya and Tanzania) | 562230 | 0.99155 | 0.00041 | 1.0 | 0.0 | | 0.00088 |
| lus | Lushai | 568212 | 0.99301 | 0.00025 | 0.99653 | 6e-05 | 0.93548 | 0.00088 |
| lut | Lushootseed | 59 | 1.0 | 0.0 | | | | |
| lvs | Standard Latvian | 3176411 | 0.96455 | 0.00182 | 0.99655 | 0.00034 | 0.94118 | 0.00164 |
| lwo | Luwo | 7810 | 0.999 | 3e-05 | | | | |
| lww | Lewo | 7830 | 0.999 | 6e-05 | | | | |
| lzh | Literary Chinese | 17606 | 0.95639 | 0.00014 | | | | |
| lzz | Laz | 75 | 0.125 | 3e-05 | | | | |
| maa | San Jerónimo Tecóatl Mazatec | 23955 | 0.9995 | 0.0 | | | | |
| mad | Madurese | 8060 | 0.9985 | 6e-05 | | 0.00028 | 0.92174 | 0.0 |
| maf | Mafa | 7943 | 0.998 | 3e-05 | | | | |
| mag | Magahi | 6208 | 0.96204 | 3e-05 | 0.95459 | 0.00136 | 0.75385 | 0.00044 |
| mah | Marshallese | 532466 | 0.99651 | 0.00019 | | | 0.96063 | 0.00055 |
| mai | Maithili | 32796 | 0.95854 | 0.00017 | 0.97366 | 6e-05 | 0.83099 | 0.0 |
| maj | Jalapa De Díaz Mazatec | 7883 | 0.999 | 3e-05 | | | | |
| mak | Makasar | 7860 | 0.9985 | 0.0 | | | | |
| mal | Malayalam | 737267 | 1.0 | 0.0 | 1.0 | 0.0 | 1.0 | 0.0 |
| mam | Mam | 244791 | 0.99601 | 0.00017 | | | 0.93913 | 0.00022 |
| maq | Chiquihuitlán Mazatec | 7930 | 0.999 | 3e-05 | | | | |
| mar | Marathi | 1382828 | 0.9896 | 0.00055 | 1.0 | 0.0 | 0.99174 | 0.00011 |
| mas | Masai | 31306 | 0.999 | 0.0 | | | | 0.00055 |
| mau | Huautla Mazatec | 197845 | 0.9985 | 6e-05 | | | | 0.00372 |
| mav | Sateré-Mawé | 15290 | 0.85929 | 0.00171 | | | | 0.0023 |
| maw | Mampruli | 7890 | 0.999 | 6e-05 | | | | 0.00296 |
| max | North Moluccan Malay | 427 | 0.9011 | 0.0 | | | | |
| maz | Central Mazahua | 9655 | 0.93082 | 0.00055 | | | 0.83582 | 0.00186 |
| mbb | Western Bukidnon Manobo | 7852 | 0.9985 | 3e-05 | | | | |
| mbc | Macushi | 9275 | 0.92423 | 0.00022 | | 6e-05 | | 0.00022 |
| mbd | Dibabawon Manobo | 7818 | 0.9945 | 0.00014 | | | | |
| mbf | Baba Malay | 7930 | 0.99449 | 8e-05 | | 6e-05 | | |
| mbh | Mangseng | 7897 | 0.9995 | 3e-05 | | | | |
| mbi | Ilianen Manobo | 7894 | 0.9985 | 6e-05 | | | | |
| mbj | Nadëb | 7842 | 1.0 | 0.0 | | | | |
| mbl | Maxakalí | 7908 | 1.0 | 0.0 | | | | |
| mbs | Sarangani Manobo | 9330 | 1.0 | 0.0 | | | | |
| mbt | Matigsalug Manobo | 7888 | 0.9995 | 3e-05 | | | | 0.00044 |
| mca | Maca | 7939 | 1.0 | 0.0 | | | | |
| mcb | Machiguenga | 7743 | 0.999 | 3e-05 | | | | 0.00033 |
| mcd | Sharanahua | 7472 | 0.9995 | 3e-05 | | | 0.97521 | 0.00022 |
| mcf | Matsés | 7847 | 1.0 | 0.0 | | | 1.0 | 0.0 |
| mck | Mbunda | 157207 | 0.99451 | 0.00022 | | 0.00119 | | 0.00788 |
| mcn | Masana | 30987 | 0.99651 | 0.00014 | | 6e-05 | | |
| mco | Coatlán Mixe | 230569 | 0.98645 | 0.00028 | | | | 0.00449 |
| mcp | Makaa | 13175 | 0.999 | 3e-05 | | 6e-05 | | 0.00033 |
| mcq | Ese | 7924 | 1.0 | 0.0 | | | | |
| mcu | Cameroon Mambila | 7894 | 0.9995 | 0.0 | | | | |
| mda | Mada (Nigeria) | 7931 | 0.999 | 0.0 | | | | |
| mdf | Moksha | 86 | 0.5 | 3e-05 | | | | |
| mdy | Male (Ethiopia) | 37003 | 1.0 | 0.0 | | | | |
| med | Melpa | 7510 | 1.0 | 0.0 | | | | |
| mee | Mengen | 7874 | 0.9995 | 3e-05 | | | | |
| meh | Southwestern Tlaxiaco Mixtec | 1543 | 0.10196 | 0.00014 | | | | 0.00022 |
| mej | Meyah | 7842 | 0.999 | 0.0 | | | | |
| mek | Mekeo | 7799 | 0.999 | 3e-05 | | | | |
| men | Mende (Sierra Leone) | 11481 | 0.99649 | 3e-05 | | | 0.9771 | 0.00033 |
| meq | Merey | 7903 | 0.998 | 8e-05 | | | | |
| mer | Meru | 3946 | 0.95789 | 0.0 | | 0.00011 | | |
| meu | Motu | 119511 | 0.98073 | 0.00014 | | | | |
| mev | Mano | 390 | 0.95726 | 3e-05 | | | | 0.00011 |
| mfa | Pattani Malay | 319 | 0.925 | 0.0 | | | | 0.00011 |
| mfe | Morisyen | 445479 | 0.99006 | 0.00044 | | | | 0.00022 |

Table 17: Performance of GlotLID-M on GlotLID-C test, FLORES-200 and UDHR benchmarks (part 13)

| iso639-3 | Language | |Sentences| | GlotLID-C | | FLORES-200 | | UDHR | |
|---|---|---|---|---|---|---|---|---|
| | | | F1↑ | FPR↓ | F1↑ | FPR↓ | F1↑ | FPR↓ |
| mfh | Matal | 7949 | 1.0 | 0.0 | | | | |
| mfi | Wandala | 7930 | 1.0 | 0.0 | | | | 0.00011 |
| mfk | North Mofu | 7937 | 0.99346 | 3e-05 | | | | |
| mfq | Moba | 9584 | 0.9965 | 8e-05 | | 0.00045 | 0.88722 | 0.00164 |
| mfy | Mayo | 10521 | 0.99548 | 3e-05 | | | | 0.00011 |
| mfz | Mabaan | 7874 | 1.0 | 0.0 | | | | |
| mgc | Morokodo | 3827 | 0.99911 | 0.0 | | | | |
| mgh | Makhuwa-Meetto | 78022 | 0.997 | 8e-05 | | 0.0004 | | 0.00033 |
| mgm | Mambae | 209 | 0.68421 | 6e-05 | | | | 0.00011 |
| mgo | Meta' | 111 | 0.92 | 0.0 | | | | 0.00022 |
| mgr | Mambwe-Lungu | 164338 | 0.99007 | 0.00047 | | 0.00182 | | 0.00022 |
| mgv | Matengo | 70 | 0.6 | 0.0 | | | | |
| mhi | Ma'di | 7941 | 0.9985 | 3e-05 | | | | |
| mhl | Mauwake | 7578 | 0.999 | 3e-05 | | | | |
| mhr | Eastern Mari | 22422 | 0.98593 | 0.00025 | | | | 0.00022 |
| mhw | Mbukushu | 609 | 0.97727 | 0.0 | | 0.00028 | | 0.00011 |
| mhx | Maru | 7918 | 0.9995 | 0.0 | | | | |
| mhy | Ma'anyan | 7889 | 0.99448 | 3e-05 | | | | |
| mib | Atatláhuca Mixtec | 7887 | 1.0 | 0.0 | | | | |
| mic | Mi'kmaq | 7923 | 0.9995 | 0.0 | | | 0.15625 | 0.0 |
| mie | Ocotepec Mixtec | 7899 | 1.0 | 0.0 | | | | |
| mif | Mofu-Gudur | 7875 | 0.9985 | 3e-05 | | | | |
| mig | San Miguel El Grande Mixtec | 7940 | 0.9995 | 0.0 | | | | 0.00011 |
| mih | Chayuco Mixtec | 7897 | 0.998 | 0.00011 | | | | |
| mik | Mikasuki | 104 | 0.72727 | 0.0 | | | | |
| mil | Peñoles Mixtec | 7894 | 0.9995 | 3e-05 | | | | 0.00022 |
| min | Minangkabau | 132106 | 0.99248 | 0.00014 | 0.6616 | 0.00017 | 0.87591 | 0.00175 |
| mio | Pinotepa Nacional Mixtec | 7914 | 0.9985 | 6e-05 | | | | |
| miq | Mískito | 86121 | 0.97908 | 0.00069 | | | 0.832 | 0.0 |
| mir | Isthmus Mixe | 7505 | 1.0 | 0.0 | | | | |
| mit | Southern Puebla Mixtec | 7779 | 0.999 | 3e-05 | | | | |
| miy | Ayutla Mixtec | 7915 | 1.0 | 0.0 | | | | |
| miz | Coatzospan Mixtec | 7953 | 1.0 | 0.0 | | | | 0.00011 |
| mjc | San Juan Colorado Mixtec | 7915 | 0.9995 | 0.0 | | | | |
| mjw | Karbi | 7953 | 0.9995 | 0.0 | | | | |
| mkd | Macedonian | 809994 | 0.99253 | 0.00033 | 1.0 | 0.0 | 0.99174 | 0.0 |
| mkl | Mokole | 7866 | 0.9995 | 0.0 | | | | |
| mkn | Kupang Malay | 9069 | 0.998 | 3e-05 | | | | |
| mks | Silacayoapan Mixtec | 7949 | 0.9985 | 3e-05 | | | | |
| mkz | Makasae | 1720 | 0.98266 | 0.0 | | 6e-05 | | 0.00033 |
| mlg | Malagasy | 30062 | 1.0 | 0.0 | | | | |
| mlh | Mape | 7925 | 0.9985 | 6e-05 | | | | |
| mlp | Bargam | 7729 | 1.0 | 0.0 | | 0.00017 | | 0.00011 |
| mlt | Maltese | 2281035 | 0.97815 | 0.0008 | 0.97401 | 0.00307 | 0.77419 | 0.00383 |
| mlu | To'abaita | 1036 | 0.99375 | 3e-05 | | 6e-05 | | 0.00011 |
| mmn | Mamanwa | 7829 | 0.999 | 3e-05 | | | | |
| mmo | Mangga Buang | 7937 | 0.9985 | 3e-05 | | | | |
| mmx | Madak | 10379 | 0.999 | 0.0 | | | | |
| mna | Mbula | 13167 | 0.998 | 3e-05 | | | | |
| mnb | Muna | 7924 | 1.0 | 0.0 | | | | 0.00022 |
| mnc | Manchu | 2 | | | | | | |
| mnf | Mundani | 7866 | 0.9995 | 0.0 | | | | |
| mni | Manipuri | 48249 | 0.9899 | 0.0 | 0.99901 | 6e-05 | | |
| mnk | Mandinka | 7913 | 0.9985 | 3e-05 | | | | 0.00011 |
| mnr | Mono (USA) | 3 | | | | | | |
| mnw | Mon | 9 | | | | | | |
| mnx | Manikion | 7376 | 0.9995 | 3e-05 | | | | |
| mny | Manyawa | 50297 | 0.97712 | 0.00017 | | | | |
| moa | Mwan | 7939 | 0.9995 | 0.0 | | | | |
| moc | Mocoví | 16176 | 0.9995 | 3e-05 | | | | |
| mog | Mongondow | 7903 | 1.0 | 0.0 | | | | |
| moh | Mohawk | 953 | 0.99678 | 0.0 | | | | |
| mon | Mongolian | 102788 | 0.98521 | 0.0008 | | | | |
| mop | Mopán Maya | 8965 | 0.999 | 0.0 | | | | |
| mor | Moro | 7935 | 1.0 | 0.0 | | | 0.9916 | 0.0 |
| mos | Mossi | 626622 | 0.9925 | 0.00019 | 0.98138 | 0.0 | 0.97015 | 0.00044 |
| mox | Molima | 7830 | 0.998 | 6e-05 | | | | |
| mpg | Marba | 7909 | 0.9985 | 3e-05 | | | | |
| mph | Maung | 598 | 1.0 | 0.0 | | | | |
| mpm | Yosondúa Mixtec | 7900 | 0.999 | 6e-05 | | | | 0.00011 |
| mpp | Migabac | 3959 | 1.0 | 0.0 | | | | |
| mps | Dadibi | 30278 | 0.9995 | 0.0 | | | | |
| mpt | Mian | 7692 | 0.999 | 3e-05 | | | | 0.00241 |
| mpx | Misima-Panaeati | 8788 | 0.998 | 3e-05 | | | | 0.00022 |
| mqb | Mbuko | 7873 | 0.9995 | 0.0 | | | | |

Table 18: Performance of GlotLID-M on GlotLID-C test, FLORES-200 and UDHR benchmarks (part 14)

|  |  |  | GlotLID-C | | FLORES-200 | | UDHR | |
| --- | --- | --- | --- | --- | --- | --- | --- | --- |
| iso639-3 | Language | \|Sentences\| | F1↑ | FPR↓ | F1↑ | FPR↓ | F1↑ | FPR↓ |
| mqj | Mamasa | 8069 | 0.992 | 0.00022 |  |  |  | 0.00011 |
| mqy | Manggarai | 7953 | 0.9985 | 3e-05 |  |  |  |  |
| mrg | Mising | 6373 | 0.9995 | 0.0 |  |  |  | 0.00022 |
| mri | Maori | 79437 | 0.98948 | 0.00025 | 0.99901 | 6e-05 | 0.8227 | 0.00022 |
| mrj | Western Mari | 10083 | 0.99041 | 0.0 |  |  |  | 0.00011 |
| mrq | North Marquesan | 1007 | 0.96774 | 0.0 |  | 6e-05 |  | 0.00011 |
| mrv | Mangareva | 187 | 0.8 | 0.0 |  |  |  |  |
| mrw | Maranao | 7932 | 0.9985 | 3e-05 |  |  |  |  |
| msa | Malay (macrolanguage) | 53819 | 0.998 | 3e-05 |  | 6e-05 |  | 0.00011 |
| msb | Masbatenyo | 7900 | 0.99549 | 8e-05 |  |  |  |  |
| msc | Sankaran Maninka | 1783 | 0.98641 | 6e-05 |  |  |  | 0.00077 |
| mse | Musey | 7917 | 0.9975 | 3e-05 |  |  |  |  |
| msk | Mansaka | 7775 | 0.99101 | 0.00028 |  |  |  |  |
| msm | Agusan Manobo | 7776 | 0.995 | 0.00014 |  |  |  |  |
| msy | Aruamu | 7694 | 0.9995 | 3e-05 |  |  |  |  |
| mta | Cotabato Manobo | 7857 | 1.0 | 0.0 |  |  |  |  |
| mtg | Una | 7834 | 0.999 | 0.0 |  |  |  |  |
| mti | Maiwa (Papua New Guinea) | 7899 | 0.99699 | 3e-05 |  |  |  |  |
| mtj | Moskona | 7846 | 0.9985 | 6e-05 |  |  |  |  |
| mto | Totontepec Mixe | 7906 | 1.0 | 0.0 |  |  | 0.0 | 0.00088 |
| mtp | Wichí Lhamtés Nocten | 7934 | 1.0 | 0.0 |  |  |  | 0.00011 |
| mua | Mundang | 7908 | 0.9995 | 0.0 |  |  |  |  |
| mug | Musgu | 30824 | 0.99451 | 0.00019 |  |  |  |  |
| muh | Mündü | 7469 | 1.0 | 0.0 |  |  |  |  |
| mur | Murle | 7816 | 0.999 | 0.0 |  |  |  | 0.00011 |
| mus | Creek | 598 | 0.9172 | 8e-05 |  |  |  |  |
| mux | Bo-Ung | 7692 | 0.999 | 3e-05 |  |  |  |  |
| muy | Muyang | 7876 | 0.9995 | 3e-05 |  |  |  | 0.00011 |
| mva | Manam | 7947 | 0.999 | 0.0 |  |  |  |  |
| mvn | Minaveha | 7827 | 0.99699 | 0.0 |  |  |  |  |
| mvp | Duri | 7814 | 0.996 | 0.00011 |  |  |  |  |
| mvv | Tagal Murut | 28 |  |  |  |  |  |  |
| mwc | Are | 1150 | 0.99342 | 3e-05 |  |  |  |  |
| mwf | Murrinh-Patha | 316 | 1.0 | 0.0 |  |  |  |  |
| mwl | Mirandese | 33797 | 0.99247 | 8e-05 |  |  |  | 0.00033 |
| mwm | Sar | 30621 | 0.999 | 6e-05 |  |  |  |  |
| mwn | Nyamwanga | 54496 | 0.99347 | 6e-05 |  | 0.0004 |  | 0.00011 |
| mwp | Kala Lagaw Ya | 4301 | 0.9984 | 3e-05 |  |  |  |  |
| mwq | Mün Chin | 7927 | 1.0 | 0.0 |  |  |  | 0.00022 |
| mwv | Mentawai | 7901 | 0.999 | 3e-05 |  |  |  |  |
| mww | Hmong Daw | 8021 | 0.99346 | 0.0 |  |  |  |  |
| mxb | Tezoatlán Mixtec | 7938 | 1.0 | 0.0 |  |  |  | 0.00022 |
| mxp | Tlahuitoltepec Mixe | 7924 | 0.9995 | 0.0 |  |  |  | 0.00011 |
| mxq | Juquila Mixe | 7933 | 1.0 | 0.0 |  |  |  | 0.00011 |
| mxt | Jamiltepec Mixtec | 7916 | 0.9975 | 3e-05 |  |  |  |  |
| mxv | Metlatónoc Mixtec | 141566 | 0.99151 | 0.00028 |  |  | 0.14925 | 0.0 |
| mya | Burmese | 498977 | 0.998 | 0.00011 | 1.0 | 0.0 | 0.66292 | 0.00646 |
| myb | Mbay | 7908 | 0.9975 | 3e-05 |  |  |  | 0.00011 |
| myk | Mamara Senoufo | 7920 | 0.999 | 3e-05 |  |  |  |  |
| myu | Mundurukú | 7683 | 0.9995 | 3e-05 |  |  |  |  |
| myv | Erzya | 18314 | 0.992 | 0.00022 |  |  |  |  |
| myw | Muyuw | 6727 | 1.0 | 0.0 |  |  |  |  |
| myx | Masaaba | 7954 | 0.999 | 0.0 |  |  |  |  |
| myy | Macuna | 7837 | 0.9995 | 0.0 |  |  |  |  |
| mza | Santa María Zacatepec Mixtec | 7940 | 1.0 | 0.0 |  |  |  | 0.00088 |
| mzh | Wichí Lhamtés Güisnay | 29905 | 0.996 | 0.00014 |  |  |  | 0.00011 |
| mzk | Nigeria Mambila | 7917 | 0.9995 | 0.0 |  |  |  |  |
| mzl | Mazatlán Mixe | 7907 | 1.0 | 0.0 |  |  |  | 0.00011 |
| mzm | Mumuye | 7945 | 1.0 | 0.0 |  |  |  |  |
| mzn | Mazanderani | 30000 | 0.96612 | 0.00019 |  | 0.00295 |  |  |
| mzw | Deg | 7889 | 0.999 | 6e-05 |  |  |  | 0.00077 |
| mzz | Maiadomu | 652 | 1.0 | 0.0 |  |  |  |  |
| nab | Southern Nambikuára | 7609 | 1.0 | 0.0 |  |  |  |  |
| naf | Nabak | 7692 | 0.9995 | 3e-05 |  |  |  | 0.00011 |
| nah | Nahuatl languages | 212 | 0.68293 | 3e-05 |  |  |  | 0.00011 |
| nak | Nakanai | 7892 | 0.9995 | 0.0 |  |  |  |  |
| nan | Min Nan Chinese | 37894 | 1.0 | 0.0 |  |  |  |  |
| nan | Min Nan Chinese | 37894 | 1.0 | 0.0 |  |  | 0.0 | 0.00011 |
| nap | Neapolitan | 10002 | 0.97646 | 0.0 |  |  |  |  |
| naq | Khoekhoe | 106437 | 0.9995 | 0.0 |  |  |  |  |
| nas | Naasioi | 7985 | 1.0 | 0.0 |  |  |  |  |
| nau | Nauru | 11 |  |  |  |  |  |  |
| nav | Navajo | 91313 | 0.999 | 6e-05 |  |  | 0.9916 | 0.00011 |
| naw | Nawuri | 7948 | 1.0 | 0.0 |  |  |  |  |
| nba | Nyemba | 162410 | 0.98861 | 0.00058 |  | 0.00227 |  |  |

Table 19: Performance of GlotLID-M on GlotLID-C test, FLORES-200 and UDHR benchmarks (part 15)

| iso639-3 | Language | \|Sentences\| | GlotLID-C F1↑ | GlotLID-C FPR↓ | FLORES-200 F1↑ | FLORES-200 FPR↓ | UDHR F1↑ | UDHR FPR↓ |
|---|---|---|---|---|---|---|---|---|
| nbc | Chang Naga | 7899 | 0.998 | 6e-05 | | | | |
| nbe | Konyak Naga | 7951 | 0.9995 | 0.0 | | | | |
| nbl | South Ndebele | 238555 | 0.9679 | 0.0008 | | | 0.0 | 0.00055 |
| nbq | Nggem | 3714 | 1.0 | 0.0 | | | | 0.00033 |
| nbu | Rongmei Naga | 30767 | 0.99198 | 0.00017 | | | | |
| nca | Iyo | 7866 | 0.999 | 3e-05 | | | | |
| nch | Central Huasteca Nahuatl | 191991 | 0.97793 | 0.00116 | | | | 0.00668 |
| ncj | Northern Puebla Nahuatl | 192101 | 0.99303 | 0.0003 | | | | 0.00033 |
| ncl | Michoacán Nahuatl | 7934 | 0.998 | 6e-05 | | | | |
| nct | Chothe Naga | 7936 | 0.998 | 3e-05 | | | | |
| ncu | Chumburung | 7654 | 1.0 | 0.0 | | | | |
| ncx | Central Puebla Nahuatl | 161081 | 0.97718 | 0.00085 | | | | 0.00011 |
| ndc | Ndau | 197046 | 0.99601 | 0.00019 | | 6e-05 | | 0.00011 |
| nde | North Ndebele | 234610 | 0.96552 | 0.00096 | | 0.00273 | | 0.06624 |
| ndh | Ndali | 743 | 0.96257 | 3e-05 | | 0.00068 | | 0.00011 |
| ndi | Samba Leko | 7924 | 1.0 | 0.0 | | | | |
| ndj | Ndamba | 7955 | 0.9965 | 0.00011 | | 6e-05 | | 0.00022 |
| ndo | Ndonga | 481932 | 0.92503 | 0.0005 | | 0.00136 | 0.86765 | 0.00197 |
| ndp | Ndo | 7946 | 1.0 | 0.0 | | | | |
| nds | Low German | 128598 | 0.98218 | 0.00077 | | | 1.0 | 0.0 |
| ndz | Ndogo | 7937 | 0.9995 | 0.0 | | | | |
| neb | Toura (Côte d'Ivoire) | 7864 | 1.0 | 0.0 | | | | |
| nep | Nepali (macrolanguage) | 137937 | 0.99649 | 0.0 | | | | |
| new | Newari | 30017 | 0.97093 | 0.00025 | | 0.00051 | | 0.00011 |
| nfa | Dhao | 7649 | 1.0 | 0.0 | | | | |
| nfr | Nafaanra | 7930 | 0.9985 | 0.0 | | | | |
| ngb | Northern Ngbandi | 3998 | 0.99164 | 3e-05 | | | | 0.00011 |
| ngc | Ngombe (Democratic Republic of Congo) | 7941 | 0.99601 | 0.00017 | | | | |
| ngl | Lomwe | 177993 | 0.97725 | 0.00094 | | 6e-05 | | 0.00033 |
| ngp | Ngulu | 7956 | 0.97885 | 0.00039 | | 0.00011 | | |
| ngt | Kriang | 19 | | | | | | |
| ngu | Guerrero Nahuatl | 118903 | 0.9935 | 0.00017 | | | | 0.00011 |
| nhd | Chiripá | 15822 | 0.83343 | 0.00033 | | | | |
| nhe | Eastern Huasteca Nahuatl | 7941 | 0.94 | 0.00165 | | | | |
| nhg | Tetelcingo Nahuatl | 7906 | 0.998 | 0.0 | | | | |
| nhi | Zacatlán-Ahuacatlán-Tepetzintla Nahuatl | 7911 | 0.99749 | 0.0 | | | | |
| nhk | Isthmus-Cosoleacaque Nahuatl | 4466 | 0.8986 | 0.00014 | | | | 0.00055 |
| nho | Takuu | 7908 | 0.9995 | 3e-05 | | | | |
| nhr | Naro | 7906 | 1.0 | 0.0 | | | | |
| nhu | Noone | 7919 | 0.999 | 3e-05 | | | | |
| nhw | Western Huasteca Nahuatl | 7942 | 0.93259 | 0.00146 | | | | |
| nhx | Isthmus-Mecayapan Nahuatl | 9954 | 0.999 | 6e-05 | | | | |
| nhy | Northern Oaxaca Nahuatl | 7931 | 0.996 | 0.00011 | | | | |
| nia | Nias | 205435 | 0.9985 | 6e-05 | | | | 0.00022 |
| nif | Nek | 3713 | 1.0 | 0.0 | | | | |
| nii | Nii | 7794 | 1.0 | 0.0 | | | | |
| nij | Ngaju | 7879 | 0.99399 | 0.00011 | | | | |
| nim | Nilamba | 7948 | 0.998 | 3e-05 | | 6e-05 | | |
| nin | Ninzo | 7915 | 0.9995 | 3e-05 | | | | |
| niq | Nandi | 35203 | 0.9975 | 8e-05 | | | | 0.00427 |
| niu | Niuean | 565376 | 0.9975 | 6e-05 | | | 1.0 | 0.0 |
| niv | Gilyak | 20 | | | | | | |
| niy | Ngiti | 7950 | 1.0 | 0.0 | | | | |
| njb | Nocte Naga | 7888 | 0.9995 | 3e-05 | | | | |
| njm | Angami Naga | 30987 | 0.99699 | 0.0 | | | | |
| njn | Liangmai Naga | 30075 | 0.99502 | 0.00025 | | | | |
| njo | Ao Naga | 30939 | 0.997 | 6e-05 | | | 0.95312 | 0.0 |
| njz | Nyishi | 7935 | 0.9995 | 0.0 | | | | 0.00142 |
| nka | Nkoya | 73 | 0.36364 | 0.0 | | | | |
| nki | Thangal Naga | 7955 | 0.98123 | 0.00011 | | | | 0.00055 |
| nko | Nkonya | 7825 | 0.9995 | 3e-05 | | | | |
| nla | Ngombale | 312 | 0.92105 | 3e-05 | | | | 0.00022 |
| nlc | Nalca | 7873 | 1.0 | 0.0 | | | | |
| nld | Dutch | 4208335 | 0.76911 | 0.01612 | 0.99803 | 0.00023 | 0.70238 | 0.00547 |
| nlv | Orizaba Nahuatl | 14 | | | | | | |
| nma | Maram Naga | 7934 | 1.0 | 0.0 | | | | |
| nmf | Tangkhul Naga (India) | 30939 | 0.996 | 8e-05 | | 6e-05 | | |
| nmh | Monsang Naga | 7951 | 0.9985 | 0.0 | | | | |
| nmo | Moyon Naga | 7950 | 0.9985 | 6e-05 | | | | 0.00044 |
| nmw | Nimoa | 1220 | 1.0 | 0.0 | | | | |
| nmz | Nawdm | 7874 | 0.999 | 0.0 | | | | 0.00033 |
| nnb | Nande | 80966 | 0.98394 | 0.00033 | | 0.00386 | | 0.00109 |
| nng | Maring Naga | 7955 | 1.0 | 0.0 | | | | |
| nnh | Ngiemboon | 7773 | 1.0 | 0.0 | | | | |
| nnl | Northern Rengma Naga | 30056 | 0.9925 | 0.00022 | | | | |

Table 20: Performance of GlotLID-M on GlotLID-C test, FLORES-200 and UDHR benchmarks (part 16)

| iso639-3 | Language | \|Sentences\| | GlotLID-C | | FLORES-200 | | UDHR | |
|---|---|---|---|---|---|---|---|---|
| | | | F1↑ | FPR↓ | F1↑ | FPR↓ | F1↑ | FPR↓ |
| nno | Norwegian Nynorsk | 463877 | 0.98089 | 0.00036 | 0.98507 | 0.00045 | 0.95868 | 0.00033 |
| nnp | Wancho Naga | 7948 | 1.0 | 0.0 | | | | 0.00011 |
| nnq | Ngindo | 7940 | 0.99649 | 6e-05 | | | | |
| nnw | Southern Nuni | 7874 | 1.0 | 0.0 | | | | |
| noa | Woun Meu | 15760 | 0.9995 | 0.0 | | | | 0.00011 |
| nob | Norwegian Bokmål | 2890508 | 0.96898 | 0.00129 | 0.98185 | 0.00148 | 0.98462 | 0.00022 |
| nog | Nogai | 9743 | 0.99649 | 3e-05 | | | | |
| non | Old Norse | 21 | | | | | | |
| nop | Numanggang | 12584 | 0.9985 | 6e-05 | | | | |
| nor | Norwegian | 1000000 | 0.997 | 6e-05 | | | | |
| not | Nomatsiguenga | 7819 | 0.998 | 3e-05 | | | 0.97391 | 0.0 |
| nou | Ewage-Notu | 7758 | 0.9995 | 0.0 | | | | 0.00033 |
| nov | Novial | 430 | 0.71579 | 0.00011 | | | | |
| nph | Phom Naga | 7947 | 0.999 | 0.0 | | | | |
| npi | Nepali (individual language) | 99103 | 0.98559 | 0.00058 | 0.99104 | 0.00011 | 0.98214 | 0.0 |
| npl | Southeastern Puebla Nahuatl | 14454 | 0.99601 | 0.00017 | | | | |
| npo | Pochuri Naga | 7954 | 0.999 | 3e-05 | | | | |
| npy | Napu | 7750 | 0.9985 | 3e-05 | | | | |
| nre | Southern Rengma Naga | 30149 | 0.998 | 8e-05 | | | | |
| nrf | Jèrriais | 1066 | 0.98339 | 0.0 | | | | 0.00022 |
| nri | Chokri Naga | 7954 | 0.99498 | 6e-05 | | | | |
| nsa | Sangtam Naga | 7931 | 1.0 | 0.0 | | | | |
| nse | Nsenga | 106988 | 0.99047 | 0.00017 | | 0.0004 | | 0.00011 |
| nsm | Sumi Naga | 30168 | 0.9965 | 0.00011 | | | | |
| nsn | Nehan | 7887 | 0.999 | 3e-05 | | | | |
| nso | Pedi | 2010451 | 0.99253 | 0.00033 | 0.99704 | 0.00028 | 0.86957 | 0.00197 |
| nss | Nali | 2204 | 0.99849 | 0.0 | | | | |
| nst | Tase Naga | 30918 | 0.9975 | 3e-05 | | | | |
| nsu | Sierra Negra Nahuatl | 7903 | 0.99599 | 3e-05 | | | | |
| ntp | Northern Tepehuan | 7753 | 1.0 | 0.0 | | | | |
| ntr | Delo | 7896 | 0.9995 | 0.0 | | | | 0.00088 |
| nus | Nuer | 16408 | 0.9985 | 0.0 | 0.99951 | 0.0 | | 0.00011 |
| nuy | Nunggubuyu | 8201 | 0.998 | 3e-05 | | | | |
| nvm | Namiae | 7255 | 1.0 | 0.0 | | | | |
| nwb | Nyabwa | 7725 | 0.9985 | 0.0 | | | | |
| nwi | Southwest Tanna | 7454 | 0.9995 | 0.0 | | | | |
| nwx | Middle Newar | 11272 | 0.99649 | 6e-05 | | | | |
| nxd | Ngando (Democratic Republic of Congo) | 7948 | 0.9975 | 6e-05 | | | | |
| nya | Nyanja | 2582911 | 0.93803 | 0.0036 | 0.99753 | 0.00023 | 0.96414 | 0.00099 |
| nyf | Giryama | 8144 | 0.98947 | 0.00022 | | 6e-05 | | |
| nyk | Nyaneka | 297246 | 0.98166 | 0.00074 | | 0.0046 | | 0.00066 |
| nyn | Nyankole | 236252 | 0.98166 | 0.00074 | | 6e-05 | 0.85938 | 0.0 |
| nyo | Nyoro | 7946 | 0.98286 | 0.00025 | | | | |
| nyu | Nyungwe | 176938 | 0.99052 | 0.00033 | | | | |
| nyy | Nyakyusa-Ngonde | 110073 | 0.99102 | 0.0003 | | 0.00034 | | 0.00055 |
| nzb | Njebi | 70 | 0.6 | | | 6e-05 | | 0.00011 |
| nzi | Nzima | 384511 | 0.9985 | 8e-05 | | | 1.0 | 0.0 |
| nzm | Zeme Naga | 30978 | 0.9975 | 3e-05 | | | | |
| oar | Old Aramaic (up to 700 BCE) | 22 | | | | | | |
| obo | Obo Manobo | 7651 | 0.9975 | 3e-05 | | | | |
| oci | Occitan (post 1500) | 135269 | 0.98549 | 0.00039 | 0.99951 | 6e-05 | 0.41101 | 0.0 |
| ofs | Old Frisian | 13 | | | | | | |
| ogo | Khana | 61553 | 0.999 | 6e-05 | | | | |
| ojb | Northwestern Ojibwa | 36949 | 0.70851 | 0.01862 | | | 0.81481 | 0.0 |
| oji | Ojibwa | 7989 | 0.999 | 0.0 | | | | |
| ojs | Severn Ojibwa | 7937 | | | | | | 0.00066 |
| oke | Okpe (Southwestern Edo) | 64083 | 0.9843 | 8e-05 | | | | 0.00011 |
| okv | Orokaiva | 7780 | 1.0 | 0.0 | | | | 0.00011 |
| old | Mochi | 7955 | 0.98808 | 0.00052 | | 0.00023 | | 0.00044 |
| omw | South Tairora | 9414 | 0.9995 | 0.0 | | | | |
| ong | Olo | 7881 | 0.9995 | 3e-05 | | | | |
| ons | Ono | 15076 | 1.0 | 0.0 | | | | 0.00011 |
| ood | Tohono O'odham | 7563 | 0.99699 | 0.0 | | 6e-05 | | |
| opm | Oksapmin | 7657 | 0.999 | 0.0 | | | | |
| ori | Oriya (macrolanguage) | 100397 | 0.9995 | 0.0 | | | | |
| orm | Oromo | 489160 | 0.9995 | 3e-05 | | 6e-05 | | |
| orv | Old Russian | 1307 | 0.94602 | 3e-05 | | | | |
| ory | Odia | 122527 | 0.9995 | 3e-05 | 1.0 | 0.0 | | |
| osp | Old Spanish | 23 | | | | | | |
| oss | Ossetian | 630164 | 0.99502 | 0.00028 | | | 0.5 | 0.00679 |
| ota | Ottoman Turkish (1500-1928) | 2287 | 0.81188 | 8e-05 | | 0.00267 | | 0.00011 |
| ote | Mezquital Otomi | 58835 | 0.92369 | 0.00237 | | | 0.0 | 0.00372 |
| otm | Eastern Highland Otomi | 7870 | 1.0 | 0.0 | | | | |
| otn | Tenango Otomi | 7870 | 1.0 | 0.0 | | | | 0.00011 |
| otq | Querétaro Otomi | 7943 | 1.0 | 0.0 | | | | |

Table 21: Performance of GlotLID-M on GlotLID-C test, FLORES-200 and UDHR benchmarks (part 17)

| iso639-3 | Language | \|Sentences\| | GlotLID-C | | FLORES-200 | | UDHR | |
|---|---|---|---|---|---|---|---|---|
| | | | F1↑ | FPR↓ | F1↑ | FPR↓ | F1↑ | FPR↓ |
| ots | Estado de México Otomi | 13242 | 0.95737 | 0.00041 | | | | 0.0046 |
| otw | Ottawa | 108 | 1.0 | 0.0 | | | | 0.00044 |
| oym | Wayampi | 7693 | 0.9995 | 3e-05 | | | | |
| ozm | Koonzime | 7891 | 1.0 | 0.0 | | | | |
| pab | Parecís | 7936 | 0.9995 | 3e-05 | | | | |
| pad | Paumarí | 7771 | 1.0 | 0.0 | | | | |
| pag | Pangasinan | 1255155 | 0.99452 | 0.00025 | 0.99852 | 0.0 | | 0.00033 |
| pah | Tenharim | 7699 | 0.9995 | 0.0 | | | | |
| pam | Pampanga | 19422 | 0.98532 | 6e-05 | | | 1.0 | 0.0 |
| pan | Panjabi | 722260 | 1.0 | 0.0 | 1.0 | 0.0 | 1.0 | 0.0 |
| pao | Northern Paiute | 7028 | 0.999 | 6e-05 | | | | |
| pap | Papiamento | 1601687 | 0.97782 | 0.00102 | 0.99069 | 0.00102 | 0.79195 | 0.00339 |
| pau | Palauan | 181209 | 0.99751 | 0.00014 | | | 0.97436 | 0.0 |
| pbb | Páez | 12673 | 0.81626 | 0.00333 | | | 0.7125 | 0.00482 |
| pbc | Patamona | 7939 | 0.9995 | 0.0 | | | | |
| pbi | Parkwa | 7849 | 1.0 | 0.0 | | | | 0.00011 |
| pbl | Mak (Nigeria) | 7924 | 0.38695 | 0.00242 | | | | 0.00011 |
| pbt | Southern Pashto | 63256 | 0.96663 | 0.00061 | 0.81486 | 0.00051 | | 0.00109 |
| pcd | Picard | 1348 | 0.89552 | 0.0 | | | | |
| pck | Paite Chin | 30968 | 0.97284 | 0.0011 | | | | 0.00033 |
| pcm | Nigerian Pidgin | 8364 | 0.97864 | 0.00011 | | | 0.71739 | 0.0 |
| pdc | Pennsylvania German | 11954 | 0.94379 | 0.00066 | | | | |
| pdt | Plautdietsch | 152305 | 0.998 | 8e-05 | | | | |
| pem | Phende | 9968 | 0.99649 | 3e-05 | | 0.00034 | | 0.00044 |
| pes | Iranian Persian | 2814370 | 0.80032 | 0.01356 | 0.57435 | 0.08493 | 0.65922 | 0.00668 |
| pfe | Pere | 10404 | 0.9995 | 3e-05 | | | | 0.00044 |
| pfl | Pfaelzisch | 10003 | 0.96516 | 0.00028 | | | | 0.00011 |
| phm | Phimbi | 45602 | 0.98943 | 0.00011 | | 0.00017 | | 0.00011 |
| phn | Phoenician | 15 | | | | | | |
| pib | Yine | 7937 | 0.999 | 0.0 | | | | |
| pid | Piaroa | 7255 | 0.47033 | 0.00085 | | | | |
| pio | Piapoco | 7655 | 1.0 | 0.0 | | | | |
| pir | Piratapuyo | 7740 | 0.9995 | 3e-05 | | | | |
| pis | Pijin | 703022 | 0.998 | 0.00011 | | | 0.9916 | 0.0 |
| pjt | Pitjantjatjara | 10949 | 1.0 | 0.0 | | | | 0.00547 |
| pkb | Pokomo | 7824 | 0.9985 | 6e-05 | | | | 0.00022 |
| plg | Pilagá | 8729 | 1.0 | 0.0 | | | | 0.00011 |
| pli | Pali | 2 | | | | | | |
| pls | San Marcos Tlacoyalco Popoloca | 23439 | 0.86827 | 0.00286 | | | | 0.00011 |
| plt | Plateau Malagasy | 202954 | 0.99552 | 0.00022 | 0.99852 | 0.0 | 0.98182 | 0.0 |
| plu | Palikúr | 8749 | 0.999 | 0.0 | | 0.00011 | | 0.0012 |
| plw | Brooke's Point Palawano | 7940 | 0.9995 | 3e-05 | | | | |
| pma | Paama | 15067 | 0.999 | 6e-05 | | | | |
| pmf | Pamona | 7956 | 0.9985 | 0.0 | | | | |
| pms | Piemontese | 30824 | 0.98993 | 8e-05 | | | | 0.00033 |
| pmx | Poumei Naga | 30182 | 0.99549 | 6e-05 | | | | |
| pnb | Western Panjabi | 300035 | 0.9762 | 0.0003 | | 0.00102 | 0.65969 | 0.00668 |
| pne | Western Penan | 7928 | 0.999 | 0.0 | | | | |
| poe | San Juan Atzingo Popoloca | 19275 | 0.9995 | 0.0 | | | | |
| poh | Poqomchi' | 41239 | 0.998 | 0.0 | | | | 0.00011 |
| poi | Highland Popoluca | 17342 | 0.7267 | 0.00613 | | | | 0.00011 |
| pol | Polish | 4592867 | 0.9621 | 0.00187 | 0.9907 | 0.00108 | 0.7362 | 0.00471 |
| pon | Pohnpeian | 431877 | 0.997 | 0.00011 | | | 1.0 | 0.0 |
| por | Portuguese | 5403043 | 0.82411 | 0.01166 | 0.99655 | 0.0004 | 0.84806 | 0.00471 |
| pot | Potawatomi | 2078 | 1.0 | 0.0 | | | | |
| pov | Upper Guinea Crioulo | 28501 | 0.99147 | 0.00014 | | 0.00233 | 0.96552 | 0.0 |
| poy | Pogolo | 7955 | 0.9995 | 8e-05 | | | | |
| ppk | Uma | 7760 | 0.999 | 3e-05 | | | | |
| ppl | Pipil | 59 | 0.4 | 0.0 | | | 0.42105 | 0.0 |
| ppo | Folopa | 7679 | 1.0 | 0.0 | | | | |
| pps | San Luís Temalacayuca Popoloca | 7950 | 0.9995 | 3e-05 | | | | 0.00055 |
| prf | Paranan | 7865 | 0.9985 | 0.0 | | | | |
| prg | Prussian | 1088 | 0.96067 | 0.0 | | | | |
| pri | Paicî | 6986 | 1.0 | 0.0 | | | | |
| prk | Parauk | 8909 | 0.63062 | 0.0014 | | | | |
| prs | Dari | 93926 | 0.80441 | 0.00083 | 0.14688 | 0.0192 | 0.0 | 0.00547 |
| pse | Central Malay | 7905 | 0.9985 | 0.0 | | | | |
| ptp | Patep | 7918 | 0.999 | 0.0 | | | | |
| ptu | Bambam | 7808 | 0.99649 | 6e-05 | | | | 0.00077 |
| pua | Western Highland Purepecha | 7950 | 0.52827 | 0.00077 | | | | |
| pus | Pushto | 30046 | 0.88156 | 3e-05 | | | | |
| pwg | Gapapaiwa | 7914 | 0.99551 | 0.00019 | | | | |
| pww | Pwo Northern Karen | 30880 | 1.0 | 0.0 | | | | |
| qub | Huallaga Huánuco Quechua | 64459 | 0.81356 | 0.00715 | | 6e-05 | | 0.02704 |
| quc | K'iche' | 173500 | 0.99402 | 0.00025 | | | 0.95575 | 0.00011 |

Table 22: Performance of GlotLID-M on GlotLID-C test, FLORES-200 and UDHR benchmarks (part 18)

| iso639-3 | Language | \|Sentences\| | GlotLID-C F1↑ | GlotLID-C FPR↓ | FLORES-200 F1↑ | FLORES-200 FPR↓ | UDHR F1↑ | UDHR FPR↓ |
|---|---|---|---|---|---|---|---|---|
| que | Quechua | 694846 | 0.97263 | 0.0014 | | | 0.9916 | 0.0 |
| quf | Lambayeque Quechua | 8072 | 0.98582 | 3e-05 | | 6e-05 | | 0.00394 |
| qug | Chimborazo Highland Quichua | 160594 | 0.84767 | 0.00787 | | | 0.0375 | 0.0 |
| quh | South Bolivian Quechua | 46618 | 0.81818 | 0.00861 | | 0.00028 | 0.92174 | 0.00044 |
| qul | North Bolivian Quechua | 15852 | 0.77368 | 0.00223 | | 6e-05 | | 0.00011 |
| qup | Southern Pastaza Quechua | 7700 | 0.99701 | 0.00014 | | | | 0.0093 |
| qus | Santiago del Estero Quichua | 6856 | 0.99649 | 0.0 | | | | 0.00208 |
| quw | Tena Lowland Quichua | 33203 | 0.67937 | 0.01062 | | | | 0.00055 |
| quy | Ayacucho Quechua | 457040 | 0.97855 | 0.00066 | 0.75904 | 0.0 | 0.67403 | 0.00646 |
| quz | Cusco Quechua | 347402 | 0.9414 | 0.00281 | | 0.02176 | 0.54822 | 0.00898 |
| qva | Ambo-Pasco Quechua | 7824 | 0.98297 | 0.00041 | | | 0.0 | 0.00011 |
| qvc | Cajamarca Quechua | 7799 | 0.998 | 6e-05 | | 6e-05 | 0.92035 | 0.00033 |
| qve | Eastern Apurímac Quechua | 7924 | 0.98167 | 0.0 | | | | |
| qvh | Huamalíes-Dos de Mayo Huánuco Quechua | 7304 | 0.98799 | 0.0003 | | | 0.0 | 0.00022 |
| qvi | Imbabura Highland Quichua | 227687 | 0.97923 | 0.00088 | | | | 0.00055 |
| qvm | Margos-Yarowilca-Lauricocha Quechua | 7299 | 0.9824 | 0.00033 | | | 0.15385 | 0.0 |
| qvn | North Junín Quechua | 7948 | 0.9945 | 0.00017 | | | 0.31884 | 0.0 |
| qvo | Napo Lowland Quechua | 7853 | 0.99198 | 0.00017 | | | | 0.00011 |
| qvs | San Martín Quechua | 7872 | 0.99649 | 3e-05 | | | | 0.00252 |
| qvw | Huaylla Wanca Quechua | 16232 | 0.77036 | 0.00239 | | | | 0.00055 |
| qvz | Northern Pastaza Quichua | 8730 | 0.94858 | 6e-05 | | | | 0.00011 |
| qwh | Huaylas Ancash Quechua | 7834 | 0.997 | 6e-05 | | | 0.48276 | 0.00066 |
| qxh | Panao Huánuco Quechua | 7727 | 0.99247 | 0.00011 | | | | |
| qxl | Salasaca Highland Quichua | 7876 | 0.9995 | 3e-05 | | | | |
| qxn | Northern Conchucos Ancash Quechua | 7937 | 0.98087 | 0.00033 | | | 0.03333 | 0.00011 |
| qxo | Southern Conchucos Ancash Quechua | 7421 | 0.97364 | 0.00088 | | | | |
| qxq | Qashqa'i | 14 | | | | | | |
| qxr | Cañar Highland Quichua | 9639 | 0.9646 | 0.00025 | | | | |
| qya | Quenya | 144 | 0.28571 | 3e-05 | | | | |
| rad | Rade | 30874 | 0.9975 | 6e-05 | | | | |
| rai | Ramoaaina | 7828 | 0.9995 | 0.0 | | | | |
| rap | Rapanui | 16603 | 0.57556 | 0.00776 | | | | 0.00011 |
| rar | Rarotongan | 920894 | 0.98958 | 0.0005 | | | 0.99174 | 0.00011 |
| rcf | Réunion Creole French | 13290 | 0.98891 | 8e-05 | | 0.00045 | | 0.00022 |
| rhg | Rohingya | 3850 | 0.98712 | 3e-05 | | 0.00028 | | 0.00142 |
| ria | Riang (India) | 7947 | 0.9985 | 0.0 | | | | |
| rif | Tarifit | 227 | 0.43137 | 6e-05 | | | | |
| rim | Nyaturu | 7954 | 0.9995 | 3e-05 | | | | |
| rkb | Rikbaktsa | 7766 | 0.999 | 3e-05 | | | | |
| rmc | Carpathian Romani | 8938 | 0.99649 | 6e-05 | | | | |
| rme | Angloromani | 168 | 0.76923 | 0.0 | | | | |
| rml | Baltic Romani | 4828 | 0.73838 | 0.00028 | | 6e-05 | | 0.00077 |
| rmn | Balkan Romani | 338459 | 0.997 | 8e-05 | | 6e-05 | 0.86636 | 0.00011 |
| rmo | Sinte Romani | 11235 | 0.99749 | 0.0 | | 0.00011 | | |
| rmq | Caló | 2273 | 0.99541 | 0.0 | | | | |
| rmy | Vlax Romani | 96254 | 0.91895 | 0.00184 | | 6e-05 | | 0.00394 |
| rnd | Ruund | 29279 | 0.999 | 3e-05 | | 0.00011 | | |
| rng | Ronga | 77778 | 0.9853 | 3e-05 | | | | 0.00044 |
| rnl | Ranglong | 10406 | 0.994 | 0.00017 | | | | |
| roh | Romansh | 30149 | 0.9945 | 0.00017 | | | 0.99268 | 0.0 |
| rom | Romany | 876 | 0.44602 | 0.00828 | | | | |
| ron | Romanian | 1542662 | 0.98327 | 0.00091 | 0.99951 | 0.0 | 0.80992 | 0.00493 |
| roo | Rotokas | 7890 | 1.0 | 0.0 | | | | |
| rop | Kriol | 29167 | 0.998 | 6e-05 | | | | |
| rro | Waima | 7946 | 0.999 | 3e-05 | | | | 0.00011 |
| rtm | Rotuman | 11052 | 0.9985 | 0.0 | | | | 0.00011 |
| rub | Gungu | 7912 | 0.999 | 0.0 | | | | |
| rue | Rusyn | 10117 | 0.95084 | 8e-05 | | | | |
| ruf | Luguru | 7951 | 0.99649 | 6e-05 | | 6e-05 | | |
| run | Rundi | 1361196 | 0.96652 | 0.00094 | 0.92541 | 0.00881 | 0.87591 | 0.00186 |
| rup | Macedo-Romanian | 2219 | 0.99573 | 3e-05 | | | 0.125 | 0.0 |
| rus | Russian | 9074266 | 0.65357 | 0.02904 | 0.99901 | 6e-05 | 0.43321 | 0.01718 |
| rwo | Rawa | 15237 | 0.9995 | 3e-05 | | | | |
| ryu | Central Okinawan | 42 | | | | | | |
| sab | Buglere | 7507 | 0.999 | 0.0 | | | | |
| sag | Sango | 1017526 | 0.99552 | 0.00025 | 0.99901 | 0.0 | 0.81553 | 0.0 |
| sah | Yakut | 115824 | 0.98601 | 0.00041 | | | 0.53881 | 0.01106 |
| san | Sanskrit | 147128 | 0.97656 | 0.00072 | 0.99104 | 6e-05 | 0.66667 | 0.0 |
| sas | Sasak | 7890 | 0.9985 | 3e-05 | | | | |
| sat | Santali | 21927 | 1.0 | 0.0 | 1.0 | 0.0 | | |
| sba | Ngambay | 30900 | 0.9995 | 0.0 | | | | |
| sbd | Southern Samo | 7909 | 0.9975 | 3e-05 | | | | 0.00011 |
| sbe | Saliba | 6215 | 0.998 | 6e-05 | | | | |
| sbl | Botolan Sambal | 7848 | 1.0 | 0.0 | | | | |
| sbs | Subiya | 328 | 0.83871 | 0.0 | | | | |

Table 23: Performance of GlotLID-M on GlotLID-C test, FLORES-200 and UDHR benchmarks (part 19)

| iso639-3 | Language | \|Sentences\| | GlotLID-C F1↑ | GlotLID-C FPR↓ | FLORES-200 F1↑ | FLORES-200 FPR↓ | UDHR F1↑ | UDHR FPR↓ |
|---|---|---|---|---|---|---|---|---|
| sby | Soli | 208 | 0.95833 | 0.0 | | | | 0.00022 |
| sck | Sadri | 29 | | | | | | |
| scn | Sicilian | 62549 | 0.96888 | 0.00074 | 0.99802 | 6e-05 | | 0.00263 |
| sco | Scots | 101211 | 0.97563 | 0.00025 | | 0.00028 | 0.92683 | 0.00044 |
| sda | Toraja-Sa'dan | 8864 | 0.99198 | 0.00017 | | | | 0.00011 |
| sdh | Southern Kurdish | 1048 | 0.93968 | 0.0 | | | | |
| sdo | Bukar-Sadung Bidayuh | 787 | 0.98851 | 3e-05 | | | | |
| seh | Sena | 301781 | 0.98462 | 0.00063 | | 0.00034 | | |
| ses | Koyraboro Senni Songhai | 12790 | 0.9955 | 0.00011 | | | | |
| sey | Secoya | 7912 | 1.0 | 0.0 | | | 0.02985 | 0.00044 |
| sfw | Sehwi | 10131 | 0.99548 | 0.0 | | | | 0.00033 |
| sgb | Mag-antsi Ayta | 7857 | 0.998 | 8e-05 | | | | |
| sgh | Shughni | 717 | 1.0 | 0.0 | | | | 0.00077 |
| sgs | Samogitian | 10047 | 0.98889 | 3e-05 | | 6e-05 | | |
| sgw | Sebat Bet Gurage | 7909 | 0.99749 | 0.0 | | | | |
| sgz | Sursurunga | 7745 | 0.9995 | 3e-05 | | | | |
| shi | Tachelhit | 10669 | 0.98226 | 0.00011 | | 0.00011 | | |
| shk | Shilluk | 7907 | 1.0 | 0.0 | | | 0.88889 | 0.0 |
| shn | Shan | 24569 | 1.0 | 0.0 | 1.0 | 0.0 | 0.99145 | 0.0 |
| shp | Shipibo-Conibo | 9806 | 0.9995 | 3e-05 | | | 0.27397 | 0.0 |
| shr | Shi | 14876 | 0.98656 | 0.0005 | | 0.00011 | | |
| shs | Shuswap | 95 | 0.8125 | 0.0 | | | | |
| shu | Chadian Arabic | 7923 | 0.998 | 6e-05 | | 0.00835 | | 0.0012 |
| shy | Tachawit | 247 | 0.27907 | 0.0 | | | | |
| sid | Sidamo | 130089 | 0.999 | 6e-05 | | | 0.90625 | 0.00131 |
| sig | Paasaal | 7893 | 0.998 | 3e-05 | | | | 0.00011 |
| sil | Tumulung Sisaala | 7868 | 0.99649 | 0.0 | | | | |
| sim | Mende (Papua New Guinea) | 7790 | 0.9995 | 3e-05 | | | | 0.00033 |
| sin | Sinhala | 554048 | 1.0 | 0.0 | 1.0 | 0.0 | 1.0 | 0.0 |
| sja | Epena | 7870 | 0.9995 | 0.0 | | | | |
| sjn | Sindarin | 92 | 0.33333 | 3e-05 | | | | |
| skg | Sakalava Malagasy | 56311 | 0.9955 | 0.00011 | | 0.00011 | | 0.00022 |
| skr | Saraiki | 134 | 0.51429 | 3e-05 | | | | |
| sld | Sissala | 7906 | 0.999 | 6e-05 | | | | 0.0035 |
| slk | Slovak | 3544374 | 0.97476 | 0.00099 | 0.99852 | 6e-05 | 0.86957 | 0.00197 |
| sll | Salt-Yui | 7869 | 0.999 | 3e-05 | | | | |
| slv | Slovenian | 4072739 | 0.96881 | 0.0016 | 0.99459 | 0.00062 | 0.88889 | 0.00164 |
| sma | Southern Sami | 59 | 0.83333 | 0.0 | | | | |
| sme | Northern Sami | 18205 | 0.99448 | 6e-05 | | 6e-05 | 0.96667 | 0.00022 |
| smk | Bolinao | 7859 | 0.9985 | 3e-05 | | | | |
| sml | Central Sama | 7891 | 0.9995 | 0.0 | | 6e-05 | | |
| smo | Samoan | 1640628 | 0.99352 | 0.00025 | 0.99603 | 0.0 | 1.0 | 0.0 |
| smt | Simte | 7953 | 0.99247 | 8e-05 | | | | |
| sna | Shona | 2150482 | 0.98521 | 0.0008 | 0.99901 | 0.00011 | 0.93846 | 0.00088 |
| snc | Sinaugoro | 7926 | 1.0 | 0.0 | | | | |
| snd | Sindhi | 132171 | 0.9985 | 0.0 | 0.99362 | 0.00074 | | |
| snf | Noon | 7907 | 0.99599 | 3e-05 | | | | 0.0012 |
| snn | Siona | 7892 | 1.0 | 0.0 | | | 0.60773 | 0.0 |
| snp | Siane | 15669 | 0.999 | 0.0 | | | | 0.00022 |
| snw | Selee | 7890 | 1.0 | 0.0 | | | | |
| sny | Saniyo-Hiyewe | 7848 | 1.0 | 0.0 | | | | |
| soe | Songomeno | 1127 | 0.96988 | 3e-05 | | | | 0.00011 |
| som | Somali | 227769 | 0.99649 | 6e-05 | 0.96657 | 0.00398 | 0.75817 | 0.00405 |
| sop | Songe | 208326 | 0.9955 | 0.00011 | | 0.00011 | | 0.00055 |
| soq | Kanasi | 10512 | 0.9985 | 3e-05 | | | | 0.00022 |
| sot | Southern Sotho | 2131930 | 0.99305 | 0.00039 | 1.0 | 0.0 | 0.98333 | 0.00011 |
| soy | Miyobe | 7920 | 1.0 | 0.0 | | | | 0.00099 |
| spa | Spanish | 2583672 | 0.50519 | 0.05363 | 0.99508 | 0.00057 | 0.71681 | 0.00679 |
| spl | Selepet | 7031 | 0.9995 | 3e-05 | | | | |
| spm | Akukem | 1622 | 0.99785 | 0.0 | | | | |
| spp | Supyire Senoufo | 7847 | 0.9985 | 6e-05 | | | | |
| sps | Saposa | 8166 | 0.999 | 0.0 | | | | |
| spy | Sabaot | 15366 | 1.0 | 0.0 | | | | |
| sqi | Albanian | 326340 | 0.999 | 6e-05 | | | | |
| srd | Sardinian | 53845 | 0.93574 | 6e-05 | 0.99951 | 0.0 | | |
| sri | Siriano | 7808 | 0.9985 | 0.0 | | | | |
| srm | Saramaccan | 75703 | 0.9975 | 6e-05 | | | | 0.00055 |
| srn | Sranan Tongo | 1166639 | 0.99107 | 0.00047 | | | | |
| srp | Serbian | 1390294 | 0.98142 | 0.00039 | 0.99901 | 0.00011 | 0.5124 | 0.00657 |
| srq | Sirionó | 7814 | 0.9995 | 0.0 | | | | |
| srr | Serer | 3524 | 0.99191 | 0.0 | | | 0.89231 | 0.00131 |
| ssd | Siroi | 10880 | 0.9965 | 0.00011 | | | | |
| ssg | Seimat | 7891 | 0.998 | 3e-05 | | | | |
| ssw | Swati | 426339 | 0.97691 | 0.00052 | 0.99654 | 0.00023 | 0.94891 | 0.00066 |
| ssx | Samberigi | 9408 | 1.0 | 0.0 | | | | 0.00011 |

Table 24: Performance of GlotLID-M on GlotLID-C test, FLORES-200 and UDHR benchmarks (part 20)

| iso639-3 | Language | \|Sentences\| | GlotLID-C | | FLORES-200 | | UDHR | |
|---|---|---|---|---|---|---|---|---|
| | | | **F1↑** | **FPR↓** | **F1↑** | **FPR↓** | **F1↑** | **FPR↓** |
| stn | Owa | 7884 | 0.9985 | 6e-05 | | | | |
| stp | Southeastern Tepehuan | 7918 | 1.0 | 0.0 | | | | 0.00022 |
| stq | Saterfriesisch | 10507 | 0.99147 | 0.00014 | | | | |
| sua | Sulka | 7558 | 0.999 | 3e-05 | | | | |
| suc | Western Subanon | 7896 | 0.999 | 0.0 | | | | |
| sue | Suena | 7882 | 0.998 | 8e-05 | | | | |
| suk | Sukuma | 8083 | 0.99599 | 3e-05 | | 6e-05 | 0.68712 | 0.00514 |
| sun | Sundanese | 109886 | 0.98094 | 0.00044 | 0.99012 | 0.00057 | 0.9697 | 0.00033 |
| sur | Mwaghavul | 7952 | 0.999 | 3e-05 | | | | |
| sus | Susu | 11639 | 0.99449 | 8e-05 | | 6e-05 | 0.92683 | 0.00088 |
| sux | Sumerian | 183 | 0.92754 | 0.0 | | | | |
| suz | Sunwar | 30868 | 0.9985 | 0.0 | | | | |
| swa | Swahili (macrolanguage) | 100000 | 0.998 | 8e-05 | | 6e-05 | | |
| swb | Maore Comorian | 1412 | 0.97387 | 0.00011 | | | 0.79137 | 0.00274 |
| swc | Congo Swahili | 452844 | 0.9025 | 0.00454 | | 0.00409 | | 0.00285 |
| swe | Swedish | 3997074 | 0.96724 | 0.00154 | 0.99754 | 0.00028 | 0.86301 | 0.00219 |
| swg | Swabian | 9915 | 0.97342 | 0.00011 | | | | 0.00011 |
| swh | Swahili (individual language) | 370928 | 0.9401 | 0.00267 | 0.94869 | 0.00187 | 0.84956 | 0.00044 |
| swk | Malawi Sena | 7727 | 0.98992 | 6e-05 | | | | |
| swp | Suau | 11467 | 0.997 | 0.00011 | | | | |
| sxb | Suba | 7906 | 0.998 | 0.0 | | | | 0.00011 |
| sxn | Sangir | 51443 | 0.995 | 0.00014 | | | | |
| syb | Central Subanen | 7644 | 0.999 | 3e-05 | | | | |
| syc | Classical Syriac | 7926 | 1.0 | 0.0 | | | | 0.00635 |
| syl | Sylheti | 15 | | | | | | |
| szb | Ngalum | 7940 | 1.0 | 0.0 | | | | |
| szl | Silesian | 57496 | 0.99247 | 0.00011 | 0.99104 | 6e-05 | | |
| tab | Tabassaran | 7851 | 0.999 | 3e-05 | | | | |
| tac | Lowland Tarahumara | 11398 | 0.9985 | 0.0 | | | | |
| tah | Tahitian | 1185188 | 0.99255 | 0.00039 | | | 0.91892 | 0.00011 |
| taj | Eastern Tamang | 7884 | 0.9995 | 0.0 | | | 0.66667 | 0.0 |
| tam | Tamil | 1581134 | 1.0 | 0.0 | 1.0 | 0.0 | 1.0 | 0.0 |
| tap | Taabwa | 216 | 0.875 | 0.0 | | 0.00011 | | |
| taq | Tamasheq | 24410 | 0.90069 | 0.00325 | 0.80642 | 0.02022 | | 0.00438 |
| tar | Central Tarahumara | 25433 | 0.98077 | 0.00019 | | | | 0.00022 |
| tat | Tatar | 372101 | 0.98657 | 0.00052 | 1.0 | 0.0 | 0.65556 | 0.00668 |
| tav | Tatuyo | 7676 | 0.9975 | 8e-05 | | | | |
| taw | Tai | 7683 | 0.9985 | 6e-05 | | | | |
| tbc | Takia | 7836 | 1.0 | 0.0 | | | | |
| tbg | North Tairora | 19510 | 0.999 | 6e-05 | | | | |
| tbk | Calamian Tagbanwa | 7653 | 1.0 | 0.0 | | | | |
| tbl | Tboli | 7806 | 0.9985 | 0.0 | | | | 0.00011 |
| tbo | Tawala | 7895 | 0.99701 | 0.00014 | | | | 0.00011 |
| tby | Tabaru | 7878 | 0.9995 | 3e-05 | | | | |
| tbz | Ditammari | 30712 | 0.9995 | 3e-05 | | | 0.20896 | 0.0 |
| tca | Ticuna | 25611 | 0.81432 | 0.00393 | | | 0.9916 | 0.0 |
| tcc | Datooga | 7953 | 1.0 | 0.0 | | | | |
| tcf | Malinaltepec Me'phaa | 125443 | 0.997 | 0.00011 | | | | 0.00022 |
| tcs | Torres Strait Creole | 12298 | 0.99046 | 0.00014 | | | | |
| tcy | Tulu | 10000 | 0.98219 | 0.0 | | | | |
| tcz | Thado Chin | 44548 | 0.997 | 0.00011 | | | | |
| tdt | Tetun Dili | 450685 | 0.99056 | 0.00044 | | | 0.66292 | 0.00646 |
| tdx | Tandroy-Mahafaly Malagasy | 73631 | 0.9975 | 6e-05 | | 6e-05 | | |
| ted | Tepo Krumen | 7812 | 0.999 | 6e-05 | | | | |
| tee | Huehuetla Tepehua | 7938 | 1.0 | 0.0 | | | | |
| tel | Telugu | 634652 | 0.999 | 6e-05 | 1.0 | 0.0 | 1.0 | 0.0 |
| tem | Timne | 7951 | 1.0 | 0.0 | | | 0.97345 | 0.0 |
| teo | Teso | 36835 | 0.99451 | 0.00019 | | | | 0.00044 |
| ter | Tereno | 8019 | 0.999 | 3e-05 | | | | |
| tet | Tetum | 9162 | 0.99197 | 0.00011 | | | | |
| tew | Tewa (USA) | 4831 | 1.0 | 0.0 | | | | |
| tfr | Teribe | 7534 | 0.9995 | 0.0 | | | | |
| tgk | Tajik | 232287 | 0.99599 | 3e-05 | 1.0 | 0.0 | 0.67429 | 0.00624 |
| tgl | Tagalog | 1391946 | 0.85393 | 0.00897 | 0.99901 | 0.00011 | 0.9403 | 0.00077 |
| tgo | Sudest | 8042 | 0.9995 | 0.0 | | | | |
| tgp | Tangoa | 7927 | 0.999 | 0.0 | | | | |
| tha | Thai | 883065 | 0.99502 | 0.00028 | 1.0 | 0.0 | 1.0 | 0.0 |
| thk | Tharaka | 7890 | 0.9995 | 3e-05 | | 0.00011 | | |
| thv | Tahaggart Tamahaq | 589 | 0.66038 | 0.0 | | 0.00182 | | 0.00011 |
| tif | Tifal | 7682 | 0.9985 | 0.0 | | | | |
| tig | Tigre | 3874 | 0.74834 | 3e-05 | | 0.00125 | | |
| tih | Timugon Murut | 7913 | 0.9995 | 3e-05 | | | | |
| tik | Tikar | 7900 | 0.9995 | 0.0 | | | | |
| tim | Timbe | 7791 | 0.9995 | 0.0 | | | | |
| tir | Tigrinya | 917442 | 0.89658 | 0.00624 | 0.98851 | 0.0 | 1.0 | 0.0 |

Table 25: Performance of GlotLID-M on GlotLID-C test, FLORES-200 and UDHR benchmarks (part 21)

|         |                            |            | GlotLID-C | | FLORES-200 | | UDHR | |
| iso639-3 | Language                   | \|Sentences\| | F1↑ | FPR↓ | F1↑ | FPR↓ | F1↑ | FPR↓ |
|---|---|---|---|---|---|---|---|---|
| tiv | Tiv | 481805 | 0.9965 | 0.00011 | | | 0.98551 | 0.00022 |
| tiy | Tiruray | 7896 | 1.0 | 0.0 | | | | |
| tke | Takwane | 10656 | 0.99498 | 6e-05 | | | | 0.00011 |
| tkl | Tokelau | 629 | 0.91026 | 0.0 | | | | |
| tkr | Tsakhur | 912 | 0.98932 | 0.0 | | | | |
| tku | Upper Necaxa Totonac | 7936 | 0.99303 | 0.0003 | | | | |
| tlb | Tobelo | 7902 | 0.999 | 3e-05 | | | | 0.00011 |
| tlf | Telefol | 7676 | 0.9995 | 3e-05 | | | | |
| tlh | Klingon | 55689 | 0.99498 | 6e-05 | | 6e-05 | | |
| tlj | Talinga-Bwisi | 7885 | 0.999 | 3e-05 | | | | |
| tll | Tetela | 584747 | 0.99304 | 0.00036 | | 0.00017 | | 0.00011 |
| tly | Talysh | 66 | 0.57143 | 0.0 | | | 0.03175 | 0.0 |
| tmd | Haruai | 7725 | 1.0 | 0.0 | | | | |
| tmr | Jewish Babylonian Aramaic | 218 | 0.7 | 0.0 | | | | |
| tmw | Temuan | 5 | | | | | | |
| tna | Tacana | 7705 | 0.999 | 3e-05 | | | | |
| tnc | Tanimuca-Retuarã | 3025 | 1.0 | 0.0 | | | | |
| tnk | Kwamera | 7870 | 1.0 | 0.0 | | | | |
| tnn | North Tanna | 7865 | 0.9975 | 3e-05 | | | | |
| tnp | Whitesands | 7864 | 0.998 | 8e-05 | | | | |
| tob | Toba | 39496 | 0.99599 | 6e-05 | | | 0.97561 | 0.00033 |
| toc | Coyutla Totonac | 7915 | 0.9985 | 6e-05 | | | | 0.00011 |
| tod | Toma | 11025 | 0.996 | 8e-05 | | | | |
| tog | Tonga (Nyasa) | 231197 | 0.99699 | 3e-05 | | 0.00011 | | |
| toh | Gitonga | 107233 | 0.995 | 0.00014 | | | | |
| toi | Tonga (Zambia) | 746307 | 0.98961 | 0.00058 | | 0.00011 | 1.0 | 0.0 |
| toj | Tojolabal | 189077 | 0.98953 | 0.00036 | | 6e-05 | 0.83688 | 0.00022 |
| tok | Toki Pona | 52772 | 1.0 | 0.0 | | | | 0.00044 |
| ton | Tonga (Tonga Islands) | 1234253 | 0.99502 | 0.00025 | | | 1.0 | 0.0 |
| too | Xicotepec De Juárez Totonac | 7940 | 0.99297 | 8e-05 | | | | |
| top | Papantla Totonac | 238408 | 0.99601 | 0.00017 | | | 0.97561 | 0.00033 |
| tos | Highland Totonac | 7906 | 0.998 | 3e-05 | | | | 0.00022 |
| tpa | Taupota | 656 | 0.94792 | 8e-05 | | | | |
| tpi | Tok Pisin | 1846778 | 0.99155 | 0.00039 | 0.99951 | 0.0 | 0.98361 | 0.00011 |
| tpm | Tampulma | 30663 | 0.997 | 8e-05 | | | | 0.00066 |
| tpp | Pisaflores Tepehua | 7926 | 0.9995 | 3e-05 | | 0.00011 | | |
| tpt | Tlachichilco Tepehua | 7927 | 0.9995 | 0.0 | | 6e-05 | | |
| tpw | Tupí | 563 | 0.90566 | 6e-05 | | | | 0.00011 |
| tpz | Tinputz | 7846 | 1.0 | 0.0 | | | | |
| tqb | Tembé | 30084 | 0.48998 | 0.01293 | | | | |
| trc | Copala Triqui | 7881 | 1.0 | 0.0 | | | | |
| trn | Trinitario | 7840 | 1.0 | 0.0 | | | | 0.00022 |
| tro | Tarao Naga | 7952 | 0.9995 | 0.0 | | | | |
| trp | Kok Borok | 30861 | 0.998 | 8e-05 | | | | |
| trq | San Martín Itunyoso Triqui | 7937 | 1.0 | 0.0 | | | | |
| tsc | Tswa | 297653 | 0.99097 | 0.00017 | | | | 0.00011 |
| tsg | Tausug | 7892 | 0.9995 | 3e-05 | | | | |
| tsn | Tswana | 799821 | 0.94537 | 0.00303 | 0.99753 | 6e-05 | 0.98361 | 0.00022 |
| tso | Tsonga | 2723082 | 0.97416 | 0.00143 | 0.99803 | 0.00023 | 0.94158 | 0.00153 |
| tsw | Tsishingini | 7902 | 0.99699 | 3e-05 | | | | |
| tsz | Purepecha | 132907 | 0.74587 | 0.01735 | | | 0.81944 | 0.00274 |
| ttc | Tektiteko | 7954 | 1.0 | 0.0 | | | | 0.00208 |
| tte | Bwanabwana | 7734 | 0.99551 | 0.00017 | | | | |
| ttj | Tooro | 106342 | 0.97983 | 0.00102 | | 0.00017 | | 0.00175 |
| ttq | Tawallammat Tamajaq | 2766 | 0.98647 | 0.0 | | | | |
| tts | Northeastern Thai | 55 | 0.5 | 0.0 | | | | 0.00011 |
| tuc | Mutu | 15211 | 0.9995 | 3e-05 | | | | |
| tue | Tuyuca | 7812 | 0.999 | 0.0 | | | | |
| tuf | Central Tunebo | 7875 | 0.9985 | 3e-05 | | | | |
| tui | Tupuri | 30989 | 0.998 | 3e-05 | | | | |
| tuk | Turkmen | 696538 | 0.99601 | 0.00017 | 0.99803 | 0.00023 | 0.94821 | 0.00142 |
| tum | Tumbuka | 808156 | 0.99551 | 0.00019 | 0.99852 | 0.00011 | | |
| tuo | Tucano | 15602 | 1.0 | 0.0 | | | | |
| tur | Turkish | 2439747 | 0.92159 | 0.00465 | 0.9907 | 0.00108 | 0.4918 | 0.01357 |
| tuv | Turkana | 237 | 0.89655 | 3e-05 | | | | 0.00153 |
| tvk | Southeast Ambrym | 13248 | 0.9985 | 0.0 | | | | |
| tvl | Tuvalu | 520271 | 0.99206 | 0.00044 | | | | |
| twi | Twi | 1934311 | 0.98668 | 0.00074 | 0.99951 | 0.0 | 0.95349 | 0.00131 |
| twu | Termanu | 7903 | 0.9975 | 0.00011 | | | | |
| twx | Tewe | 31794 | 0.98376 | 3e-05 | | | | |
| txq | Tii | 9062 | 0.997 | 8e-05 | | | | |
| txu | Kayapó | 7661 | 1.0 | 0.0 | | | | |
| tyv | Tuvinian | 147493 | 0.9945 | 0.00014 | | | 0.98361 | 0.00022 |
| tzh | Tzeltal | 223502 | 0.98949 | 0.00028 | | | 0.98333 | 0.00022 |
| tzj | Tz'utujil | 17101 | 0.99352 | 0.00025 | | | | 0.00011 |

Table 26: Performance of GlotLID-M on GlotLID-C test, FLORES-200 and UDHR benchmarks (part 22)

| iso639-3 | Language | \|Sentences\| | GlotLID-C | | FLORES-200 | | UDHR | |
|---|---|---|---|---|---|---|---|---|
| | | | F1↑ | FPR↓ | F1↑ | FPR↓ | F1↑ | FPR↓ |
| tzl | Talossan | 343 | 0.775 | 3e-05 | | | | |
| tzm | Central Atlas Tamazight | 8142 | 0.88564 | 0.00292 | 0.94421 | 0.0054 | 0.01754 | 0.00569 |
| tzo | Tzotzil | 521363 | 0.9985 | 8e-05 | | | 0.97479 | 0.00011 |
| ubr | Ubir | 9424 | 0.9975 | 8e-05 | | | | |
| ubu | Umbu-Ungu | 15261 | 0.9985 | 3e-05 | | | | |
| udm | Udmurt | 95670 | 0.99053 | 0.00036 | | | | 0.00022 |
| udu | Uduk | 7952 | 0.9995 | 0.0 | | | 0.98361 | 0.00022 |
| uig | Uighur | 118987 | 0.9995 | 0.0 | 0.99901 | 0.00011 | 0.89916 | 0.0012 |
| ukr | Ukrainian | 2374463 | 0.95992 | 0.00212 | 0.99951 | 6e-05 | 0.98361 | 0.00022 |
| umb | Umbundu | 646290 | 0.99301 | 0.00022 | 0.88585 | 0.00045 | 0.87931 | 0.00109 |
| upv | Uripiv-Wala-Rano-Atchin | 7916 | 0.998 | 6e-05 | | | | |
| ura | Urarina | 7752 | 0.9985 | 3e-05 | | | 0.82963 | 0.0 |
| urb | Urubú-Kaapor | 7742 | 0.9995 | 0.0 | | | | |
| urd | Urdu | 775141 | 0.97065 | 0.00143 | 0.98346 | 0.00187 | 0.96522 | 0.00077 |
| urh | Urhobo | 181379 | 0.99302 | 0.00028 | | | | 0.00011 |
| uri | Urim | 2487 | 1.0 | 0.0 | | | | |
| urk | Urak Lawoi' | 7911 | 0.9995 | 0.0 | | | | |
| urt | Urat | 7816 | 0.9985 | 0.0 | | | | |
| usa | Usarufa | 8134 | 1.0 | 0.0 | | | | |
| usp | Uspanteco | 7898 | 0.999 | 0.0 | | | | |
| uvh | Uri | 7690 | 0.9995 | 0.0 | | | | |
| uvl | Lote | 7908 | 0.999 | 0.0 | | | | |
| uzb | Uzbek | 181303 | 0.999 | 6e-05 | | | | |
| uzn | Northern Uzbek | 1516837 | 0.99054 | 0.00039 | 0.96885 | 0.00364 | 0.48819 | 0.00788 |
| vag | Vagla | 7938 | 0.9995 | 0.0 | | | | 0.00011 |
| vap | Vaiphei | 30918 | 0.98504 | 0.0005 | | | | 0.00011 |
| var | Huarijio | 7954 | 0.9975 | 3e-05 | | | | |
| vec | Venetian | 124915 | 0.96296 | 0.00099 | 0.99703 | 6e-05 | 0.86957 | 0.00197 |
| ven | Venda | 806164 | 0.998 | 0.00011 | | | 1.0 | 0.0 |
| vep | Veps | 13 | | | | | | |
| vgt | Vlaamse Gebarentaal | 9618 | 0.78372 | 3e-05 | | | | |
| vid | Vidunda | 7943 | 0.99154 | 0.00036 | | 0.00017 | | 0.00022 |
| vie | Vietnamese | 2010052 | 0.99303 | 0.00033 | 0.99951 | 6e-05 | 0.66304 | 0.00011 |
| viv | Iduna | 7521 | 0.9985 | 3e-05 | | | | |
| vls | Vlaams | 30000 | 0.97166 | 0.00044 | | 0.00023 | | 0.00011 |
| vmk | Makhuwa-Shirima | 1970 | 0.88306 | 0.0 | | | | 0.00011 |
| vmw | Makhuwa | 306018 | 0.97205 | 0.00132 | | 0.00011 | 0.95798 | 0.00022 |
| vmy | Ayautla Mazatec | 7941 | 0.999 | 0.0 | | | | 0.00263 |
| vol | Volapük | 105178 | 0.99751 | 0.00014 | | | | 0.00044 |
| vro | Võro | 10015 | 0.98739 | 0.00011 | | | | 0.00328 |
| vun | Vunjo | 7951 | 0.99145 | 8e-05 | | | | |
| vut | Vute | 7912 | 1.0 | 0.0 | | | | 0.00328 |
| waj | Waffa | 7854 | 0.9985 | 3e-05 | | | | |
| wal | Wolaytta | 309841 | 0.98961 | 0.00058 | | | | 0.00033 |
| wap | Wapishana | 14453 | 0.9985 | 0.0 | | | | 0.00099 |
| war | Waray (Philippines) | 606273 | 0.99305 | 0.00039 | 0.99951 | 0.0 | 0.9916 | 0.0 |
| wat | Kaninuwa | 2596 | 0.99868 | 3e-05 | | | | |
| way | Wayana | 7923 | 0.9995 | 0.0 | | | | |
| wba | Warao | 388 | 0.97521 | 3e-05 | | | | 0.00044 |
| wbm | Wa | 30852 | 0.76979 | 0.01416 | | | | 0.00197 |
| wbp | Warlpiri | 10006 | 1.0 | 0.0 | | | | |
| wca | Yanomámi | 7788 | 0.80093 | 0.00803 | | | | |
| wed | Wedau | 1309 | 0.96658 | 0.00019 | | | | |
| wer | Weri | 7854 | 1.0 | 0.0 | | | | 0.00011 |
| wes | Cameroon Pidgin | 91652 | 0.94163 | 0.00267 | | | | 0.00252 |
| wew | Wejewa | 884 | 0.98893 | 0.0 | | | | |
| whg | North Wahgi | 977 | 0.9699 | 3e-05 | | | | 0.00011 |
| whk | Wahau Kenyah | 7945 | 0.99548 | 3e-05 | | | | |
| wib | Southern Toussian | 6982 | 0.9995 | 3e-05 | | | | |
| wim | Wik-Mungkan | 7697 | 1.0 | 0.0 | | | | |
| wiu | Wiru | 7809 | 0.9995 | 0.0 | | | | 0.00099 |
| wln | Walloon | 30053 | 0.99198 | 0.00014 | | | 0.62105 | 0.00777 |
| wls | Wallisian | 880578 | 0.99255 | 0.00039 | | | | |
| wlv | Wichí Lhamtés Vejoz | 1809 | 0.9899 | 3e-05 | | | | 0.00033 |
| wmt | Walmajarri | 1149 | 1.0 | 0.0 | | | | |
| wmw | Mwani | 7940 | 0.99549 | 8e-05 | | | | 0.00022 |
| wnc | Wantoat | 10243 | 0.999 | 0.0 | | | | |
| wnu | Usan | 7720 | 1.0 | 0.0 | | | | |
| wob | Wè Northern | 7924 | 0.9985 | 3e-05 | | | | |
| wol | Wolof | 79608 | 0.97988 | 0.00039 | 0.99852 | 0.0 | 0.79747 | 0.0035 |
| wos | Hanga Hundi | 7811 | 0.999 | 0.0 | | | | 0.00011 |
| wrk | Garrwa | 4838 | 0.99862 | 3e-05 | | | | |
| wrs | Waris | 8985 | 0.999 | 0.0 | | | | |
| wsk | Waskia | 13069 | 0.99701 | 0.00014 | | | | |
| wuu | Wu Chinese | 104765 | 0.8768 | 0.00033 | | 0.00011 | 0.35811 | 0.01905 |

Table 27: Performance of GlotLID-M on GlotLID-C test, FLORES-200 and UDHR benchmarks (part 23)

| iso639-3 | Language | \|Sentences\| | GlotLID-C | | FLORES-200 | | UDHR | |
|---|---|---|---|---|---|---|---|---|
| | | | F1↑ | FPR↓ | F1↑ | FPR↓ | F1↑ | FPR↓ |
| wuv | Wuvulu-Aua | 7938 | 0.998 | 0.0 | | | | |
| wwa | Waama | 7901 | 0.9985 | 0.0 | | | 0.88636 | 0.0 |
| xal | Kalmyk | 8727 | 0.99147 | 0.00014 | | | | 0.00022 |
| xav | Xavánte | 24284 | 0.78285 | 0.00803 | | | | 0.00022 |
| xbi | Kombio | 4326 | 0.99924 | 0.0 | | | | |
| xbr | Kambera | 7547 | 0.999 | 0.0 | | | | |
| xed | Hdi | 7945 | 0.9985 | 3e-05 | | | | 0.00011 |
| xho | Xhosa | 1011828 | 0.98372 | 0.00083 | 0.99118 | 0.00097 | 0.93846 | 0.00088 |
| xla | Kamula | 9641 | 0.9995 | 3e-05 | | | | |
| xmf | Mingrelian | 124528 | 0.99303 | 0.0003 | | | | |
| xmv | Antankarana Malagasy | 76882 | 0.99699 | 3e-05 | | 6e-05 | | 0.00011 |
| xnn | Northern Kankanay | 7735 | 0.99649 | 3e-05 | | | | |
| xog | Soga | 25897 | 0.99249 | 0.00017 | | | | |
| xon | Konkomba | 15023 | 0.99651 | 0.00019 | | | | |
| xpe | Liberia Kpelle | 31074 | 0.55157 | 0.00919 | | | | |
| xqa | Karakhanid | 12 | | | | | | |
| xrb | Eastern Karaboro | 7887 | 0.9995 | 3e-05 | | | | |
| xsb | Sambal | 7915 | 0.997 | 0.00011 | | | | |
| xsi | Sio | 10302 | 1.0 | 0.0 | | | | 0.00044 |
| xsm | Kasem | 7874 | 0.9985 | 6e-05 | | | | |
| xsr | Sherpa | 7935 | 1.0 | 0.0 | | | | |
| xsu | Sanumá | 7788 | 0.76697 | 0.00371 | | | | 0.00011 |
| xtd | Diuxi-Tilantongo Mixtec | 8182 | 0.9985 | 3e-05 | | | | 0.00044 |
| xtm | Magdalena Peñasco Mixtec | 7929 | 0.9985 | 0.0 | | | | 0.00427 |
| xtn | Northern Tlaxiaco Mixtec | 14968 | 0.9985 | 8e-05 | | | | |
| xuo | Kuo | 7884 | 0.999 | 3e-05 | | | | |
| yaa | Yaminahua | 7622 | 1.0 | 0.0 | | | | |
| yad | Yagua | 7454 | 1.0 | 0.0 | | | 0.88889 | 0.0 |
| yal | Yalunka | 7932 | 0.998 | 0.0 | | | | 0.00055 |
| yam | Yamba | 7927 | 0.9995 | 0.0 | | | | |
| yan | Mayangna | 30722 | 0.99449 | 8e-05 | | | | |
| yao | Yao | 204519 | 0.99701 | 0.00014 | | 6e-05 | 0.97345 | 0.00022 |
| yap | Yapese | 352203 | 0.99651 | 0.00017 | | | 0.96721 | 0.00022 |
| yaq | Yaqui | 7934 | 1.0 | 0.0 | | | | |
| ybb | Yemba | 8028 | 0.91398 | 0.00028 | | | | 0.00197 |
| yby | Yaweyuha | 7883 | 0.999 | 0.0 | | | | |
| ycn | Yucuna | 7857 | 0.9995 | 0.0 | | | | |
| ydd | Eastern Yiddish | 911 | 0.99661 | 0.0 | 0.99603 | 0.0 | 0.99187 | 0.0 |
| yid | Yiddish | 44101 | 0.99497 | 0.0 | | | | |
| yim | Yimchungru Naga | 30883 | 0.99548 | 3e-05 | | | | |
| yka | Yakan | 8455 | 0.9975 | 3e-05 | | | | |
| ykg | Northern Yukaghir | 20 | | | | | | |
| yle | Yele | 7585 | 0.9995 | 0.0 | | 6e-05 | | |
| yli | Angguruk Yali | 7905 | 0.999 | 3e-05 | | | | 0.00088 |
| yml | Iamalele | 7317 | 0.9985 | 0.0 | | | | |
| yom | Yombe | 77077 | 0.98856 | 0.00047 | | 6e-05 | | 0.00646 |
| yon | Yongkom | 11798 | 0.9995 | 0.0 | | | | 0.00022 |
| yor | Yoruba | 1812160 | 0.99206 | 0.00041 | 0.99406 | 0.00023 | 0.85106 | 0.0023 |
| yrb | Yareba | 7117 | 0.9975 | 8e-05 | | | | |
| yre | Yaouré | 7908 | 1.0 | 0.0 | | | | |
| yrk | Nenets | 1908 | 0.99112 | 3e-05 | | | 0.8381 | 0.00011 |
| yrl | Nhengatu | 7439 | 0.9322 | 0.00022 | | | | |
| yss | Yessan-Mayo | 15808 | 0.999 | 6e-05 | | | | |
| yua | Yucateco | 616290 | 0.99701 | 0.00014 | | | 1.0 | 0.0 |
| yue | Yue Chinese | 64647 | 0.9549 | 0.00022 | 0.00394 | 0.0 | 0.46897 | 0.00449 |
| yuj | Karkar-Yuri | 7845 | 0.9995 | 0.0 | | | | |
| yup | Yukpa | 329 | 0.96078 | 3e-05 | | | | 0.00077 |
| yut | Yopno | 10245 | 1.0 | 0.0 | | | | |
| yuw | Yau (Morobe Province) | 7846 | 0.999 | 6e-05 | | | | |
| yuz | Yuracare | 7826 | 1.0 | 0.0 | | | | |
| yva | Yawa | 7651 | 0.9995 | 3e-05 | | | | 0.00011 |
| zaa | Sierra de Juárez Zapotec | 7891 | 1.0 | 0.0 | | | | |
| zab | Western Tlacolula Valley Zapotec | 9862 | 0.99549 | 6e-05 | | | | |
| zac | Ocotlán Zapotec | 7941 | 0.9995 | 0.0 | | | | |
| zad | Cajonos Zapotec | 7938 | 1.0 | 0.0 | | | | 0.00033 |
| zae | Yareni Zapotec | 7916 | 0.999 | 6e-05 | | | | 0.00011 |
| zai | Isthmus Zapotec | 276740 | 0.98903 | 0.00039 | | | | 0.00055 |
| zam | Miahuatlán Zapotec | 7943 | 0.9995 | 3e-05 | | | 0.0 | 0.00011 |
| zao | Ozolotepec Zapotec | 7940 | 0.998 | 3e-05 | | | | 0.00109 |
| zar | Rincón Zapotec | 7919 | 0.9975 | 8e-05 | | | | |
| zas | Santo Domingo Albarradas Zapotec | 7928 | 0.9995 | 3e-05 | | | | 0.00131 |
| zat | Tabaa Zapotec | 7938 | 1.0 | 0.0 | | | | |
| zav | Yatzachi Zapotec | 11233 | 0.87575 | 0.00099 | | | | 0.00088 |
| zaw | Mitla Zapotec | 7951 | 0.9995 | 0.0 | | | | |
| zca | Coatecas Altas Zapotec | 7938 | 1.0 | 0.0 | | | | 0.00022 |

Table 28: Performance of GlotLID-M on GlotLID-C test, FLORES-200 and UDHR benchmarks (part 24)

| iso639-3 | Language | \|Sentences\| | GlotLID-C | | FLORES-200 | | UDHR | |
|---|---|---|---|---|---|---|---|---|
| | | | F1↑ | FPR↓ | F1↑ | FPR↓ | F1↑ | FPR↓ |
| zdj | Ngazidja Comorian | 2097 | 0.97484 | 6e-05 | | 0.00011 | 0.78125 | 0.00208 |
| zea | Zeeuws | 10012 | 0.91037 | 0.00025 | | 0.00045 | | |
| zgh | Standard Moroccan Tamazight | 3285 | 0.93333 | 0.00039 | | 0.00068 | 0.17647 | 0.0 |
| zho | Chinese | 4212269 | 0.998 | 0.0 | 1.0 | 0.0 | | |
| zia | Zia | 7893 | 0.99749 | 0.0 | | | | |
| ziw | Zigula | 7957 | 0.97857 | 0.00069 | | | | |
| zlm | Malay (individual language) | 2260 | 0.86341 | 3e-05 | | | 0.0 | 0.00011 |
| zne | Zande (individual language) | 315310 | 0.99053 | 0.00036 | | | | 0.00219 |
| zom | Zou | 30810 | 0.64431 | 0.0192 | | | | 0.0012 |
| zos | Francisco León Zoque | 7953 | 0.999 | 0.0 | | | | |
| zpa | Lachiguiri Zapotec | 67840 | 0.75362 | 0.01389 | | | | 0.00055 |
| zpc | Choapan Zapotec | 7914 | 0.999 | 0.0 | | | | |
| zpd | Southeastern Ixtlán Zapotec | 106 | 0.82927 | 3e-05 | | | | 0.0023 |
| zpf | San Pedro Quiatoni Zapotec | 94 | 1.0 | 0.0 | | | | 0.00208 |
| zpg | Guevea De Humboldt Zapotec | 1020 | 0.98462 | 0.0 | | | | |
| zpi | Santa María Quiegolani Zapotec | 7856 | 0.999 | 6e-05 | | | | 0.00011 |
| zpj | Quiavicuzas Zapotec | 124 | 0.89362 | 0.0 | | | | 0.00077 |
| zpl | Lachixío Zapotec | 7953 | 0.999 | 3e-05 | | | | |
| zpm | Mixtepec Zapotec | 7695 | 0.9995 | 0.0 | | | | 0.00033 |
| zpo | Amatlán Zapotec | 7913 | 0.9985 | 3e-05 | | | | 0.00033 |
| zpq | Zoogocho Zapotec | 7889 | 0.99649 | 3e-05 | | | | |
| zpt | San Vicente Coatlán Zapotec | 7929 | 1.0 | 0.0 | | | | 0.00383 |
| zpu | Yalálag Zapotec | 7935 | 0.999 | 0.0 | | | | |
| zpv | Chichicapan Zapotec | 7946 | 1.0 | 0.0 | | | | 0.00438 |
| zpz | Texmelucan Zapotec | 7936 | 0.9995 | 0.0 | | | | |
| zsm | Standard Malay | 445673 | 0.9223 | 0.00168 | 0.93506 | 0.00307 | | 0.0081 |
| zsr | Southern Rincon Zapotec | 7918 | 0.998 | 0.0 | | | | |
| ztq | Quioquitani-Quierí Zapotec | 7898 | 1.0 | 0.0 | | | | 0.00011 |
| zty | Yatee Zapotec | 7908 | 0.999 | 6e-05 | | | | |
| zul | Zulu | 990448 | 0.99295 | 0.0 | 0.96893 | 0.0 | 0.98305 | 0.0 |
| zyb | Yongbei Zhuang | 10631 | 0.9985 | 0.0 | | | 0.912 | 0.00066 |
| zyp | Zyphe Chin | 7934 | 0.999 | 0.0 | | | | 0.00011 |
| zza | Zaza | 1713 | 0.936 | 0.0 | | | | |

Table 29: Performance of GlotLID-M on GlotLID-C test, FLORES-200 and UDHR benchmarks (part 25)

| | | | | | | | with confidence threshold θ | | | | | | | |
|---|---|---|---|---|---|---|---|---|---|---|---|---|---|---|
| | | | GlotLID-M | | CLD3 | | GlotLID-M θ=.3 | | GlotLID-M θ=.5 | | CLD3 θ=.5 | | CLD3 θ=.7 | |
| iso639-3 | FLORES Code(s) | CLD3 Code(s) | F1↑ | FPR↓ | F1↑ | FPR↓ | F1↑ | FPR↓ | F1↑ | FPR↓ | F1↑ | FPR↓ | F1↑ | FPR↓ |
| ace | ace | - | **0.95757** | 0.01099 | | | 0.95732 | 0.00984 | 0.95689 | 0.00788 | | | | |
| afr | afr | af | **1.0** | 0.0 | 0.86863 | 0.00308 | 1.0 | 0.0 | 1.0 | 0.0 | 0.88042 | 0.00275 | 0.91091 | 0.00188 |
| aka | aka/twi | - | 0.99876 | 0.00012 | | | 0.99901 | 0.0 | 0.99901 | 0.0 | | | | |
| amh | amh | am | **0.99951** | 0.00012 | 0.66579 | 0.01046 | 0.99951 | 0.00011 | 0.99951 | 0.0001 | 0.66579 | 0.01043 | 0.66579 | 0.0102 |
| ara | arz/ars/acm/ary/aeb/acq/apc/arb/ajp | ar | **0.93244** | 0.06061 | 0.88458 | 0.0137 | 0.92725 | 0.0542 | 0.92631 | 0.04535 | 0.89253 | 0.01176 | 0.90924 | 0.00751 |
| asm | asm | - | **1.0** | 0.0 | | | 1.0 | 0.0 | 1.0 | 0.0 | | | | |
| ast | ast | - | 0.99209 | 0.00099 | | | **0.99308** | 0.00066 | 0.99257 | 0.00049 | | | | |
| awa | awa | - | 0.38951 | 0.00012 | | | **0.38982** | 0.0 | 0.35313 | 0.0 | | | | |
| ayr | ayr | - | **0.99556** | 0.00086 | | | 0.96738 | 0.00011 | 0.93193 | 0.0 | | | | |
| aze | azb/azj | az | **0.71808** | 0.00049 | 0.45528 | 0.01446 | 0.71546 | 0.00033 | 0.69627 | 0.00019 | 0.48299 | 0.0118 | 0.53112 | 0.00768 |
| bak | bak | - | **1.0** | 0.0 | | | 1.0 | 0.0 | 1.0 | 0.0 | | | | |
| bam | bam | - | 0.52563 | 0.11122 | | | 0.52664 | 0.0991 | **0.53084** | 0.08467 | | | | |
| ban | ban | - | 0.97521 | 0.00012 | | | **0.97571** | 0.0 | 0.97467 | 0.0 | | | | |
| bel | bel | be | **1.0** | 0.0 | 0.98827 | 0.00024 | 1.0 | 0.0 | 1.0 | 0.0 | 0.98972 | 0.00021 | 0.99312 | 0.00013 |
| bem | bem | - | 0.9906 | 0.00099 | | | **0.99256** | 0.00044 | 0.99256 | 0.00039 | | | | |
| ben | ben | bn | **0.99852** | 0.00012 | 0.5005 | 0.02059 | 0.99852 | 0.00011 | 0.99852 | 0.0001 | 0.5005 | 0.02053 | 0.5005 | 0.02006 |
| bho | bho | - | 0.94329 | 0.00963 | | | 0.94374 | 0.00852 | **0.94846** | 0.00477 | | | | |
| bod | bod | - | **0.94589** | 0.00012 | | | 0.94589 | 0.00011 | 0.94589 | 0.0001 | | | | |
| bos | bos | bs | **0.58206** | 0.00679 | 0.53913 | 0.00518 | 0.57353 | 0.00608 | 0.49605 | 0.00331 | 0.55291 | 0.00431 | 0.00197 | 1e-05 |
| bug | bug | - | **0.99802** | 0.00012 | | | 0.99404 | 0.0 | 0.99404 | 0.0 | | | | |
| bul | bul | bg-Latn/bg | **0.99951** | 0.0 | 0.93506 | 0.00141 | 0.99951 | 0.0 | 0.99951 | 0.0 | 0.96413 | 0.00073 | 0.98051 | 0.00034 |
| cat | cat | ca | **1.0** | 0.0 | 0.54988 | 0.017 | 1.0 | 0.0 | 1.0 | 0.0 | 0.56949 | 0.01565 | 0.60701 | 0.01305 |
| ceb | ceb | ceb | **0.99503** | 0.0 | 0.65466 | 0.01081 | 0.99454 | 0.0 | 0.99404 | 0.0 | 0.67137 | 0.00999 | 0.68434 | 0.00908 |
| ces | ces | cs | **0.99951** | 0.00012 | 0.89569 | 0.0024 | 0.99951 | 0.00011 | 0.99901 | 0.0001 | 0.92435 | 0.00166 | 0.95071 | 0.00096 |
| cjk | cjk | - | **0.84493** | 0.00012 | | | 0.83429 | 0.00011 | 0.79834 | 0.0001 | | | | |
| ckb | ckb | - | **1.0** | 0.0 | | | 1.0 | 0.0 | 1.0 | 0.0 | | | | |
| cos | - | co | | 0.00037 | | 0.02364 | | 0.00033 | | 0.00019 | | 0.02132 | | 0.0177 |
| crh | crh | - | **0.98851** | 0.0 | | | 0.988 | 0.0 | 0.988 | 0.0 | | | | |
| cym | cym | cy | 0.99951 | 0.00012 | 0.84123 | 0.00396 | 0.99951 | 0.00011 | **1.0** | 0.0 | 0.89947 | 0.00232 | 0.93828 | 0.00133 |
| dan | dan | da | 0.99357 | 0.00074 | 0.93422 | 0.00118 | **0.99554** | 0.00022 | 0.99554 | 0.00019 | 0.95082 | 0.00079 | 0.96071 | 0.00046 |
| deu | deu | de | **0.99901** | 0.0 | 0.98011 | 0.0004 | 0.99901 | 0.0 | 0.99852 | 0.0 | 0.98826 | 0.00023 | 0.99016 | 0.00014 |
| dik | dik | - | **0.99653** | 0.00012 | | | 0.99653 | 0.0 | 0.99454 | 0.0 | | | | |
| dyu | dyu | - | **0.12435** | 0.03148 | | | 0.11878 | 0.0282 | 0.11186 | 0.02472 | | | | |
| dzo | dzo | - | **0.9496** | 0.01271 | | | 0.9491 | 0.01139 | 0.9491 | 0.01002 | | | | |
| ell | ell | el-Latn/el | **1.0** | 0.0 | 0.90811 | 0.00207 | 1.0 | 0.0 | 1.0 | 0.0 | 0.94382 | 0.0012 | 0.96691 | 0.00066 |
| eng | eng | en | 0.9878 | 0.00309 | 0.95943 | 0.00081 | 0.99215 | 0.00166 | **0.99556** | 0.00058 | 0.97473 | 0.00044 | 0.98132 | 0.00024 |
| epo | epo | eo | 0.99901 | 0.00025 | 0.91773 | 0.00178 | 0.99951 | 0.00011 | **1.0** | 0.0 | 0.95121 | 0.00098 | 0.97423 | 0.00043 |
| est | est | et | 0.99852 | 0.00037 | 0.92477 | 0.00166 | **0.99951** | 0.00011 | 0.99951 | 0.0001 | 0.95905 | 0.00084 | 0.9781 | 0.00038 |
| eus | eus | eu | **0.99951** | 0.0 | 0.86818 | 0.00317 | 0.99951 | 0.0 | 0.99951 | 0.0 | 0.93137 | 0.00153 | 0.96099 | 0.00081 |
| ewe | ewe | - | **1.0** | 0.0 | | | 1.0 | 0.0 | 1.0 | 0.0 | | | | |
| fao | fao | - | **0.99951** | 0.0 | | | 0.99951 | 0.0 | 0.99951 | 0.0 | | | | |
| fas | pes/prs | fa | 0.79937 | 0.12542 | 0.62121 | 0.02544 | 0.80032 | 0.11171 | **0.82343** | 0.08448 | 0.64803 | 0.02256 | 0.71721 | 0.01595 |
| fij | fij | - | **0.99901** | 0.0 | | | 0.99901 | 0.0 | 0.99901 | 0.0 | | | | |
| fil | - | fil | | | | 0.01989 | | | | | | 0.01719 | | 0.01419 |
| fin | fin | fi | 0.99951 | 0.00012 | 0.92449 | 0.00169 | **1.0** | 0.0 | 1.0 | 0.0 | 0.95825 | 0.00089 | 0.97817 | 0.00041 |
| fon | fon | - | **0.99752** | 0.0 | | | 0.99752 | 0.0 | 0.99703 | 0.0 | | | | |
| fra | fra | fr | **0.99951** | 0.00012 | 0.82909 | 0.00428 | 0.99951 | 0.00011 | 0.99852 | 0.0001 | 0.85641 | 0.00345 | 0.89511 | 0.00233 |
| fry | - | fy | | | | 0.00228 | | | | | | 0.00168 | | 0.00099 |
| fur | fur | - | 0.99951 | 0.00012 | | | 0.99951 | 0.00011 | **1.0** | 0.0 | | | | |
| fuv | fuv | - | **0.96738** | 0.00012 | | | 0.96099 | 0.0 | 0.94693 | 0.0 | | | | |
| gaz | gaz | - | **0.99312** | 0.0016 | | | 0.90232 | 0.00055 | 0.78878 | 0.00029 | | | | |
| gla | gla | gd | 0.99951 | 0.00012 | 0.82382 | 0.00446 | **1.0** | 0.0 | 1.0 | 0.0 | 0.86919 | 0.00312 | 0.91561 | 0.00185 |
| gle | gle | ga | **1.0** | 0.0 | 0.92696 | 0.00162 | 1.0 | 0.0 | 1.0 | 0.0 | 0.95234 | 0.00101 | 0.97961 | 0.00039 |
| glg | glg | gl | **0.99703** | 0.00025 | 0.76147 | 0.0063 | 0.99703 | 0.00022 | 0.99703 | 0.0001 | 0.79904 | 0.00501 | 0.84291 | 0.0035 |
| grn | grn | - | **1.0** | 0.0 | | | 1.0 | 0.0 | 1.0 | 0.0 | | | | |
| guj | guj | gu | **1.0** | 0.0 | 0.99703 | 0.0 | 1.0 | 0.0 | 1.0 | 0.0 | 0.99703 | 0.0 | 0.99703 | 0.0 |
| hat | hat | ht | 0.99852 | 0.00037 | 0.63645 | 0.01196 | 0.99852 | 0.00033 | **0.99901** | 0.00019 | 0.70798 | 0.00861 | 0.78066 | 0.0057 |
| hau | hau | ha | 0.95924 | 0.01062 | 0.44685 | 0.02592 | 0.98348 | 0.00376 | **0.99313** | 0.00136 | 0.53795 | 0.0179 | 0.64867 | 0.01101 |
| haw | - | haw | | | | 0.01547 | | | | | | 0.01097 | | 0.00655 |
| heb | heb | iw | **1.0** | 0.0 | 0.99555 | 4e-05 | 1.0 | 0.0 | 1.0 | 0.0 | 0.99555 | 4e-05 | 0.99604 | 3e-05 |
| hin | hin | hi-Latn/hi | 0.67624 | 0.11961 | 0.21078 | 0.07843 | 0.67692 | 0.10685 | **0.69697** | 0.08564 | 0.22372 | 0.07248 | 0.25377 | 0.05996 |
| hmn | - | hmn | | | | 0.00069 | | | | | | 0.00029 | | 0.00014 |
| hne | hne | - | 0.90296 | 0.00555 | | | **0.90343** | 0.00487 | 0.898 | 0.0035 | | | | |
| hrv | hrv | hr | 0.75157 | 0.0711 | 0.56313 | 0.0049 | 0.75533 | 0.06216 | **0.768** | 0.04389 | 0.56847 | 0.00456 | | 0.0 |
| hun | hun | hu | **1.0** | 0.0 | 0.71804 | 0.00822 | 1.0 | 0.0 | 0.99951 | 0.0 | 0.79889 | 0.00525 | 0.88105 | 0.00275 |
| hye | hye | hy | **1.0** | 0.0 | 0.99703 | 0.0 | 1.0 | 0.0 | 1.0 | 0.0 | 0.99703 | 0.0 | 0.99703 | 0.0 |
| ibo | ibo | ig | 0.99951 | 0.00012 | 0.66843 | 0.01036 | 0.99951 | 0.00011 | **1.0** | 0.0 | 0.75769 | 0.00665 | 0.85581 | 0.0034 |
| ilo | ilo | - | **0.99951** | 0.00012 | | | 0.99951 | 0.00011 | 0.99951 | 0.0001 | | | | |
| isl | isl | is | 0.99901 | 0.00025 | 0.62928 | 0.01231 | 0.99951 | 0.00011 | 0.99951 | 0.0001 | 0.64432 | 0.01148 | 0.6569 | 0.01061 |
| ita | ita | it | 0.99803 | 0.00037 | 0.53638 | 0.01807 | 0.99852 | 0.00022 | **0.99901** | 0.0001 | 0.56196 | 0.01622 | 0.61908 | 0.01244 |
| jav | jav | jv | 0.98442 | 0.00383 | 0.57216 | 0.0156 | 0.99166 | 0.00177 | **0.99213** | 0.00127 | 0.62126 | 0.01266 | 0.67272 | 0.00972 |
| jpn | jpn | ja-Latn/ja | **1.0** | 0.0 | 0.67627 | 0.00998 | 1.0 | 0.0 | 1.0 | 0.0 | 0.76672 | 0.00631 | 0.854 | 0.00345 |
| kab | kab | - | 0.86127 | 0.02605 | | | 0.87886 | 0.01858 | **0.90909** | 0.00954 | | | | |
| kac | kac | - | **1.0** | 0.0 | | | 1.0 | 0.0 | 1.0 | 0.0 | | | | |
| kam | kam | - | **0.91658** | 0.00012 | | | 0.91183 | 0.0 | 0.87368 | 0.0 | | | | |
| kan | kan | kn | **1.0** | 0.0 | 0.99153 | 0.0 | 1.0 | 0.0 | 1.0 | 0.0 | 0.99153 | 0.0 | 0.99153 | 0.0 |
| kas | kas | - | **0.97649** | 0.0 | | | 0.97597 | 0.0 | 0.96945 | 0.0 | | | | |
| kat | kat | ka | **1.0** | 0.0 | 0.99354 | 0.0 | 1.0 | 0.0 | 1.0 | 0.0 | 0.99354 | 0.0 | 0.99354 | 0.0 |
| kaz | kaz | kk | **0.99951** | 0.0 | 0.71835 | 0.00819 | 0.99951 | 0.0 | 0.99951 | 0.0 | 0.72531 | 0.00788 | 0.75968 | 0.00643 |
| kbp | kbp | - | **0.99901** | 0.00012 | | | 0.99901 | 0.00011 | 0.99901 | 0.0001 | | | | |
| kea | kea | - | **0.95238** | 0.0 | | | 0.9513 | 0.0 | 0.93586 | 0.0 | | | | |
| khm | khm | km | **0.99951** | 0.0 | 0.99404 | 0.0 | 0.99951 | 0.0 | 0.99951 | 0.0 | 0.99404 | 0.0 | 0.99404 | 0.0 |
| kik | kik | - | **0.96562** | 0.00876 | | | 0.96509 | 0.00774 | 0.96456 | 0.00672 | | | | |
| kin | kin | - | **0.91471** | 0.00074 | | | 0.91471 | 0.00066 | 0.91471 | 0.00058 | | | | |
| kir | kir | ky | **1.0** | 0.0 | 0.63286 | 0.01214 | 1.0 | 0.0 | 1.0 | 0.0 | 0.63803 | 0.01182 | 0.67267 | 0.00987 |
| kmb | kmb | - | 0.96321 | 0.00901 | | | 0.96923 | 0.00664 | **0.97713** | 0.0038 | | | | |
| kmr | kmr | ku (Latn) | **0.99901** | 0.0 | 0.51859 | 0.01944 | 0.99901 | 0.0 | 0.99901 | 0.0 | 0.60922 | 0.01339 | 0.72318 | 0.00781 |
| knc | knc | - | 0.86459 | 0.00272 | | | 0.86869 | 0.00055 | **0.86966** | 0.0001 | | | | |
| kon | kon | - | 0.99359 | 0.00111 | | | 0.99408 | 0.00088 | **0.99703** | 0.00019 | | | | |
| kor | kor | ko | **1.0** | 0.0 | 0.98657 | 7e-05 | 1.0 | 0.0 | 1.0 | 0.0 | 0.98805 | 4e-05 | 0.98853 | 2e-05 |
| lao | lao | lo | **1.0** | 0.0 | 0.97726 | 0.0 | 0.99901 | 0.0 | 0.99802 | 0.0 | 0.97726 | 0.0 | 0.97726 | 0.0 |
| lat | - | la | | | | 0.00042 | | | | | | 0.00028 | | 0.00013 |

Table 30: Comparison of GlotLID vs CLD3 on FLORES-200 benchmark (part 1)

| | | | GlotLID-M | | CLD3 | | GlotLID-M θ=.3 | | GlotLID-M θ=.5 | | CLD3 θ=.5 | | CLD3 θ=.7 | |
|---|---|---|---|---|---|---|---|---|---|---|---|---|---|---|
| iso639-3 | FLORES Code(s) | CLD3 Code(s) | F1↑ | FPR↓ | F1↑ | FPR↓ | F1↑ | FPR↓ | F1↑ | FPR↓ | F1↑ | FPR↓ | F1↑ | FPR↓ |
| lav | lvs/ltg | lv | 0.99951 | 0.00012 | 0.89202 | 0.00348 | **0.99975** | 0.0 | 0.99951 | 0.0 | 0.91429 | 0.00225 | 0.92211 | 0.00122 |
| lij | lij | - | **0.99901** | 0.00012 | | | 0.99852 | 0.00011 | 0.99753 | 0.0001 | | | | |
| lim | lim | - | **0.99253** | 0.0 | | | 0.99253 | 0.0 | 0.99153 | 0.0 | | | | |
| lin | lin | - | **0.99901** | 0.00025 | | | 0.99852 | 0.00022 | 0.99901 | 0.0001 | | | | |
| lit | lit | lt | **0.99951** | 0.0 | 0.78767 | 0.00561 | 0.99951 | 0.0 | 0.99951 | 0.0 | 0.84719 | 0.00373 | 0.91057 | 0.00196 |
| lmo | lmo | - | **0.99554** | 0.00025 | | | 0.99554 | 0.00022 | 0.99504 | 0.0001 | | | | |
| ltz | ltz | lb | **0.99951** | 0.0 | 0.93872 | 0.00136 | 0.99901 | 0.0 | 0.99901 | 0.0 | 0.96011 | 0.00086 | 0.97821 | 0.00043 |
| lua | lua | - | **0.99653** | 0.00012 | | | 0.99553 | 0.0 | 0.99404 | 0.0 | | | | |
| lug | lug | - | **0.99603** | 0.0 | | | 0.99603 | 0.0 | 0.99454 | 0.0 | | | | |
| luo | luo | - | **1.0** | 0.0 | | | 1.0 | 0.0 | 0.99951 | 0.0 | | | | |
| lus | lus | - | 0.99653 | 0.00012 | | | **0.99703** | 0.0 | 0.99653 | 0.0 | | | | |
| mag | mag | - | 0.95459 | 0.00296 | | | **0.95507** | 0.00254 | 0.95408 | 0.00127 | | | | |
| mai | mai | - | **0.97366** | 0.00012 | | | 0.97366 | 0.00011 | 0.97102 | 0.0 | | | | |
| mal | mal | ml | **1.0** | 0.0 | 0.99653 | 0.0 | 1.0 | 0.0 | 1.0 | 0.0 | 0.99653 | 0.0 | 0.99653 | 0.0 |
| mar | mar | mr | **1.0** | 0.0 | 0.47801 | 0.02287 | 1.0 | 0.0 | 1.0 | 0.0 | 0.50855 | 0.02018 | 0.59325 | 0.01397 |
| mkd | mkd | mk | **1.0** | 0.0 | 0.99407 | 6e-05 | 1.0 | 0.0 | 1.0 | 0.0 | 0.99357 | 5e-05 | 0.99305 | 2e-05 |
| mlg | plt | mg | **0.99951** | 0.00012 | 0.89399 | 0.00249 | 0.99951 | 0.00011 | 0.99951 | 0.0001 | 0.94757 | 0.00116 | 0.97352 | 0.00055 |
| mlt | mlt | mt | 0.97731 | 0.0058 | 0.55065 | 0.01708 | 0.99216 | 0.00177 | **0.99803** | 0.00039 | 0.62752 | 0.01237 | 0.72662 | 0.00765 |
| mni | mni | - | **0.99901** | 0.00012 | | | 0.99901 | 0.00011 | 0.99901 | 0.0001 | | | | |
| mon | khk | mn | **1.0** | 0.0 | 0.99508 | 0.0001 | 1.0 | 0.0 | 0.99951 | 0.0 | 0.99704 | 6e-05 | 0.99901 | 2e-05 |
| mos | mos | - | **0.98138** | 0.0 | | | 0.97415 | 0.0 | 0.96418 | 0.0 | | | | |
| mri | mri | mi | 0.99901 | 0.00012 | 0.60242 | 0.01228 | **0.99951** | 0.0 | 0.99901 | 0.0 | 0.67426 | 0.00878 | 0.7575 | 0.00525 |
| msa | min/ind/bjn/zsm | ms/id | 0.97122 | 0.02271 | 0.64861 | 0.01076 | **0.97515** | 0.01482 | 0.97478 | 0.01051 | 0.64377 | 0.00722 | 0.54044 | 0.00419 |
| mya | mya | my | **1.0** | 0.0 | 0.67006 | 0.00983 | 1.0 | 0.0 | 1.0 | 0.0 | 0.67006 | 0.0098 | 0.67006 | 0.00958 |
| nep | npi | ne | 0.99852 | 0.00025 | 0.34321 | 0.03993 | 0.99852 | 0.00022 | **0.99951** | 0.0 | 0.36071 | 0.03687 | 0.41741 | 0.02834 |
| nld | nld | nl | 0.99803 | 0.00049 | 0.77945 | 0.00588 | **0.99901** | 0.00022 | 0.99901 | 0.0001 | 0.78535 | 0.00565 | 0.80854 | 0.00471 |
| nor | nno/nob | no | 0.99729 | 0.00111 | 0.95306 | 0.0011 | 0.99753 | 0.00088 | **0.99778** | 0.00068 | 0.96076 | 0.0007 | 0.95885 | 0.00038 |
| nso | nso | - | **0.99704** | 0.00062 | | | 0.99704 | 0.00055 | 0.99655 | 0.00049 | | | | |
| nus | nus | - | **0.99951** | 0.0 | | | 0.99951 | 0.0 | 0.99951 | 0.0 | | | | |
| nya | nya | ny | 0.99753 | 0.00049 | 0.37034 | 0.03552 | 0.99803 | 0.00033 | **0.99852** | 0.0001 | 0.41506 | 0.02935 | 0.50325 | 0.01996 |
| oci | oci | - | 0.99951 | 0.00012 | | | **1.0** | 0.0 | 0.99951 | 0.0 | | | | |
| ory | ory | - | **1.0** | 0.0 | | | 0.80519 | 0.0 | 0.66314 | 0.0 | | | | |
| pag | pag | - | **0.99852** | 0.0 | | | 0.99852 | 0.0 | 0.99852 | 0.0 | | | | |
| pan | pan | pa | **1.0** | 0.0 | 0.99553 | 0.0 | 1.0 | 0.0 | 1.0 | 0.0 | 0.99553 | 0.0 | 0.99553 | 0.0 |
| pap | pap | - | 0.99069 | 0.00222 | | | 0.99118 | 0.00188 | **0.99557** | 0.00078 | | | | |
| pol | pol | pl | 0.9907 | 0.00235 | 0.585 | 0.01483 | **0.99167** | 0.00188 | 0.99167 | 0.00165 | 0.61774 | 0.01289 | 0.64081 | 0.01137 |
| por | por | pt | 0.99655 | 0.00049 | 0.77611 | 0.00597 | **0.99704** | 0.00066 | 0.99704 | 0.00049 | 0.81641 | 0.0046 | 0.86528 | 0.00305 |
| pus | pus | ps | 0.88805 | 0.00198 | 0.66315 | 0.01051 | 0.88853 | 0.00166 | **0.89142** | 0.00088 | 0.70238 | 0.00869 | 0.80064 | 0.00494 |
| quy | quy | - | **0.75676** | 0.0 | | | 0.625 | 0.0 | 0.57163 | 0.0 | | | | |
| ron | ron | ro | **0.99951** | 0.0 | 0.82339 | 0.00442 | 0.99951 | 0.0 | 0.99951 | 0.0 | 0.87289 | 0.00297 | 0.91857 | 0.00172 |
| run | run | - | 0.92541 | 0.01913 | | | 0.92584 | 0.01703 | **0.92627** | 0.01489 | | | | |
| rus | rus | ru-Latn/ru | **0.99901** | 0.00012 | 0.94255 | 0.00124 | 0.99901 | 0.00011 | 0.99901 | 0.0001 | 0.96459 | 0.00072 | 0.97521 | 0.00042 |
| sag | sag | - | **0.99901** | 0.0 | | | 0.99901 | 0.0 | 0.99901 | 0.0 | | | | |
| san | san | - | **0.99104** | 0.00012 | | | 0.99104 | 0.00011 | 0.99103 | 0.0 | | | | |
| sat | sat | - | **1.0** | 0.0 | | | 1.0 | 0.0 | 1.0 | 0.0 | | | | |
| scn | scn | - | 0.99802 | 0.00012 | | | 0.99802 | 0.00011 | **0.99852** | 0.0 | | | | |
| shn | shn | - | **1.0** | 0.0 | | | 1.0 | 0.0 | 1.0 | 0.0 | | | | |
| sin | sin | si | **1.0** | 0.0 | 0.99354 | 0.0 | 1.0 | 0.0 | 1.0 | 0.0 | 0.99354 | 0.0 | 0.99354 | 0.0 |
| slk | slk | sk | 0.99852 | 0.00012 | 0.75414 | 0.00668 | 0.99852 | 0.00011 | **0.99901** | 0.0 | 0.81083 | 0.00474 | 0.87511 | 0.00279 |
| slv | slv | sl | 0.99508 | 0.00123 | 0.97107 | 0.00057 | 0.99655 | 0.00077 | **0.99951** | 0.0001 | 0.97955 | 0.00037 | 0.98528 | 0.00022 |
| smo | smo | sm | **0.99603** | 0.0 | 0.90934 | 0.00198 | 0.99603 | 0.0 | 0.99603 | 0.0 | 0.944 | 0.00114 | 0.97187 | 0.00048 |
| sna | sna | sn | 0.99951 | 0.00012 | 0.61449 | 0.01309 | 0.99951 | 0.00011 | **1.0** | 0.0 | 0.66955 | 0.01024 | 0.75336 | 0.00662 |
| snd | snd | sd | 0.99606 | 0.00099 | 0.47115 | 0.02287 | 0.99606 | 0.00088 | **0.99704** | 0.00058 | 0.49206 | 0.02092 | 0.57735 | 0.01439 |
| som | som | so | 0.97028 | 0.00765 | 0.52313 | 0.01912 | 0.98973 | 0.00232 | **0.99803** | 0.00039 | 0.57094 | 0.01571 | 0.61892 | 0.01256 |
| sot | sot | st | **1.0** | 0.0 | 0.5043 | 0.02015 | 1.0 | 0.0 | 1.0 | 0.0 | 0.52382 | 0.01851 | 0.54545 | 0.01644 |
| spa | spa | es | **0.99508** | 0.00123 | 0.63533 | 0.01193 | 0.99508 | 0.00111 | 0.99508 | 0.00088 | 0.64905 | 0.0112 | 0.67561 | 0.00971 |
| sqi | als | sq | 0.99951 | 0.00012 | 0.79984 | 0.00523 | **1.0** | 0.0 | 1.0 | 0.0 | 0.88762 | 0.00263 | 0.94382 | 0.00117 |
| srd | srd | - | **0.99951** | 0.0 | | | 0.99901 | 0.0 | 0.99901 | 0.0 | | | | |
| srp | srp | sp | 0.99901 | 0.00025 | 0.99753 | 2e-05 | 0.99951 | 0.00011 | **1.0** | 0.0 | 0.99802 | 1e-05 | 0.99653 | 1e-05 |
| ssw | ssw | - | **0.99455** | 0.00037 | | | 0.99455 | 0.00033 | 0.99455 | 0.00029 | | | | |
| sun | sun | su | 0.9906 | 0.00099 | 0.45112 | 0.02448 | **0.99306** | 0.00033 | 0.99304 | 0.0001 | 0.4952 | 0.02034 | 0.55461 | 0.01544 |
| swa | swh | sw | 0.96611 | 0.00876 | 0.26502 | 0.05802 | 0.97542 | 0.00564 | **0.98635** | 0.00273 | 0.29567 | 0.04969 | 0.35717 | 0.03665 |
| swe | swe | sv | 0.99803 | 0.00049 | 0.96728 | 0.00063 | 0.99951 | 0.00011 | **1.0** | 0.0 | 0.97431 | 0.00048 | 0.97991 | 0.00029 |
| szl | szl | - | **0.99104** | 0.00012 | | | 0.99104 | 0.00011 | 0.99104 | 0.0001 | | | | |
| tam | tam | ta | **1.0** | 0.0 | 0.99303 | 0.0 | 1.0 | 0.0 | 1.0 | 0.0 | 0.99303 | 0.0 | 0.99303 | 0.0 |
| taq | taq | - | 0.80906 | 0.04234 | | | 0.83861 | 0.02168 | **0.84449** | 0.00876 | | | | |
| tat | tat | - | **1.0** | 0.0 | | | 1.0 | 0.0 | 1.0 | 0.0 | | | | |
| tel | tel | te | **1.0** | 0.0 | 0.97881 | 0.0 | 1.0 | 0.0 | 1.0 | 0.0 | 0.97881 | 0.0 | 0.97881 | 0.0 |
| tgk | tgk | tg | **1.0** | 0.0 | 0.99411 | 0.00012 | 1.0 | 0.0 | 1.0 | 0.0 | 0.99754 | 5e-05 | 0.99803 | 3e-05 |
| tgl | tgl | - | **0.99951** | 0.00012 | | | 0.99901 | 0.00011 | 0.99901 | 0.0001 | | | | |
| tha | tha | th | **1.0** | 0.0 | 0.98749 | 0.0 | 1.0 | 0.0 | 1.0 | 0.0 | 0.98749 | 0.0 | 0.98749 | 0.0 |
| tir | tir | - | **0.98851** | 0.0 | | | 0.98851 | 0.0 | 0.988 | 0.0 | | | | |
| tpi | tpi | - | **0.99951** | 0.0 | | | 0.99951 | 0.0 | 0.99901 | 0.0 | | | | |
| tsn | tsn | - | **0.99753** | 0.0 | | | 0.99753 | 0.0001 | 0.99753 | 0.0001 | | | | |
| tso | tso | - | 0.99753 | 0.00049 | | | 0.99753 | 0.00044 | **0.99901** | 0.0001 | | | | |
| tuk | tuk | - | **0.99951** | 0.00012 | | | 0.99951 | 0.00011 | 0.99951 | 0.0001 | | | | |
| tum | tum | - | 0.99852 | 0.00025 | | | **0.99901** | 0.00011 | 0.99901 | 0.0 | | | | |
| tur | tur | tr | **0.99119** | 0.00222 | 0.54852 | 0.01718 | 0.99119 | 0.00199 | 0.99119 | 0.00175 | 0.56606 | 0.01595 | 0.59982 | 0.01354 |
| tzm | tzm | - | 0.94421 | 0.01173 | | | 0.94421 | 0.01051 | **0.94524** | 0.00837 | | | | |
| uig | uig | - | **1.0** | 0.0 | | | 1.0 | 0.0 | 1.0 | 0.0 | | | | |
| ukr | ukr | uk | **0.99951** | 0.00012 | 0.9936 | 0.0001 | 0.99951 | 0.00011 | 0.99951 | 0.0001 | 0.99409 | 9e-05 | 0.99309 | 8e-05 |
| umb | umb | - | **0.88585** | 0.0 | | | 0.8781 | 0.00055 | 0.834 | 0.00029 | | | | |
| urd | urd | ur | 0.98346 | 0.00407 | 0.64759 | 0.0113 | 0.98346 | 0.00365 | **0.99021** | 0.00185 | 0.66866 | 0.01026 | 0.73611 | 0.00724 |
| uzb | uzn | uz | 0.96015 | 0.01037 | 0.63806 | 0.01187 | 0.98396 | 0.00365 | **0.99411** | 0.00117 | 0.70282 | 0.00882 | 0.77099 | 0.00604 |
| vec | vec | - | **0.99703** | 0.0 | | | 0.99703 | 0.0001 | 0.99653 | 0.0 | | | | |
| vie | vie | vi | 0.99951 | 0.00012 | 0.27281 | 0.0559 | **1.0** | 0.0 | 1.0 | 0.0 | 0.30667 | 0.04728 | 0.37036 | 0.03475 |
| war | war | - | **0.99951** | 0.0 | | | 0.99951 | 0.0 | 0.99951 | 0.0 | | | | |
| wol | wol | - | **0.99852** | 0.0 | | | 0.99802 | 0.0 | 0.99802 | 0.0 | | | | |
| xho | xho | xh | 0.98918 | 0.00198 | 0.58763 | 0.0121 | 0.98918 | 0.00177 | **0.99113** | 0.00117 | 0.64438 | 0.00924 | 0.71054 | 0.00591 |
| yid | ydd | yi | **1.0** | 0.0 | 0.99304 | 1e-05 | 1.0 | 0.0 | 1.0 | 0.0 | 0.99304 | 1e-05 | 0.99254 | 1e-05 |
| yor | yor | yo | **0.99455** | 0.00037 | 0.16841 | 0.00856 | 0.99355 | 0.00033 | 0.99053 | 0.0001 | 0.1928 | 0.00562 | 0.21661 | 0.00333 |
| zho | yue/zho | zh-Latn/zh | **1.0** | 0.0 | 0.88912 | 0.00752 | 1.0 | 0.0 | 1.0 | 0.0 | 0.93055 | 0.00435 | 0.9619 | 0.00208 |
| zul | zul | zu | 0.96158 | 0.00185 | 0.41074 | 0.02908 | **0.96353** | 0.00122 | 0.96293 | 0.00088 | 0.4457 | 0.02503 | 0.50587 | 0.01868 |

Table 31: Comparison of GlotLID vs CLD3 on FLORES-200 benchmark (part 2)

| | | | with confidence threshold θ | | | | | | | |
|---|---|---|---|---|---|---|---|---|---|---|
| | | | GlotLID-M | | FT176 | | GlotLID-M θ=.3 | | GlotLID-M θ=.5 | | FT176 θ=.3 | | FT176 θ=.5 | |
| iso639-3 | FLORES Code(s) | FT176 Code(s) | F1↑ | FPR↓ | F1↑ | FPR↓ | F1↑ | FPR↓ | F1↑ | FPR↓ | F1↑ | FPR↓ | F1↑ | FPR↓ |
|---|---|---|---|---|---|---|---|---|---|---|---|---|---|---|
| ace | ace | - | **0.95757** | 0.01253 | | | 0.95732 | 0.0121 | 0.95689 | 0.01 | | | | |
| afr | afr | af | **1.0** | 0.0 | | | 1.0 | 0.0 | 1.0 | 0.0 | | | | |
| aka | aka/twi | - | 0.99876 | 0.00014 | | | **0.99901** | 0.0 | 0.99901 | 0.0 | | | | |
| amh | amh | am | **0.99951** | 0.00014 | 0.66689 | 0.01108 | 0.99951 | 0.00012 | 0.99951 | 0.0 | 0.66689 | 0.01069 | **0.66799** | 0.00991 |
| ara | arz/ars/acm/ary/aeb/acq/apc/arb/ajp | ar/arz | **0.93243** | 0.06899 | 0.78539 | 0.04342 | 0.92725 | 0.0666 | 0.92631 | 0.05753 | 0.7875 | 0.04127 | **0.80111** | 0.03454 |
| asm | asm | as | **1.0** | 0.0 | **0.99553** | 0.0 | 1.0 | 0.0 | 1.0 | 0.0 | 0.99553 | 0.0 | 0.99553 | 0.0 |
| ast | ast | ast | 0.99209 | 0.00113 | 0.47151 | 0.00067 | **0.99308** | 0.00082 | 0.99257 | 0.00062 | **0.47655** | 0.00012 | 0.36055 | 2e-05 |
| awa | awa | - | 0.38951 | 0.00014 | | | **0.38982** | 0.0 | 0.35313 | 0.0 | | | | |
| aym | ayr | - | 0.99655 | 0.00099 | | | **0.99951** | 0.00014 | 0.99951 | 0.00012 | | | | |
| aze | azb/azj | az/azb | 0.71808 | 0.00056 | 0.74843 | 0.0041 | 0.71546 | 0.00041 | 0.69627 | 0.00025 | 0.76746 | 0.00295 | **0.77402** | 0.002 |
| bak | bak | ba | **1.0** | 0.0 | 0.98705 | 5e-05 | 1.0 | 0.0 | 1.0 | 0.0 | **0.98754** | 4e-05 | 0.98751 | 1e-05 |
| bam | bam | - | 0.52563 | 0.12687 | | | 0.52664 | 0.12179 | **0.53084** | 0.10741 | | | | |
| ban | ban | - | 0.97521 | 0.00014 | | | **0.97571** | 0.0 | 0.97467 | 0.0 | | | | |
| bar | - | bar | | | | 4e-05 | | | | | | | 0 | 0 |
| bcl | - | bcl | | | | 0.00018 | | | | | | | 1e-05 | 0 |
| bel | bel | be | **1.0** | 0.0 | **0.99951** | 0.0 | | | | | 0.99951 | 0.0 | 0.99901 | 0.0 |
| bem | bem | - | 0.9906 | 0.00113 | | | **0.99256** | 0.00054 | 0.99256 | 0.00049 | | | | |
| ben | ben | bn | **0.99852** | 0.00014 | 0.66187 | 0.01133 | 0.99852 | 0.00014 | 0.99852 | 0.00012 | **0.6647** | 0.0108 | 0.6647 | 0.01006 |
| bho | bho | - | 0.94329 | 0.01098 | | | 0.94374 | 0.01047 | **0.94846** | 0.00605 | | | | |
| bih | - | bh | | | | 0.00118 | | | | | | 0.00113 | | 0.00054 |
| bod | bod | bo | **0.94589** | 0.00014 | 0.66667 | 0.01109 | 0.94589 | 0.00014 | 0.94589 | 0.00012 | **0.66755** | 0.01066 | 0.66755 | 0.00993 |
| bre | - | br | | | | 0.00278 | | | | | | 0.00022 | | 5e-05 |
| bug | bug | - | **0.99802** | 0.00014 | | | 0.99703 | 0.0 | 0.99404 | 0.0 | | | | |
| bul | bul | bg | **0.99951** | 0.0 | **0.99802** | 2e-05 | 0.99951 | 0.0 | 0.99951 | 0.0 | 0.99802 | 2e-05 | 0.99753 | 1e-05 |
| cat | cat | ca | **1.0** | 0.0 | 0.67245 | 0.01069 | 1.0 | 0.0 | 1.0 | 0.0 | 0.7861 | 0.00574 | **0.87855** | 0.00249 |
| cbk | - | cbk | | | | 0.00035 | | | | | | 3e-05 | | 0 |
| ceb | ceb | ceb | **0.99503** | 0.0 | 0.85532 | 0.00265 | 0.99454 | 0.0 | 0.99404 | 0.0 | 0.8684 | 0.00221 | **0.89669** | 0.00118 |
| ces | ces | cs | **0.99951** | 0.00014 | 0.94535 | 0.00128 | 0.99951 | 0.00014 | 0.99901 | 0.00012 | 0.98015 | 0.00043 | **0.98971** | 0.00019 |
| che | - | ce | | | | 5e-05 | | | | | | 0 | | 0 |
| cjk | cjk | - | **0.84493** | 0.00014 | | | 0.83429 | 0.00014 | 0.79834 | 0.00012 | | | | |
| ckb | ckb | ckb | **1.0** | 0.0 | 0.99803 | 3e-05 | 1.0 | 0.0 | 1.0 | 0.0 | **0.99951** | 0.0 | 0.99951 | 0.0 |
| cor | - | kw | | | | 0.00264 | | | | | | 4e-05 | | 0 |
| cos | - | co | | 0.00042 | | 9e-05 | | 0.00041 | | 0.00025 | | 1e-05 | | 0 |
| crh | crh | - | **0.98851** | 0.0 | | | 0.988 | 0.0 | 0.988 | 0.0 | | | | |
| cym | cym | cy | 0.99951 | 0.00014 | 0.74279 | 0.00755 | 0.99951 | 0.00014 | **1.0** | 0.0 | **0.97985** | 0.00027 | 0.97881 | 0.0 |
| dan | dan | da | 0.9931 | 0.00127 | 0.91718 | 0.00145 | 0.99505 | 0.00054 | **0.99554** | 0.00025 | 0.93032 | 0.00107 | **0.93853** | 0.00049 |
| deu | deu | de | **0.99901** | 0.00014 | 0.63409 | 0.01279 | 0.99901 | 0.0 | 0.99852 | 0.0 | 0.96935 | 0.00068 | **0.98732** | 0.00026 |
| din | din | dik | **0.99802** | 0.00014 | | | 0.99802 | 0.0 | 0.99752 | 0.0 | | | | |
| diq | - | diq | | 0.00028 | | 0.00026 | | 0.00027 | | 0.00025 | | 0 | | 0 |
| dsb | - | dsb | | | | 2e-05 | | | | | | 0 | | 0 |
| dyu | dyu | - | **0.12435** | 0.03591 | | | 0.11878 | 0.03466 | 0.11186 | 0.03136 | | | | |
| dzo | dzo | - | **0.9496** | 0.0145 | | | 0.9491 | 0.014 | 0.9491 | 0.01272 | | | | |
| ell | ell | el | **1.0** | 0.0 | 0.99167 | 0.00019 | 1.0 | 0.0 | 1.0 | 0.0 | **1.0** | 0.0 | 1.0 | 0.0 |
| eml | - | eml | | 0.00113 | | 0.00104 | | 0.00082 | | 0.00062 | | 0.0001 | | 0 |
| eng | eng | en | 0.9878 | 0.00352 | 0.09376 | 0.21432 | 0.99215 | 0.00204 | **0.99556** | 0.00074 | 0.42352 | 0.02914 | **0.82545** | 0.00422 |
| epo | epo | eo | 0.99901 | 0.00028 | 0.58053 | 0.01599 | 0.99951 | 0.0 | **1.0** | 0.0 | 0.96148 | 0.00085 | **0.99606** | 6e-05 |
| est | est | et | 0.99852 | 0.00042 | 0.86428 | 0.00335 | **0.99951** | 0.00014 | 0.99951 | 0.00012 | 0.97997 | 0.00034 | **0.98756** | 5e-05 |
| eus | eus | eu | **0.99951** | 0.0 | 0.8848 | 0.00286 | 0.99951 | 0.0 | 0.99951 | 0.0 | 0.98531 | 0.00025 | **0.98905** | 4e-05 |
| ewe | ewe | - | **1.0** | 0.0 | | | 1.0 | 0.0 | 1.0 | 0.0 | | | | |
| fao | fao | - | **0.99951** | 0.0 | | | 0.99951 | 0.0 | 0.99951 | 0.0 | | | | |
| fas | pes/prs | fa | 0.79937 | 0.14306 | 0.85221 | 0.00769 | 0.80032 | 0.13728 | **0.82343** | 0.10716 | 0.85491 | 0.00727 | **0.87129** | 0.00589 |
| fij | fij | - | **0.99901** | 0.0 | | | 0.99901 | 0.0 | 0.99901 | 0.0 | | | | |
| fin | fin | fi | 0.99951 | 0.00014 | 0.46242 | 0.02578 | **1.0** | 0.0 | 1.0 | 0.0 | 0.69102 | 0.00957 | **0.85981** | 0.00325 |
| fon | fon | - | **0.99752** | 0.0 | | | 0.99752 | 0.0 | 0.99703 | 0.0 | | | | |
| fra | fra | fr | **0.99951** | 0.00014 | 0.49378 | 0.02273 | 0.99951 | 0.0 | 0.99852 | 0.00012 | 0.81844 | 0.00475 | **0.94619** | 0.00112 |
| frr | - | frr | | 0.00042 | | 4e-05 | | 0.00041 | | 0.00025 | | 1e-05 | | 0 |
| fry | - | fy | | | | 0.00022 | | | | | | 1e-05 | | 0 |
| fur | fur | - | 0.99951 | 0.00014 | | | 0.99951 | 0.00014 | **1.0** | 0.0 | | | | |
| fuv | fuv | - | **0.96738** | 0.00014 | | | 0.96099 | 0.0 | 0.94693 | 0.0 | | | | |
| gla | gla | gd | 0.99951 | 0.00014 | 0.95846 | 0.00018 | **1.0** | 0.0 | 1.0 | 0.0 | 0.91609 | 2e-05 | 0.71818 | 0.0 |
| gle | gle | ga | **1.0** | 0.0 | 0.85702 | 0.00354 | 1.0 | 0.0 | 1.0 | 0.0 | 0.88839 | 0.00231 | **0.90054** | 0.00098 |
| glg | glg | gl | **0.99703** | 0.00028 | 0.889 | 0.00081 | 0.99703 | 0.00027 | 0.99703 | 0.00012 | **0.91426** | 0.00021 | 0.88375 | 1e-05 |
| glv | - | gv | | | | 0.00018 | | | | | | 2e-05 | | 0 |
| gom | - | gom | | | | 0.00019 | | | | | | 0 | | 0 |
| grn | grn | gn | **1.0** | 0.0 | **0.77784** | 0.00064 | 1.0 | 0.0 | 1.0 | 0.0 | 0.60893 | 0.0 | 0.2891 | 0.0 |
| gsw | - | als | | 0.00028 | | 0.00042 | | 0.00027 | | 0.00025 | | 5e-05 | | 0 |
| guj | guj | gu | **1.0** | 0.0 | 0.99754 | 5e-05 | 1.0 | 0.0 | 1.0 | 0.0 | **1.0** | 0.0 | 1.0 | 0.0 |
| hat | hat | ht | 0.99852 | 0.00042 | 0.48932 | 0.0017 | 0.99852 | 0.00041 | **0.99901** | 0.0 | 0.22877 | 0.00011 | 0.05374 | 2e-05 |
| hau | hau | - | 0.95924 | 0.01211 | | | 0.98348 | 0.00462 | **0.99313** | 0.00173 | | | | |
| hbs | srp/bos/hrv | bs/hr/sh/sr | 0.99868 | 0.00099 | 0.80096 | 0.01623 | 0.99901 | 0.00068 | **0.99984** | 0.0 | **0.97202** | 0.00092 | 0.77693 | 7e-05 |
| heb | heb | he | **1.0** | 0.0 | **1.0** | 0.0 | 1.0 | 0.0 | 1.0 | 0.0 | 1.0 | 0.0 | 1.0 | 0.0 |
| hif | - | hif | | 0.00084 | | 2e-05 | | 0.00068 | | 0.00049 | | 0 | | 0 |
| hin | hin | hi | 0.67624 | 0.13644 | 0.2581 | 0.06373 | 0.67692 | 0.1313 | **0.69697** | 0.10864 | 0.25813 | 0.06152 | **0.26337** | 0.05577 |
| hne | hne | - | 0.90296 | 0.00634 | | | **0.90343** | 0.00598 | 0.898 | 0.00444 | | | | |
| hsb | - | hsb | | | | 7e-05 | | | | | | 1e-05 | | 0 |
| hun | hun | hu | **1.0** | 0.0 | 0.88772 | 0.0028 | 1.0 | 0.0 | 0.99951 | 0.0 | 0.98828 | 0.00025 | **0.99704** | 5e-05 |
| hye | hye | hy | **1.0** | 0.0 | 0.99754 | 5e-05 | 1.0 | 0.0 | 1.0 | 0.0 | **1.0** | 0.0 | 1.0 | 0.0 |
| ibo | ibo | - | 0.99951 | 0.00014 | | | **1.0** | 0.0 | 1.0 | 0.0 | | | | |
| ido | - | io | | | | 0.003 | | | | | | 0.00015 | | 2e-05 |
| ile | - | ie | | | | 0.00022 | | | | | | 0 | | 0 |
| ilo | ilo | ilo | **0.99951** | 0.00014 | 0.79124 | 0.00553 | 0.99951 | 0.00014 | 0.99951 | 0.00012 | **0.91814** | 0.00125 | 0.88235 | 0.00014 |
| ina | - | ia | | | | 0.00045 | | | | | | 4e-05 | | 0 |
| isl | isl | is | 0.99901 | 0.00028 | 0.69613 | 0.0096 | **0.99951** | 0.00014 | 0.99951 | 0.00012 | 0.70219 | 0.009 | **0.72313** | 0.00071 |
| ita | ita | it | 0.99803 | 0.00042 | 0.3149 | 0.04818 | 0.99852 | 0.00027 | **0.99901** | 0.00012 | 0.46925 | 0.02418 | **0.63128** | 0.01162 |
| jav | jav | jv | 0.98442 | 0.00436 | **0.70892** | 0.00342 | 0.99166 | 0.00217 | **0.99213** | 0.0016 | 0.69871 | 0.00047 | 0.39526 | 3e-05 |
| jbo | - | jbo | | | | 0.00049 | | | | | | 5e-05 | | 0 |
| jpn | jpn | ja | 0.99951 | 0.0 | 0.86607 | 0.00343 | 0.99951 | 0.0 | 0.99951 | 0.0 | 0.99803 | 4e-05 | **1.0** | 0.0 |
| kab | kab | - | 0.86127 | 0.02971 | | | 0.87886 | 0.02284 | **0.90909** | 0.0121 | | | | |
| kac | kac | - | **1.0** | 0.0 | | | 1.0 | 0.0 | 1.0 | 0.0 | | | | |
| kam | kam | - | **0.91658** | 0.00014 | | | 0.91183 | 0.0 | 0.87368 | 0.0 | | | | |
| kan | kan | kn | **1.0** | 0.0 | 0.92632 | 0.00176 | 1.0 | 0.0 | 1.0 | 0.0 | 0.99951 | 1e-05 | **1.0** | 0.0 |
| kas | kas | - | **0.97649** | 0.0 | | | 0.97597 | 0.0 | 0.96945 | 0.0 | | | | |
| kat | kat | ka | **1.0** | 0.0 | **1.0** | 0.0 | 1.0 | 0.0 | 1.0 | 0.0 | 1.0 | 0.0 | 1.0 | 0.0 |
| kaz | kaz | kk | **0.99951** | 0.0 | 0.99655 | 7e-05 | 0.99951 | 0.0 | 0.99951 | 0.0 | 0.99704 | 5e-05 | **0.99753** | 3e-05 |
| kbp | kbp | - | **0.99901** | 0.00014 | | | 0.99901 | 0.00014 | 0.99901 | 0.00012 | | | | |
| kea | kea | - | **0.95238** | 0.0 | | | 0.9513 | 0.0 | 0.93586 | 0.0 | | | | |
| khm | khm | km | **0.99951** | 0.0 | **1.0** | 0.0 | 0.99951 | 0.0 | 0.99951 | 0.0 | 1.0 | 0.0 | 0.99951 | 0.0 |
| kik | kik | - | **0.96562** | 0.01 | | | 0.96509 | 0.00951 | 0.96456 | 0.00852 | | | | |
| kin | kin | - | **0.91471** | 0.00084 | | | 0.91471 | 0.00082 | 0.91471 | 0.00074 | | | | |
| kir | kir | ky | **1.0** | 0.0 | **0.99752** | 0.0 | 1.0 | 0.0 | 1.0 | 0.0 | 0.99752 | 0.0 | 0.99603 | 0.0 |
| kmb | kmb | - | 0.96321 | 0.01028 | | | 0.96923 | 0.00816 | **0.97713** | 0.00481 | | | | |
| kmr | kmr | ku | **0.99901** | 0.0 | 0.94046 | 0.00129 | 0.99901 | 0.0 | 0.99901 | 0.0 | **0.98717** | 0.00015 | 0.98193 | 2e-05 |
| knc | knc | - | 0.86459 | 0.0031 | | | 0.86869 | 0.00068 | **0.86966** | 0.00068 | | | | |
| kom | - | kv | | | | 0.00027 | | | | | | 0 | | 0 |
| kon | kon | - | 0.99359 | 0.00127 | | | 0.99408 | 0.00109 | **0.99703** | 0.00025 | | | | |
| kor | kor | ko | **1.0** | 0.0 | 0.99901 | 2e-05 | 1.0 | 0.0 | 1.0 | 0.0 | **1.0** | 0.0 | 1.0 | 0.0 |
| lao | lao | lo | **1.0** | 0.0 | 0.99362 | 0.00014 | 0.99901 | 0.0 | 0.99802 | 0.0 | **0.99802** | 0.0 | 0.99653 | 0.0 |
| lat | - | la | | | | 0.0042 | | | | | | 0.00017 | | 2e-05 |
| lav | lvs/ltg | lv | 0.99926 | 0.00014 | 0.87141 | 0.00369 | **0.99951** | 0.0 | 0.99951 | 0.0 | **0.90388** | 0.00145 | 0.87915 | 0.00034 |
| lez | - | lez | | | | 1e-05 | | | | | | 0 | | 0 |
| lij | lij | - | **0.99901** | 0.0 | | | 0.99852 | 0.00014 | 0.99753 | 0.00012 | | | | |
| lim | lim | li | **0.99253** | 0.0 | **0.31** | 2e-05 | 0.99253 | 0.0 | 0.99153 | 0.0 | 0.20408 | 0.0 | 0.03876 | 0.0 |
| lin | lin | - | **0.99901** | 0.00028 | | | 0.99852 | 0.00027 | 0.99901 | 0.00012 | | | | |
| lit | lit | lt | **0.99951** | 0.0 | 0.85921 | 0.0036 | 0.99951 | 0.0 | 0.99951 | 0.0 | 0.94393 | 0.00125 | **0.97632** | 0.00046 |
| lmo | lmo | lmo | **0.99554** | 0.00028 | **0.27942** | 0.00133 | 0.99554 | 0.00027 | 0.99504 | 0.00012 | 0.11296 | 7e-05 | 0.01763 | 0.0 |
| lrc | - | lrc | | | | 1e-05 | | | | | | 0 | | 0 |

Table 32: Comparison of GlotLID vs FT176 on FLORES-200 benchmark (part 1)

| | | | GlotLID-M | | FT176 | | with confidence threshold $\theta$ | | | | | | | |
| | | | | | | | GlotLID-M $\theta=.3$ | | GlotLID-M $\theta=.5$ | | FT176 $\theta=.3$ | | FT176 $\theta=.5$ | |
| iso639-3 | FLORES Code(s) | FT176 Code(s) | F1↑ | FPR↓ | F1↑ | FPR↓ | F1↑ | FPR↓ | F1↑ | FPR↓ | F1↑ | FPR↓ | F1↑ | FPR↓ |
|---|---|---|---|---|---|---|---|---|---|---|---|---|---|---|
| ltz | ltz | lb | **0.99951** | 0.0 | 0.95608 | 0.00011 | 0.99901 | 0.0 | 0.99901 | 0.0 | 0.94966 | 0.0 | 0.89035 | 0.0 |
| lua | lua | - | **0.99653** | 0.00014 | | | 0.99553 | 0.0 | 0.99404 | 0.0 | | | | |
| lug | lug | - | **0.99603** | 0.0 | | | 0.99603 | 0.0 | 0.99454 | 0.0 | | | | |
| luo | luo | - | **1.0** | 0.0 | | | 1.0 | 0.0 | 0.99951 | 0.0 | | | | |
| lus | lus | - | 0.99653 | 0.00014 | | | **0.99703** | 0.0 | 0.99653 | 0.0 | | | | |
| mag | mag | - | 0.95459 | 0.00338 | | | **0.95507** | 0.00313 | 0.95408 | 0.0016 | | | | |
| mai | mai | mai | **0.97366** | 0.00014 | 0.32838 | 1e-05 | 0.97366 | 0.00014 | 0.97102 | 0.0 | 0.32838 | 1e-05 | 0.16998 | 0.0 |
| mal | mal | ml | **1.0** | 0.0 | 0.97683 | 0.00053 | 1.0 | 0.0 | 1.0 | 0.0 | 0.99951 | 1e-05 | **1.0** | 0.0 |
| mar | mar | mr | **1.0** | 0.0 | 0.9907 | 0.00021 | 1.0 | 0.0 | 1.0 | 0.0 | 0.9907 | 0.0002 | 0.99655 | 7e-05 |
| mkd | mkd | mk | **1.0** | 0.0 | 0.99556 | 7e-05 | 1.0 | 0.0 | 1.0 | 0.0 | 0.99654 | 4e-05 | 0.99654 | 4e-05 |
| mlg | plt | mg | **0.99951** | 0.00014 | 0.79112 | 0.00055 | 0.99951 | 0.00014 | 0.99951 | 0.00012 | 0.51211 | 2e-05 | 0.20885 | 0.0 |
| mlt | mlt | mt | 0.97778 | 0.00648 | 0.91161 | 0.00139 | 0.99216 | 0.00217 | **0.99803** | 0.00049 | 0.90527 | 5e-05 | 0.75599 | 0.0 |
| mni | mni | - | **0.99901** | 0.00014 | | | 0.99901 | 0.00014 | 0.99901 | 0.00012 | | | | |
| mon | khk | mn | **1.0** | 0.0 | 0.99951 | 1e-05 | 1.0 | 0.0 | 0.99951 | 0.0 | **1.0** | 0.0 | 0.99951 | 0.0 |
| mos | mos | - | **0.98138** | 0.0 | | | 0.97415 | 0.0 | 0.96418 | 0.0 | | | | |
| mri | mri | - | 0.99901 | 0.00014 | | | **0.99951** | 0.0 | 0.99901 | 0.0 | | | | |
| mrj | | mrj | | | | 3e-05 | | | | | | 0 | | 0 |
| msa | min/ind/bjn/zsm | id/min/ms | 0.97122 | 0.02591 | 0.52247 | 0.05449 | **0.97515** | 0.01821 | 0.97478 | 0.01333 | 0.65691 | 0.01246 | 0.57549 | 0.00236 |
| mwl | - | mwl | | | | 3e-05 | | | | | | 0 | | 0 |
| mya | mya | my | **1.0** | 0.0 | 0.66689 | 0.01108 | 1.0 | 0.0 | 1.0 | 0.0 | 0.66755 | 0.01066 | 0.6702 | 0.00981 |
| mzn | - | mzn | | 0.00197 | | 0.00019 | | 0.00177 | | 0.00062 | | 6e-05 | | 1e-05 |
| nah | - | nah | | | | 5e-05 | | | | | | 0 | | 0 |
| nap | - | nap | | | | 0.00026 | | | | | | 1e-05 | | 1e-05 |
| nds | - | nds | | | | 0.00016 | | | | | | 7e-05 | | 3e-05 |
| nep | npi | ne | 0.99852 | 0.00028 | 0.98671 | 0.00019 | 0.99852 | 0.00027 | **0.99951** | 0.0 | 0.98768 | 0.00016 | 0.9901 | 8e-05 |
| nld | nld | nl | 0.99803 | 0.00056 | 0.60661 | 0.01433 | **0.99901** | 0.00027 | 0.99901 | 0.00012 | 0.69464 | 0.00937 | 0.77509 | 0.00572 |
| nno | nno | nn | 0.98507 | 0.00113 | 0.65506 | 0.00034 | **0.98606** | 0.00082 | 0.986 | 0.00025 | 0.66016 | 0.00018 | 0.59654 | 2e-05 |
| nob | nob | no | **0.98185** | 0.00366 | 0.71096 | 0.00772 | 0.95835 | 0.00353 | 0.89931 | 0.00259 | 0.74733 | 0.00605 | 0.76308 | 0.00435 |
| nso | nso | - | **0.99704** | 0.0007 | | | 0.99704 | 0.00068 | 0.99655 | 0.00062 | | | | |
| nus | nus | - | **0.99951** | 0.0 | | | 0.99951 | 0.0 | 0.99951 | 0.0 | | | | |
| nya | nya | - | 0.99753 | 0.00056 | | | 0.99803 | 0.00041 | **0.99852** | 0.00012 | | | | |
| oci | oci | oc | 0.99951 | 0.00014 | 0.70333 | 0.0017 | **1.0** | 0.0 | 0.99951 | 0.0 | 0.73798 | 0.0004 | 0.63818 | 8e-05 |
| ori | ory | or | **1.0** | 0.0 | **1.0** | 0.0 | 1.0 | 0.0 | 1.0 | 0.0 | 1.0 | 0.0 | 1.0 | 0.0 |
| orm | gaz | - | 0.99167 | 0.00239 | | | 0.99655 | 0.00095 | **0.99803** | 0.00049 | | | | |
| pag | pag | - | **0.99852** | 0.0 | | | 0.99852 | 0.0 | 0.99852 | 0.0 | | | | |
| pam | - | pam | | | | 0.00031 | | | | | | 2e-05 | | 0 |
| pan | pan | pa | **1.0** | 0.0 | 0.99951 | 1e-05 | 1.0 | 0.0 | 1.0 | 0.0 | **1.0** | 0.0 | 1.0 | 0.0 |
| pap | pap | - | 0.99069 | 0.00253 | | | 0.99118 | 0.00231 | **0.99557** | 0.00099 | | | | |
| pfl | - | pfl | | | | 2e-05 | | | | | | 0 | | 0 |
| pms | - | pms | | | | 0.00162 | | | | | | 0.00042 | | 0.00012 |
| pnb | - | pnb | | 0.00253 | | 0.0026 | | 0.00245 | | 0.00136 | | 0.00246 | | 0.00145 |
| pol | pol | pl | 0.9907 | 0.00268 | 0.49198 | 0.0229 | **0.99167** | 0.00231 | 0.99167 | 0.0021 | 0.6529 | 0.01138 | 0.66601 | 0.01 |
| por | por | pt | 0.99655 | 0.00099 | 0.62192 | 0.01343 | **0.99704** | 0.00082 | 0.99704 | 0.00062 | 0.81947 | 0.00469 | 0.92062 | 0.00168 |
| pus | pbt | ps | 0.88805 | 0.00225 | 0.9861 | 0.0001 | 0.88853 | 0.00204 | 0.89142 | 0.00111 | **0.98757** | 6e-05 | 0.98295 | 2e-05 |
| que | quy | qu | 0.99901 | 0.00014 | 0.78925 | 0.00273 | **0.99951** | 0.0 | 0.99901 | 0.0 | 0.73444 | 0.00016 | 0.38209 | 0.0 |
| roh | - | rm | | | | 0.00138 | | | | | | 0.0001 | | 1e-05 |
| ron | ron | ro | **0.99951** | 0.0 | 0.75522 | 0.00719 | 0.99951 | 0.0 | 0.99951 | 0.0 | 0.93791 | 0.00142 | 0.9902 | 0.00018 |
| run | run | - | 0.92541 | 0.02182 | | | 0.92584 | 0.02093 | **0.92627** | 0.01889 | | | | |
| rus | rus | ru | **0.99901** | 0.00014 | 0.91501 | 0.00206 | 0.99901 | 0.00014 | 0.99901 | 0.00012 | 0.99216 | 0.00017 | 0.99557 | 9e-05 |
| sag | sag | - | **0.99901** | 0.0 | | | 0.99901 | 0.0 | 0.99901 | 0.0 | | | | |
| san | san | sa | 0.99104 | 0.00014 | 0.95791 | 3e-05 | 0.99104 | 0.00014 | 0.99103 | 0.0 | 0.95791 | 3e-05 | 0.95189 | 1e-05 |
| sat | sat | - | **1.0** | 0.0 | | | 1.0 | 0.0 | 1.0 | 0.0 | | | | |
| scn | scn | scn | 0.99802 | 0.00014 | 0.48256 | 0.00059 | 0.99802 | 0.00014 | **0.99852** | 0.0 | 0.33826 | 0.0 | 0.12419 | 0.0 |
| sco | - | sco | | 0.0007 | | 0.00024 | | 0.00041 | | 0.00025 | | 1e-05 | | 0 |
| shn | shn | - | **1.0** | 0.0 | | | 1.0 | 0.0 | 1.0 | 0.0 | | | | |
| sin | sin | si | **1.0** | 0.0 | 0.99951 | 1e-05 | 1.0 | 0.0 | 1.0 | 0.0 | **1.0** | 0.0 | 1.0 | 0.0 |
| slk | slk | sk | 0.99852 | 0.00014 | 0.97443 | 0.00034 | 0.99852 | 0.00014 | **0.99901** | 0.0 | 0.98753 | 3e-05 | 0.9824 | 0.0 |
| slv | slv | sl | 0.99508 | 0.00141 | 0.6687 | 0.01048 | 0.99655 | 0.00095 | **0.99951** | 0.00012 | 0.94732 | 0.00081 | 0.93215 | 0.00011 |
| smo | smo | - | **0.99603** | 0.0 | | | 0.99603 | 0.0 | 0.99603 | 0.0 | | | | |
| sna | sna | - | 0.99951 | 0.00014 | | | 0.99951 | 0.00014 | **1.0** | 0.0 | | | | |
| snd | snd | sd | 0.99606 | 0.00113 | 0.98623 | 0.00021 | 0.99606 | 0.00109 | **0.99704** | 0.00074 | 0.98964 | 0.00013 | 0.99405 | 3e-05 |
| som | som | so | 0.97028 | 0.00873 | 0.56777 | 0.00055 | 0.98973 | 0.00285 | **0.99803** | 0.00049 | 0.10467 | 2e-05 | 0.00197 | 0.0 |
| sot | sot | - | **1.0** | 0.0 | | | 1.0 | 0.0 | 1.0 | 0.0 | | | | |
| spa | spa | es | **0.99508** | 0.00141 | 0.30413 | 0.05073 | 0.99508 | 0.00136 | 0.99508 | 0.00111 | 0.57615 | 0.01575 | 0.72832 | 0.00744 |
| sqi | als | sq | 0.99901 | 0.00028 | 0.82767 | 0.0046 | **1.0** | 0.0 | 1.0 | 0.0 | 0.92795 | 0.00165 | 0.95549 | 0.0009 |
| srd | srd | sc | **0.99951** | 0.0 | 0.02822 | 0.00039 | 0.99901 | 0.0 | 0.99901 | 0.0 | 0.0 | 2e-05 | 0.0 | 0.0 |
| ssw | ssw | - | **0.99455** | 0.00042 | | | 0.99455 | 0.00041 | 0.99455 | 0.00037 | | | | |
| sun | sun | su | 0.9906 | 0.00113 | 0.51105 | 0.00995 | **0.99306** | 0.00041 | 0.99304 | 0.00012 | 0.65131 | 0.00044 | 0.41875 | 0.0 |
| swa | swh | sw | 0.96611 | 0.01 | 0.27392 | 0.05511 | 0.97542 | 0.00693 | **0.98635** | 0.00346 | 0.53494 | 0.0138 | 0.62159 | 0.00248 |
| swe | swe | sv | 0.99803 | 0.00056 | 0.95652 | 0.00101 | 0.99951 | 0.00014 | **1.0** | 0.0 | 0.98538 | 0.00031 | 0.99357 | 6e-05 |
| szl | szl | - | **0.99104** | 0.00014 | | | 0.99104 | 0.00014 | 0.99104 | 0.00012 | | | | |
| tam | tam | ta | **1.0** | 0.0 | 0.99557 | 0.0001 | 1.0 | 0.0 | 1.0 | 0.0 | **1.0** | 0.0 | 1.0 | 0.0 |
| taq | taq | - | 0.80906 | 0.0483 | | | 0.83861 | 0.02664 | **0.84449** | 0.01111 | | | | |
| tat | tat | tt | **1.0** | 0.0 | 0.98583 | 0.00028 | 1.0 | 0.0 | 1.0 | 0.0 | 0.98922 | 0.0002 | 0.98917 | 0.00015 |
| tel | tel | te | **1.0** | 0.0 | 0.99901 | 2e-05 | 1.0 | 0.0 | 1.0 | 0.0 | **1.0** | 0.0 | 1.0 | 0.0 |
| tgk | tgk | tg | **1.0** | 0.0 | 0.99556 | 7e-05 | 1.0 | 0.0 | 1.0 | 0.0 | 0.99802 | 1e-05 | 0.99752 | 0.0 |
| tgl | tgl | tl | **0.99951** | 0.00014 | 0.27747 | 0.0576 | 0.99901 | 0.00014 | 0.99901 | 0.00012 | 0.77129 | 0.00631 | 0.94112 | 0.00119 |
| tha | tha | th | **1.0** | 0.0 | 0.99754 | 5e-05 | 1.0 | 0.0 | 1.0 | 0.0 | **1.0** | 0.0 | 1.0 | 0.0 |
| tir | tir | - | **0.98851** | 0.0 | | | 0.98851 | 0.0 | 0.988 | 0.0 | | | | |
| tpi | tpi | - | **0.99951** | 0.0 | | | 0.99951 | 0.0 | 0.99901 | 0.0 | | | | |
| tsn | tsn | - | **0.99753** | 0.00014 | | | 0.99753 | 0.00014 | 0.99753 | 0.00012 | | | | |
| tso | tso | - | 0.99753 | 0.00056 | | | 0.99753 | 0.00054 | **0.99901** | 0.00012 | | | | |
| tuk | tuk | tk | **0.99951** | 0.00014 | 0.92876 | 3e-05 | 0.99951 | 0.0 | 0.99951 | 0.00012 | 0.91416 | 0.0 | 0.82394 | 0.0 |
| tum | tum | - | 0.99852 | 0.00028 | | | **0.99901** | 0.00014 | 0.99901 | 0.0 | | | | |
| tur | tur | tr | **0.99119** | 0.00253 | 0.56113 | 0.01734 | 0.99119 | 0.00245 | 0.99119 | 0.00222 | 0.64768 | 0.01164 | 0.67692 | 0.00952 |
| tzm | tzm | - | 0.94421 | 0.01338 | | | 0.94421 | 0.01291 | **0.94524** | 0.01062 | | | | |
| uig | uig | ug | **1.0** | 0.0 | 0.99901 | 2e-05 | 1.0 | 0.0 | 1.0 | 0.0 | **1.0** | 0.0 | 1.0 | 0.0 |
| ukr | ukr | uk | **0.99951** | 0.00014 | 0.96657 | 0.00077 | 0.99951 | 0.00014 | 0.99951 | 0.00012 | **1.0** | 0.0 | 1.0 | 0.0 |
| umb | umb | - | **0.88585** | 0.00113 | | | 0.8781 | 0.00068 | 0.834 | 0.00037 | | | | |
| urd | urd | ur | 0.98346 | 0.00465 | 0.69461 | 0.00973 | 0.98346 | 0.00449 | **0.99021** | 0.00235 | 0.73985 | 0.00751 | 0.77087 | 0.00591 |
| uzb | uzn | uz | 0.96015 | 0.01183 | 0.88813 | 0.00144 | 0.98396 | 0.00449 | **0.99411** | 0.00148 | 0.87171 | 0.00018 | 0.64847 | 3e-05 |
| vec | vec | vec | **0.99703** | 0.00014 | 0.3007 | 0.0008 | 0.99703 | 0.00014 | 0.99653 | 0.0 | 0.23345 | 2e-05 | 0.07054 | 0.0 |
| vep | - | vep | | | | 5e-05 | | | | | | 3e-05 | | 2e-05 |
| vie | vie | vi | 0.99951 | 0.00014 | 0.81449 | 0.00505 | **1.0** | 0.0 | 1.0 | 0.0 | 0.92126 | 0.00183 | 0.97074 | 0.0006 |
| vls | - | vls | | 0.00056 | | 1e-05 | | 0.00054 | | 0.00037 | | 0 | | 0 |
| vol | - | vo | | | | 7e-05 | | | | | | 0 | | 0 |
| war | war | war | **0.99951** | 0.0 | 0.56436 | 0.00767 | 0.99951 | 0.0 | 0.99951 | 0.0 | 0.7045 | 0.0002 | 0.65677 | 0.00029 |
| wln | - | wa | | | | 0.00062 | | | | | | 3e-05 | | 0 |
| wol | wol | - | **0.99852** | 0.0 | | | 0.99802 | 0.0 | 0.99802 | 0.0 | | | | |
| xal | - | xal | | | | 2e-05 | | | | | | 0 | | 0 |
| xho | xho | - | 0.98918 | 0.00225 | | | 0.98918 | 0.00217 | **0.99113** | 0.00148 | | | | |
| yid | ydd | yi | **1.0** | 0.0 | **1.0** | 0.0 | 1.0 | 0.0 | 1.0 | 0.0 | 1.0 | 0.0 | 1.0 | 0.0 |
| yor | yor | yo | **0.99455** | 0.00042 | 0.70831 | 0.00024 | 0.99355 | 0.00041 | 0.99053 | 0.00012 | 0.67366 | 0.0 | 0.5491 | 0.0 |
| zho | yue/zho | zh/wuu/yue | **1.0** | 0.0 | 0.82283 | 0.01431 | 1.0 | 0.0 | 1.0 | 0.0 | 0.9762 | 0.00155 | 0.99835 | 8e-05 |
| zul | zul | - | 0.96206 | 0.00197 | | | **0.96353** | 0.0015 | 0.96293 | 0.00111 | | | | |

Table 33: Comparison of GlotLID vs FT176 on FLORES-200 benchmark (part 2)

| | | | with confidence threshold $\theta$ | | | | | | | | | | | |
| | | | GlotLID-M | | OpenLID | | GlotLID-M $\theta$=.3 | | GlotLID-M $\theta$=.5 | | OpenLID $\theta$=.3 | | OpenLID $\theta$=.5 | |
| iso639-3 | FLORES Code(s) | OpenLID Code(s) | F1↑ | FPR↓ | F1↑ | FPR↓ | F1↑ | FPR↓ | F1↑ | FPR↓ | F1↑ | FPR↓ | F1↑ | FPR↓ |
|---|---|---|---|---|---|---|---|---|---|---|---|---|---|---|
| ace | ace | ace | 0.95503 | 0.00579 | 0.96012 | 0.00734 | 0.955 | 0.00426 | 0.95689 | 0.00299 | 0.96012 | 0.00734 | 0.96099 | 0.0067 |
| acm | acm | acm | 0.01562 | 0.00023 | 0.03279 | 0.00051 | 0.00784 | 0.00017 | 0.00393 | 0.00011 | 0.03282 | 0.00051 | 0.02713 | 0.00038 |
| acq | acq | acq | 0.00197 | 6e-05 | 0.00197 | 0.0 | 0.00197 | 4e-05 | 0.00197 | 0.0 | 0.00197 | 0.0 | 0.00197 | 0.0 |
| aeb | aeb | aeb | 0.28501 | 0.00199 | 0.33982 | 0.00624 | 0.20348 | 0.00089 | 0.15064 | 0.00026 | 0.34333 | 0.00624 | 0.32223 | 0.00444 |
| afr | afr | afr | 1.0 | 0.0 | 0.99951 | 0.0 | 1.0 | 0.0 | 1.0 | 0.0 | 0.99951 | 0.0 | 0.99951 | 0.0 |
| ajp | ajp | ajp | 0.10836 | 0.00102 | 0.19064 | 0.00206 | 0.09328 | 0.00042 | 0.05369 | 0.00011 | 0.18924 | 0.00206 | 0.14808 | 0.00163 |
| aka | aka/twi | twi | 0.99852 | 0.0 | 0.99852 | 0.0 | 0.99852 | 0.0 | 0.99852 | 0.0 | 0.99852 | 0.0 | 0.99802 | 0.0 |
| als | als | als | 0.99852 | 0.00011 | 0.99951 | 6e-05 | 0.67539 | 0.0 | 0.44785 | 0.0 | 1.0 | 6e-05 | 1.0 | 0.0 |
| amh | amh | amh | 0.99951 | 6e-05 | 0.99951 | 6e-05 | 0.99951 | 4e-05 | 0.99951 | 4e-05 | 0.99951 | 6e-05 | 0.99951 | 6e-05 |
| apc | apc | apc | 0.17857 | 0.00915 | 0.23324 | 0.01287 | 0.13223 | 0.00498 | 0.09179 | 0.00255 | 0.23229 | 0.01287 | 0.21749 | 0.01001 |
| arb | arb | arb | 0.25705 | 0.21515 | 0.24409 | 0.14709 | 0.12788 | 0.06529 | 0.09585 | 0.03457 | 0.24413 | 0.14709 | 0.24699 | 0.13319 |
| ars | ars | ars | 0.00894 | 0.00574 | 0.01843 | 0.0179 | 0.00377 | 0.00202 | 0.00387 | 0.00074 | 0.01846 | 0.0179 | 0.01784 | 0.01314 |
| ary | ary | ary | 0.56588 | 0.05164 | 0.48937 | 0.09958 | 0.69151 | 0.00578 | 0.71835 | 0.00218 | 0.50306 | 0.09958 | 0.50306 | 0.08957 |
| arz | arz | arz | 0.46309 | 0.09806 | 0.42345 | 0.14168 | 0.57422 | 0.03125 | 0.58652 | 0.01647 | 0.42387 | 0.14168 | 0.43993 | 0.12643 |
| asm | asm | asm | 1.0 | 0.0 | 1.0 | 0.0 | 1.0 | 0.0 | 1.0 | 0.0 | 1.0 | 0.0 | 1.0 | 0.0 |
| ast | ast | ast | 0.9916 | 0.00051 | 0.99308 | 0.00058 | 0.99308 | 0.00025 | 0.99257 | 0.00018 | 0.99011 | 0.00058 | 0.99009 | 0.00044 |
| awa | awa | awa | 0.38951 | 6e-05 | 0.67704 | 0.00051 | 0.38982 | 0.0 | 0.35313 | 0.0 | 0.67704 | 0.00051 | 0.64761 | 0.00031 |
| ayr | ayr | ayr | 0.99557 | 0.00045 | 0.99852 | 0.00019 | 0.96738 | 4e-05 | 0.93193 | 0.0 | 0.99852 | 0.00019 | 0.99951 | 6e-05 |
| azb | azb | azb | 0.36583 | 0.00011 | 0.75139 | 0.0 | 0.25065 | 0.0 | 0.13112 | 0.0 | 0.75139 | 0.0 | 0.74286 | 0.0 |
| azj | azj | azj | 0.99901 | 0.0 | 0.99901 | 6e-05 | 0.49963 | 0.0 | 0.27304 | 0.0 | 0.99901 | 6e-05 | 0.99901 | 6e-05 |
| bak | bak | bak | 1.0 | 0.0 | 1.0 | 0.0 | 1.0 | 0.0 | 1.0 | 0.0 | 1.0 | 0.0 | 1.0 | 0.0 |
| bam | bam | bam | 0.52563 | 0.05119 | 0.61044 | 0.06424 | 0.52664 | 0.03779 | 0.53084 | 0.03213 | 0.61123 | 0.06424 | 0.61308 | 0.0614 |
| ban | ban | ban | 0.97521 | 6e-05 | 0.97887 | 0.00019 | 0.97571 | 0.0 | 0.97467 | 0.0 | 0.97937 | 0.00019 | 0.97885 | 0.00013 |
| bel | bel | bel | 1.0 | 0.0 | 1.0 | 0.0 | 1.0 | 0.0 | 1.0 | 0.0 | 1.0 | 0.0 | 1.0 | 0.0 |
| bem | bem | bem | 0.9906 | 0.00045 | 0.97961 | 0.00251 | 0.99256 | 0.00017 | 0.99256 | 0.00015 | 0.98152 | 0.00251 | 0.99116 | 0.00094 |
| ben | ben | ben | 0.99852 | 6e-05 | 0.99253 | 0.0 | 0.99852 | 4e-05 | 0.99852 | 4e-05 | 0.99253 | 0.0 | 0.99253 | 0.0 |
| bho | bho | bho | 0.94329 | 0.00443 | 0.89206 | 0.01481 | 0.94374 | 0.00325 | 0.94846 | 0.00181 | 0.89206 | 0.01481 | 0.90785 | 0.01195 |
| bjn | bjn | bjn | 0.79496 | 0.05329 | 0.79638 | 0.06225 | 0.7956 | 0.03927 | 0.79547 | 0.03335 | 0.79638 | 0.06225 | 0.79741 | 0.05965 |
| bod | bod | bod | 0.94589 | 6e-05 | 0.80449 | 0.0 | 0.94589 | 4e-05 | 0.94589 | 4e-05 | 0.80449 | 0.0 | 0.80378 | 0.0 |
| bos | bos | bos | 0.58206 | 0.00312 | 0.69239 | 0.01229 | 0.57353 | 0.00232 | 0.49605 | 0.00126 | 0.69239 | 0.01229 | 0.69172 | 0.01183 |
| bug | bug | bug | 0.99802 | 6e-05 | 0.99654 | 0.00013 | 0.99703 | 0.0 | 0.99404 | 0.0 | 0.99653 | 0.00013 | 0.99703 | 0.0 |
| bul | bul | bul | 0.99951 | 0.0 | 1.0 | 0.0 | 0.99951 | 0.0 | 0.99951 | 0.0 | 1.0 | 0.0 | 0.99951 | 0.0 |
| cat | cat | cat | 1.0 | 0.0 | 1.0 | 0.0 | 1.0 | 0.0 | 1.0 | 0.0 | 1.0 | 0.0 | 0.99951 | 0.0 |
| ceb | ceb | ceb | 0.99503 | 6e-05 | 0.99951 | 6e-05 | 0.99454 | 0.0 | 0.99404 | 0.0 | 0.99951 | 6e-05 | 1.0 | 0.0 |
| ces | ces | ces | 0.99951 | 6e-05 | 0.99753 | 0.00019 | 0.99951 | 4e-05 | 0.99901 | 4e-05 | 0.99704 | 0.00019 | 0.99704 | 0.00019 |
| cjk | cjk | cjk | 0.84493 | 6e-05 | 0.90232 | 0.00032 | 0.83429 | 4e-05 | 0.79834 | 4e-05 | 0.89554 | 0.00032 | 0.8716 | 0.00019 |
| ckb | ckb | ckb | 0.99901 | 0.00011 | 1.0 | 0.0 | 1.0 | 0.0 | 1.0 | 0.0 | 1.0 | 0.0 | 1.0 | 0.0 |
| crh | crh | crh | 0.98902 | 6e-05 | 0.99204 | 6e-05 | 0.988 | 0.0 | 0.988 | 0.0 | 0.99204 | 6e-05 | 0.99204 | 6e-05 |
| cym | cym | cym | 0.99951 | 6e-05 | 1.0 | 0.0 | 1.0 | 0.0 | 1.0 | 0.0 | 1.0 | 0.0 | 1.0 | 0.0 |
| dan | dan | dan | 0.9931 | 0.00051 | 0.98661 | 0.00064 | 0.99505 | 0.00017 | 0.99554 | 7e-05 | 0.98758 | 0.00064 | 0.98757 | 0.00038 |
| deu | deu | deu | 0.99901 | 0.0 | 1.0 | 0.0 | 0.99901 | 0.0 | 0.99852 | 0.0 | 1.0 | 0.0 | 1.0 | 0.0 |
| dik | dik | dik | 0.99653 | 6e-05 | 0.99951 | 0.0 | 0.99653 | 0.0 | 0.99454 | 0.0 | 0.99951 | 0.0 | 0.99802 | 0.0 |
| dyu | dyu | dyu | 0.12435 | 0.01449 | 0.04212 | 0.00367 | 0.11878 | 0.01075 | 0.11186 | 0.00938 | 0.04212 | 0.00367 | 0.04033 | 0.00357 |
| dzo | dzo | dzo | 0.9496 | 0.00585 | 0.85848 | 0.02131 | 0.9491 | 0.00434 | 0.9491 | 0.0038 | 0.85848 | 0.02131 | 0.85848 | 0.02072 |
| ell | ell | ell | 1.0 | 0.0 | 1.0 | 0.0 | 1.0 | 0.0 | 1.0 | 0.0 | 1.0 | 0.0 | 1.0 | 0.0 |
| eng | eng | eng | 0.98732 | 0.00148 | 0.99263 | 0.00084 | 0.99215 | 0.00063 | 0.99556 | 0.00022 | 0.99308 | 0.00084 | 0.99256 | 0.00025 |
| epo | epo | epo | 0.99852 | 0.00017 | 0.99901 | 0.00013 | 0.99951 | 4e-05 | 1.0 | 0.0 | 0.99901 | 0.00013 | 0.99951 | 0.0 |
| est | est | est | 1.0 | 0.0 | 0.99753 | 0.00026 | 1.0 | 0.0 | 1.0 | 0.0 | 0.99901 | 0.00026 | 0.99802 | 0.0 |
| eus | eus | eus | 0.99951 | 0.0 | 0.99704 | 0.00032 | 0.99951 | 0.0 | 0.99951 | 0.0 | 0.99852 | 0.00032 | 0.99951 | 0.0 |
| ewe | ewe | ewe | 1.0 | 0.0 | 0.99803 | 0.00026 | 1.0 | 0.0 | 1.0 | 0.0 | 0.99803 | 0.00026 | 0.99803 | 0.00025 |
| fao | fao | fao | 0.99951 | 0.0 | 1.0 | 0.0 | 0.99951 | 0.0 | 0.99951 | 0.0 | 1.0 | 0.0 | 1.0 | 0.0 |
| fij | fij | fij | 0.99951 | 0.0 | 0.99852 | 6e-05 | 0.99901 | 0.0 | 0.99901 | 0.0 | 0.99852 | 6e-05 | 0.99852 | 6e-05 |
| fin | fin | fin | 0.99901 | 0.00011 | 0.99951 | 6e-05 | 1.0 | 0.0 | 1.0 | 0.0 | 1.0 | 6e-05 | 0.99901 | 0.0 |
| fon | fon | fon | 0.99752 | 0.0 | 0.99802 | 0.0 | 0.99752 | 0.0 | 0.99703 | 0.0 | 0.99802 | 0.0 | 0.99802 | 0.0 |
| fra | fra | fra | 0.99951 | 6e-05 | 0.99503 | 0.0 | 0.99951 | 4e-05 | 0.99852 | 4e-05 | 0.99454 | 0.0 | 0.99253 | 0.0 |
| fur | fur | fur | 0.99951 | 6e-05 | 0.99852 | 0.00019 | 0.99951 | 4e-05 | 1.0 | 0.0 | 0.99901 | 0.00019 | 1.0 | 0.0 |
| fuv | fuv | fuv | 0.96843 | 6e-05 | 0.98649 | 6e-05 | 0.96099 | 0.0 | 0.94693 | 0.0 | 0.98699 | 6e-05 | 0.98189 | 0.0 |
| gaz | gaz | gaz | 0.99411 | 0.00068 | 0.99264 | 0.00097 | 0.90281 | 0.00017 | 0.78925 | 7e-05 | 0.99508 | 0.00097 | 0.99852 | 0.00019 |
| gla | gla | gla | 0.99951 | 6e-05 | 0.99704 | 0.00039 | 1.0 | 0.0 | 1.0 | 0.0 | 0.99754 | 0.00039 | 0.99901 | 0.00013 |
| gle | gle | gle | 1.0 | 0.0 | 1.0 | 0.0 | 1.0 | 0.0 | 1.0 | 0.0 | 1.0 | 0.0 | 1.0 | 0.0 |
| glg | glg | glg | 0.99703 | 0.00011 | 0.99704 | 0.00032 | 0.99703 | 8e-05 | 0.99703 | 4e-05 | 0.99704 | 0.00032 | 0.99704 | 0.00031 |
| grn | grn | grn | 1.0 | 0.0 | 0.99754 | 0.00032 | 1.0 | 0.0 | 1.0 | 0.0 | 0.99803 | 0.00032 | 0.99901 | 0.00013 |
| guj | guj | guj | 1.0 | 0.0 | 1.0 | 0.0 | 1.0 | 0.0 | 1.0 | 0.0 | 1.0 | 0.0 | 1.0 | 0.0 |
| hat | hat | hat | 0.99852 | 0.00017 | 0.99655 | 0.00045 | 0.99852 | 0.00013 | 0.99901 | 7e-05 | 0.99754 | 0.00045 | 0.99852 | 0.00019 |
| hau | hau | hau | 0.95457 | 0.00551 | 0.90196 | 0.01416 | 0.98348 | 0.00143 | 0.99313 | 0.00052 | 0.93143 | 0.01416 | 0.97495 | 0.00325 |
| heb | heb | heb | 0.99606 | 0.00045 | 0.99901 | 0.00013 | 1.0 | 0.0 | 1.0 | 0.0 | 0.99901 | 0.00013 | 0.99901 | 0.00013 |
| hin | hin | hin | 0.67444 | 0.05551 | 0.84774 | 0.02279 | 0.6767 | 0.04078 | 0.69697 | 0.0325 | 0.8481 | 0.02279 | 0.86814 | 0.01859 |
| hne | hne | hne | 0.90296 | 0.00256 | 0.93617 | 0.00406 | 0.90343 | 0.00186 | 0.898 | 0.00133 | 0.93617 | 0.00406 | 0.9314 | 0.00344 |
| hrv | hrv | hrv | 0.75157 | 0.03272 | 0.74355 | 0.02427 | 0.75573 | 0.0237 | 0.768 | 0.01666 | 0.74389 | 0.02427 | 0.74309 | 0.02347 |
| hun | hun | hun | 1.0 | 0.0 | 1.0 | 0.0 | 1.0 | 0.0 | 0.99951 | 0.0 | 1.0 | 0.0 | 0.99951 | 0.0 |
| hye | hye | hye | 1.0 | 0.0 | 1.0 | 0.0 | 1.0 | 0.0 | 1.0 | 0.0 | 1.0 | 0.0 | 1.0 | 0.0 |
| ibo | ibo | ibo | 0.99951 | 6e-05 | 0.99951 | 6e-05 | 1.0 | 0.0 | 0.99951 | 4e-05 | 0.99951 | 6e-05 | 0.99951 | 6e-05 |
| ilo | ilo | ilo | 0.99951 | 6e-05 | 0.99901 | 0.00013 | 0.99951 | 4e-05 | 0.99951 | 4e-05 | 0.99901 | 0.00013 | 0.99951 | 6e-05 |
| ind | ind | ind | 0.91929 | 0.00983 | 0.92788 | 0.00566 | 0.92775 | 0.00645 | 0.93204 | 0.00528 | 0.92788 | 0.00566 | 0.92728 | 0.00545 |
| isl | isl | isl | 0.99951 | 0.00011 | 0.99951 | 0.0 | 0.99951 | 4e-05 | 1.0 | 0.0 | 1.0 | 0.0 | 1.0 | 0.0 |
| ita | ita | ita | 0.99803 | 0.00017 | 0.99404 | 0.0 | 0.99852 | 8e-05 | 0.99901 | 4e-05 | 0.99404 | 0.0 | 0.99203 | 0.0 |
| jav | jav | jav | 0.98346 | 0.00187 | 0.96103 | 0.00521 | 0.99166 | 0.00067 | 0.99213 | 0.00048 | 0.96978 | 0.00521 | 0.98394 | 0.002 |
| jpn | jpn | jpn | 1.0 | 0.0 | 0.99951 | 0.0 | 1.0 | 0.0 | 1.0 | 0.0 | 0.99951 | 0.0 | 0.99951 | 0.0 |
| kab | kab | kab | 0.85967 | 0.01221 | 0.83636 | 0.02549 | 0.87886 | 0.00709 | 0.90909 | 0.00362 | 0.8404 | 0.02549 | 0.84887 | 0.02247 |
| kac | kac | kac | 1.0 | 0.0 | 1.0 | 0.0 | 1.0 | 0.0 | 1.0 | 0.0 | 1.0 | 0.0 | 1.0 | 0.0 |
| kam | kam | kam | 0.92406 | 6e-05 | 0.90011 | 6e-05 | 0.91416 | 0.0 | 0.87368 | 0.0 | 0.89651 | 6e-05 | 0.87382 | 6e-05 |
| kan | kan | kan | 1.0 | 0.0 | 1.0 | 0.0 | 1.0 | 0.0 | 1.0 | 0.0 | 1.0 | 0.0 | 1.0 | 0.0 |
| kas | kas | kas | 0.97674 | 0.0 | 0.98497 | 0.00013 | 0.97597 | 0.0 | 0.96945 | 0.0 | 0.98497 | 0.00013 | 0.98445 | 6e-05 |
| kat | kat | kat | 1.0 | 0.0 | 1.0 | 0.0 | 1.0 | 0.0 | 1.0 | 0.0 | 1.0 | 0.0 | 1.0 | 0.0 |
| kaz | kaz | kaz | 0.99951 | 0.0 | 0.99951 | 0.0 | 0.99951 | 0.0 | 0.99951 | 0.0 | 0.99951 | 0.0 | 0.99951 | 0.0 |
| kbp | kbp | kbp | 0.99901 | 6e-05 | 1.0 | 0.0 | 0.99901 | 4e-05 | 0.99901 | 4e-05 | 1.0 | 0.0 | 1.0 | 0.0 |
| kea | kea | kea | 0.95238 | 0.0 | 0.96524 | 0.0 | 0.9513 | 0.0 | 0.93586 | 0.0 | 0.96258 | 0.0 | 0.95455 | 0.0 |
| khk | khk | khk | 1.0 | 0.0 | 1.0 | 0.0 | 0.36246 | 0.0 | 0.21674 | 0.0 | 1.0 | 0.0 | 1.0 | 0.0 |
| khm | khm | khm | 0.99951 | 0.0 | 0.99951 | 0.0 | 0.99951 | 0.0 | 0.99951 | 0.0 | 0.99951 | 0.0 | 0.99951 | 0.0 |
| kik | kik | kik | 0.96562 | 0.00403 | 0.96282 | 0.00489 | 0.96509 | 0.00295 | 0.96456 | 0.00255 | 0.96374 | 0.00489 | 0.96512 | 0.00444 |
| kin | kin | kin | 0.91471 | 0.00034 | 0.8872 | 0.0009 | 0.91471 | 0.00025 | 0.91471 | 0.00022 | 0.8872 | 0.0009 | 0.88768 | 0.00081 |
| kir | kir | kir | 1.0 | 0.0 | 1.0 | 0.0 | 1.0 | 0.0 | 1.0 | 0.0 | 1.0 | 0.0 | 1.0 | 0.0 |
| kmb | kmb | kmb | 0.96321 | 0.00415 | 0.93939 | 0.00695 | 0.96923 | 0.00253 | 0.97713 | 0.00144 | 0.94291 | 0.00695 | 0.95349 | 0.00426 |
| kmr | kmr | kmr | 0.99901 | 0.0 | 0.99852 | 0.00013 | 0.99901 | 0.0 | 0.99901 | 0.0 | 0.99852 | 0.00013 | 0.99852 | 6e-05 |
| knc | knc | knc | 0.8634 | 0.00153 | 0.86942 | 0.00013 | 0.86869 | 0.00021 | 0.86966 | 4e-05 | 0.86935 | 0.00013 | 0.86935 | 6e-05 |
| kon | kon | kon | 0.99802 | 0.0 | 0.99458 | 0.00058 | 0.99802 | 0.0 | 0.99802 | 0.0 | 0.99507 | 0.00058 | 0.99654 | 0.00013 |
| kor | kor | kor | 1.0 | 0.0 | 1.0 | 0.0 | 1.0 | 0.0 | 1.0 | 0.0 | 1.0 | 0.0 | 1.0 | 0.0 |

Table 34: Comparison of GlotLID vs OpenLID on FLORES-200 benchmark (part 1)

| iso639-3 | FLORES Code(s) | OpenLID Code(s) | GlotLID-M | | OpenLID | | GlotLID-M $\theta$=.3 | | GlotLID-M $\theta$=.5 | | OpenLID $\theta$=.3 | | OpenLID $\theta$=.5 | |
|---|---|---|---|---|---|---|---|---|---|---|---|---|---|---|
| | | | F1↑ | FPR↓ | F1↑ | FPR↓ | F1↑ | FPR↓ | F1↑ | FPR↓ | F1↑ | FPR↓ | F1↑ | FPR↓ |
| lao | lao | lao | 1.0 | 0.0 | 1.0 | 0.0 | 0.99901 | 0.0 | 0.99802 | 0.0 | 1.0 | 0.0 | 1.0 | 0.0 |
| lij | lij | lij | 0.99901 | 6e-05 | 0.99803 | 0.00019 | 0.99852 | 4e-05 | 0.99753 | 4e-05 | 0.99803 | 0.00019 | 0.99852 | 0.00013 |
| lim | lim | lim | 0.99253 | 0.0 | 0.99654 | 0.00019 | 0.99253 | 0.0 | 0.99153 | 0.0 | 0.99703 | 0.00019 | 0.99703 | 0.00013 |
| lin | lin | lin | 0.99901 | 0.00011 | 0.99901 | 0.00013 | 0.99852 | 8e-05 | 0.99901 | 4e-05 | 0.99901 | 0.00013 | 0.99802 | 0.00013 |
| lit | lit | lit | 0.99991 | 0.0 | 0.99951 | 0.0 | 0.99951 | 0.0 | 0.99951 | 0.0 | 0.99852 | 0.00013 | 0.99901 | 6e-05 |
| lmo | lmo | lmo | 0.99554 | 0.00011 | 0.99753 | 0.00026 | 0.99554 | 8e-05 | 0.99504 | 4e-05 | 0.99753 | 0.00026 | 0.99802 | 6e-05 |
| ltg | ltg | ltg | 0.99653 | 0.0 | 0.99852 | 0.0 | 0.99653 | 0.0 | 0.99503 | 0.0 | 0.99852 | 0.0 | 0.99802 | 0.0 |
| ltz | ltz | ltz | 0.99951 | 0.0 | 0.99951 | 0.0 | 0.99901 | 0.0 | 0.99901 | 0.0 | 0.99951 | 0.0 | 0.99951 | 0.0 |
| lua | lua | lua | 0.99653 | 6e-05 | 0.99604 | 6e-05 | 0.99553 | 0.0 | 0.99404 | 0.0 | 0.99554 | 6e-05 | 0.99553 | 0.0 |
| lug | lug | lug | 0.99653 | 0.0 | 0.99409 | 0.00058 | 0.99603 | 0.0 | 0.99458 | 0.0 | 0.99458 | 0.00058 | 0.99605 | 0.00031 |
| luo | luo | luo | 1.0 | 0.0 | 0.99852 | 0.00019 | 1.0 | 0.0 | 0.99951 | 0.0 | 0.99951 | 0.00019 | 0.99951 | 6e-05 |
| lus | lus | lus | 0.99653 | 6e-05 | 0.99852 | 0.0 | 0.99653 | 0.0 | 0.99653 | 0.0 | 0.99852 | 0.0 | 0.99802 | 0.0 |
| lvs | lvs | lvs | 0.99655 | 0.00034 | 0.99901 | 6e-05 | 0.92495 | 0.00021 | 0.80189 | 0.00015 | 0.99901 | 6e-05 | 0.99901 | 6e-05 |
| mag | mag | mag | 0.95459 | 0.00136 | 0.96204 | 0.00174 | 0.95507 | 0.00097 | 0.95408 | 0.00048 | 0.96204 | 0.00174 | 0.96393 | 0.00138 |
| mai | mai | mai | 0.97401 | 6e-05 | 0.98802 | 0.00013 | 0.97366 | 4e-05 | 0.97102 | 0.0 | 0.98802 | 0.00013 | 0.98701 | 0.00013 |
| mal | mal | mal | 1.0 | 0.0 | 1.0 | 0.0 | 1.0 | 0.0 | 1.0 | 0.0 | 1.0 | 0.0 | 1.0 | 0.0 |
| mar | mar | mar | 1.0 | 0.0 | 0.99901 | 0.00013 | 1.0 | 0.0 | 1.0 | 0.0 | 0.99901 | 0.00013 | 1.0 | 0.0 |
| min | min | min | 0.6616 | 0.00017 | 0.66183 | 0.00039 | 0.66182 | 8e-05 | 0.66116 | 4e-05 | 0.66183 | 0.00039 | 0.66205 | 0.00019 |
| mkd | mkd | mkd | 1.0 | 0.0 | 0.99951 | 6e-05 | 1.0 | 0.0 | 1.0 | 0.0 | 0.99951 | 6e-05 | 0.99951 | 6e-05 |
| mlt | mlt | mlt | 0.97401 | 0.00307 | 0.93143 | 0.00959 | 0.99216 | 0.00067 | 0.99803 | 0.00015 | 0.99803 | 0.00959 | 0.98684 | 0.00169 |
| mni | mni | mni | 0.99901 | 6e-05 | 0.99411 | 0.00077 | 0.99901 | 4e-05 | 0.99901 | 4e-05 | 0.99411 | 0.00077 | 0.99411 | 0.00075 |
| mos | mos | mos | 0.98138 | 0.0 | 0.9814 | 6e-05 | 0.97415 | 0.0 | 0.96418 | 0.0 | 0.97881 | 6e-05 | 0.96997 | 0.0 |
| mri | mri | mri | 0.99901 | 6e-05 | 0.99951 | 6e-05 | 0.99951 | 0.0 | 0.99901 | 0.0 | 1.0 | 6e-05 | 1.0 | 0.0 |
| mya | mya | mya | 1.0 | 0.0 | 1.0 | 0.0 | 1.0 | 0.0 | 1.0 | 0.0 | 1.0 | 0.0 | 1.0 | 0.0 |
| nld | nld | nld | 0.99803 | 0.00023 | 0.99704 | 0.00019 | 0.99901 | 8e-05 | 0.99901 | 4e-05 | 0.99704 | 0.00019 | 0.99704 | 0.00019 |
| nno | nno | nno | 0.98507 | 0.00045 | 0.98277 | 0.00135 | 0.98606 | 0.00025 | 0.986 | 7e-05 | 0.98277 | 0.00135 | 0.98374 | 0.00119 |
| nob | nob | nob | 0.98185 | 0.00148 | 0.97188 | 0.00193 | 0.95835 | 0.0011 | 0.89931 | 0.00078 | 0.97086 | 0.00193 | 0.96883 | 0.00188 |
| npi | npi | npi | 0.99104 | 0.00011 | 0.99803 | 0.00026 | 0.53362 | 8e-05 | 0.30628 | 0.0 | 0.99803 | 0.00026 | 0.99803 | 0.00019 |
| nso | nso | nso | 0.99704 | 0.00028 | 0.9868 | 0.00154 | 0.99704 | 0.00021 | 0.99655 | 0.00018 | 0.9868 | 0.00154 | 0.98776 | 0.00138 |
| nus | nus | nus | 0.99951 | 0.0 | 0.99951 | 0.0 | 0.99951 | 0.0 | 0.99951 | 0.0 | 0.99951 | 0.0 | 0.99951 | 0.0 |
| nya | nya | nya | 0.99753 | 0.00023 | 0.99606 | 0.00051 | 0.99803 | 0.00013 | 0.99852 | 4e-05 | 0.99704 | 0.00051 | 0.99754 | 0.00031 |
| oci | oci | oci | 0.99951 | 6e-05 | 0.9941 | 0.00071 | 1.0 | 0.0 | 0.99951 | 0.0 | 0.9941 | 0.00071 | 0.99557 | 0.0005 |
| ory | ory | ory | 1.0 | 0.0 | 1.0 | 0.0 | 0.80519 | 0.0 | 0.66314 | 0.0 | 1.0 | 0.0 | 1.0 | 0.0 |
| pag | pag | pag | 0.99852 | 0.0 | 0.99901 | 6e-05 | 0.99852 | 0.0 | 0.99852 | 0.0 | 0.99901 | 6e-05 | 0.99901 | 6e-05 |
| pan | pan | pan | 1.0 | 0.0 | 1.0 | 0.0 | 1.0 | 0.0 | 1.0 | 0.0 | 1.0 | 0.0 | 1.0 | 0.0 |
| pap | pap | pap | 0.99069 | 0.00102 | 0.97681 | 0.00303 | 0.99118 | 0.00072 | 0.99557 | 0.0003 | 0.9787 | 0.00303 | 0.9815 | 0.00213 |
| pbt | pbt | pbt | 0.81486 | 0.00051 | 0.99704 | 0.00032 | 0.76381 | 0.00025 | 0.68523 | 0.00011 | 0.99704 | 0.00032 | 0.99753 | 0.00025 |
| pes | pes | pes | 0.57435 | 0.08493 | 0.54791 | 0.06997 | 0.60647 | 0.04812 | 0.57502 | 0.02962 | 0.54829 | 0.06997 | 0.55288 | 0.06641 |
| plt | plt | plt | 0.99852 | 0.0 | 1.0 | 0.0 | 0.99802 | 0.0 | 0.99603 | 0.0 | 1.0 | 0.0 | 1.0 | 0.0 |
| pol | pol | pol | 0.9907 | 0.00108 | 0.99411 | 0.00077 | 0.99167 | 0.00072 | 0.99167 | 0.00063 | 0.99606 | 0.00077 | 0.99606 | 0.0005 |
| por | por | por | 0.99655 | 0.0004 | 0.99408 | 0.00051 | 0.99704 | 0.00025 | 0.99704 | 0.00018 | 0.99457 | 0.00051 | 0.99654 | 0.00019 |
| prs | prs | prs | 0.14688 | 0.0192 | 0.51273 | 0.01571 | 0.14348 | 0.01135 | 0.1171 | 0.00716 | 0.51273 | 0.01571 | 0.51395 | 0.01502 |
| quy | quy | quy | 0.75904 | 0.0 | 0.99951 | 6e-05 | 0.625 | 0.0 | 0.57163 | 0.0 | 0.99951 | 6e-05 | 1.0 | 0.0 |
| ron | ron | ron | 0.99951 | 0.0 | 0.99754 | 0.00032 | 0.99951 | 0.0 | 0.99951 | 0.0 | 0.99852 | 0.00032 | 0.99852 | 0.00019 |
| run | run | run | 0.92541 | 0.00881 | 0.9044 | 0.01268 | 0.92584 | 0.0065 | 0.92627 | 0.00565 | 0.9044 | 0.01268 | 0.90563 | 0.01214 |
| rus | rus | rus | 0.99901 | 6e-05 | 0.99901 | 6e-05 | 0.99901 | 4e-05 | 0.99901 | 4e-05 | 0.99901 | 6e-05 | 0.99901 | 0.0 |
| sag | sag | sag | 0.99901 | 0.0 | 0.99901 | 0.0 | 0.99901 | 0.0 | 0.99901 | 0.0 | 0.99901 | 0.0 | 0.99901 | 0.0 |
| san | san | san | 0.99104 | 6e-05 | 0.99002 | 0.0 | 0.99104 | 4e-05 | 0.99103 | 0.0 | 0.99002 | 0.0 | 0.98749 | 0.0 |
| sat | sat | sat | 1.0 | 0.0 | 1.0 | 0.0 | 1.0 | 0.0 | 1.0 | 0.0 | 1.0 | 0.0 | 1.0 | 0.0 |
| scn | scn | scn | 0.99802 | 6e-05 | 0.99507 | 0.00051 | 0.99802 | 4e-05 | 0.99852 | 0.0 | 0.99556 | 0.00051 | 0.99556 | 0.00038 |
| shn | shn | shn | 1.0 | 0.0 | 1.0 | 0.0 | 1.0 | 0.0 | 1.0 | 0.0 | 1.0 | 0.0 | 1.0 | 0.0 |
| sin | sin | sin | 1.0 | 0.0 | 1.0 | 0.0 | 1.0 | 0.0 | 1.0 | 0.0 | 1.0 | 0.0 | 1.0 | 0.0 |
| slk | slk | slk | 0.99852 | 6e-05 | 0.99654 | 0.00019 | 0.99852 | 4e-05 | 0.99901 | 0.0 | 0.99703 | 0.00019 | 0.99753 | 6e-05 |
| slv | slv | slv | 0.99459 | 0.00062 | 0.99606 | 0.00045 | 0.99606 | 0.0003 | 0.99951 | 4e-05 | 0.99753 | 0.00045 | 0.99951 | 0.0 |
| smo | smo | smo | 0.99603 | 0.0 | 0.99852 | 0.00013 | 0.99603 | 0.0 | 0.99603 | 0.0 | 0.99901 | 0.00013 | 0.99951 | 0.0 |
| sna | sna | sna | 0.99901 | 0.00011 | 0.99951 | 0.00077 | 0.99951 | 4e-05 | 1.0 | 0.0 | 0.99704 | 0.00077 | 0.99852 | 0.00019 |
| snd | snd | snd | 0.99362 | 0.00074 | 0.99901 | 0.0 | 0.99508 | 0.00042 | 0.99704 | 0.00022 | 0.99901 | 0.0 | 0.99901 | 0.0 |
| som | som | som | 0.96657 | 0.00398 | 0.97683 | 0.00309 | 0.98973 | 0.00089 | 0.99803 | 0.00015 | 0.98828 | 0.00309 | 0.99557 | 0.00056 |
| sot | sot | sot | 1.0 | 0.0 | 0.9567 | 0.0 | 1.0 | 0.0 | 1.0 | 0.0 | 0.9567 | 0.0 | 0.95401 | 0.0 |
| spa | spa | spa | 0.99508 | 0.00057 | 0.99211 | 0.00064 | 0.99508 | 0.00042 | 0.99508 | 0.00033 | 0.9921 | 0.00064 | 0.99259 | 0.0005 |
| srd | srd | srd | 0.99951 | 0.0 | 0.99606 | 0.00039 | 0.99901 | 0.0 | 0.99901 | 0.0 | 0.99704 | 0.00039 | 0.99704 | 0.00025 |
| srp | srp | srp | 0.99901 | 0.00011 | 0.99951 | 0.0 | 0.99951 | 4e-05 | 1.0 | 0.0 | 0.99951 | 0.0 | 0.99951 | 0.0 |
| ssw | ssw | ssw | 0.99654 | 0.00023 | 0.99106 | 0.00026 | 0.99456 | 0.00017 | 0.99455 | 0.00011 | 0.99205 | 0.00026 | 0.99254 | 6e-05 |
| sun | sun | sun | 0.99012 | 0.00057 | 0.99014 | 0.00077 | 0.99355 | 0.00013 | 0.99304 | 4e-05 | 0.99112 | 0.00077 | 0.99259 | 0.00044 |
| swe | swe | swe | 0.99754 | 0.00028 | 0.99852 | 0.00019 | 0.99951 | 4e-05 | 1.0 | 0.0 | 1.0 | 0.00019 | 1.0 | 0.0 |
| swh | swh | swh | 0.94869 | 0.00187 | 0.92378 | 0.01075 | 0.91684 | 0.00072 | 0.74599 | 0.00018 | 0.93015 | 0.01075 | 0.95247 | 0.00632 |
| szl | szl | szl | 0.99104 | 6e-05 | 0.99504 | 0.00013 | 0.99104 | 4e-05 | 0.99104 | 4e-05 | 0.99554 | 0.00013 | 0.99553 | 0.0 |
| tam | tam | tam | 1.0 | 0.0 | 1.0 | 0.0 | 1.0 | 0.0 | 1.0 | 0.0 | 1.0 | 0.0 | 1.0 | 0.0 |
| taq | taq | taq | 0.80642 | 0.02022 | 0.82365 | 0.01635 | 0.83861 | 0.00827 | 0.84449 | 0.00332 | 0.83486 | 0.01635 | 0.85028 | 0.00814 |
| tat | tat | tat | 1.0 | 0.0 | 1.0 | 0.0 | 1.0 | 0.0 | 1.0 | 0.0 | 1.0 | 0.0 | 1.0 | 0.0 |
| tel | tel | tel | 1.0 | 0.0 | 0.99901 | 0.0 | 1.0 | 0.0 | 1.0 | 0.0 | 0.99901 | 0.0 | 0.99901 | 0.0 |
| tgk | tgk | tgk | 1.0 | 0.0 | 1.0 | 0.0 | 1.0 | 0.0 | 1.0 | 0.0 | 1.0 | 0.0 | 1.0 | 0.0 |
| tgl | tgl | tgl | 0.99901 | 0.00011 | 1.0 | 0.0 | 0.99901 | 4e-05 | 0.99901 | 4e-05 | 1.0 | 0.0 | 0.99951 | 0.0 |
| tha | tha | tha | 1.0 | 0.0 | 1.0 | 0.0 | 1.0 | 0.0 | 1.0 | 0.0 | 1.0 | 0.0 | 1.0 | 0.0 |
| tir | tir | tir | 0.98851 | 0.0 | 0.99951 | 0.0 | 0.98851 | 0.0 | 0.988 | 0.0 | 0.99951 | 0.0 | 0.99951 | 0.0 |
| tpi | tpi | tpi | 0.99951 | 0.0 | 1.0 | 0.0 | 0.99951 | 0.0 | 0.99901 | 0.0 | 0.99951 | 0.0 | 0.99951 | 0.0 |
| tsn | tsn | tsn | 0.99753 | 6e-05 | 0.96932 | 0.00406 | 0.99753 | 4e-05 | 0.99753 | 4e-05 | 0.96932 | 0.00406 | 0.97345 | 0.00344 |
| tso | tso | tso | 0.99803 | 0.00023 | 0.99606 | 0.00045 | 0.99753 | 0.00017 | 0.99901 | 4e-05 | 0.99655 | 0.00045 | 0.99852 | 0.00013 |
| tuk | tuk | tuk | 0.99803 | 0.00023 | 0.99951 | 6e-05 | 0.99951 | 4e-05 | 0.99951 | 4e-05 | 1.0 | 6e-05 | 1.0 | 0.0 |
| tum | tum | tum | 0.99852 | 0.00011 | 0.99556 | 0.00045 | 0.99901 | 4e-05 | 0.99901 | 0.0 | 0.99655 | 0.00045 | 0.99802 | 0.0 |
| tur | tur | tur | 0.9907 | 0.00108 | 0.99362 | 0.00084 | 0.9907 | 0.0008 | 0.99119 | 0.00066 | 0.99362 | 0.00084 | 0.99362 | 0.00081 |
| tzm | tzm | tzm | 0.94421 | 0.0054 | 0.95352 | 0.00515 | 0.94421 | 0.00401 | 0.94524 | 0.00318 | 0.95352 | 0.00515 | 0.95302 | 0.00501 |
| uig | uig | uig | 0.99901 | 0.00011 | 1.0 | 0.0 | 1.0 | 0.0 | 1.0 | 0.0 | 1.0 | 0.0 | 1.0 | 0.0 |
| ukr | ukr | ukr | 0.99951 | 6e-05 | 0.99951 | 6e-05 | 0.99951 | 4e-05 | 0.99951 | 4e-05 | 0.99951 | 6e-05 | 0.99951 | 6e-05 |
| umb | umb | umb | 0.88585 | 0.00045 | 0.97762 | 0.00103 | 0.8781 | 0.00021 | 0.834 | 0.00011 | 0.97906 | 0.00103 | 0.97433 | 0.00044 |
| urd | urd | urd | 0.98346 | 0.00187 | 0.98491 | 0.002 | 0.98346 | 0.00139 | 0.99021 | 0.0007 | 0.98491 | 0.002 | 0.98684 | 0.00169 |
| uzn | uzn | uzn | 0.96885 | 0.00364 | 0.92844 | 0.01004 | 0.87839 | 0.00089 | 0.76439 | 0.0003 | 0.94934 | 0.01004 | 0.97731 | 0.00294 |
| vec | vec | vec | 0.99703 | 6e-05 | 0.99605 | 0.00026 | 0.99703 | 4e-05 | 0.99653 | 0.0 | 0.99605 | 0.00026 | 0.99605 | 0.00025 |
| vie | vie | vie | 0.99951 | 6e-05 | 0.99951 | 6e-05 | 1.0 | 0.0 | 1.0 | 0.0 | 0.99951 | 6e-05 | 1.0 | 0.0 |
| war | war | war | 0.99951 | 0.0 | 1.0 | 0.0 | 0.99951 | 0.0 | 0.99951 | 0.0 | 1.0 | 0.0 | 1.0 | 0.0 |
| wol | wol | wol | 0.99852 | 0.0 | 0.99704 | 0.00026 | 0.99802 | 0.0 | 0.99802 | 0.0 | 0.99753 | 0.00026 | 0.99852 | 0.0 |
| xho | xho | xho | 0.99118 | 0.00097 | 0.98581 | 0.00154 | 0.98968 | 0.00067 | 0.99113 | 0.00044 | 0.98725 | 0.00154 | 0.98968 | 0.001 |
| ydd | ydd | ydd | 0.99603 | 0.0 | 0.99901 | 0.0 | | | | | 0.99901 | 0.0 | 0.99901 | 0.0 |
| yor | yor | yor | 0.99406 | 0.00023 | 0.99901 | 0.00013 | 0.99355 | 0.00013 | 0.99053 | 4e-05 | 0.99951 | 0.00013 | 0.99951 | 0.0 |
| yue | yue | yue | 0.00394 | 0.0 | 0.00588 | 0.00032 | | | | | 0.00588 | 0.00032 | | |
| zho | zho | zho | 1.0 | 0.0 | 0.79873 | 0.06508 | 1.0 | 0.0 | 1.0 | 0.0 | 0.79873 | 0.06508 | 0.79881 | 0.06315 |
| zsm | zsm | zsm | 0.93506 | 0.00307 | 0.92745 | 0.00766 | 0.94641 | 0.00127 | 0.94972 | 0.00081 | 0.93589 | 0.00766 | 0.94665 | 0.00463 |
| zul | zul | zul | 0.96893 | 0.0 | 0.98277 | 0.00135 | 0.96893 | 0.0 | 0.96735 | 0.0 | 0.98325 | 0.00135 | 0.98519 | 0.001 |

Table 35: Comparison of GlotLID vs OpenLID on FLORES-200 benchmark (part 2)

| | | | GlotLID-M | | NLLB | | with confidence threshold θ | | | | | | | |
|---|---|---|---|---|---|---|---|---|---|---|---|---|---|---|
| | | | | | | | GlotLID-M θ=.3 | | GlotLID-M θ=.5 | | NLLB θ=.3 | | NLLB θ=.5 | |
| iso639-3 | FLORES Code(s) | NLLB Code(s) | F1↑ | FPR↓ | F1↑ | FPR↓ | F1↑ | FPR↓ | F1↑ | FPR↓ | F1↑ | FPR↓ | F1↑ | FPR↓ |
| ace | ace | ace | 0.95503 | 0.00579 | 0.93532 | 0.01209 | 0.955 | 0.00426 | **0.95689** | 0.00299 | 0.93532 | 0.01209 | 0.9379 | 0.01076 |
| afr | afr | afr | **1.0** | 0.0 | 0.99852 | 0.00011 | 1.0 | 0.0 | 1.0 | 0.0 | 0.99901 | 0.00011 | 0.99951 | 0.0 |
| aka | aka | aka | **0.99852** | 0.0 | 0.82334 | 0.0058 | 0.99852 | 0.0 | 0.99852 | 0.0 | 0.82334 | 0.0058 | 0.82272 | 0.00554 |
| als | als | als | 0.99852 | 0.00011 | 0.99803 | 0.00022 | 0.67539 | 0.0 | 0.44785 | 0.0 | 0.99901 | 0.00022 | 0.99951 | 5e-05 |
| amh | amh | amh | **0.99951** | 6e-05 | 0.99901 | 0.00011 | 0.99951 | 4e-05 | 0.99951 | 4e-05 | **0.99951** | 0.00011 | 0.99951 | 5e-05 |
| arb | arb | arb | 0.25705 | 0.21515 | 0.31812 | 0.46765 | 0.12788 | 0.06529 | 0.09585 | 0.03457 | **0.31814** | 0.46765 | 0.31769 | 0.44516 |
| asm | asm | asm | **1.0** | 0.0 | 1.0 | 0.0 | 1.0 | 0.0 | 1.0 | 0.0 | 1.0 | 0.0 | 1.0 | 0.0 |
| ast | ast | ast | 0.9916 | 0.00051 | 0.99016 | 0.00076 | **0.99308** | 0.00025 | 0.99257 | 0.00018 | 0.99065 | 0.00076 | 0.9926 | 0.00047 |
| awa | awa | awa | 0.38951 | 0.0 | 0.96113 | 0.00092 | 0.38982 | 0.0 | 0.35313 | 0.0 | 0.96113 | 0.00092 | **0.96304** | 0.00062 |
| ayr | ayr | ayr | 0.99557 | 0.00045 | 0.99802 | 5e-05 | 0.96738 | 4e-05 | 0.93193 | 0.0 | 0.99802 | 5e-05 | **0.99852** | 0.0 |
| azb | azb | azb | 0.36583 | 0.00011 | 0.8767 | 0.00119 | 0.25065 | 0.0 | 0.13112 | 0.0 | 0.87956 | 0.00119 | **0.88136** | 0.00057 |
| azj | azj | azj | **0.99901** | 0.0 | 0.99704 | 0.00033 | 0.49963 | 0.0 | 0.27304 | 0.0 | 0.99704 | 0.00033 | 0.99704 | 0.00031 |
| bak | bak | bak | **1.0** | 0.0 | 0.99901 | 5e-05 | 1.0 | 0.0 | 1.0 | 0.0 | 0.99901 | 5e-05 | 0.99901 | 5e-05 |
| bam | bam | bam | 0.52563 | 0.05119 | 0.61944 | 0.05293 | 0.52664 | 0.03779 | 0.53084 | 0.03213 | 0.61987 | 0.05293 | **0.62064** | 0.04998 |
| ban | ban | ban | 0.97521 | 6e-05 | 0.9712 | 0.00033 | **0.97571** | 0.0 | 0.97467 | 0.0 | 0.9712 | 0.00033 | 0.97117 | 0.00026 |
| bel | bel | bel | **1.0** | 0.0 | **1.0** | 0.0 | 1.0 | 0.0 | 1.0 | 0.0 | 1.0 | 0.0 | 1.0 | 0.0 |
| bem | bem | bem | 0.9906 | 0.0 | 0.97394 | 0.00277 | **0.99256** | 0.00017 | 0.99256 | 0.00015 | 0.97677 | 0.00277 | 0.98922 | 0.00098 |
| ben | ben | ben | 0.99852 | 6e-05 | **0.99951** | 5e-05 | 0.99852 | 4e-05 | 0.99852 | 4e-05 | 0.99951 | 5e-05 | 0.99951 | 5e-05 |
| bho | bho | bho | 0.94329 | 0.00443 | 0.93354 | 0.00168 | 0.94374 | 0.00325 | **0.94846** | 0.00181 | 0.93354 | 0.00168 | **0.93416** | 0.00124 |
| bjn | bjn | bjn | 0.79496 | 0.05329 | 0.75225 | 0.06312 | **0.7956** | 0.03927 | 0.79547 | 0.03335 | 0.75225 | 0.06312 | **0.75747** | 0.05764 |
| bod | bod | bod | 0.94589 | 6e-05 | 0.96512 | 0.00385 | 0.94589 | 4e-05 | 0.94589 | 4e-05 | **0.9678** | 0.00385 | 0.96569 | 0.00222 |
| bos | bos | bos | 0.58206 | 0.00312 | **0.5954** | 0.0064 | 0.57353 | 0.00232 | 0.49605 | 0.00126 | 0.5954 | 0.0064 | 0.5949 | 0.00605 |
| bug | bug | bug | **0.99802** | 6e-05 | 0.97649 | 0.0006 | 0.99703 | 0.0 | 0.99404 | 0.0 | **0.97747** | 0.0006 | 0.97742 | 0.00036 |
| bul | bul | bul | **0.99951** | 0.0 | 0.99951 | 0.0 | 0.99951 | 0.0 | 0.99951 | 0.0 | 0.99951 | 0.0 | 0.99951 | 0.0 |
| cat | cat | cat | **1.0** | 0.0 | 0.98732 | 0.00141 | 1.0 | 0.0 | 1.0 | 0.0 | 0.98828 | 0.00141 | 0.99459 | 0.00057 |
| ceb | ceb | ceb | 0.99503 | 0.0 | **0.99951** | 0.0 | 0.99454 | 0.0 | 0.99404 | 0.0 | 0.99951 | 0.0 | 0.99951 | 0.0 |
| ces | ces | ces | **0.99951** | 6e-05 | 0.99901 | 0.00011 | 0.99951 | 4e-05 | 0.99901 | 4e-05 | 0.99951 | 0.00011 | **1.0** | 0.0 |
| cjk | cjk | cjk | 0.84493 | 6e-05 | **0.86875** | 0.00098 | 0.83429 | 4e-05 | 0.79834 | 4e-05 | 0.8611 | 0.00098 | 0.83995 | 0.00052 |
| ckb | ckb | ckb | 0.99901 | 0.00011 | **1.0** | 0.0 | **1.0** | 0.0 | 1.0 | 0.0 | 1.0 | 0.0 | 1.0 | 0.0 |
| crh | crh | crh | **0.98902** | 6e-05 | 0.98291 | 0.0 | 0.988 | 0.0 | 0.988 | 0.0 | 0.98291 | 0.0 | 0.98138 | 0.0 |
| cym | cym | cym | 0.99951 | 6e-05 | 0.99951 | 5e-05 | 0.99951 | 4e-05 | **1.0** | 0.0 | **1.0** | 5e-05 | 1.0 | 0.0 |
| dan | dan | dan | 0.9931 | 0.00051 | 0.99456 | 0.00022 | 0.99505 | 0.00017 | **0.99554** | 7e-05 | 0.99505 | 0.00022 | **0.99604** | 5e-05 |
| deu | deu | deu | 0.99901 | 0.0 | 0.9907 | 0.00103 | 0.99901 | 0.0 | 0.99852 | 0.0 | **1.0** | 0.00103 | **1.0** | 0.0 |
| dik | dik | dik | **0.99653** | 6e-05 | 0.99253 | 0.0 | 0.99653 | 0.0 | 0.99454 | 0.0 | 0.99203 | 0.0 | 0.99002 | 0.0 |
| dyu | dyu | dyu | **0.12435** | 0.01449 | 0.04797 | 0.00249 | 0.11878 | 0.01075 | 0.11186 | 0.00938 | 0.04797 | 0.00249 | 0.04621 | 0.00233 |
| dzo | dzo | dzo | 0.9496 | 0.00585 | **0.96791** | 5e-05 | 0.9491 | 0.00434 | 0.9491 | 0.0038 | 0.96685 | 5e-05 | 0.95405 | 5e-05 |
| ell | ell | ell | **1.0** | 0.0 | **1.0** | 0.0 | 1.0 | 0.0 | 1.0 | 0.0 | 1.0 | 0.0 | 1.0 | 0.0 |
| eng | eng | eng | 0.98732 | 0.00148 | 0.97825 | 0.00244 | 0.99215 | 0.00063 | **0.99556** | 0.00022 | 0.98925 | 0.00244 | **0.99362** | 0.00067 |
| epo | epo | epo | 0.99852 | 0.00017 | 0.99704 | 0.00033 | 0.99951 | 4e-05 | **1.0** | 0.0 | 0.99803 | 0.00033 | 0.99901 | 0.0001 |
| est | est | est | **1.0** | 0.0 | 0.99852 | 0.00016 | 1.0 | 0.0 | 1.0 | 0.0 | 0.99901 | 0.00016 | **0.99951** | 0.0 |
| eus | eus | eus | **0.99951** | 0.0 | 0.99852 | 0.00016 | 0.99951 | 0.0 | 0.99951 | 0.0 | 0.99901 | 0.00016 | **0.99951** | 5e-05 |
| ewe | ewe | ewe | **1.0** | 0.0 | 0.99704 | 0.00033 | 1.0 | 0.0 | 1.0 | 0.0 | 0.99704 | 0.00033 | 0.99704 | 0.00031 |
| fao | fao | fao | **0.99951** | 0.0 | 0.50517 | 0.0 | 0.99951 | 0.0 | 0.99951 | 0.0 | 0.50517 | 0.0 | 0.49852 | 0.0 |
| fij | fij | fij | 0.99951 | 0.0 | **1.0** | 0.0 | 0.99901 | 0.0 | 0.99901 | 0.0 | 1.0 | 0.0 | 0.99951 | 0.0 |
| fin | fin | fin | 0.99901 | 0.00011 | 0.99951 | 5e-05 | **1.0** | 0.0 | 1.0 | 0.0 | **1.0** | 5e-05 | 1.0 | 0.0 |
| fon | fon | fon | **0.99752** | 0.0 | 0.99703 | 0.0 | 0.99752 | 0.0 | 0.99703 | 0.0 | 0.99703 | 0.0 | 0.99703 | 0.0 |
| fra | fra | fra | **0.99951** | 6e-05 | 0.99606 | 0.00038 | 0.99951 | 4e-05 | **0.99852** | 4e-05 | 0.99852 | 0.00038 | **0.99901** | 5e-05 |
| fur | fur | fur | 0.99951 | 6e-05 | 0.99802 | 0.0 | 0.99951 | 4e-05 | **1.0** | 0.0 | 0.99802 | 0.0 | 0.99752 | 0.0 |
| fuv | fuv | fuv | 0.96843 | 6e-05 | **0.98102** | 0.00043 | 0.96099 | 0.0 | 0.94693 | 0.0 | 0.97842 | 0.00043 | 0.97578 | 0.00016 |
| gaz | gaz | gaz | 0.99411 | 0.00068 | **0.99951** | 5e-05 | 0.90281 | 0.00017 | 0.78925 | 7e-05 | 0.99951 | 5e-05 | 0.99951 | 5e-05 |
| gla | gla | gla | 0.99951 | 6e-05 | 0.99803 | 0.00016 | **1.0** | 0.0 | 1.0 | 0.0 | **0.99901** | 0.00016 | 0.99901 | 5e-05 |
| gle | gle | gle | **1.0** | 0.0 | 0.99803 | 0.00022 | 1.0 | 0.0 | 1.0 | 0.0 | **0.99951** | 0.00022 | 0.99951 | 5e-05 |
| glg | glg | glg | **0.99703** | 0.00011 | 0.9931 | 0.00054 | 0.99703 | 8e-05 | 0.99703 | 4e-05 | 0.99457 | 0.00054 | **0.99605** | 0.00021 |
| grn | grn | grn | **1.0** | 0.0 | 0.99654 | 0.00016 | 1.0 | 0.0 | 1.0 | 0.0 | **0.99703** | 0.00016 | 0.99703 | 0.0 |
| guj | guj | guj | **1.0** | 0.0 | 1.0 | 0.0 | 1.0 | 0.0 | 1.0 | 0.0 | 1.0 | 0.0 | 1.0 | 0.0 |
| hat | hat | hat | 0.99852 | 0.00017 | **0.99852** | 5e-05 | 0.99852 | 0.00013 | **0.99901** | 7e-05 | 0.99852 | 5e-05 | 0.99852 | 0.0 |
| hau | hau | hau | 0.95427 | 0.00551 | 0.99704 | 0.00027 | 0.98348 | 0.00143 | **0.99313** | 0.00052 | 0.99704 | 0.00027 | **0.99802** | 0.0001 |
| heb | heb | heb | 0.99606 | 0.00045 | **1.0** | 0.0 | **1.0** | 0.0 | 1.0 | 0.0 | 1.0 | 0.0 | 1.0 | 0.0 |
| hin | hin | hin | 0.67444 | 0.05551 | 0.87219 | 0.01594 | 0.6767 | 0.04078 | **0.69697** | 0.0325 | 0.87295 | 0.01594 | **0.88558** | 0.0134 |
| hne | hne | hne | 0.90296 | 0.00146 | **0.92997** | 0.00146 | **0.90343** | 0.00186 | 0.898 | 0.00133 | 0.92713 | 0.00146 | 0.92713 | 0.00135 |
| hrv | hrv | hrv | 0.75157 | 0.03272 | 0.73352 | 0.02901 | 0.75573 | 0.0237 | **0.768** | 0.01666 | **0.73382** | 0.02901 | 0.73361 | 0.02758 |
| hun | hun | hun | **1.0** | 0.0 | 0.99264 | 0.00081 | 1.0 | 0.0 | 0.99951 | 0.0 | 0.99557 | 0.00081 | **1.0** | 0.0 |
| hye | hye | hye | **1.0** | 0.0 | 1.0 | 0.0 | 1.0 | 0.0 | 1.0 | 0.0 | 1.0 | 0.0 | 1.0 | 0.0 |
| ibo | ibo | ibo | 0.99951 | 6e-05 | 0.99951 | 5e-05 | **1.0** | 0.0 | 1.0 | 0.0 | **1.0** | 5e-05 | 0.99951 | 0.0 |
| ilo | ilo | ilo | **0.99951** | 0.0 | 0.99852 | 0.00016 | 0.99951 | 4e-05 | 0.99951 | 4e-05 | 0.99852 | 0.00016 | **0.99951** | 5e-05 |
| ind | ind | ind | 0.91929 | 0.00983 | 0.81942 | 0.02294 | 0.92775 | 0.00645 | **0.93204** | 0.00528 | 0.82348 | 0.02294 | **0.8455** | 0.018 |
| isl | isl | isl | 0.99901 | 0.00011 | 0.76205 | 0.03427 | **0.99951** | 4e-05 | 0.99951 | 4e-05 | 0.76205 | 0.03427 | **0.76522** | 0.03213 |
| ita | ita | ita | 0.99803 | 0.00011 | 0.97212 | 0.00309 | 0.99852 | 8e-05 | **0.99901** | 4e-05 | 0.97634 | 0.00309 | **0.98346** | 0.00171 |
| jav | jav | jav | 0.98346 | 0.00187 | 0.97674 | 0.00239 | 0.99166 | 0.00067 | **0.99213** | 0.00048 | 0.97769 | 0.00239 | **0.98244** | 0.0016 |
| jpn | jpn | jpn | **1.0** | 0.0 | 0.98268 | 0.00087 | 1.0 | 0.0 | 1.0 | 0.0 | 0.97702 | 0.00087 | 0.96175 | 0.00031 |
| kab | kab | kab | 0.85967 | 0.01221 | 0.85787 | 0.01811 | 0.87886 | 0.00709 | **0.90909** | 0.00362 | 0.8586 | 0.01811 | **0.86043** | 0.01692 |
| kac | kac | kac | **1.0** | 0.0 | **1.0** | 0.0 | 1.0 | 0.0 | 1.0 | 0.0 | 1.0 | 0.0 | 1.0 | 0.0 |
| kam | kam | kam | **0.92406** | 6e-05 | 0.75811 | 0.00011 | 0.91416 | 0.0 | 0.87368 | 0.0 | 0.74892 | 0.00011 | 0.70415 | 0.0001 |
| kan | kan | kan | **1.0** | 0.0 | **1.0** | 0.0 | 1.0 | 0.0 | 1.0 | 0.0 | 1.0 | 0.0 | 1.0 | 0.0 |
| kas | kas | kas | 0.97674 | 0.0 | **0.97753** | 5e-05 | 0.97597 | 0.0 | 0.96945 | 0.0 | 0.97752 | 5e-05 | 0.97545 | 0.0 |
| kat | kat | kat | **1.0** | 0.0 | 1.0 | 0.0 | 1.0 | 0.0 | 1.0 | 0.0 | 1.0 | 0.0 | 1.0 | 0.0 |
| kaz | kaz | kaz | **0.99951** | 0.0 | **0.99951** | 0.0 | 0.99951 | 0.0 | 0.99951 | 0.0 | 0.99951 | 0.0 | 0.99951 | 0.0 |
| kbp | kbp | kbp | 0.99901 | 6e-05 | **1.0** | 0.0 | 0.99901 | 4e-05 | 0.99901 | 4e-05 | 0.99951 | 0.0 | 0.99901 | 0.0 |
| kea | kea | kea | 0.95238 | 0.0 | **0.96099** | 0.0 | 0.9513 | 0.0 | 0.93586 | 0.0 | 0.95831 | 0.0 | 0.94693 | 0.0 |
| khk | khk | khk | **1.0** | 0.0 | **1.0** | 0.0 | 0.36246 | 0.0 | 0.21674 | 0.0 | 1.0 | 0.0 | 1.0 | 0.0 |
| khm | khm | khm | **0.99951** | 0.0 | 0.99901 | 0.0 | 0.99951 | 0.0 | 0.99951 | 0.0 | 0.99901 | 0.0 | 0.99901 | 0.0 |
| kik | kik | kik | **0.96562** | 0.00403 | 0.96357 | 0.00374 | 0.96509 | 0.00295 | 0.96456 | 0.00255 | 0.96257 | 0.00374 | 0.96158 | 0.00357 |
| kin | kin | kin | 0.91471 | 0.00034 | **0.97881** | 0.0013 | 0.91471 | 0.00025 | 0.91471 | 0.00022 | 0.97879 | 0.0013 | 0.97879 | 0.00119 |
| kir | kir | kir | **1.0** | 0.0 | 1.0 | 0.0 | 1.0 | 0.0 | 1.0 | 0.0 | 1.0 | 0.0 | 1.0 | 0.0 |
| kmb | kmb | kmb | 0.96321 | 0.00415 | 0.93613 | 0.00564 | 0.96923 | 0.00253 | **0.97713** | 0.00144 | 0.94106 | 0.00564 | **0.94588** | 0.00357 |
| kmr | kmr | kmr | **0.99901** | 0.0 | 0.99508 | 0.00054 | 0.99901 | 0.0 | 0.99901 | 0.0 | 0.99655 | 0.00054 | **0.99704** | 0.00026 |
| knc | knc | knc | 0.8634 | 0.00153 | 0.86855 | 0.00016 | 0.86869 | 0.00021 | **0.86966** | 4e-05 | 0.86784 | 0.00016 | 0.86745 | 5e-05 |
| kon | kon | kon | **0.99802** | 0.0 | 0.9936 | 0.00054 | 0.99802 | 0.0 | 0.99802 | 0.0 | 0.99507 | 0.00054 | **0.99605** | 0.00021 |
| kor | kor | kor | **1.0** | 0.0 | 0.99606 | 0.00043 | 1.0 | 0.0 | 1.0 | 0.0 | 0.99951 | 0.00043 | **1.0** | 0.0 |
| lao | lao | lao | **1.0** | 0.0 | 0.99951 | 0.0 | 0.99901 | 0.0 | 0.99802 | 0.0 | 0.99901 | 0.0 | 0.99852 | 0.0 |
| lij | lij | lij | **0.99901** | 6e-05 | 0.97738 | 0.00027 | 0.99852 | 4e-05 | 0.99753 | 4e-05 | 0.97735 | 0.00027 | 0.97423 | 0.00016 |
| lim | lim | lim | **0.99253** | 0.0 | 0.98701 | 0.00011 | 0.99253 | 0.0 | 0.99253 | 0.0 | 0.98701 | 0.00011 | 0.98701 | 0.0001 |
| lin | lin | lin | **0.99901** | 0.00011 | 0.99556 | 0.00033 | 0.99852 | 8e-05 | 0.99901 | 4e-05 | 0.99605 | 0.00033 | **0.99654** | 0.00016 |
| lit | lit | lit | **0.99951** | 0.0 | 0.99901 | 0.00011 | 0.99951 | 0.0 | 0.99951 | 0.0 | **0.99951** | 0.00011 | 0.99901 | 5e-05 |

Table 36: Comparison of GlotLID vs NLLB on FLORES-200 benchmark (part 1)

| iso639-3 | FLORES Code(s) | NLLB Code(s) | GlotLID-M | | NLLB | | GlotLID-M θ=.3 | | GlotLID-M θ=.5 | | NLLB θ=.3 | | NLLB θ=.5 | |
|---|---|---|---|---|---|---|---|---|---|---|---|---|---|---|
| | | | F1↑ | FPR↓ | F1↑ | FPR↓ | F1↑ | FPR↓ | F1↑ | FPR↓ | F1↑ | FPR↓ | F1↑ | FPR↓ |
| lmo | lmo | lmo | **0.99554** | 0.00011 | 0.96961 | 0.00119 | 0.99554 | 8e-05 | 0.99504 | 4e-05 | 0.97003 | 0.00119 | 0.96569 | 0.00067 |
| ltg | ltg | ltg | **0.99653** | 0.0 | 0.99203 | 0.0 | 0.99653 | 0.0 | 0.99503 | 0.0 | 0.99203 | 0.0 | 0.99153 | 0.0 |
| ltz | ltz | ltz | **0.99951** | 0.0 | **0.99951** | 0.0 | 0.99901 | 0.0 | 0.99901 | 0.0 | 0.99951 | 0.0 | 0.99951 | 0.0 |
| lua | lua | lua | **0.99653** | 6e-05 | 0.99358 | 0.00038 | 0.99553 | 0.0 | 0.99404 | 0.0 | 0.99357 | 0.00038 | 0.99554 | 0.0001 |
| lug | lug | lug | **0.99653** | 0.0 | 0.99214 | 0.00076 | 0.99603 | 0.0 | 0.99454 | 0.0 | 0.99311 | 0.00076 | 0.99458 | 0.00041 |
| luo | luo | luo | **1.0** | 0.0 | 0.99753 | 5e-05 | 1.0 | 0.0 | 0.99951 | 0.0 | 0.99753 | 5e-05 | 0.99703 | 0.0 |
| lus | lus | lus | 0.99653 | 6e-05 | 0.99454 | 5e-05 | **0.99703** | 0.0 | 0.99653 | 0.0 | 0.99404 | 5e-05 | 0.99303 | 0.0 |
| lvs | lvs | lvs | **0.99655** | 0.00034 | 0.99362 | 0.0007 | 0.92495 | 0.00021 | 0.80189 | 0.00015 | 0.99362 | 0.0007 | 0.99411 | 0.00062 |
| mag | mag | mag | 0.95459 | 0.00136 | 0.9311 | 0.00233 | **0.95507** | 0.00097 | 0.95408 | 0.00048 | 0.9311 | 0.00233 | 0.93218 | 0.00181 |
| mai | mai | mai | 0.97366 | 6e-05 | **0.98709** | 0.00043 | 0.97366 | 4e-05 | 0.97102 | 0.0 | 0.98709 | 0.00043 | 0.98706 | 0.00031 |
| mal | mal | mal | **1.0** | 0.0 | **1.0** | 0.0 | 1.0 | 0.0 | 1.0 | 0.0 | 1.0 | 0.0 | 1.0 | 0.0 |
| mar | mar | mar | **1.0** | 0.0 | 0.99508 | 0.00054 | 1.0 | 0.0 | 1.0 | 0.0 | 0.99655 | 0.00054 | 0.99901 | 0.0001 |
| min | min | min | 0.6616 | 0.00017 | 0.29545 | 5e-05 | **0.66182** | 8e-05 | 0.66116 | 4e-05 | 0.29558 | 5e-05 | 0.26341 | 0.0 |
| mkd | mkd | mkd | **1.0** | 0.0 | **1.0** | 0.0 | 1.0 | 0.0 | 1.0 | 0.0 | 1.0 | 0.0 | 1.0 | 0.0 |
| mlt | mlt | mlt | 0.97401 | 0.00307 | 0.99901 | 0.00011 | 0.99216 | 0.00067 | 0.99803 | 0.00015 | **0.99951** | 0.00011 | 0.99951 | 5e-05 |
| mni | mni | mni | 0.99901 | 6e-05 | **0.99951** | 0.0 | 0.99901 | 4e-05 | 0.99901 | 4e-05 | 0.99951 | 0.0 | 0.99951 | 0.0 |
| mos | mos | mos | **0.98138** | 0.0 | 0.9684 | 0.0 | 0.97415 | 0.0 | 0.96418 | 0.0 | 0.96629 | 0.0 | 0.95992 | 0.0 |
| mri | mri | mri | 0.99901 | 6e-05 | 0.99852 | 5e-05 | **0.99951** | 0.0 | 0.99901 | 0.0 | 0.99852 | 5e-05 | 0.99852 | 0.0 |
| mya | mya | mya | **1.0** | 0.0 | **1.0** | 0.0 | 1.0 | 0.0 | 1.0 | 0.0 | 1.0 | 0.0 | 1.0 | 0.0 |
| nld | nld | nld | 0.99803 | 0.00023 | 0.983 | 0.0019 | **0.99901** | 8e-05 | 0.99901 | 4e-05 | 0.98587 | 0.0019 | 0.98828 | 0.00124 |
| nno | nno | nno | 0.98507 | 0.00045 | 0.9697 | 0.00228 | **0.98606** | 0.00025 | 0.986 | 7e-05 | 0.9697 | 0.00228 | 0.97491 | 0.00155 |
| nob | nob | nob | 0.98185 | 0.00148 | 0.98289 | 0.00152 | 0.95835 | 0.0011 | 0.89931 | 0.00078 | 0.98385 | 0.00152 | **0.98481** | 0.00124 |
| npi | npi | npi | 0.99104 | 0.00011 | 0.99803 | 0.00022 | 0.53362 | 8e-05 | 0.30628 | 0.0 | 0.99803 | 0.00022 | **0.99852** | 0.00016 |
| nso | nso | nso | **0.99704** | 0.00028 | 0.98386 | 0.00146 | 0.99704 | 0.00021 | 0.99655 | 0.00018 | 0.98386 | 0.00146 | 0.98579 | 0.00119 |
| nus | nus | nus | **0.99951** | 0.0 | 0.99803 | 0.00016 | 0.99951 | 0.0 | 0.99951 | 0.0 | 0.99803 | 0.00016 | 0.99852 | 0.0001 |
| nya | nya | nya | 0.99753 | 0.00023 | 0.94604 | 0.00179 | 0.99803 | 0.00013 | **0.99852** | 4e-05 | 0.94636 | 0.00179 | 0.95175 | 0.00047 |
| oci | oci | oci | 0.99951 | 6e-05 | 0.98346 | 0.00179 | **1.0** | 0.0 | 0.99951 | 0.0 | 0.98634 | 0.00179 | 0.99118 | 0.00088 |
| ory | ory | ory | **1.0** | 0.0 | **1.0** | 0.0 | 0.80519 | 0.0 | 0.66314 | 0.0 | 1.0 | 0.0 | 1.0 | 0.0 |
| pag | pag | pag | **0.99852** | 0.0 | 0.99703 | 0.00011 | 0.99852 | 0.0 | 0.99852 | 0.0 | 0.99753 | 0.00011 | 0.99653 | 0.0 |
| pan | pan | pan | **1.0** | 0.0 | **1.0** | 0.0 | 1.0 | 0.0 | 1.0 | 0.0 | 1.0 | 0.0 | 1.0 | 0.0 |
| pap | pap | pap | 0.99069 | 0.00102 | 0.98394 | 0.00174 | 0.99118 | 0.00072 | **0.99557** | 0.0003 | 0.98538 | 0.00174 | 0.98681 | 0.00129 |
| pbt | pbt | pbt | 0.81486 | 0.00051 | **0.99654** | 0.00016 | 0.76381 | 0.00025 | 0.68523 | 0.00011 | 0.99654 | 0.00016 | 0.99555 | 0.00016 |
| pes | pes | pes | 0.57435 | 0.08493 | 0.68739 | 0.04815 | 0.60647 | 0.04812 | 0.57502 | 0.02962 | 0.68763 | 0.04815 | **0.6893** | 0.04553 |
| plt | plt | plt | 0.99852 | 0.0 | **1.0** | 0.0 | 0.99802 | 0.0 | 0.99603 | 0.0 | 1.0 | 0.0 | 1.0 | 0.0 |
| pol | pol | pol | 0.9907 | 0.00108 | 0.98396 | 0.00179 | **0.99167** | 0.00072 | 0.99167 | 0.00063 | 0.9878 | 0.00179 | 0.98925 | 0.00114 |
| por | por | por | 0.99655 | 0.0 | 0.98538 | 0.00157 | **0.99704** | 0.00025 | 0.99704 | 0.00018 | 0.98924 | 0.00157 | 0.99312 | 0.00067 |
| prs | prs | prs | 0.14688 | 0.0192 | **0.49305** | 0.00098 | 0.14348 | 0.01135 | 0.1171 | 0.00716 | 0.49305 | 0.00098 | 0.49305 | 0.00093 |
| quy | quy | quy | 0.75904 | 0.0 | **1.0** | 0.0 | 0.625 | 0.0 | 0.57163 | 0.0 | 1.0 | 0.0 | 1.0 | 0.0 |
| ron | ron | ron | **0.99951** | 0.0 | 0.99852 | 0.00016 | 0.99951 | 0.0 | 0.99951 | 0.0 | 0.99852 | 0.00016 | 0.99852 | 0.00016 |
| run | run | run | 0.92541 | 0.00881 | 0.97824 | 0.00114 | 0.92584 | 0.0065 | 0.92627 | 0.00565 | 0.97824 | 0.00114 | **0.97919** | 0.00093 |
| rus | rus | rus | **0.99901** | 6e-05 | **0.99901** | 0.00011 | 0.99901 | 4e-05 | 0.99901 | 4e-05 | 0.99901 | 0.00011 | 0.99901 | 0.0001 |
| sag | sag | sag | **0.99901** | 0.0 | 0.99703 | 5e-05 | 0.99901 | 0.0 | 0.99901 | 0.0 | 0.99752 | 5e-05 | 0.99703 | 0.0 |
| san | san | san | **0.99104** | 6e-05 | 0.98853 | 0.00011 | 0.99104 | 4e-05 | 0.99103 | 0.0 | 0.98902 | 0.00011 | 0.98495 | 0.0 |
| sat | sat | sat | **1.0** | 0.0 | **1.0** | 0.0 | 1.0 | 0.0 | 1.0 | 0.0 | 1.0 | 0.0 | 1.0 | 0.0 |
| scn | scn | scn | 0.99802 | 6e-05 | 0.99361 | 0.0006 | 0.99802 | 4e-05 | **0.99852** | 0.0 | 0.99458 | 0.0006 | 0.99458 | 0.00041 |
| shn | shn | shn | **1.0** | 0.0 | 0.99852 | 0.0 | 1.0 | 0.0 | 1.0 | 0.0 | 0.99802 | 0.0 | 0.99503 | 0.0 |
| sin | sin | sin | **1.0** | 0.0 | **1.0** | 0.0 | 1.0 | 0.0 | 1.0 | 0.0 | 1.0 | 0.0 | 1.0 | 0.0 |
| slk | slk | slk | 0.99852 | 6e-05 | 0.99951 | 5e-05 | 0.99852 | 4e-05 | **0.99901** | 0.0 | **1.0** | 5e-05 | 1.0 | 0.0 |
| slv | slv | slv | 0.99459 | 0.00062 | 0.99852 | 0.00016 | 0.99655 | 0.0003 | **0.99951** | 4e-05 | 0.99951 | 0.00016 | **1.0** | 0.0 |
| smo | smo | smo | 0.99603 | 0.0 | 0.99852 | 0.00011 | 0.99603 | 0.0 | 0.99603 | 0.0 | **0.99951** | 0.00011 | 0.99951 | 0.0 |
| sna | sna | sna | 0.99901 | 0.00011 | 0.99411 | 0.00065 | 0.99951 | 4e-05 | **1.0** | 0.0 | 0.99508 | 0.00065 | **0.99655** | 0.00031 |
| snd | snd | snd | 0.99362 | 0.00011 | 0.99704 | 0.00033 | 0.99508 | 0.00042 | **0.99704** | 0.00022 | 0.99704 | 0.00033 | **0.99852** | 0.00016 |
| som | som | som | 0.96657 | 0.00398 | **1.0** | 0.0 | 0.98973 | 0.00089 | **0.99803** | 0.00015 | 1.0 | 0.0 | 1.0 | 0.0 |
| sot | sot | sot | **1.0** | 0.0 | 0.75523 | 0.0 | 1.0 | 0.0 | 1.0 | 0.0 | 0.75523 | 0.0 | 0.75062 | 0.0 |
| spa | spa | spa | 0.99508 | 0.00057 | 0.99215 | 0.00081 | 0.99508 | 0.00042 | 0.99508 | 0.00033 | 0.99361 | 0.00081 | **0.99655** | 0.00031 |
| srd | srd | srd | **0.99951** | 0.0 | 0.97726 | 0.0 | 0.99901 | 0.0 | 0.99901 | 0.0 | 0.97519 | 0.0 | 0.96735 | 0.0 |
| srp | srp | srp | 0.99901 | 0.00011 | **1.0** | 0.0 | 0.99951 | 4e-05 | **1.0** | 0.0 | 1.0 | 0.0 | 1.0 | 0.0 |
| ssw | ssw | ssw | **0.99654** | 0.00023 | 0.99155 | 0.00016 | 0.99456 | 0.00017 | 0.99455 | 0.00011 | 0.99155 | 0.00016 | 0.99204 | 5e-05 |
| sun | sun | sun | 0.99012 | 0.00057 | 0.95988 | 0.00277 | **0.99355** | 0.00013 | 0.99304 | 4e-05 | 0.96129 | 0.00277 | 0.96781 | 0.00155 |
| swe | swe | swe | 0.99754 | 0.0 | 0.99901 | 0.0 | 0.99951 | 4e-05 | **1.0** | 0.0 | 0.99951 | 5e-05 | 0.99951 | 0.0 |
| swh | swh | swh | **0.94869** | 0.00187 | 0.88153 | 0.01475 | 0.91684 | 0.00072 | 0.74599 | 0.00018 | 0.88811 | 0.01475 | 0.91328 | 0.00988 |
| szl | szl | szl | **0.99104** | 6e-05 | 0.98753 | 0.00016 | 0.99104 | 4e-05 | 0.99104 | 4e-05 | 0.98852 | 0.00016 | 0.98901 | 0.0 |
| tam | tam | tam | **1.0** | 0.0 | **1.0** | 0.0 | 1.0 | 0.0 | 1.0 | 0.0 | 1.0 | 0.0 | 1.0 | 0.0 |
| taq | taq | taq | 0.80642 | 0.02022 | 0.82223 | 0.0 | 0.83861 | 0.00827 | **0.84449** | 0.00332 | 0.82189 | 0.0 | 0.81776 | 0.0 |
| tat | tat | tat | **1.0** | 0.0 | 0.99951 | 0.0 | 1.0 | 0.0 | 1.0 | 0.0 | 0.99951 | 0.0 | 0.99951 | 0.0 |
| tel | tel | tel | **1.0** | 0.0 | **1.0** | 0.0 | 1.0 | 0.0 | 1.0 | 0.0 | 1.0 | 0.0 | 1.0 | 0.0 |
| tgk | tgk | tgk | **1.0** | 0.0 | **1.0** | 0.0 | 1.0 | 0.0 | 1.0 | 0.0 | 1.0 | 0.0 | 1.0 | 0.0 |
| tgl | tgl | tgl | **0.99901** | 0.00011 | 0.99704 | 0.00027 | 0.99901 | 4e-05 | 0.99901 | 4e-05 | 0.99803 | 0.00027 | 0.99852 | 0.0001 |
| tha | tha | tha | **1.0** | 0.0 | **1.0** | 0.0 | 1.0 | 0.0 | 1.0 | 0.0 | 1.0 | 0.0 | 1.0 | 0.0 |
| tir | tir | tir | 0.98851 | 0.0 | **0.99951** | 0.0 | 0.98851 | 0.0 | 0.988 | 0.0 | 0.99951 | 0.0 | 0.99951 | 0.0 |
| tpi | tpi | tpi | **0.99951** | 0.0 | 0.99802 | 0.0 | 0.99951 | 0.0 | 0.99901 | 0.0 | 0.99802 | 0.0 | 0.99752 | 0.0 |
| tsn | tsn | tsn | **0.99753** | 6e-05 | 0.84237 | 0.02039 | 0.99753 | 4e-05 | 0.99753 | 4e-05 | 0.84259 | 0.02039 | 0.84577 | 0.01888 |
| tso | tso | tso | 0.99803 | 0.00023 | 0.99069 | 0.00098 | 0.99753 | 0.00017 | **0.99901** | 4e-05 | 0.99214 | 0.00098 | 0.99606 | 0.00031 |
| tuk | tuk | tuk | 0.99803 | 0.00023 | **1.0** | 0.0 | 0.99951 | 0.0 | 0.99951 | 4e-05 | 1.0 | 0.0 | 1.0 | 0.0 |
| tum | tum | tum | 0.99852 | 0.00011 | 0.98155 | 0.00201 | **0.99901** | 4e-05 | 0.99901 | 0.0 | 0.98251 | 0.00201 | 0.98972 | 0.00103 |
| tur | tur | tur | 0.9907 | 0.00108 | 0.98348 | 0.00184 | 0.9907 | 0.0008 | **0.99119** | 0.00066 | 0.98491 | 0.00184 | 0.98539 | 0.00155 |
| twi | twi | twi | **0.99951** | 0.0 | 0.8426 | 0.01231 | 0.99752 | 0.0 | 0.99503 | 0.0 | 0.84299 | 0.01231 | 0.84284 | 0.01164 |
| tzm | tzm | tzm | 0.94421 | 0.0054 | 0.88539 | 0.01421 | 0.94421 | 0.00401 | **0.94524** | 0.00318 | 0.8849 | 0.01421 | 0.8849 | 0.01356 |
| uig | uig | uig | 0.99901 | 0.00011 | 0.99951 | 5e-05 | **1.0** | 0.0 | 1.0 | 0.0 | 0.99951 | 5e-05 | 0.99951 | 5e-05 |
| ukr | ukr | ukr | 0.99951 | 6e-05 | **1.0** | 0.0 | 0.99951 | 4e-05 | 0.99951 | 4e-05 | 1.0 | 0.0 | 1.0 | 0.0 |
| umb | umb | umb | 0.88615 | 0.00117 | 0.96869 | 0.00228 | 0.8781 | 0.00021 | 0.834 | 0.00011 | 0.97247 | 0.00228 | **0.98681** | 0.00098 |
| urd | urd | urd | 0.98346 | 0.00187 | 0.97354 | 0.00298 | 0.98346 | 0.00139 | **0.99021** | 0.0007 | 0.97401 | 0.00298 | 0.97495 | 0.00269 |
| uzn | uzn | uzn | 0.96885 | 0.00364 | 0.99852 | 0.00016 | 0.87839 | 0.00089 | 0.76439 | 0.0003 | 0.99852 | 0.00016 | **0.99951** | 5e-05 |
| vec | vec | vec | **0.99703** | 0.0 | 0.99159 | 0.00038 | 0.99703 | 4e-05 | 0.99653 | 0.0 | 0.99208 | 0.00038 | 0.99206 | 0.00016 |
| vie | vie | vie | 0.99951 | 6e-05 | 0.98925 | 0.00119 | **1.0** | 0.0 | 1.0 | 0.0 | 0.99951 | 0.00119 | **1.0** | 0.0 |
| war | war | war | 0.99951 | 0.0 | 0.99901 | 0.00011 | 0.99951 | 0.0 | 0.99951 | 0.0 | 0.99951 | 0.00011 | **1.0** | 0.0 |
| wol | wol | wol | **0.99852** | 0.0 | 0.99554 | 0.00011 | 0.99802 | 0.0 | 0.99802 | 0.0 | 0.99554 | 0.00011 | 0.99554 | 5e-05 |
| xho | xho | xho | **0.99118** | 0.00097 | 0.97793 | 0.00163 | 0.98968 | 0.00067 | 0.99113 | 0.00044 | 0.97937 | 0.00163 | 0.98566 | 0.00072 |
| ydd | ydd | ydd | 0.99603 | 0.0 | | | | | | | 1.0 | 0.0 | 1.0 | 0.0 |
| yor | yor | yor | 0.99406 | 0.00023 | 0.99556 | 0.00033 | 0.99355 | 0.00013 | 0.99053 | 4e-05 | 0.99654 | 0.00033 | **0.99752** | 0.0 |
| yue | yue | yue | 0.00394 | 0.0 | **0.4777** | 0.03395 | | | | | 0.47231 | 0.03395 | 0.46292 | 0.02752 |
| zho | zho | zho | **1.0** | 0.0 | 0.69715 | 0.02543 | 1.0 | 0.0 | 1.0 | 0.0 | 0.65912 | 0.02543 | 0.617 | 0.01821 |
| zsm | zsm | zsm | 0.93506 | 0.00307 | 0.93459 | 0.00342 | 0.94641 | 0.00127 | **0.94972** | 0.00081 | 0.93924 | 0.00342 | 0.94869 | 0.00171 |
| zul | zul | zul | 0.96893 | 0.0 | 0.96955 | 0.00293 | 0.96893 | 0.0 | 0.96735 | 0.0 | 0.97379 | 0.00293 | **0.97901** | 0.00176 |

Table 37: Comparison of GlotLID vs NLLB on FLORES-200 benchmark (part 2)

| iso639-3 | UDHR Code(s) | CLD3 Code(s) | GlotLID-M F1↑ | GlotLID-M FPR↓ | CLD3 F1↑ | CLD3 FPR↓ | GlotLID-M θ=.3 F1↑ | GlotLID-M θ=.3 FPR↓ | GlotLID-M θ=.5 F1↑ | GlotLID-M θ=.5 FPR↓ | CLD3 θ=.5 F1↑ | CLD3 θ=.5 FPR↓ | CLD3 θ=.7 F1↑ | CLD3 θ=.7 FPR↓ |
|---|---|---|---|---|---|---|---|---|---|---|---|---|---|---|
| aar | aar | - | | | | | | | | | | | | |
| abk | abk | - | | | | | | | | | | | | |
| ace | ace | - | **0.91603** | 0.00125 | | | 0.91603 | 0.00122 | 0.91603 | 0.00118 | | | | |
| acu | acu | - | **0.58635** | 0.0 | | | 0.58635 | 0.0 | 0.58065 | 0.0 | | | | |
| ada | ada | - | 0.91045 | 0.0015 | | | 0.9313 | 0.0011 | **0.93846** | 0.00094 | | | | |
| ady | ady | - | **0.83495** | 0.0 | | | 0.83495 | 0.0 | | | | | | |
| afr | afr | af | 0.95238 | 0.00075 | 0.85106 | 0.00096 | 0.96774 | 0.00049 | **0.97561** | 0.00035 | 0.89552 | 0.00064 | 0.93023 | 0.00041 |
| agr | agr | - | 0.81429 | 0.00037 | | | **0.82014** | 0.00024 | 0.82014 | 0.00024 | | | | |
| aii | aii | - | | | | | | | | | | | | |
| ajg | ajg | - | 0.64516 | 0.00822 | | | 0.65217 | 0.0078 | **0.65574** | 0.00743 | | | | |
| alt | alt | - | 0.95495 | 0.00037 | | | **0.96364** | 0.00024 | 0.96364 | 0.00024 | | | | |
| amc | amc | - | | | | | | | | | | | | |
| ame | ame | - | **1.0** | 0.0 | | | 1.0 | 0.0 | 1.0 | 0.0 | | | | |
| amh | amh | am | **1.0** | 0.0 | 0.7 | 0.00274 | 1.0 | 0.0 | 1.0 | 0.0 | 0.7 | 0.00273 | 0.7 | 0.00272 |
| ami | ami | - | **0.20896** | 0.0 | | | 0.125 | 0.0 | 0.09524 | 0.0 | | | | |
| amr | amr | - | **1.0** | 0.0 | | | 0.99187 | 0.0 | 0.99187 | 0.0 | | | | |
| ara | arb | ar | 0.89552 | 0.00174 | 0.83333 | 0.00109 | 0.93023 | 0.0011 | **0.98361** | 0.00024 | 0.85106 | 0.00096 | 0.92308 | 0.00045 |
| arl | arl | - | **0.99187** | 0.0 | | | 0.99187 | 0.0 | 0.99187 | 0.0 | | | | |
| arn | arn | - | 0.93913 | 0.00012 | | | **0.94737** | 0.0 | 0.94737 | 0.0 | | | | |
| ast | ast | - | 0.97521 | 0.00012 | | | 0.97521 | 0.00012 | **0.98333** | 0.0 | | | | |
| auc | auc | - | **0.01504** | 0.0 | | | | | | | | | | |
| ayr | ayr | - | **0.99174** | 0.00012 | | | 0.91071 | 0.00012 | 0.84615 | 0.0 | | | | |
| aze | azj/azb | az | 0.6413 | 0.0081 | 0.21034 | 0.01446 | 0.6413 | 0.00792 | **0.64658** | 0.00731 | 0.22736 | 0.01253 | 0.25708 | 0.00973 |
| bam | bam | - | 0.49682 | 0.00723 | | | 0.5098 | 0.00658 | **0.55319** | 0.00495 | | | | |
| ban | ban | - | **0.98361** | 0.0 | | | 0.98361 | 0.0 | 0.97521 | 0.0 | | | | |
| bax | bax | - | | | | | | | | | | | | |
| bba | bba | - | 0.92187 | 0.00112 | | | 0.92913 | 0.00098 | **0.93548** | 0.00071 | | | | |
| bci | bci | - | 0.97521 | 0.00037 | | | 0.9916 | 0.00012 | **1.0** | 0.0 | | | | |
| bcl | bcl | - | **1.0** | 0.0 | | | 1.0 | 0.0 | 0.9916 | 0.0 | | | | |
| bel | bel | be | **0.98333** | 0.0 | 0.50633 | 0.00529 | 0.98333 | 0.0 | 0.98333 | 0.0 | 0.53571 | 0.00469 | 0.61856 | 0.0033 |
| bem | bem | - | **0.98333** | 0.00025 | | | 0.98333 | 0.00024 | 0.98333 | 0.00024 | | | | |
| ben | ben | bn | **1.0** | 0.0 | **1.0** | 0.0 | 1.0 | 0.0 | 1.0 | 0.0 | 1.0 | 0.0 | 1.0 | 0.0 |
| bfa | bfa | - | | | | | | | | | | | | |
| bho | bho | - | **0.78519** | 0.00037 | | | 0.78519 | 0.00037 | 0.77273 | 0.00024 | | | | |
| bin | bin | - | 0.9927 | 0.00012 | | | **1.0** | 0.0 | 1.0 | 0.0 | | | | |
| bis | bis | - | **1.0** | 0.0 | | | 1.0 | 0.0 | 1.0 | 0.0 | | | | |
| blt | blt | - | | | | | | | | | | | | |
| boa | boa | - | 0.99213 | 0.00012 | | | 0.99213 | 0.00012 | **1.0** | 0.0 | | | | |
| bod | bod | - | **0.89091** | 0.00012 | | | 0.89091 | 0.00012 | 0.89091 | 0.00012 | | | | |
| bos | bos | bs | 0.18103 | 0.01134 | 0.23913 | 0.00561 | 0.18605 | 0.00914 | 0.14607 | 0.00531 | **0.27386** | 0.00401 | 0.0 | 5e-05 |
| bre | bre | - | **0.98361** | 0.0 | | | 0.98361 | 0.0 | 0.98361 | 0.0 | | | | |
| buc | buc | - | | | | | | | | | | | | |
| bug | bug | - | 0.95312 | 0.00062 | | | 0.96063 | 0.00049 | **0.976** | 0.00024 | | | | |
| bul | bul | bg/bg-Latn | 0.96 | 0.00062 | 0.52212 | 0.00488 | 0.97561 | 0.00061 | **0.97561** | 0.00035 | 0.59 | 0.00369 | 0.67052 | 0.00249 |
| bum | bum | - | **0.54762** | 0.00025 | | | 0.53659 | 0.00024 | 0.46154 | 0.00012 | | | | |
| cab | cab | - | **1.0** | 0.0 | | | 1.0 | 0.0 | 1.0 | 0.0 | | | | |
| cak | cak | - | **1.0** | 0.0 | | | 1.0 | 0.0 | 1.0 | 0.0 | | | | |
| cat | cat | ca | 0.9375 | 0.001 | 0.24691 | 0.0167 | 0.95238 | 0.00073 | **0.96774** | 0.00047 | 0.27397 | 0.01449 | 0.3352 | 0.01077 |
| cbi | cbi | - | 0.9771 | 0.00025 | | | **0.99225** | 0.0 | 0.99225 | 0.0 | | | | |
| cbr | cbr | - | **0.7033** | 0.0 | | | 0.65909 | 0.0 | 0.54321 | 0.0 | | | | |
| cbs | cbs | - | **0.67308** | 0.00037 | | | 0.66 | 0.00012 | 0.61053 | 0.0 | | | | |
| cbt | cbt | - | 0.97479 | 0.00012 | | | 0.97479 | 0.00012 | **0.98305** | 0.0 | | | | |
| cbu | cbu | - | **0.18462** | 0.0 | | | 0.09677 | 0.0 | 0.03333 | 0.0 | | | | |
| ccp | ccp | - | | | | | | | | | | | | |
| ceb | ceb | ceb | 0.96721 | 0.0005 | 0.43182 | 0.00675 | 0.9916 | 0.00012 | **1.0** | 0.0 | 0.456 | 0.00611 | 0.48718 | 0.00534 |
| ces | ces | cs | **0.98387** | 0.0 | 0.6776 | 0.00265 | 0.98387 | 0.0 | 0.98387 | 0.0 | 0.74699 | 0.00187 | 0.79487 | 0.0014 |
| cfm | cfm | - | **0.85714** | 0.0 | | | 0.85714 | 0.0 | 0.84615 | 0.0 | | | | |
| cha | cha | - | **0.8381** | 0.00012 | | | 0.82353 | 0.0 | 0.8 | 0.0 | | | | |
| chj | chj | - | 0.12727 | 0.00536 | | | 0.14286 | 0.00378 | **0.15385** | 0.00141 | | | | |
| chk | chk | - | 0.97521 | 0.00012 | | | **0.98333** | 0.0 | 0.97479 | 0.0 | | | | |
| chr | chr | - | **0.03333** | 0.0 | | | | | | | | | | |
| chv | chv | - | **0.86154** | 0.0 | | | 0.86154 | 0.0 | 0.86154 | 0.0 | | | | |
| cic | cic | - | | | | | | | | | | | | |
| cjk | cjk | - | **0.92641** | 0.00037 | | | 0.92641 | 0.00037 | 0.92035 | 0.00012 | | | | |
| cjs | cjs | - | **0.65217** | 0.0 | | | 0.65217 | 0.0 | 0.63736 | 0.0 | | | | |
| ckb | ckb | - | | | | | | | | | | | | |
| cnh | cnh | - | 0.93023 | 0.00112 | | | 0.9375 | 0.00098 | **0.94488** | 0.00083 | | | | |
| cni | cni | - | 0.90226 | 0.00162 | | | 0.90909 | 0.00146 | **0.91603** | 0.0013 | | | | |
| cnr | cnr | - | | | | | | | | | | | | |
| cof | cof | - | **0.74747** | 0.0 | | | 0.63736 | 0.0 | 0.54118 | 0.0 | | | | |
| cos | cos | co | 0.95082 | 0.0005 | 0.30227 | 0.01264 | **0.98305** | 0.0 | 0.98305 | 0.0 | 0.33241 | 0.01098 | 0.41812 | 0.00756 |
| cot | cot | - | 0.96774 | 0.00025 | | | 0.96774 | 0.00024 | **0.97561** | 0.00012 | | | | |
| cpu | cpu | - | **0.89908** | 0.0 | | | 0.89908 | 0.0 | 0.89908 | 0.0 | | | | |
| crh | crh | - | **0.98361** | 0.00025 | | | 0.98361 | 0.00024 | 0.98361 | 0.00024 | | | | |
| cri | cri | - | 0.84404 | 0.00037 | | | **0.85185** | 0.00024 | 0.80769 | 0.00024 | | | | |
| crs | crs | - | **1.0** | 0.0 | | | 1.0 | 0.0 | 1.0 | 0.0 | | | | |
| csa | csa | - | | | | | | | | | | | | |
| csw | csw | - | **0.0** | 0.00199 | | | 0.0 | 0.00049 | | | | | | |
| ctd | ctd | - | **0.78431** | 0.0 | | | 0.78431 | 0.0 | 0.74747 | 0.0 | | | | |
| cym | cym | cy | **1.0** | 0.0 | 0.46792 | 0.00643 | 1.0 | 0.0 | 1.0 | 0.0 | 0.55111 | 0.0046 | 0.63918 | 0.00317 |
| dag | dag | - | | | | | | | | | | | | |
| dan | dan | da | 0.85714 | 0.00262 | 0.91473 | 0.00032 | 0.91304 | 0.00146 | **0.98437** | 0.00024 | 0.93651 | 0.00018 | 0.95161 | 9e-05 |
| ddn | ddn | - | | | | | | | | | | | | |
| deu | deu | de | **0.98745** | 0.00012 | 0.92549 | 0.00078 | 0.98745 | 0.00012 | 0.98745 | 0.00012 | 0.944 | 0.00055 | 0.95547 | 0.00041 |
| dga | dga | - | 0.71166 | 0.00548 | | | 0.73885 | 0.00463 | **0.8** | 0.00307 | | | | |
| dip | dip | - | | | | | | | | | | | | |
| div | div | - | **0.96774** | 0.0 | | | 0.96774 | 0.0 | 0.93333 | 0.0 | | | | |
| duu | duu | - | | | | | | | | | | | | |
| dyo | dyo | - | **0.97391** | 0.0 | | | 0.97391 | 0.0 | 0.96491 | 0.0 | | | | |
| dyu | dyu | - | **0.23188** | 0.00785 | | | 0.22059 | 0.00756 | 0.17323 | 0.00672 | | | | |
| dzo | dzo | - | **0.90769** | 0.00137 | | | 0.90769 | 0.00134 | 0.90769 | 0.0013 | | | | |
| ell | ell | el/el-Latn | **0.97908** | 0.0 | 0.81879 | 0.00246 | 0.97908 | 0.0 | 0.97908 | 0.0 | 0.91045 | 0.00109 | 0.95686 | 0.0005 |
| emk | emk | - | | | | | | | | | | | | |
| eng | eng | en | 0.85294 | 0.00224 | 0.40956 | 0.00789 | 0.87218 | 0.00183 | **0.8855** | 0.00153 | 0.40972 | 0.0077 | 0.42143 | 0.00729 |
| epo | epo | eo | 0.96825 | 0.0005 | 0.35882 | 0.00995 | **0.976** | 0.00037 | 0.976 | 0.00035 | 0.39228 | 0.00861 | 0.43636 | 0.00697 |
| ese | ese | - | 0.75 | 0.00012 | | | 0.73118 | 0.0 | **0.75789** | 0.0 | | | | |
| est | ekk | et | 0.59701 | 0.01009 | 0.40816 | 0.00794 | 0.63492 | 0.00841 | **0.7362** | 0.00507 | 0.46332 | 0.00633 | 0.59406 | 0.00371 |
| eus | eus | eu | 0.9313 | 0.00112 | 0.36747 | 0.00958 | 0.96825 | 0.00049 | **0.98387** | 0.00024 | 0.488 | 0.00583 | 0.60396 | 0.00362 |

Table 38: Comparison of GlotLID vs CLD3 on UDHR benchmark (part 1)

| iso639-3 | UDHR Code(s) | CLD3 Code(s) | GlotLID-M | | CLD3 | | GlotLID-M $\theta$=.3 | | GlotLID-M $\theta$=.5 | | CLD3 $\theta$=.5 | | CLD3 $\theta$=.7 | |
|---|---|---|---|---|---|---|---|---|---|---|---|---|---|---|
| | | | F1↑ | FPR↓ | F1↑ | FPR↓ | F1↑ | FPR↓ | F1↑ | FPR↓ | F1↑ | FPR↓ | F1↑ | FPR↓ |
| eve | eve | - | **0.27586** | 0.00361 | | | 0.27451 | 0.00207 | 0.22222 | 0.00012 | | | | |
| evn | evn | - | **0.21154** | 0.00424 | | | 0.2029 | 0.00037 | 0.03333 | 0.0 | | | | |
| ewe | ewe | - | **0.98361** | 0.00025 | | | 0.98361 | 0.00024 | 0.98361 | 0.00024 | | | | |
| fao | fao | - | **0.98305** | 0.0 | | | 0.98305 | 0.0 | 0.98305 | 0.0 | | | | |
| fas | pes/prs | fa | 0.90152 | 0.00311 | 0.86331 | 0.00173 | **0.90494** | 0.00293 | 0.90494 | 0.00283 | 0.89888 | 0.00123 | **0.94118** | 0.00068 |
| fat | fat | - | **0.97521** | 0.00025 | | | 0.97521 | 0.00024 | 0.97521 | 0.00024 | | | | |
| fij | fij | - | **1.0** | 0.0 | | | 1.0 | 0.0 | 1.0 | 0.0 | | | | |
| fil | - | fil | | | | 0.01186 | | | | | | 0.01039 | | 0.00851 |
| fin | fin | fi | 0.38066 | 0.02554 | 0.27391 | 0.01524 | 0.39252 | 0.02377 | **0.41311** | 0.0211 | 0.3 | 0.0134 | 0.33871 | 0.01113 |
| fkv | fkv | - | **0.28571** | 0.0 | | | 0.28571 | 0.0 | 0.23529 | 0.0 | | | | |
| fon | fon | - | 0.94118 | 0.0005 | | | **0.95726** | 0.00024 | | | | | | |
| fra | fra | fr | 0.95238 | 0.00075 | 0.37855 | 0.00899 | 0.95935 | 0.00049 | **0.9661** | 0.00012 | 0.41379 | 0.00775 | 0.48583 | 0.00575 |
| fry | fry | fy | 0.99174 | 0.00012 | 0.67039 | 0.00269 | **1.0** | 0.0 | 1.0 | 0.0 | 0.74074 | 0.00191 | 0.83333 | 0.00109 |
| fuf | fuf | - | **0.04762** | 0.0005 | | | 0.01613 | 0.00049 | 0.0 | 0.00035 | | | | |
| fur | fur | - | 0.91473 | 0.00125 | | | 0.95935 | 0.00049 | **0.96721** | 0.00035 | | | | |
| fuv | fuv | - | 0.77912 | 0.00237 | | | 0.80833 | 0.00122 | **0.81197** | 0.00071 | | | | |
| fvr | fvr | - | | | | | | | | | | | | |
| gaa | gaa | - | 0.93846 | 0.001 | | | 0.96825 | 0.00049 | **0.98387** | 0.00024 | | | | |
| gag | gag | - | **0.93103** | 0.0 | | | 0.93103 | 0.0 | 0.93103 | 0.0 | | | | |
| gaz | gaz | - | 0.83221 | 0.00311 | | | **0.88235** | 0.00171 | 0.83761 | 0.00071 | | | | |
| gjn | gjn | - | 0.90476 | 0.001 | | | 0.93443 | 0.00049 | **0.95798** | 0.00012 | | | | |
| gkp | gkp | - | 0.92063 | 0.00125 | | | 0.93548 | 0.00098 | **0.96667** | 0.00047 | | | | |
| gla | gla | gd | 0.94574 | 0.00025 | 0.50769 | 0.00584 | 0.95312 | 0.00012 | **0.96063** | 0.0 | 0.6055 | 0.00392 | 0.71351 | 0.0024 |
| gld | gld | - | | | | | | | | | | | | |
| gle | gle | ga | 0.95 | 0.001 | 0.55882 | 0.00324 | **0.96815** | 0.00061 | 0.96815 | 0.00059 | 0.63333 | 0.00214 | 0.74026 | 0.00095 |
| glg | glg | gl | **0.98305** | 0.00012 | 0.57711 | 0.00383 | 0.98305 | 0.00012 | 0.98305 | 0.00012 | 0.64804 | 0.00283 | 0.73885 | 0.00181 |
| glv | glv | - | **1.0** | 0.0 | | | 1.0 | 0.0 | 1.0 | 0.0 | | | | |
| gsw | gsw | - | 0.98333 | 0.00012 | | | **0.9916** | 0.0 | 0.9916 | 0.0 | | | | |
| guc | guc | - | 0.96063 | 0.00062 | | | 0.96825 | 0.00049 | **0.98387** | 0.00024 | | | | |
| gug | gug | - | **0.80992** | 0.0 | | | 0.7027 | 0.0 | 0.62857 | 0.0 | | | | |
| guj | guj | gu | **1.0** | 0.0 | **1.0** | 0.0 | 1.0 | 0.0 | 1.0 | 0.0 | 1.0 | 0.0 | 1.0 | 0.0 |
| guk | guk | | | | | | | | | | | | | |
| guu | guu | | | | | | | | | | | | | |
| gyr | gyr | - | **0.62745** | 0.0 | | | 0.57143 | 0.0 | 0.44444 | 0.0 | | | | |
| hat | hat | ht | 0.90706 | 0.00311 | 0.37931 | 0.01802 | 0.9313 | 0.00219 | **0.95312** | 0.00141 | 0.41724 | 0.01536 | 0.48303 | 0.01168 |
| hau | hau | ha | 0.94488 | 0.00262 | 0.32727 | 0.03376 | 0.96 | 0.00183 | **0.97297** | 0.00118 | 0.39173 | 0.02548 | 0.49587 | 0.01656 |
| haw | haw | haw | **1.0** | 0.0 | 0.17101 | 0.0261 | 1.0 | 0.0 | 1.0 | 0.0 | 0.20205 | 0.02124 | 0.26818 | 0.01457 |
| heb | heb | iw | **1.0** | 0.0 | 0.98305 | 9e-05 | 1.0 | 0.0 | 1.0 | 0.0 | 0.99145 | 5e-05 | **1.0** | 0.0 |
| hil | hil | - | **1.0** | 0.0 | | | 1.0 | 0.0 | 1.0 | 0.0 | | | | |
| hin | hin | hi/hi-Latn | 0.62 | 0.00947 | 0.17175 | 0.02728 | 0.62312 | 0.00914 | **0.62944** | 0.00861 | 0.19528 | 0.02329 | 0.23664 | 0.0181 |
| hlt | hlt | - | **0.92982** | 0.0 | | | 0.92982 | 0.0 | 0.89091 | 0.0 | | | | |
| hmn | hms/hnj/hea | hmn | 0.02198 | 0.0 | **0.41538** | 0.00119 | 0.02198 | 0.0 | | | 0.329 | 0.00059 | 0.21596 | 0.00045 |
| hna | hna | - | | | | | | | | | | | | |
| hni | hni | | | | | | | | | | | | | |
| hns | hns | - | 0.89655 | 0.00025 | | | **0.91228** | 0.0 | 0.91228 | 0.0 | | | | |
| hrv | hrv | hr | 0.60302 | 0.00984 | 0.44324 | 0.00383 | 0.62176 | 0.0089 | **0.69364** | 0.00625 | 0.45055 | 0.00369 | | |
| hsb | hsb | - | **1.0** | 0.0 | | | 1.0 | 0.0 | 1.0 | 0.0 | | | | |
| hun | hun | hu | 0.82192 | 0.00324 | 0.21779 | 0.01966 | 0.84507 | 0.00268 | **0.89552** | 0.00165 | 0.2649 | 0.01518 | 0.33803 | 0.01063 |
| hus | hus | - | 0.98082 | 0.00037 | | | **0.98615** | 0.0 | 0.97479 | 0.0 | | | | |
| huu | huu | - | **0.98305** | 0.0 | | | 0.96552 | 0.0 | 0.96552 | 0.0 | | | | |
| hye | hye | hy | **1.0** | 0.0 | **1.0** | 0.0 | 1.0 | 0.0 | 1.0 | 0.0 | 1.0 | 0.0 | 1.0 | 0.0 |
| ibb | ibb | - | | | | | | | | | | | | |
| ibo | ibo | ig | 0.98718 | 0.00025 | 0.29903 | 0.01647 | 0.99355 | 0.00012 | **1.0** | 0.0 | 0.35484 | 0.01276 | 0.46526 | 0.00801 |
| ido | ido | - | **0.95** | 0.00012 | | | 0.94915 | 0.00012 | 0.94915 | 0.0 | | | | |
| idu | idu | - | **0.0** | 0.00012 | | | 0.0 | 0.00012 | | | | | | |
| iii | iii | - | | | | | | | | | | | | |
| ijs | ijs | - | | | | | | | | | | | | |
| ike | ike | - | **0.97872** | 0.00025 | | | 0.97872 | 0.00024 | 0.97872 | 0.00024 | | | | |
| ilo | ilo | - | 0.91339 | 0.00137 | | | 0.928 | 0.0011 | **0.97479** | 0.00035 | | | | |
| ina | ina | - | 0.84892 | 0.00249 | | | 0.90769 | 0.00134 | **0.944** | 0.00071 | | | | |
| isl | isl | is | **0.9916** | 0.00012 | 0.53953 | 0.00447 | 0.9916 | 0.00012 | 0.9916 | 0.00012 | 0.59184 | 0.0036 | 0.60733 | 0.00335 |
| ita | ita | it | 0.78947 | 0.00386 | 0.30227 | 0.01259 | 0.83916 | 0.00268 | **0.86957** | 0.002 | 0.32698 | 0.01121 | 0.39216 | 0.00837 |
| jav | jav | jv | **0.97581** | 0.0005 | 0.22139 | 0.01601 | 0.97581 | 0.00049 | 0.97561 | 0.00035 | 0.26517 | 0.01199 | 0.31892 | 0.00851 |
| jiv | jiv | - | 0.48583 | 0.01557 | | | **0.53881** | 0.0139 | 0.53881 | 0.01155 | | | | |
| jpn | jpn | ja/ja-Latn | 0.7861 | 0.00984 | 0.36616 | 0.02277 | **0.79245** | 0.00926 | 0.79245 | 0.00896 | 0.44207 | 0.01654 | 0.50699 | 0.01263 |
| kaa | kaa | - | 0.96667 | 0.00025 | | | 0.96667 | 0.00024 | **0.97479** | 0.00012 | | | | |
| kal | kal | - | **0.98305** | 0.0 | | | 0.98305 | 0.0 | 0.98305 | 0.0 | | | | |
| kan | kan | kn | **1.0** | 0.0 | **1.0** | 0.0 | 1.0 | 0.0 | 1.0 | 0.0 | 1.0 | 0.0 | 1.0 | 0.0 |
| kat | kat | ka | **1.0** | 0.0 | **1.0** | 0.0 | 1.0 | 0.0 | 1.0 | 0.0 | 1.0 | 0.0 | 1.0 | 0.0 |
| kaz | kaz | kk | 0.96721 | 0.00037 | 0.39604 | 0.00835 | **0.97521** | 0.00024 | 0.96667 | 0.00024 | 0.41522 | 0.0077 | 0.46154 | 0.00634 |
| kbd | kbd | - | **0.87692** | 0.00199 | | | 0.87692 | 0.00195 | 0.87692 | 0.00189 | | | | |
| kbp | kbp | - | 0.85714 | 0.00249 | | | 0.90909 | 0.00146 | **0.9375** | 0.00094 | | | | |
| kbr | kbr | - | 0.99174 | 0.00012 | | | **1.0** | 0.0 | 1.0 | 0.0 | | | | |
| kde | kde | - | **0.58491** | 0.0 | | | 0.58491 | 0.0 | 0.54369 | 0.0 | | | | |
| kdh | kdh | - | **0.77551** | 0.0 | | | 0.76289 | 0.0 | 0.73684 | 0.0 | | | | |
| kea | kea | - | **0.72727** | 0.00125 | | | 0.72727 | 0.00122 | 0.69811 | 0.00106 | | | | |
| kek | kek | - | 0.97521 | 0.00037 | | | **0.9916** | 0.00012 | 0.9916 | 0.00012 | | | | |
| kha | kha | - | 0.97521 | 0.00025 | | | **0.98333** | 0.00012 | 0.97479 | 0.00012 | | | | |
| khm | khm | km | **1.0** | 0.0 | **1.0** | 0.0 | 1.0 | 0.0 | 1.0 | 0.0 | 1.0 | 0.0 | 1.0 | 0.0 |
| kin | kin | - | 0.76336 | 0.00262 | | | 0.81967 | 0.00146 | **0.8547** | 0.00083 | | | | |
| kir | kir | ky | 0.95238 | 0.00037 | 0.22599 | 0.01875 | 0.95238 | 0.00073 | **0.96774** | 0.00047 | 0.23211 | 0.01809 | 0.26374 | 0.01516 |
| kjh | kjh | - | 0.83099 | 0.00287 | | | 0.83099 | 0.0028 | **0.84892** | 0.00236 | | | | |
| kkh | kkh | - | | | | | | | | | | | | |
| kmb | kmb | - | **0.99194** | 0.00012 | | | 0.99194 | 0.00012 | 0.99194 | 0.00012 | | | | |
| kmr | kmr | ku (Latn) | **0.66667** | 0.00735 | 0.15506 | 0.02934 | 0.66667 | 0.00719 | 0.66667 | 0.00696 | 0.19125 | 0.02274 | 0.25764 | 0.01539 |
| knc | knc | - | 0.97059 | 0.00037 | | | **0.97778** | 0.00024 | 0.97778 | 0.00024 | | | | |
| kng | kng | - | **0.0** | 0.00037 | | | 0.0 | 0.00012 | 0.0 | 0.00012 | | | | |
| koi | koi | - | **0.96667** | 0.00037 | | | 0.95652 | 0.00012 | 0.94737 | 0.00012 | | | | |
| koo | koo | - | **0.79389** | 0.00012 | | | 0.79389 | 0.0 | 0.79389 | 0.0 | | | | |
| kor | kor | ko | 0.95238 | 0.00075 | **1.0** | 0.0 | 0.96774 | 0.00049 | **0.99174** | 0.00012 | 1.0 | 0.0 | 1.0 | 0.0 |
| kqn | kqn | - | **1.0** | 0.0 | | | 1.0 | 0.0 | 1.0 | 0.0 | | | | |
| kqs | kqs | - | **0.20896** | 0.0 | | | 0.09524 | 0.0 | 0.03279 | 0.0 | | | | |
| kri | kri | - | **0.96875** | 0.0 | | | 0.96875 | 0.0 | 0.96063 | 0.0 | | | | |
| krl | krl | - | **0.02581** | 0.0 | | | 0.02581 | 0.0 | | | | | | |
| ktu | ktu | - | **0.78912** | 0.00723 | | | 0.57944 | 0.0039 | 0.54369 | 0.00354 | | | | |
| kwi | kwi | - | **0.837** | 0.0 | | | 0.78809 | 0.0 | 0.70588 | 0.0 | | | | |
| lad | lad | - | 0.92174 | 0.00025 | | | 0.92174 | 0.00024 | **0.92982** | 0.00012 | | | | |
| lao | lao | lo | **1.0** | 0.0 | **1.0** | 0.0 | 1.0 | 0.0 | 1.0 | 0.0 | 1.0 | 0.0 | 1.0 | 0.0 |

Table 39: Comparison of GlotLID vs CLD3 on UDHR benchmark (part 2)

| | | | GlotLID-M | | CLD3 | | GlotLID-M θ=.3 | | GlotLID-M θ=.5 | | CLD3 θ=.5 | | CLD3 θ=.7 | |
|---|---|---|---|---|---|---|---|---|---|---|---|---|---|---|
| iso639-3 | UDHR Code(s) | CLD3 Code(s) | F1↑ | FPR↓ | F1↑ | FPR↓ | F1↑ | FPR↓ | F1↑ | FPR↓ | F1↑ | FPR↓ | F1↑ | FPR↓ |
| lat | lat | la | **0.975** | 0.00012 | 0.68222 | 0.00474 | 0.975 | 0.00012 | 0.97071 | 0.00012 | 0.70732 | 0.0041 | 0.76159 | 0.00294 |
| lav | lvs | lv | 0.93385 | 0.00212 | 0.84507 | 0.00201 | 0.95618 | 0.00134 | **0.96386** | 0.00106 | 0.89888 | 0.00123 | 0.94118 | 0.00068 |
| lia | lia | - | **0.94643** | 0.0 | | | 0.94643 | 0.0 | 0.92727 | 0.0 | | | | |
| lij | lij | - | 0.49785 | 0.01445 | | | 0.53953 | 0.01195 | **0.64804** | 0.00731 | | | | |
| lin | lin | - | 0.99145 | 0.00025 | | | **1.0** | 0.0 | 1.0 | 0.0 | | | | |
| lit | lit | lt | 0.9375 | 0.001 | 0.43382 | 0.00698 | 0.96774 | 0.00049 | **0.98361** | 0.00024 | 0.53153 | 0.00469 | 0.63784 | 0.00299 |
| lld | lld | - | 0.53465 | 0.00174 | | | **0.5567** | 0.00122 | 0.51685 | 0.00071 | | | | |
| lns | lns | - | | | | | | | | | | | | |
| lob | lob | - | 0.92857 | 0.00025 | | | **0.93694** | 0.00012 | 0.93578 | 0.0 | | | | |
| lot | lot | - | | | | | | | | | | | | |
| loz | loz | - | **0.84892** | 0.0 | | | 0.84892 | 0.0 | 0.84892 | 0.0 | | | | |
| ltz | ltz | lb | **0.98305** | 0.0 | 0.54054 | 0.00465 | 0.98305 | 0.0 | 0.98305 | 0.0 | 0.57282 | 0.00396 | 0.63441 | 0.00303 |
| lua | lua | - | 0.72414 | 0.00174 | | | 0.75 | 0.00122 | **0.78846** | 0.00035 | | | | |
| lue | lue | - | **0.99145** | 0.0 | | | 0.99145 | 0.0 | 0.99145 | 0.0 | | | | |
| lug | lug | - | 0.9589 | 0.00075 | | | 0.97222 | 0.00049 | **0.98592** | 0.00024 | | | | |
| lun | lun | - | 0.81752 | 0.00037 | | | 0.8209 | 0.00012 | **0.82707** | 0.0 | | | | |
| lus | lus | - | 0.94309 | 0.00087 | | | 0.95868 | 0.00061 | **0.97479** | 0.00035 | | | | |
| mad | mad | - | **0.92174** | 0.0 | | | 0.92174 | 0.0 | 0.89286 | 0.0 | | | | |
| mag | mag | - | 0.75385 | 0.0005 | | | **0.76562** | 0.00024 | 0.76562 | 0.00024 | | | | |
| mah | mah | - | 0.96063 | 0.00062 | | | **1.0** | 0.0 | 1.0 | 0.0 | | | | |
| mai | mai | - | **0.83099** | 0.0 | | | 0.83099 | 0.0 | 0.83099 | 0.0 | | | | |
| mal | mal | ml | **1.0** | 0.0 | **1.0** | 0.0 | 1.0 | 0.0 | 1.0 | 0.0 | 1.0 | 0.0 | 1.0 | 0.0 |
| mam | mam | - | 0.93913 | 0.00025 | | | **0.95575** | 0.0 | 0.94643 | 0.0 | | | | |
| mar | mar | mr | **0.99174** | 0.00012 | 0.28362 | 0.01328 | 0.99174 | 0.00012 | 0.99174 | 0.00012 | 0.30287 | 0.01208 | 0.35258 | 0.00955 |
| maz | maz | - | 0.83582 | 0.00212 | | | 0.85496 | 0.00171 | **0.896** | 0.00094 | | | | |
| mcd | mcd | - | 0.97521 | 0.0 | | | **0.9916** | 0.0 | 0.98305 | 0.0 | | | | |
| mcf | mcf | - | **1.0** | 0.0 | | | 1.0 | 0.0 | 1.0 | 0.0 | | | | |
| men | men | - | 0.9771 | 0.00037 | | | 0.9771 | 0.00037 | **0.99225** | 0.00012 | | | | |
| mfq | mfq | - | 0.88722 | 0.00187 | | | 0.95935 | 0.00061 | **0.9916** | 0.00012 | | | | |
| mic | mic | - | **0.15625** | 0.0 | | | 0.03333 | 0.0 | 0.0 | 0.0 | | | | |
| miq | miq | - | **0.832** | 0.0 | | | 0.832 | 0.0 | 0.832 | 0.0 | | | | |
| mkd | mkd | mk | **0.99174** | 0.0 | 0.95238 | 0.00023 | 0.99174 | 0.0 | 0.99174 | 0.0 | 0.96 | 0.00018 | 0.98361 | 5e-05 |
| mlg | plt | mg | 0.93913 | 0.00062 | 0.70886 | 0.0021 | 0.96429 | 0.00024 | **0.97297** | 0.00012 | 0.77778 | 0.00146 | 0.86822 | 0.00077 |
| mlt | mlt | mt | 0.77419 | 0.00436 | 0.15267 | 0.03038 | 0.78947 | 0.0039 | **0.82759** | 0.00295 | 0.17857 | 0.02516 | 0.22599 | 0.0186 |
| mnw | mnw | - | | | | | | | | | | | | |
| mon | khk | mn | 0.88276 | 0.00199 | 0.27928 | 0.01446 | 0.9771 | 0.00024 | **0.98462** | 0.00012 | 0.29314 | 0.01349 | 0.33983 | 0.01054 |
| mor | mor | - | **0.9916** | 0.0 | | | 0.9916 | 0.0 | 0.9916 | 0.0 | | | | |
| mos | mos | - | 0.97015 | 0.0005 | | | 0.98485 | 0.00024 | **0.99237** | 0.00012 | | | | |
| mri | mri | mi | 0.8227 | 0.00025 | 0.2 | 0.02053 | **0.82857** | 0.00012 | 0.82857 | 0.00012 | 0.23695 | 0.01632 | 0.28502 | 0.0124 |
| msa | zlm/min/ind | id/ms | 0.86842 | 0.0076 | 0.44054 | 0.01537 | 0.89535 | 0.00549 | **0.91124** | 0.00424 | 0.5088 | 0.0103 | 0.45511 | 0.00588 |
| mto | mto | - | **0.0** | 0.001 | | | 0.0 | 0.00037 | 0.0 | 0.00012 | | | | |
| mxi | mxi | - | | | | | | | | | | | | |
| mxv | mxv | - | **0.14925** | 0.0 | | | 0.12121 | 0.0 | 0.0625 | 0.0 | | | | |
| mya | mya | my | 0.66667 | 0.00723 | 0.5042 | 0.00538 | 0.66667 | 0.00707 | **0.67045** | 0.00672 | 0.5042 | 0.00538 | 0.5042 | 0.00534 |
| mzi | mzi | - | | | | | | | | | | | | |
| nav | nav | - | 0.9916 | 0.00012 | | | **1.0** | 0.0 | 1.0 | 0.0 | | | | |
| nba | nba | - | | | | | | | | | | | | |
| ndo | ndo | - | 0.8806 | 0.00199 | | | 0.90076 | 0.00158 | **0.95161** | 0.00071 | | | | |
| nds | nds | - | **1.0** | 0.0 | | | 1.0 | 0.0 | 1.0 | 0.0 | | | | |
| nep | npi | ne | **0.99115** | 0.0 | 0.24675 | 0.01588 | 0.99115 | 0.0 | 0.99115 | 0.0 | 0.26207 | 0.01463 | 0.29867 | 0.01186 |
| nhn | nhn | - | | | | | | | | | | | | |
| nio | nio | - | | | | | | | | | | | | |
| niu | niu | - | **1.0** | 0.0 | | | 1.0 | 0.0 | 1.0 | 0.0 | | | | |
| niv | niv | - | | | | | | | | | | | | |
| njo | njo | - | **0.95312** | 0.0 | | | 0.95312 | 0.0 | 0.95312 | 0.0 | | | | |
| nku | nku | - | | | | | | | | | | | | |
| nld | nld | nl | 0.70659 | 0.00611 | 0.80272 | 0.00132 | 0.71084 | 0.00585 | 0.71515 | 0.00554 | 0.80822 | 0.00128 | **0.84286** | 0.001 |
| nor | nob/nno | no | 0.97619 | 0.00062 | 0.87455 | 0.00151 | 0.984 | 0.00037 | **0.99194** | 0.00012 | 0.88727 | 0.00132 | 0.88971 | 0.00122 |
| not | not | - | **0.97391** | 0.0 | | | 0.97391 | 0.0 | 0.97391 | 0.0 | | | | |
| nso | nso | - | 0.86957 | 0.00224 | | | 0.87591 | 0.00207 | **0.88235** | 0.00189 | | | | |
| nya | nya | ny | 0.96414 | 0.00112 | 0.21838 | 0.03914 | 0.97581 | 0.00073 | **0.99588** | 0.00012 | 0.24948 | 0.03286 | 0.31957 | 0.02308 |
| nym | nym | - | | | | | | | | | | | | |
| nyn | nyn | - | **0.85938** | 0.0 | | | 0.85938 | 0.0 | 0.85938 | 0.0 | | | | |
| nzi | nzi | - | **1.0** | 0.0 | | | 1.0 | 0.0 | 1.0 | 0.0 | | | | |
| oaa | oaa | - | | | | | | | | | | | | |
| oci | oci | - | **0.41101** | 0.0 | | | 0.40516 | 0.0 | 0.38131 | 0.0 | | | | |
| ojb | ojb | - | **0.7027** | 0.0 | | | 0.7027 | 0.0 | 0.66667 | 0.0 | | | | |
| oki | oki | - | | | | | | | | | | | | |
| orh | orh | - | | | | | | | | | | | | |
| oss | oss | - | 0.50273 | 0.0076 | | | 0.50549 | 0.00731 | **0.50829** | 0.00696 | | | | |
| ote | ote | - | **0.0** | 0.00411 | | | 0.0 | 0.00329 | 0.0 | 0.0013 | | | | |
| pam | pam | - | **1.0** | 0.0 | | | 1.0 | 0.0 | 1.0 | 0.0 | | | | |
| pan | pan | pa | **1.0** | 0.0 | **1.0** | 0.0 | 1.0 | 0.0 | 1.0 | 0.0 | 1.0 | 0.0 | 1.0 | 0.0 |
| pap | pap | - | 0.79195 | 0.00386 | | | 0.83099 | 0.00293 | **0.86765** | 0.00212 | | | | |
| pau | pau | - | **0.97436** | 0.0 | | | 0.97436 | 0.0 | 0.97436 | 0.0 | | | | |
| pbb | pbb | - | 0.71698 | 0.00536 | | | 0.76056 | 0.00354 | **0.816** | 0.00177 | | | | |
| pcd | pcd | - | | | | | | | | | | | | |
| pcm | pcm | - | **0.71739** | 0.0 | | | 0.71739 | 0.0 | 0.71739 | 0.0 | | | | |
| pis | pis | - | **0.9916** | 0.0 | | | 0.9916 | 0.0 | 0.9916 | 0.0 | | | | |
| piu | piu | - | | | | | | | | | | | | |
| pnb | pnb | - | **0.65969** | 0.0076 | | | 0.65969 | 0.00744 | 0.65969 | 0.00719 | | | | |
| pol | pol | pl | 0.74074 | 0.00523 | 0.40134 | 0.00817 | 0.76923 | 0.00439 | **0.81633** | 0.00318 | 0.47244 | 0.00611 | 0.56872 | 0.00412 |
| pon | pon | - | **1.0** | 0.0 | | | 1.0 | 0.0 | 1.0 | 0.0 | | | | |
| por | por | pt | 0.85106 | 0.00523 | 0.65217 | 0.00584 | 0.87273 | 0.00427 | **0.89219** | 0.00342 | 0.70381 | 0.0046 | 0.7619 | 0.00339 |
| pov | pov | - | **0.96552** | 0.0 | | | 0.96552 | 0.0 | 0.95652 | 0.0 | | | | |
| ppl | ppl | - | **0.42105** | 0.0 | | | 0.20896 | 0.0 | 0.125 | 0.0 | | | | |
| pus | pbu | ps | 0.75 | 0.0 | 0.90769 | 0.0005 | 0.75 | 0.0 | 0.75 | 0.0 | 0.92187 | 0.00041 | **0.95935** | 0.00018 |
| que | que/qxn/qvh/quh/qvc/qxu/quz/qvn/qug/qwh/qvm/quy/quc/qva | - | 0.919 | 0.01146 | | | 0.93737 | 0.0067 | **0.9491** | 0.00365 | | | | |
| rar | rar | - | **0.99174** | 0.00012 | | | 0.99174 | 0.00012 | 0.99174 | 0.00012 | | | | |
| rgn | rgn | - | | | | | | | | | | | | |
| rmn | rmn | - | **0.86636** | 0.00012 | | | 0.86111 | 0.00012 | 0.85047 | 0.00012 | | | | |
| roh | roh | - | **0.99268** | 0.0 | | | 0.99145 | 0.0 | 0.98775 | 0.0 | | | | |
| ron | ron | ro | 0.81328 | 0.00548 | 0.66443 | 0.00456 | 0.80833 | 0.00536 | **0.82906** | 0.00448 | 0.73606 | 0.00324 | 0.79839 | 0.00226 |
| run | run | - | 0.87591 | 0.00212 | | | 0.88889 | 0.00183 | **0.90909** | 0.00141 | | | | |
| rup | rup | - | **0.125** | 0.0 | | | 0.125 | 0.0 | 0.09524 | 0.0 | | | | |
| rus | rus | ru/ru-Latn | 0.43478 | 0.01944 | 0.32345 | 0.01145 | 0.47431 | 0.01621 | **0.51064** | 0.01356 | 0.35191 | 0.01007 | 0.4 | 0.00815 |
| sag | sag | - | **0.81553** | 0.0 | | | 0.80392 | 0.0 | 0.79208 | 0.0 | | | | |
| sah | sah | - | 0.54128 | 0.01246 | | | 0.56459 | 0.01109 | **0.60513** | 0.00908 | | | | |

Table 40: Comparison of GlotLID vs CLD3 on UDHR benchmark (part 3)

|  |  |  | GlotLID-M | | CLD3 | | GlotLID-M θ=.3 | | GlotLID-M θ=.5 | | CLD3 θ=.5 | | CLD3 θ=.7 | |
|---|---|---|---|---|---|---|---|---|---|---|---|---|---|---|
| iso639-3 | UDHR Code(s) | CLD3 Code(s) | F1↑ | FPR↓ | F1↑ | FPR↓ | F1↑ | FPR↓ | F1↑ | FPR↓ | F1↑ | FPR↓ | F1↑ | FPR↓ |
| san | san | - | **0.66667** | 0.0 | | | 0.66667 | 0.0 | 0.66667 | 0.0 | | | | |
| sco | sco | - | 0.92683 | 0.0005 | | | **0.95** | 0.00012 | 0.94118 | 0.00012 | | | | |
| sey | sey | - | **0.02985** | 0.0005 | | | 0.0 | 0.00037 | | | | | | |
| shk | shk | - | **0.88889** | 0.0 | | | 0.88889 | 0.0 | 0.86792 | 0.0 | | | | |
| shn | shn | - | **0.99145** | 0.0 | | | 0.99145 | 0.0 | 0.99145 | 0.0 | | | | |
| shp | shp | - | **0.27397** | 0.0 | | | 0.25 | 0.0 | 0.14706 | 0.0 | | | | |
| sid | sid | - | 0.90625 | 0.0015 | | | 0.90625 | 0.00146 | **0.94309** | 0.00083 | | | | |
| sin | sin | si | **1.0** | 0.0 | **1.0** | 0.0 | 1.0 | 0.0 | 1.0 | 0.0 | 1.0 | 0.0 | 1.0 | 0.0 |
| skr | skr | - | | | | | | | | | | | | |
| slk | slk | sk | 0.88889 | 0.00187 | 0.6 | 0.00365 | 0.9375 | 0.00098 | **0.94488** | 0.00083 | 0.67416 | 0.00264 | 0.78146 | 0.00145 |
| slr | slr | - | | | | | | | | | | | | |
| slv | slv | sl | 0.90909 | 0.0015 | 0.67045 | 0.0026 | 0.95238 | 0.00073 | **0.97561** | 0.00035 | 0.71084 | 0.00214 | 0.78667 | 0.0014 |
| sme | sme | - | **0.96667** | 0.00025 | | | 0.96667 | 0.00024 | 0.96667 | 0.00024 | | | | |
| smo | smo | sm | **1.0** | 0.0 | 0.6087 | 0.0037 | 1.0 | 0.0 | 1.0 | 0.0 | 0.65969 | 0.00296 | 0.7875 | 0.00154 |
| sna | sna | sn | 0.93846 | 0.001 | 0.23282 | 0.01834 | 0.96063 | 0.00061 | **0.98387** | 0.00024 | 0.30576 | 0.01262 | 0.40132 | 0.00824 |
| snd | - | sd | | | | 0.00233 | | | | | | 0.00223 | | 0.00158 |
| snk | snk | - | | | | | | | | | | | | |
| snn | snn | - | **0.60773** | 0.0 | | | 0.57627 | 0.0 | 0.425 | 0.0 | | | | |
| som | som | so | 0.75817 | 0.00461 | 0.12141 | 0.03759 | 0.80556 | 0.00341 | **0.89231** | 0.00165 | 0.13919 | 0.03208 | 0.1699 | 0.02516 |
| sot | sot | st | 0.98333 | 0.00012 | 0.35398 | 0.00999 | **0.9916** | 0.0 | 0.9916 | 0.0 | 0.4 | 0.0082 | 0.4461 | 0.00674 |
| spa | spa | es | 0.72321 | 0.00748 | 0.38835 | 0.01136 | 0.75701 | 0.0061 | **0.78** | 0.0046 | 0.40506 | 0.01057 | 0.43011 | 0.00946 |
| sqi | als | sq | 0.85714 | 0.00249 | 0.29703 | 0.01296 | 0.86131 | 0.00219 | **0.89394** | 0.00153 | 0.36036 | 0.00971 | 0.41667 | 0.0076 |
| src | src | - | | | | | | | | | | | | |
| srp | srp | sr | **0.5124** | 0.00748 | 0.45455 | 0.00383 | 0.4958 | 0.00719 | 0.48945 | 0.00696 | 0.45455 | 0.00383 | 0.45802 | 0.00371 |
| srr | srr | - | 0.89231 | 0.0015 | | | 0.93443 | 0.00061 | **0.9661** | 0.00012 | | | | |
| ssw | ssw | - | 0.94891 | 0.00075 | | | 0.97015 | 0.00037 | **0.99237** | 0.0 | | | | |
| suk | suk | - | 0.68712 | 0.00586 | | | 0.69136 | 0.00561 | **0.71895** | 0.00448 | | | | |
| sun | sun | su | 0.9697 | 0.00037 | 0.3416 | 0.01077 | **0.9771** | 0.00024 | 0.9771 | 0.00024 | 0.3949 | 0.00852 | 0.48062 | 0.00593 |
| sus | sus | - | 0.92683 | 0.001 | | | 0.94215 | 0.00073 | **0.95798** | 0.00047 | | | | |
| swa | swh | sw | 0.85315 | 0.00262 | 0.09772 | 0.0505 | 0.86525 | 0.00232 | **0.89051** | 0.00177 | 0.11101 | 0.04375 | 0.14118 | 0.03299 |
| swb | swb | - | 0.80292 | 0.00287 | | | 0.84615 | 0.00195 | **0.86885** | 0.00118 | | | | |
| swe | swe | sv | 0.86897 | 0.00237 | 0.89362 | 0.00068 | 0.93333 | 0.0011 | **1.0** | 0.0 | 0.92647 | 0.00046 | 0.95455 | 0.00027 |
| tah | tah | - | 0.91892 | 0.00012 | | | 0.91892 | 0.00012 | **0.92727** | 0.0 | | | | |
| taj | taj | - | **0.66667** | 0.0 | | | 0.66667 | 0.0 | | | | | | |
| tam | tam | ta | **1.0** | 0.0 | **1.0** | 0.0 | 1.0 | 0.0 | 1.0 | 0.0 | 1.0 | 0.0 | 1.0 | 0.0 |
| tat | tat | - | 0.68208 | 0.00673 | | | 0.68182 | 0.00658 | **0.69822** | 0.00589 | | | | |
| tbz | tbz | - | **0.20896** | 0.0 | | | 0.18182 | 0.09524 | | | | | | |
| tca | tca | - | **0.9916** | 0.0 | | | 0.9916 | 0.0 | 0.9916 | 0.0 | | | | |
| tdt | tdt | - | **0.66292** | 0.00735 | | | 0.66292 | 0.00719 | 0.65537 | 0.00696 | | | | |
| tel | tel | te | **1.0** | 0.0 | **1.0** | 0.0 | 1.0 | 0.0 | 1.0 | 0.0 | 1.0 | 0.0 | 1.0 | 0.0 |
| tem | tem | - | **0.97345** | 0.0 | | | 0.97345 | 0.0 | 0.94545 | 0.0 | | | | |
| tet | tet | - | | | | | | | | | | | | |
| tgk | tgk | tg | 0.85507 | 0.00249 | 0.464 | 0.00607 | 0.92913 | 0.0011 | **0.95082** | 0.00059 | 0.48739 | 0.00551 | 0.52252 | 0.00475 |
| tgl | tgl | - | 0.9403 | 0.00087 | | | 0.95455 | 0.00061 | **0.97674** | 0.00024 | | | | |
| tha | tha | th | **1.0** | 0.0 | **1.0** | 0.0 | 1.0 | 0.0 | 1.0 | 0.0 | 1.0 | 0.0 | 1.0 | 0.0 |
| tir | tir | - | **1.0** | 0.0 | | | 1.0 | 0.0 | 1.0 | 0.0 | | | | |
| tiv | tiv | - | 0.98551 | 0.00025 | | | **1.0** | 0.0 | 1.0 | 0.0 | | | | |
| tly | tly | - | | | | | | | | | | | | |
| tob | tob | - | 0.97561 | 0.00037 | | | 0.98361 | 0.00024 | **0.99174** | 0.00012 | | | | |
| toi | toi | - | **1.0** | 0.0 | | | 1.0 | 0.0 | 1.0 | 0.0 | | | | |
| toj | toj | - | 0.83688 | 0.00025 | | | 0.83688 | 0.00024 | **0.84892** | 0.0 | | | | |
| ton | ton | - | **1.0** | 0.0 | | | 1.0 | 0.0 | 1.0 | 0.0 | | | | |
| top | top | - | **0.99174** | 0.00012 | | | 0.99174 | 0.00012 | 0.99174 | 0.00012 | | | | |
| tpi | tpi | - | **0.98361** | 0.00012 | | | 0.98361 | 0.00012 | 0.98361 | 0.00012 | | | | |
| tsn | tsn | - | 0.98361 | 0.00025 | | | **0.99174** | 0.00012 | 0.99174 | 0.00012 | | | | |
| tso | tso | - | 0.94158 | 0.00174 | | | 0.94158 | 0.00171 | **0.95139** | 0.0013 | | | | |
| tsz | tsz | - | 0.82517 | 0.00299 | | | 0.86765 | 0.00207 | **0.90625** | 0.00118 | | | | |
| tuk | tuk | - | 0.96748 | 0.001 | | | 0.96748 | 0.00098 | **0.98347** | 0.00047 | | | | |
| tur | tur | tr | 0.49383 | 0.01533 | 0.25641 | 0.01588 | 0.50209 | 0.01451 | **0.51502** | 0.01332 | 0.27907 | 0.01413 | 0.31088 | 0.01204 |
| twi | twi | - | 0.95349 | 0.0015 | | | 0.96094 | 0.00122 | **0.9685** | 0.00094 | | | | |
| tyv | tyv | - | 0.98361 | 0.00025 | | | 0.99174 | 0.00012 | **1.0** | 0.0 | | | | |
| tzh | tzh | - | 0.98333 | 0.00025 | | | 0.98333 | 0.00024 | **0.9916** | 0.00012 | | | | |
| tzm | tzm | - | 0.01754 | 0.00648 | | | 0.01754 | 0.00634 | **0.0177** | 0.00601 | | | | |
| tzo | tzo | - | 0.97479 | 0.00012 | | | **0.98305** | 0.0 | 0.98305 | 0.0 | | | | |
| udu | udu | - | 0.98361 | 0.00025 | | | **0.99174** | 0.00012 | 0.99174 | 0.00012 | | | | |
| uig | uig | - | **0.91892** | 0.0 | | | 0.91403 | 0.0 | 0.89908 | 0.0 | | | | |
| ukr | ukr | uk | 0.98361 | 0.00025 | 0.65169 | 0.00274 | 0.98361 | 0.00024 | **0.99174** | 0.00012 | 0.68639 | 0.00232 | 0.80556 | 0.00118 |
| umb | umb | - | **0.87931** | 0.00125 | | | 0.87931 | 0.00122 | 0.87611 | 0.00083 | | | | |
| ura | ura | - | **0.82963** | 0.0 | | | 0.82963 | 0.0 | 0.82963 | 0.0 | | | | |
| urd | urd | ur | **0.96522** | 0.00087 | 0.62678 | 0.00589 | 0.96522 | 0.00085 | 0.96522 | 0.00083 | 0.63218 | 0.00574 | 0.63218 | 0.0057 |
| uzb | uzn | uz | 0.70414 | 0.01233 | 0.1326 | 0.03308 | 0.78033 | 0.00805 | **0.86545** | 0.00424 | 0.15504 | 0.02707 | 0.19017 | 0.02041 |
| vai | vai | - | | | | | | | | | | | | |
| vec | vec | - | 0.88889 | 0.00187 | | | 0.95238 | 0.00073 | **0.96774** | 0.00047 | | | | |
| ven | ven | - | **1.0** | 0.0 | | | 1.0 | 0.0 | 1.0 | 0.0 | | | | |
| vep | vep | - | | | | | | | | | | | | |
| vie | vie | vi | **0.66304** | 0.00012 | 0.07687 | 0.06405 | 0.66304 | 0.00012 | 0.66304 | 0.00012 | 0.08671 | 0.05578 | 0.10932 | 0.04222 |
| vmw | vmw | - | 0.95798 | 0.00025 | | | **0.97436** | 0.0 | 0.97436 | 0.0 | | | | |
| war | war | - | **0.9916** | 0.0 | | | 0.9916 | 0.0 | 0.9916 | 0.0 | | | | |
| wln | wln | - | 0.62105 | 0.00885 | | | 0.62434 | 0.00853 | **0.64481** | 0.00755 | | | | |
| wol | wol | - | 0.79747 | 0.00399 | | | 0.86897 | 0.00232 | **0.92537** | 0.00106 | | | | |
| wwa | wwa | - | **0.88636** | 0.0 | | | 0.88636 | 0.0 | 0.87356 | 0.0 | | | | |
| xho | xho | xh | 0.94574 | 0.00087 | 0.26339 | 0.01496 | 0.96825 | 0.00049 | **0.976** | 0.00035 | 0.29949 | 0.01249 | 0.39322 | 0.00796 |
| xsm | xsm | - | | | | | | | | | | | | |
| yad | yad | - | **0.8785** | 0.0 | | | 0.85714 | 0.0 | 0.75 | 0.0 | | | | |
| yao | yao | - | 0.97345 | 0.00025 | | | **0.98214** | 0.00012 | 0.98214 | 0.00012 | | | | |
| yap | yap | - | 0.96721 | 0.00025 | | | **0.98333** | 0.0 | 0.98333 | 0.0 | | | | |
| yid | ydd | yi | **1.0** | 0.0 | **1.0** | 0.0 | 1.0 | 0.0 | 1.0 | 0.0 | 1.0 | 0.0 | 1.0 | 0.0 |
| ykg | ykg | - | | | | | | | | | | | | |
| yor | yor | yo | 0.85106 | 0.00262 | 0.0 | 0.01323 | 0.88889 | 0.00183 | **0.93023** | 0.00106 | 0.0 | 0.00852 | 0.0 | 0.00529 |
| yrk | yrk | - | **0.8381** | 0.00012 | | | 0.83495 | 0.0 | 0.69565 | 0.0 | | | | |
| yua | yua | - | **1.0** | 0.0 | | | 1.0 | 0.0 | 1.0 | 0.0 | | | | |
| zam | zam | - | **0.0** | 0.00012 | | | 0.88496 | 0.00049 | **0.8972** | 0.0 | | | | |
| zdj | zdj | - | 0.78125 | 0.00237 | | | 0.88496 | 0.00049 | **0.8972** | 0.0 | | | | |
| zgh | zgh | - | **0.17647** | 0.0 | | | 0.17647 | 0.0 | 0.14925 | 0.0 | | | | |
| zho | cjy/hak/wuu/yue/cmn/gan/hsn/nan | zh/zh-Latn | 0.71795 | 0.07015 | 0.81359 | 0.01802 | 0.73346 | 0.0629 | 0.75177 | 0.05459 | 0.85431 | 0.0134 | **0.88293** | 0.01 |
| zro | zro | - | | | | | | | | | | | | |
| ztu | ztu | - | | | | | | | | | | | | |
| zul | zul/nbl | zu | 0.65591 | 0.00075 | 0.34138 | 0.01807 | 0.66304 | 0.00049 | **0.67033** | 0.00024 | 0.38267 | 0.01499 | 0.44492 | 0.01091 |
| zyb | zyb | - | 0.92683 | 0.0005 | | | **0.95** | 0.00012 | 0.94915 | 0.0 | | | | |

Table 41: Comparison of GlotLID vs CLD3 on UDHR benchmark (part 4)

| | | | GlotLID-M | | FT176 | | with confidence threshold $\theta$ | | | | | | | |
| | | | | | | | GlotLID-M $\theta$=.3 | | GlotLID-M $\theta$=.5 | | FT176 $\theta$=.3 | | FT176 $\theta$=.5 | |
| iso639-3 | UDHR Code(s) | FT176 Code(s) | F1↑ | FPR↓ | F1↑ | FPR↓ | F1↑ | FPR↓ | F1↑ | FPR↓ | F1↑ | FPR↓ | F1↑ | FPR↓ |
|---|---|---|---|---|---|---|---|---|---|---|---|---|---|---|
| aar | aar | - | | | | | | | | | | | | |
| abk | abk | | | | | | | | | | | | | |
| ace | ace | - | **0.91603** | 0.00129 | | | 0.91603 | 0.00126 | 0.91603 | 0.00122 | | | | |
| acu | acu | - | **0.58635** | 0.0 | | | 0.58635 | 0.0 | 0.58065 | 0.0 | | | | |
| ada | ada | - | 0.91045 | 0.00154 | | | 0.9313 | 0.00114 | **0.93846** | 0.00098 | | | | |
| ady | ady | - | **0.83495** | 0.0 | | | 0.83495 | 0.0 | 0.83495 | 0.0 | | | | |
| afr | afr | af | 0.95238 | 0.00077 | 0.84553 | 0.00052 | 0.96774 | 0.00051 | **0.97561** | 0.00037 | 0.92857 | 0.0 | 0.90909 | 0.0 |
| agr | agr | - | 0.81429 | 0.00039 | | | **0.82014** | 0.00025 | 0.82014 | 0.00024 | | | | |
| aii | aii | | | | | | | | | | | | | |
| ajg | ajg | - | 0.64516 | 0.0085 | | | 0.65217 | 0.00808 | **0.65574** | 0.0077 | | | | |
| alt | alt | - | 0.95495 | 0.00039 | | | **0.96364** | 0.00025 | 0.96364 | 0.00024 | | | | |
| amc | amc | | | | | | | | | | | | | |
| ame | ame | - | **1.0** | 0.0 | | | 1.0 | 0.0 | 1.0 | 0.0 | | | | |
| amh | amh | am | **1.0** | 0.0 | 0.35443 | 0.00283 | 1.0 | 0.0 | 1.0 | 0.0 | 0.26667 | 0.00276 | 0.25503 | 0.00268 |
| ami | ami | - | **0.20896** | 0.0 | | | 0.125 | 0.0 | 0.09524 | 0.0 | | | | |
| amr | amr | - | **1.0** | 0.0 | | | 0.99187 | 0.0 | 0.99187 | 0.0 | | | | |
| ara | arb | ar/arz | 0.89552 | 0.0018 | 0.64171 | 0.00316 | 0.93023 | 0.00114 | **0.98361** | 0.00024 | 0.66298 | 0.0028 | 0.66667 | 0.00268 |
| arg | - | an | | | | 5e-05 | | | | | | 5e-05 | | 4e-05 |
| arl | arl | - | **0.99187** | 0.0 | | | 0.99187 | 0.0 | 0.99187 | 0.0 | | | | |
| arn | arn | - | 0.93913 | 0.00013 | | | **0.94737** | 0.0 | 0.94737 | 0.0 | | | | |
| ast | ast | ast | 0.97521 | 0.00013 | 0.47059 | 0.00203 | 0.97521 | 0.00013 | **0.98333** | 0.0 | 0.62626 | 0.00032 | 0.57471 | 4e-05 |
| auc | auc | - | **0.01504** | 0.0 | | | | | | | | | | |
| ava | - | av | | | | 9e-05 | | | | | | 5e-05 | | 0 |
| ayr | ayr | - | **0.99174** | 0.00013 | | | 0.91071 | 0.00013 | 0.84615 | 0.0 | | | | |
| aze | azb/azj | az/azb | 0.6413 | 0.00837 | 0.23529 | 0.01178 | 0.6413 | 0.00821 | **0.64658** | 0.00758 | 0.26914 | 0.00864 | 0.29367 | 0.00678 |
| bak | - | ba | | 0.0009 | | 0.00113 | | 0.00088 | | 0.00086 | | 0.00096 | | 0.00076 |
| bam | bam | - | 0.49682 | 0.00747 | | | 0.5098 | 0.00682 | **0.55319** | 0.00513 | | | | |
| ban | ban | - | **0.98361** | 0.0 | | | 0.98361 | 0.0 | 0.97521 | 0.0 | | | | |
| bar | - | bar | | | | 5e-05 | | | | | | 0 | | 0 |
| bax | bax | | | | | | | | | | | | | |
| bba | bba | - | 0.92187 | 0.00116 | | | 0.92913 | 0.00101 | **0.93548** | 0.00073 | | | | |
| bci | bci | - | 0.97521 | 0.00039 | | | 0.9916 | 0.00013 | **1.0** | 0.0 | | | | |
| bcl | bcl | bcl | **1.0** | 0.0 | 0.0 | 0.0 | 1.0 | 0.0 | 0.9916 | 0.0 | 0.0 | 0.0 | 0.0 | 0.0 |
| bel | bel | be | **0.98333** | 0.0 | 0.62105 | 0.0033 | 0.98333 | 0.0 | 0.98333 | 0.0 | 0.66667 | 0.00262 | 0.83688 | 0.00094 |
| bem | bem | - | **0.98333** | 0.00026 | | | 0.98333 | 0.00025 | 0.98333 | 0.00024 | | | | |
| ben | ben | bn | **1.0** | 0.0 | 0.97674 | 0.00014 | 1.0 | 0.0 | 1.0 | 0.0 | **1.0** | 0.0 | 1.0 | 0.0 |
| bfa | bfa | - | | | | | | | | | | | | |
| bho | bho | - | **0.78519** | 0.00039 | | | 0.78519 | 0.00038 | 0.77273 | 0.00024 | | | | |
| bin | bin | - | 0.9927 | 0.00013 | | | **1.0** | 0.0 | 1.0 | 0.0 | | | | |
| bis | bis | - | **1.0** | 0.0 | | | 1.0 | 0.0 | 1.0 | 0.0 | | | | |
| blt | blt | | | | | | | | | | | | | |
| boa | boa | - | 0.99213 | 0.00013 | | | 0.99213 | 0.00013 | **1.0** | 0.0 | | | | |
| bod | bod | bo | **0.89091** | 0.00013 | 0.66667 | 0.00283 | 0.89091 | 0.00013 | 0.89091 | 0.00012 | 0.66667 | 0.00276 | 0.66667 | 0.00268 |
| bre | bre | br | **0.98361** | 0.0 | 0.60204 | 0.00354 | 0.98361 | 0.0 | 0.98361 | 0.0 | 0.86131 | 0.00074 | 0.92683 | 0.00018 |
| buc | buc | | | | | | | | | | | | | |
| bug | bug | - | 0.95312 | 0.00064 | | | 0.96063 | 0.00051 | **0.976** | 0.00024 | | | | |
| bul | bul | bg | 0.96 | 0.00064 | 0.85294 | 0.00085 | 0.96774 | 0.00063 | **0.97561** | 0.00037 | 0.89231 | 0.00055 | 0.95868 | 0.00013 |
| bum | bum | - | **0.54762** | 0.00026 | | | 0.53659 | 0.00013 | 0.46154 | 0.00012 | | | | |
| bxr | - | bxr | | | | 0.00024 | | | | | | 9e-05 | | 0 |
| cab | cab | - | **1.0** | 0.0 | | | 1.0 | 0.0 | 1.0 | 0.0 | | | | |
| cak | cak | - | **1.0** | 0.0 | | | 1.0 | 0.0 | 1.0 | 0.0 | | | | |
| cat | cat | ca | 0.9375 | 0.00103 | 0.43165 | 0.00745 | 0.95238 | 0.00076 | **0.96774** | 0.00049 | 0.64865 | 0.00299 | 0.81379 | 0.00116 |
| cbi | cbi | - | 0.9771 | 0.00026 | | | **0.99225** | 0.0 | 0.99225 | 0.0 | | | | |
| cbk | - | cbk | | 0.00039 | | 0.0008 | | 0.00038 | | 0.00024 | | 0.00032 | | 0 |
| cbr | cbr | - | **0.7033** | 0.0 | | | 0.65909 | 0.0 | 0.54321 | 0.0 | | | | |
| cbs | cbs | - | **0.67308** | 0.00039 | | | 0.66 | 0.00013 | 0.61053 | 0.0 | | | | |
| cbt | cbt | - | 0.97479 | 0.00013 | | | 0.97479 | 0.00013 | **0.98305** | 0.0 | | | | |
| cbu | cbu | - | **0.18462** | 0.0 | | | 0.09677 | 0.0 | 0.03333 | 0.0 | | | | |
| ccp | ccp | | | | | | | | | | | | | |
| ceb | ceb | ceb | 0.96721 | 0.00051 | 0.58291 | 0.00387 | 0.9916 | 0.00013 | **1.0** | 0.0 | 0.62366 | 0.00317 | 0.65909 | 0.00263 |
| ces | ces | cs | **0.98387** | 0.0 | 0.4 | 0.00834 | 0.98387 | 0.0 | 0.98387 | 0.0 | 0.60606 | 0.00345 | 0.76433 | 0.00152 |
| cfm | cfm | - | **0.85714** | 0.0 | | | 0.85714 | 0.0 | 0.84615 | 0.0 | | | | |
| cha | cha | - | **0.8381** | 0.00013 | | | 0.82353 | 0.0 | 0.8 | 0.0 | | | | |
| che | - | ce | | | | 0.00052 | | | | | | 0 | | 0 |
| chj | chj | - | 0.12727 | 0.00554 | | | 0.14286 | 0.00391 | **0.15385** | 0.00147 | | | | |
| chk | chk | - | 0.97521 | 0.00013 | | | **0.98333** | 0.0 | 0.97479 | 0.0 | | | | |
| chr | chr | - | **0.03333** | 0.0 | | | | | | | | | | |
| chv | chv | cv | **0.86154** | 0.0 | **0.86154** | 0.0 | 0.86154 | 0.0 | 0.86154 | 0.0 | 0.86154 | 0.0 | 0.84375 | 0.0 |
| cic | cic | | | | | | | | | | | | | |
| cjk | cjk | - | **0.92641** | 0.00039 | | | 0.92641 | 0.00038 | 0.92035 | 0.00012 | | | | |
| cjs | cjs | - | **0.65217** | 0.0 | | | 0.65217 | 0.0 | 0.63736 | 0.0 | | | | |
| ckb | ckb | ckb | | | **0.0** | 0.0 | | | | | 0.0 | 0.0 | 0.0 | 0.0 |
| cnh | cnh | - | 0.93023 | 0.00116 | | | 0.9375 | 0.00101 | **0.94488** | 0.00086 | | | | |
| cni | cni | - | 0.90226 | 0.00167 | | | 0.90909 | 0.00152 | **0.91603** | 0.00134 | | | | |
| cof | cof | - | **0.74747** | 0.0 | | | 0.63736 | 0.0 | 0.54118 | 0.0 | | | | |
| cor | - | kw | | | | 0.00698 | | | | | | 0.00032 | | 0 |
| cos | cos | co | **0.98305** | 0.0 | 0.03077 | 0.00019 | 0.98305 | 0.0 | 0.98305 | 0.0 | 0.0 | 0.0 | 0.0 | 0.0 |
| cot | cot | - | 0.96774 | 0.00026 | | | 0.96774 | 0.00025 | **0.97561** | 0.00012 | | | | |
| cpu | cpu | - | **0.89908** | 0.0 | | | 0.89908 | 0.0 | 0.89908 | 0.0 | | | | |
| crh | crh | - | **0.98361** | 0.00026 | | | 0.98361 | 0.00025 | 0.98361 | 0.00024 | | | | |
| cri | cri | - | **0.85185** | 0.00026 | | | 0.85185 | 0.00025 | 0.80769 | 0.00024 | | | | |
| crs | crs | - | **1.0** | 0.0 | | | 1.0 | 0.0 | 1.0 | 0.0 | | | | |
| csa | csa | | | | | | | | | | | | | |
| csw | csw | - | **0.0** | 0.00206 | | | 0.0 | 0.00051 | | | | | | |
| ctd | ctd | - | **0.78431** | 0.0 | | | 0.78431 | 0.0 | 0.74747 | 0.0 | | | | |
| cym | cym | cy | **1.0** | 0.0 | 0.22303 | 0.01999 | 1.0 | 0.0 | 1.0 | 0.0 | 0.65574 | 0.0028 | 0.85938 | 0.00049 |
| dag | dag | - | | | | | | | | | | | | |
| dan | dan | da | 0.85135 | 0.00283 | 0.65922 | 0.00269 | 0.91304 | 0.00152 | **0.98437** | 0.00024 | 0.7973 | 0.00119 | 0.86567 | 0.00058 |
| ddn | ddn | - | | | | | | | | | | | | |
| deu | deu | de | **0.98745** | 0.00013 | 0.35242 | 0.02079 | 0.98745 | 0.00013 | 0.98745 | 0.00012 | 0.73394 | 0.004 | 0.86131 | 0.00161 |
| dga | dga | - | 0.71166 | 0.00566 | | | 0.73885 | 0.0048 | **0.8** | 0.00318 | | | | |
| dip | dip | | | | | | | | | | | | | |
| diq | - | diq | | 0.00129 | | 0.00071 | | 0.00038 | | 0.00024 | | 0 | | 0 |
| div | div | dv | 0.96774 | 0.0 | **1.0** | 0.0 | 0.96774 | 0.0 | 0.93333 | 0.0 | 1.0 | 0.0 | 1.0 | 0.0 |
| duu | duu | | | | | | | | | | | | | |
| dyo | dyo | - | **0.97391** | 0.0 | | | 0.97391 | 0.0 | 0.96491 | 0.0 | | | | |
| dyu | dyu | - | **0.23188** | 0.00811 | | | 0.22059 | 0.00783 | 0.17323 | 0.00697 | | | | |
| dzo | dzo | - | **0.90769** | 0.00142 | | | 0.90769 | 0.00139 | 0.90769 | 0.00134 | | | | |
| ell | ell | el | 0.97908 | 0.0 | 0.96063 | 0.00047 | 0.97908 | 0.0 | 0.97908 | 0.0 | **0.99592** | 5e-05 | 0.98755 | 0.0 |
| emk | emk | | | | | | | | | | | | | |
| eml | - | eml | | 0.0112 | | 0.0016 | | 0.01073 | | 0.00758 | | 0.0006 | | 9e-05 |
| eng | eng | en | 0.85294 | 0.00232 | 0.0289 | 0.19006 | 0.87218 | 0.00189 | **0.8855** | 0.00159 | 0.14052 | 0.03373 | 0.31915 | 0.01141 |
| epo | epo | eo | 0.96825 | 0.00051 | 0.18155 | 0.02593 | **0.976** | 0.00038 | 0.976 | 0.00037 | 0.55455 | 0.0045 | 0.75949 | 0.00165 |
| ese | ese | - | 0.75 | 0.00013 | | | **0.75789** | 0.0 | 0.73118 | 0.0 | | | | |

Table 42: Comparison of GlotLID vs FT176 on UDHR benchmark (part 1)

with confidence threshold θ

| | | | GlotLID-M | | FT176 | | GlotLID-M θ=.3 | | GlotLID-M θ=.5 | | FT176 θ=.3 | | FT176 θ=.5 | |
|---|---|---|---|---|---|---|---|---|---|---|---|---|---|---|
| iso639-3 | UDHR Code(s) | FT176 Code(s) | F1↑ | FPR↓ | F1↑ | FPR↓ | F1↑ | FPR↓ | F1↑ | FPR↓ | F1↑ | FPR↓ | F1↑ | FPR↓ |
| est | ekk | et | 0.60302 | 0.01017 | 0.28986 | 0.01386 | 0.63492 | 0.00871 | **0.7362** | 0.00526 | 0.553 | 0.00446 | **0.84507** | 0.00098 |
| eus | eus | eu | 0.9313 | 0.00116 | 0.58586 | 0.00372 | 0.96825 | 0.0051 | **0.98387** | 0.00024 | 0.95082 | 0.00014 | **0.97479** | 0.0 |
| eve | eve | - | **0.27586** | 0.00373 | | | 0.27451 | 0.00215 | 0.22222 | 0.00012 | | | | |
| evn | evn | - | **0.21154** | 0.00438 | | | 0.2029 | 0.00038 | 0.03333 | 0.0 | | | | |
| ewe | ewe | - | **0.98361** | 0.00026 | | | 0.98361 | 0.00025 | 0.98361 | 0.00024 | | | | |
| fao | fao | - | **0.98305** | 0.0 | | | 0.98305 | 0.0 | 0.98305 | 0.0 | | | | |
| fas | prs/pes | fa | 0.90152 | 0.00322 | 0.97959 | 0.00024 | **0.90494** | 0.00303 | 0.90494 | 0.00293 | **0.99174** | 9e-05 | 0.99174 | 9e-05 |
| fat | fat | - | **0.97521** | 0.00026 | | | 0.97521 | 0.00025 | 0.97521 | 0.00024 | | | | |
| fij | fij | - | **1.0** | 0.0 | | | 1.0 | 0.0 | 1.0 | 0.0 | | | | |
| fin | fin | fi | 0.38415 | 0.026 | 0.11397 | 0.04539 | 0.39252 | 0.02462 | **0.41311** | 0.02188 | 0.20228 | 0.02242 | **0.28704** | 0.01369 |
| fkv | fkv | - | **0.28571** | 0.0 | | | 0.28571 | 0.0 | 0.23529 | 0.0 | | | | |
| fon | fon | - | 0.94118 | 0.00051 | | | 0.94915 | 0.00038 | **0.95726** | 0.00024 | | | | |
| fra | fra | fr | 0.95238 | 0.00077 | 0.17831 | 0.02607 | 0.95935 | 0.00051 | **0.9661** | 0.00012 | 0.29412 | 0.01323 | **0.40404** | 0.00789 |
| frr | - | frr | | 0.00026 | | 5e-05 | | 0.00025 | | 0.00012 | | 0 | | 0 |
| fry | fry | fy | 0.99174 | 0.00013 | 0.944 | 0.00028 | **1.0** | 0.0 | 1.0 | 0.0 | **0.9916** | 0.0 | 0.98305 | 0.0 |
| fuf | fuf | - | **0.04762** | 0.00051 | | | 0.01613 | 0.00051 | 0.0 | 0.00037 | | | | |
| fur | fur | - | 0.944 | 0.00077 | | | 0.95935 | 0.00051 | **0.96721** | 0.00037 | | | | |
| fuv | fuv | - | 0.77912 | 0.00245 | | | 0.80833 | 0.00126 | **0.81197** | 0.00073 | | | | |
| fvr | fvr | - | | | | | | | | | | | | |
| gaa | gaa | - | 0.93846 | 0.00103 | | | 0.96825 | 0.00051 | **0.98387** | 0.00024 | | | | |
| gag | gag | - | **0.93103** | 0.0 | | | 0.93103 | 0.0 | 0.93103 | 0.0 | | | | |
| gaz | gaz | - | 0.83221 | 0.00322 | | | **0.88231** | 0.00073 | 0.83761 | 0.00073 | | | | |
| gjn | gjn | - | 0.90476 | 0.00103 | | | 0.93443 | 0.00051 | **0.95798** | 0.00012 | | | | |
| gkp | gkp | - | 0.92063 | 0.00129 | | | 0.93548 | 0.00101 | **0.96667** | 0.00049 | | | | |
| gla | gla | gd | 0.94574 | 0.00026 | 0.57843 | 0.00372 | 0.95312 | 0.00013 | **0.96063** | 0.0 | 0.7451 | 0.00138 | **0.80702** | 9e-05 |
| gld | gld | - | | | | | | | | | | | | |
| gle | gle | ga | 0.95 | 0.00103 | 0.72258 | 0.00108 | **0.96815** | 0.00063 | 0.96815 | 0.00061 | 0.79433 | 0.00041 | **0.80597** | 0.00018 |
| glg | glg | gl | **0.98305** | 0.00013 | 0.78832 | 0.00113 | 0.98305 | 0.00013 | 0.98305 | 0.00012 | 0.90756 | 0.00028 | **0.94643** | 0.0 |
| glv | glv | gv | **1.0** | 0.0 | **0.08571** | 0.00038 | 1.0 | 0.0 | 1.0 | 0.0 | 0.0 | 5e-05 | 0.0 | 0.0 |
| gom | - | gom | | 0.00051 | | 0.00028 | | 0.00025 | | 0.00012 | | 0 | | 0 |
| grn | gug | gn | 0.59649 | 0.00618 | **0.5** | 0.00057 | 0.60714 | 0.00568 | **0.65359** | 0.00379 | 0.30233 | 5e-05 | 0.08 | 0.0 |
| gsw | gsw | als | 0.98333 | 0.00013 | 0.85217 | 0.00028 | **0.9916** | 0.0 | 0.9916 | 0.0 | **0.8785** | 0.0 | 0.55422 | 0.0 |
| guc | guc | - | 0.96063 | 0.00064 | | | 0.96825 | 0.00051 | **0.98387** | 0.00024 | | | | |
| guj | guj | gu | **1.0** | 0.0 | 0.99379 | 5e-05 | 1.0 | 0.0 | 1.0 | 0.0 | **1.0** | 0.0 | 1.0 | 0.0 |
| guk | guk | - | | | | | | | | | | | | |
| guu | guu | - | | | | | | | | | | | | |
| gyr | gyr | - | **0.52632** | 0.0 | | | 0.52632 | 0.0 | 0.44444 | 0.0 | | | | |
| hat | hat | ht | 0.90706 | 0.00322 | **0.45128** | 0.00137 | 0.9313 | 0.00227 | **0.95312** | 0.00147 | 0.1831 | 0.00032 | 0.032 | 4e-05 |
| hau | hau | - | 0.94488 | 0.0027 | | | 0.96 | 0.00189 | **0.97297** | 0.00122 | | | | |
| haw | haw | - | **1.0** | 0.0 | | | 1.0 | 0.0 | 1.0 | 0.0 | | | | |
| hbs | hrv/bos/srp/cnr | bs/hr/sh/sr | 0.95957 | 0.00335 | 0.68431 | 0.01466 | 0.97534 | 0.00177 | **0.98066** | 0.0011 | 0.93048 | 0.00184 | 0.73559 | 0.00058 |
| hea | hea | - | | | | | | | | | | | | |
| heb | heb | he | **1.0** | 0.0 | **1.0** | 0.0 | 1.0 | 0.0 | 1.0 | 0.0 | 1.0 | 0.0 | 1.0 | 0.0 |
| hif | - | hif | | 0.0009 | | 9e-05 | | 0.00076 | | 0.00073 | | 0 | | 0 |
| hil | hil | - | **1.0** | 0.0 | | | 1.0 | 0.0 | 1.0 | 0.0 | | | | |
| hin | hin | hi | 0.62 | 0.00978 | 0.33333 | 0.01169 | 0.62312 | 0.00947 | **0.62944** | 0.00892 | 0.33333 | 0.0114 | **0.34066** | 0.0107 |
| hlt | hlt | - | **0.92982** | 0.0 | | | 0.92982 | 0.0 | 0.89091 | 0.0 | | | | |
| hms | hms | - | | | | | | | | | | | | |
| hna | hna | - | | | | | | | | | | | | |
| hni | hni | - | | | | | | | | | | | | |
| hnj | hnj | - | | | | | | | | | | | | |
| hns | hns | - | 0.89655 | 0.00026 | | | **0.91228** | 0.0 | 0.91228 | 0.0 | | | | |
| hsb | hsb | hsb | **1.0** | 0.0 | 0.77064 | 0.00033 | 1.0 | 0.0 | 1.0 | 0.0 | **0.82353** | 0.0 | 0.68132 | 0.0 |
| hun | hun | hu | 0.82192 | 0.00335 | 0.30928 | 0.01263 | 0.84507 | 0.00278 | **0.89552** | 0.00171 | 0.57143 | 0.00414 | **0.69767** | 0.00232 |
| hus | hus | - | 0.98082 | 0.00039 | | | **0.98615** | 0.0 | 0.97479 | 0.0 | | | | |
| huu | huu | - | **0.98305** | 0.0 | | | 0.96552 | 0.0 | 0.96552 | 0.0 | | | | |
| hye | hye | hy | **1.0** | 0.0 | 0.99281 | 5e-05 | 1.0 | 0.0 | 1.0 | 0.0 | **1.0** | 0.0 | 1.0 | 0.0 |
| ibb | ibb | - | | | | | | | | | | | | |
| ibo | ibo | - | 0.98718 | 0.00026 | | | 0.99355 | 0.00013 | **1.0** | 0.0 | | | | |
| ido | ido | io | **0.95** | 0.00013 | 0.21839 | 0.00438 | 0.94915 | 0.0 | 0.94915 | 0.0 | **0.24324** | 0.00014 | 0.09231 | 0.0 |
| idu | idu | - | **0.0** | 0.00013 | | | 0.0 | 0.00013 | | | | | | |
| iii | iii | - | | | | | | | | | | | | |
| ijs | ijs | - | | | | | | | | | | | | |
| ike | ike | - | **0.97872** | 0.00026 | | | 0.97872 | 0.00025 | 0.97872 | 0.00024 | | | | |
| ile | - | ie | | | | 0.00033 | | | | | | 5e-05 | | 0 |
| ilo | ilo | ilo | 0.91339 | 0.00142 | 0.77465 | 0.00137 | 0.928 | 0.00114 | **0.97479** | 0.00037 | **0.97345** | 0.0 | 0.94545 | 0.0 |
| ina | ina | ia | 0.86131 | 0.00232 | **0.82569** | 0.00019 | 0.90769 | 0.00139 | **0.944** | 0.00073 | 0.77551 | 0.0 | 0.28571 | 0.0 |
| isl | isl | is | **0.9916** | 0.00013 | 0.74839 | 0.00179 | 0.9916 | 0.00013 | 0.9916 | 0.00012 | 0.77852 | 0.00147 | **0.78621** | 0.00129 |
| ita | ita | it | 0.8 | 0.00373 | 0.15038 | 0.03191 | 0.83916 | 0.00278 | **0.86957** | 0.00208 | 0.30769 | 0.01236 | **0.47967** | 0.00562 |
| jav | jav | jv | **0.97581** | 0.00051 | 0.43243 | 0.0024 | 0.97581 | 0.00051 | 0.97561 | 0.00037 | **0.48485** | 9e-05 | 0.16418 | 0.0 |
| jbo | - | jbo | | 0.00116 | | 0.0017 | | 0.00088 | | 0.00061 | | 0.00037 | | 4e-05 |
| jiv | jiv | - | 0.48583 | 0.01609 | | | 0.50847 | 0.0144 | **0.53881** | 0.01198 | | | | |
| jpn | jpn | ja | 0.7861 | 0.01017 | 0.5 | 0.01381 | **0.79245** | 0.0096 | 0.79245 | 0.00929 | 0.68852 | 0.00607 | **0.80109** | 0.00321 |
| kaa | kaa | - | 0.96667 | 0.00026 | | | 0.96667 | 0.00025 | **0.97479** | 0.00012 | | | | |
| kal | kal | - | **0.98305** | 0.0 | | | 0.98305 | 0.0 | 0.98305 | 0.0 | | | | |
| kan | kan | kn | **1.0** | 0.0 | 0.67836 | 0.00259 | 1.0 | 0.0 | 1.0 | 0.0 | 0.99145 | 5e-05 | **1.0** | 0.0 |
| kat | kat | ka | **1.0** | 0.0 | **1.0** | 0.0 | 1.0 | 0.0 | 1.0 | 0.0 | 1.0 | 0.0 | 1.0 | 0.0 |
| kaz | kaz | kk | 0.96721 | 0.00039 | 0.39057 | 0.00844 | **0.97521** | 0.00025 | 0.96667 | 0.00024 | 0.40845 | 0.00763 | **0.42491** | 0.00691 |
| kbd | kbd | - | **0.87692** | 0.00206 | | | 0.87692 | 0.00202 | 0.87692 | 0.00196 | | | | |
| kbp | kbp | - | 0.85714 | 0.00257 | | | 0.90909 | 0.00152 | **0.9375** | 0.00098 | | | | |
| kbr | kbr | - | 0.99174 | 0.00013 | | | **1.0** | 0.0 | 1.0 | 0.0 | | | | |
| kde | kde | - | **0.58491** | 0.0 | | | 0.58491 | 0.0 | 0.54369 | 0.0 | | | | |
| kdh | kdh | - | **0.77551** | 0.0 | | | 0.76289 | 0.0 | 0.73684 | 0.0 | | | | |
| kea | kea | - | **0.72727** | 0.00129 | | | 0.72727 | 0.00126 | 0.69811 | 0.0011 | | | | |
| kek | kek | - | 0.97521 | 0.00039 | | | **0.9916** | 0.00013 | 0.9916 | 0.00012 | | | | |
| kha | kha | - | 0.97521 | 0.00026 | | | **0.98333** | 0.00013 | 0.97479 | 0.00012 | | | | |
| khm | khm | km | **1.0** | 0.0 | **1.0** | 0.0 | 1.0 | 0.0 | 1.0 | 0.0 | 1.0 | 0.0 | 1.0 | 0.0 |
| kin | kin | - | 0.76336 | 0.0027 | | | 0.81967 | 0.00152 | **0.8547** | 0.00086 | | | | |
| kir | kir | ky | 0.95238 | 0.00077 | 0.67797 | 0.00269 | 0.95238 | 0.00076 | **0.96774** | 0.00049 | 0.69767 | 0.00239 | **0.81081** | 0.00125 |
| kjh | kjh | - | 0.83099 | 0.00296 | | | 0.83099 | 0.0029 | **0.84892** | 0.00244 | | | | |
| kkh | kkh | - | | | | | | | | | | | | |
| kmb | kmb | - | **0.99194** | 0.00013 | | | 0.99194 | 0.00013 | 0.99194 | 0.00012 | | | | |
| kmr | kmr | ku | **0.66667** | 0.0076 | 0.53881 | 0.00476 | 0.66667 | 0.00745 | 0.66667 | 0.00721 | 0.59296 | 0.00372 | **0.6** | 0.0033 |
| knc | knc | - | 0.97059 | 0.00039 | | | **0.97778** | 0.00025 | 0.97778 | 0.00024 | | | | |
| kng | kng | - | **0.0** | 0.00039 | | | 0.0 | 0.00013 | 0.0 | 0.00012 | | | | |
| kom | koi | kv | 0.95935 | 0.00064 | **0.65909** | 0.0 | 0.98333 | 0.00025 | **0.9916** | 0.00012 | 0.65909 | 0.0 | 0.525 | 0.0 |
| koo | koo | - | **0.79389** | 0.00013 | | | 0.79389 | 0.00013 | 0.79389 | 0.00012 | | | | |
| kor | kor | ko | 0.95238 | 0.00077 | 0.86331 | 0.0009 | 0.96774 | 0.00051 | **0.99174** | 0.00012 | **1.0** | 0.0 | 1.0 | 0.0 |
| kqn | kqn | - | **1.0** | 0.0 | | | 1.0 | 0.0 | 1.0 | 0.0 | | | | |
| kqs | kqs | - | **0.20896** | 0.0 | | | 0.09524 | 0.0 | 0.03279 | 0.0 | | | | |
| krc | - | krc | | 0.00132 | | | | | | | | 0.0011 | | 0.0004 |
| kri | kri | - | **0.96875** | 0.0 | | | 0.96875 | 0.0 | 0.96063 | 0.0 | | | | |
| krl | krl | - | **0.02581** | 0.0 | | | 0.02581 | 0.0 | 0.0 | 0.0 | | | | |
| ktu | ktu | - | **0.78912** | 0.00747 | | | 0.57944 | 0.00404 | 0.54369 | 0.00367 | | | | |

Table 43: Comparison of GlotLID vs FT176 on UDHR benchmark (part 2)

| | | | GlotLID-M | | FT176 | | with confidence threshold θ | | | | | | | |
| | | | | | | | GlotLID-M θ=.3 | | GlotLID-M θ=.5 | | FT176 θ=.3 | | FT176 θ=.5 | |
| iso639-3 | UDHR Code(s) | FT176 Code(s) | F1↑ | FPR↓ | F1↑ | FPR↓ | F1↑ | FPR↓ | F1↑ | FPR↓ | F1↑ | FPR↓ | F1↑ | FPR↓ |
|---|---|---|---|---|---|---|---|---|---|---|---|---|---|---|
| kwi | kwi | - | **0.837** | 0.0 | | | 0.78899 | 0.0 | 0.70588 | 0.0 | | | | |
| lad | lad | - | 0.92174 | 0.00026 | | | 0.92174 | 0.00025 | **0.92982** | 0.00012 | | | | |
| lao | lao | lo | **1.0** | 0.0 | 0.84932 | 0.00104 | 1.0 | 0.0 | 1.0 | 0.0 | 0.98413 | 9e-05 | **1.0** | 0.0 |
| lat | lat | la | **0.975** | 0.00013 | 0.6568 | 0.00495 | 0.975 | 0.00013 | 0.97071 | 0.00012 | 0.92174 | 9e-05 | 0.78 | 4e-05 |
| lav | lvs | lv | 0.94118 | 0.00193 | 0.88148 | 0.00146 | 0.95618 | 0.00139 | 0.96386 | 0.0011 | 0.96356 | 0.00037 | **0.99167** | 4e-05 |
| lez | | lez | | | | 0.00019 | | | | | | 0.00014 | | 4e-05 |
| lia | lia | - | **0.94643** | 0.0 | | | 0.94643 | 0.0 | 0.92727 | 0.0 | | | | |
| lij | lij | - | 0.49785 | 0.01493 | | | 0.53953 | 0.01238 | **0.64804** | 0.00758 | | | | |
| lin | lin | - | 0.99145 | 0.00026 | | | **1.0** | 0.0 | 1.0 | 0.0 | | | | |
| lit | lit | lt | 0.9375 | 0.00103 | 0.68208 | 0.00255 | 0.96774 | 0.00051 | 0.98361 | 0.00024 | 0.97521 | 9e-05 | **0.9916** | 0.0 |
| lld | lld | - | 0.53465 | 0.0018 | | | **0.5567** | 0.00126 | 0.51685 | 0.00073 | | | | |
| lmo | | lmo | | 0.00991 | | 0.00212 | | 0.00897 | | 0.00733 | | 0.0006 | | 4e-05 |
| lns | lns | - | | | | | | | | | | | | |
| lob | lob | - | 0.92857 | 0.00026 | | | **0.93694** | 0.00013 | 0.93578 | 0.0 | | | | |
| lot | lot | - | | | | | | | | | | | | |
| loz | loz | - | **0.84892** | 0.0 | | | 0.84892 | 0.0 | 0.84892 | 0.0 | | | | |
| ltz | ltz | lb | **0.98305** | 0.0 | 0.90435 | 0.00014 | 0.98305 | 0.0 | 0.98305 | 0.0 | 0.90265 | 9e-05 | 0.88889 | 0.0 |
| lua | lua | - | 0.72414 | 0.0018 | | | 0.75 | 0.00126 | **0.78846** | 0.00037 | | | | |
| lue | lue | - | **0.99145** | 0.0 | | | 0.99145 | 0.0 | 0.99145 | 0.0 | | | | |
| lug | lug | - | 0.9589 | 0.00077 | | | 0.97222 | 0.00051 | **0.98592** | 0.00024 | | | | |
| lun | lun | - | 0.81752 | 0.00039 | | | 0.8209 | 0.00013 | **0.82707** | 0.0 | | | | |
| lus | lus | - | 0.94309 | 0.0009 | | | 0.95868 | 0.00063 | **0.97479** | 0.00037 | | | | |
| mad | mad | - | **0.92174** | 0.0 | | | 0.92174 | 0.0 | 0.89286 | 0.0 | | | | |
| mag | mag | - | 0.75385 | 0.00051 | | | **0.76562** | 0.00025 | 0.76562 | 0.00024 | | | | |
| mah | mah | - | 0.96063 | 0.00064 | | | **1.0** | 0.0 | 1.0 | 0.0 | | | | |
| mai | mai | mai | **0.83099** | 0.0 | 0.06977 | 0.0 | 0.83099 | 0.0 | 0.83099 | 0.0 | 0.06977 | 0.0 | 0.02381 | 0.0 |
| mal | mal | ml | **1.0** | 0.0 | 0.9145 | 0.00108 | 1.0 | 0.0 | 1.0 | 0.0 | **1.0** | 0.0 | 1.0 | 0.0 |
| mam | mam | - | 0.93913 | 0.00026 | | | **0.95575** | 0.0 | 0.94643 | 0.0 | | | | |
| mar | mar | mr | **0.99174** | 0.00013 | 0.98361 | 9e-05 | 0.99174 | 0.00013 | 0.99174 | 0.00012 | 0.98361 | 9e-05 | 0.98361 | 9e-05 |
| maz | maz | - | 0.83582 | 0.00219 | | | 0.85496 | 0.00177 | **0.896** | 0.00098 | | | | |
| mcd | mcd | - | 0.97521 | 0.00026 | | | **0.9916** | 0.0 | 0.98305 | 0.0 | | | | |
| mcf | mcf | - | **1.0** | 0.0 | | | 1.0 | 0.0 | 1.0 | 0.0 | | | | |
| men | men | - | 0.9771 | 0.00039 | | | 0.9771 | 0.00038 | **0.99225** | 0.00012 | | | | |
| mfq | mfq | - | 0.88722 | 0.00193 | | | 0.95935 | 0.00063 | **0.9916** | 0.00012 | | | | |
| mhr | | mhr | | 0.00013 | | 5e-05 | | 0.00013 | | 0.00012 | 0 | | 0 | |
| mic | mic | - | **0.15625** | 0.0 | | | 0.03333 | 0.0 | | | | | | |
| miq | miq | - | **0.832** | 0.0 | | | 0.832 | 0.0 | 0.832 | 0.0 | | | | |
| mkd | mkd | mk | **0.99174** | 0.0 | 0.96774 | 0.00014 | 0.99174 | 0.0 | 0.99174 | 0.0 | 0.97561 | 9e-05 | **0.99174** | 0.0 |
| mlg | plt | mg | 0.94737 | 0.00051 | 0.80342 | 0.00066 | 0.96429 | 0.00025 | 0.97297 | 0.00012 | 0.83673 | 5e-05 | 0.76923 | 0.0 |
| mlt | mlt | mt | 0.77419 | 0.00451 | 0.71951 | 0.00212 | 0.78947 | 0.00404 | 0.82759 | 0.00306 | **0.9916** | 0.0 | 0.98305 | 0.0 |
| mnw | mnw | - | | | | | | | | | | | | |
| mon | khk | mn | 0.88276 | 0.00206 | 0.63317 | 0.00335 | 0.9771 | 0.00025 | **0.98462** | 0.00012 | 0.68108 | 0.00262 | 0.85714 | 0.00085 |
| mor | mor | - | **0.9916** | 0.0 | | | 0.9916 | 0.0 | 0.9916 | 0.0 | | | | |
| mos | mos | - | 0.97015 | 0.00051 | | | 0.98485 | 0.00025 | **0.99237** | 0.00012 | | | | |
| mri | mri | - | 0.8227 | 0.00026 | | | **0.82857** | 0.00013 | 0.82857 | 0.00012 | | | | |
| mrj | | mrj | | | | 9e-05 | | | | | 0 | | 0 | |
| msa | min/ind/zlm | id/min/ms | 0.86842 | 0.00785 | 0.23222 | 0.05256 | 0.89535 | 0.00568 | **0.91124** | 0.0044 | 0.63004 | 0.00616 | **0.73232** | 0.00049 |
| mto | mto | - | **0.0** | 0.00103 | | | 0.0 | 0.00038 | 0.0 | 0.00012 | | | | |
| mwl | | mwl | | 0.00039 | | 0.00019 | | 0.00013 | | 0.00012 | | 5e-05 | 0 | |
| mxi | mxi | - | | | | | | | | | | | | |
| mxv | mxv | - | **0.12121** | 0.0 | | | 0.12121 | 0.0 | 0.0625 | 0.0 | | | | |
| mya | mya | my | 0.66667 | 0.00747 | 0.5042 | 0.00556 | 0.66667 | 0.00732 | **0.67045** | 0.00697 | 0.5042 | 0.00542 | 0.5042 | 0.00526 |
| myv | | myv | | | | 9e-05 | | | | | 0 | | 0 | |
| mzi | mzi | - | | | | | | | | | | | | |
| nah | | nah | | | | 0.00042 | | | | | | 9e-05 | 0 | |
| nap | | nap | | | | 0.00061 | | | | | | 5e-05 | | 4e-05 |
| nav | nav | - | 0.9916 | 0.00013 | | | **1.0** | 0.0 | 1.0 | 0.0 | | | | |
| nba | nba | - | | | | | | | | | | | | |
| ndo | ndo | - | 0.8806 | 0.00206 | | | 0.90076 | 0.00164 | **0.95161** | 0.00073 | | | | |
| nds | nds | nds | **1.0** | 0.0 | 0.92308 | 0.00019 | 1.0 | 0.0 | 1.0 | 0.0 | 0.93913 | 9e-05 | 0.93913 | 9e-05 |
| nep | npi | ne | **0.99115** | 0.0 | 0.96552 | 0.00014 | 0.99115 | 0.0 | 0.99115 | 0.0 | 0.96552 | 0.00014 | 0.98246 | 4e-05 |
| nhn | nhn | - | | | | | | | | | | | | |
| nio | nio | - | | | | | | | | | | | | |
| niu | niu | - | **1.0** | 0.0 | | | 1.0 | 0.0 | 1.0 | 0.0 | | | | |
| niv | niv | - | | | | | | | | | | | | |
| njo | njo | - | **0.95312** | 0.0 | | | 0.95312 | 0.0 | 0.95312 | 0.0 | | | | |
| nku | nku | - | | | | | | | | | | | | |
| nld | nld | nl | 0.70659 | 0.00631 | 0.39189 | 0.00844 | 0.71084 | 0.00606 | 0.71515 | 0.00575 | 0.77333 | 0.00152 | **0.89231** | 0.00058 |
| nno | nno | nn | 0.95868 | 0.00039 | 0.4 | 0.00019 | **0.96667** | 0.00025 | 0.9661 | 0.00012 | 0.41558 | 5e-05 | 0.35616 | 0.0 |
| nob | nob | no | 0.98462 | 0.00026 | 0.53744 | 0.00481 | **0.99225** | 0.00013 | 0.98438 | 0.00012 | 0.63212 | 0.00312 | 0.66286 | 0.00236 |
| not | not | - | **0.97391** | 0.0 | | | 0.97391 | 0.0 | 0.97391 | 0.0 | | | | |
| nso | nso | - | 0.86957 | 0.00232 | | | 0.87591 | 0.00215 | **0.88235** | 0.00196 | | | | |
| nya | nya | - | 0.96414 | 0.00116 | | | 0.97581 | 0.00076 | **0.99588** | 0.00012 | | | | |
| nym | nym | - | | | | | | | | | | | | |
| nyn | nyn | - | **0.85938** | 0.0 | | | 0.85938 | 0.0 | 0.85938 | 0.0 | | | | |
| nzi | nzi | - | **1.0** | 0.0 | | | 1.0 | 0.0 | 1.0 | 0.0 | | | | |
| oaa | oaa | - | | | | | | | | | | | | |
| oci | oci | oc | **0.41101** | 0.0 | 0.22008 | 0.00132 | 0.40516 | 0.0 | 0.38131 | 0.0 | 0.17073 | 0.00078 | 0.09565 | 0.00022 |
| ojb | ojb | - | **0.7027** | 0.0 | | | 0.7027 | 0.0 | 0.66667 | 0.0 | | | | |
| oki | oki | - | | | | | | | | | | | | |
| orh | orh | - | | | | | | | | | | | | |
| oss | oss | os | 0.50273 | 0.00785 | 0.05128 | 0.0 | 0.50549 | 0.00758 | **0.50829** | 0.00721 | 0.02597 | 0.0 | 0.0 | 0.0 |
| ote | ote | - | **0.0** | 0.00425 | | | 0.0 | 0.00341 | 0.0 | 0.00134 | | | | |
| pam | pam | pam | **1.0** | 0.0 | **0.0** | 0.00052 | 1.0 | 0.0 | 1.0 | 0.0 | 0.0 | 0.0 | 0.0 | 0.0 |
| pan | pan | pa | **1.0** | 0.0 | **1.0** | 0.0 | 1.0 | 0.0 | 1.0 | 0.0 | 1.0 | 0.0 | 1.0 | 0.0 |
| pap | pap | - | 0.7973 | 0.00386 | | | 0.83099 | 0.00303 | **0.86765** | 0.0022 | | | | |
| pau | pau | - | **0.97436** | 0.0 | | | 0.97436 | 0.0 | 0.97436 | 0.0 | | | | |
| pbb | pbb | - | 0.71698 | 0.00554 | | | 0.76056 | 0.00366 | **0.816** | 0.00183 | | | | |
| pcd | pcd | - | | | | | | | | | | | | |
| pcm | pcm | - | **0.71739** | 0.0 | | | 0.71739 | 0.0 | 0.71739 | 0.0 | | | | |
| pfl | | pfl | | | | 0.00014 | | | | | 0 | | 0 | |
| pis | pis | - | **0.9916** | 0.0 | | | 0.9916 | 0.0 | 0.9916 | 0.0 | | | | |
| piu | piu | - | | | | | | | | | | | | |
| pms | | pms | | 0.00039 | | 0.00085 | | 0.00038 | | 0.00012 | | 0.00014 | 0 | |
| pnb | pnb | pnb | 0.65969 | 0.00785 | 0.65625 | 0.00292 | 0.65969 | 0.0077 | 0.65969 | 0.00746 | **0.66667** | 0.00271 | 0.65957 | 0.00263 |
| pol | pol | pl | 0.74074 | 0.00541 | 0.26966 | 0.01532 | 0.76923 | 0.00455 | **0.81633** | 0.0033 | 0.62176 | 0.00335 | 0.7284 | 0.00192 |
| pon | pon | - | **1.0** | 0.0 | | | 1.0 | 0.0 | 1.0 | 0.0 | | | | |
| por | por | pt | 0.86957 | 0.00463 | 0.4829 | 0.01211 | 0.87273 | 0.00442 | **0.89219** | 0.00355 | 0.71642 | 0.00437 | 0.83624 | 0.0021 |
| pov | pov | - | **0.96104** | 0.0 | | | 0.96104 | 0.0 | 0.95652 | 0.0 | | | | |
| ppl | ppl | - | **0.42105** | 0.0 | | | 0.20896 | 0.0 | 0.125 | 0.0 | | | | |
| pus | pbu | ps | 0.75 | 0.0 | **0.98305** | 0.0 | 0.75 | 0.0 | 0.75 | 0.0 | 0.98305 | 0.0 | 0.98305 | 0.0 |
| que | que/qxn/qvh/quh/qvc/qxu/quz/qvn/qug/qwh/qvm/quy/quc/qva | qu | 0.92242 | 0.01094 | 0.51086 | 0.00306 | 0.93787 | 0.00682 | **0.9491** | 0.00379 | 0.22307 | 0.00014 | 0.05846 | 4e-05 |
| rar | rar | - | **0.99174** | 0.00013 | | | 0.99174 | 0.00013 | 0.99174 | 0.00012 | | | | |
| rgn | rgn | - | | | | | | | | | | | | |

Table 44: Comparison of GlotLID vs FT176 on UDHR benchmark (part 3)

| iso639-3 | UDHR Code(s) | FT176 Code(s) | GlotLID-M | | FT176 | | GlotLID-M θ=.3 | | GlotLID-M θ=.5 | | FT176 θ=.3 | | FT176 θ=.5 | |
|---|---|---|---|---|---|---|---|---|---|---|---|---|---|---|
| | | | F1↑ | FPR↓ | F1↑ | FPR↓ | F1↑ | FPR↓ | F1↑ | FPR↓ | F1↑ | FPR↓ | F1↑ | FPR↓ |
| rmn | rmn | - | **0.86636** | 0.00013 | | | 0.86111 | 0.00013 | 0.85047 | 0.00012 | | | | |
| roh | roh | rm | **0.99268** | 0.0 | 0.63764 | 0.00118 | 0.99145 | 0.0 | 0.98775 | 0.0 | 0.29857 | 0.00014 | 0.09217 | 4e-05 |
| ron | ron | ro | 0.81328 | 0.00566 | 0.57647 | 0.00674 | 0.80833 | 0.00556 | 0.82906 | 0.00465 | 0.80992 | 0.00207 | **0.91943** | 0.00067 |
| run | run | - | 0.87591 | 0.00219 | | | 0.88889 | 0.00189 | **0.90909** | 0.00147 | | | | |
| rup | rup | - | **0.125** | | | | 0.125 | | 0.09524 | | | | | |
| rus | rus | ru | 0.43478 | 0.02008 | 0.21352 | 0.02084 | 0.47431 | 0.0168 | **0.51064** | 0.01406 | 0.22901 | 0.01856 | 0.27972 | 0.01378 |
| sag | sag | - | **0.81553** | 0.0 | | | 0.80392 | 0.0 | 0.79208 | 0.0 | | | | |
| sah | sah | sah | 0.54128 | 0.01287 | 0.62105 | 0.00339 | 0.56459 | 0.01149 | 0.60513 | 0.00941 | 0.63441 | 0.00312 | **0.67836** | 0.00241 |
| san | san | sa | **0.66667** | 0.0 | 0.58333 | 5e-05 | 0.66667 | 0.0 | 0.66667 | 0.0 | 0.58333 | 5e-05 | 0.51707 | 0.0 |
| scn | - | scn | | 0.00129 | | 0.00061 | | 0.00101 | | 0.00024 | | 0 | | 0 |
| sco | sco | sco | 0.92683 | 0.00051 | 0.0 | 0.00042 | **0.95** | 0.00013 | 0.94118 | 0.00012 | 0.0 | 0.0 | 0.0 | 0.0 |
| sey | sey | - | 0.02985 | 0.00051 | | | 0.0 | 0.00038 | | | | | | |
| shk | shk | - | **0.88889** | 0.0 | | | 0.88889 | 0.0 | 0.86792 | 0.0 | | | | |
| shn | shn | - | **0.99145** | 0.0 | | | 0.99145 | 0.0 | 0.99145 | 0.0 | | | | |
| shp | shp | - | **0.27397** | 0.0 | | | 0.25 | 0.0 | 0.14706 | 0.0 | | | | |
| sid | sid | - | 0.90625 | 0.00154 | | | 0.90625 | 0.00152 | **0.94309** | 0.00086 | | | | |
| sin | sin | si | **1.0** | 0.0 | 0.99174 | 5e-05 | 1.0 | 0.0 | 1.0 | 0.0 | **1.0** | 0.0 | 1.0 | 0.0 |
| skr | skr | - | | | | | | | | | | | | |
| slk | slk | sk | 0.88889 | 0.00193 | 0.88722 | 0.00066 | 0.9375 | 0.00101 | 0.94488 | 0.00086 | 0.96667 | 9e-05 | **0.97479** | 4e-05 |
| slr | slr | - | | | | | | | | | | | | |
| slv | slv | sl | 0.92308 | 0.00129 | 0.26866 | 0.01358 | 0.96 | 0.00063 | **0.97561** | 0.00037 | 0.72483 | 0.00161 | 0.8595 | 0.0004 |
| sme | sme | - | **0.96667** | 0.00026 | | | 0.96667 | 0.00025 | 0.96667 | 0.00024 | | | | |
| smo | smo | - | **1.0** | 0.0 | | | 1.0 | 0.0 | 1.0 | 0.0 | | | | |
| sna | sna | - | 0.93846 | 0.00103 | | | 0.96063 | 0.00063 | **0.98387** | 0.00024 | | | | |
| snk | snk | - | | | | | | | | | | | | |
| snn | snn | - | **0.56818** | 0.0 | | | 0.56818 | 0.0 | 0.425 | 0.0 | | | | |
| som | som | so | 0.75817 | 0.00476 | 0.71318 | 0.00118 | 0.80556 | 0.00354 | **0.89231** | 0.00171 | 0.29412 | 0.0 | 0.0339 | 0.0 |
| sot | sot | - | 0.98333 | 0.00013 | | | **0.9916** | 0.0 | 0.9916 | 0.0 | | | | |
| spa | spa | es | 0.75 | 0.00669 | 0.09529 | 0.07429 | 0.76415 | 0.00606 | **0.78** | 0.00477 | 0.24093 | 0.02403 | 0.36281 | 0.01239 |
| sqi | als | sq | 0.85714 | 0.00257 | 0.3224 | 0.01164 | 0.86131 | 0.00227 | **0.89394** | 0.00159 | 0.49789 | 0.00542 | 0.60825 | 0.00334 |
| srd | src | sc | 0.88722 | 0.0018 | 0.08955 | 0.00019 | 0.93651 | 0.00088 | **0.97521** | 0.00024 | 0.0 | 0.0 | 0.0 | 0.0 |
| srr | srr | - | 0.89231 | 0.00154 | | | 0.93443 | 0.00063 | **0.9661** | 0.00012 | | | | |
| ssw | ssw | - | 0.94891 | 0.00077 | | | 0.97015 | 0.00038 | **0.99237** | 0.0 | | | | |
| suk | suk | - | 0.68712 | 0.00605 | | | 0.69136 | 0.00581 | **0.71895** | 0.00465 | | | | |
| sun | sun | su | 0.9697 | 0.00039 | 0.26891 | 0.0115 | **0.9771** | 0.00025 | 0.9771 | 0.00024 | **0.68519** | 0.00028 | 0.35443 | 0.0 |
| sus | sus | - | 0.92683 | 0.00103 | | | 0.94215 | 0.00076 | **0.95798** | 0.00049 | | | | |
| swa | swh | sw | 0.85315 | 0.0027 | 0.10546 | 0.04455 | 0.86525 | 0.0024 | **0.89051** | 0.00183 | 0.30321 | 0.01057 | 0.49425 | 0.00312 |
| swb | swb | - | 0.80292 | 0.00296 | | | 0.84615 | 0.00202 | **0.86885** | 0.00122 | | | | |
| swe | swe | sv | 0.88112 | 0.00219 | 0.86713 | 0.00085 | 0.93333 | 0.00114 | **1.0** | 0.0 | 0.96875 | 0.00014 | 0.98413 | 4e-05 |
| tah | tah | - | 0.91892 | 0.00013 | | | 0.91892 | 0.00013 | **0.92727** | 0.0 | | | | |
| taj | taj | - | **0.66667** | 0.0 | | | 0.66667 | 0.0 | 0.66667 | 0.0 | | | | |
| tam | tam | ta | **1.0** | 0.0 | 0.95238 | 0.00057 | 1.0 | 0.0 | 1.0 | 0.0 | **1.0** | 0.0 | 1.0 | 0.0 |
| tat | tat | tt | 0.68208 | 0.00695 | 0.4055 | 0.00811 | 0.68208 | 0.00682 | **0.69822** | 0.00611 | 0.42754 | 0.00721 | 0.50862 | 0.00504 |
| tbz | tbz | - | **0.20896** | 0.0 | | | 0.18182 | 0.0 | 0.09524 | 0.0 | | | | |
| tca | tca | - | **0.9916** | 0.0 | | | 0.9916 | 0.0 | 0.9916 | 0.0 | | | | |
| tdt | tdt | - | **0.66292** | 0.0076 | | | 0.66292 | 0.00745 | 0.65537 | 0.00721 | | | | |
| tel | tel | te | **1.0** | 0.0 | 0.9916 | 5e-05 | 1.0 | 0.0 | 1.0 | 0.0 | **1.0** | 0.0 | 1.0 | 0.0 |
| tem | tem | - | **0.97345** | 0.0 | | | 0.97345 | 0.0 | 0.94545 | 0.0 | | | | |
| tet | tet | - | | | | | | | | | | | | |
| tgk | tgk | tg | 0.86131 | 0.00245 | 0.67429 | 0.00269 | 0.92913 | 0.00114 | **0.95082** | 0.00061 | 0.68605 | 0.00248 | 0.71951 | 0.00205 |
| tgl | tgl | tl | 0.9403 | 0.0009 | 0.09104 | 0.05826 | 0.95455 | 0.00063 | **0.97674** | 0.00024 | 0.49799 | 0.00565 | 0.83562 | 0.00094 |
| tha | tha | th | **1.0** | 0.0 | 0.99574 | 5e-05 | 1.0 | 0.0 | 1.0 | 0.0 | **1.0** | 0.0 | 1.0 | 0.0 |
| tir | tir | - | **1.0** | 0.0 | | | 1.0 | 0.0 | 1.0 | 0.0 | | | | |
| tiv | tiv | - | 0.98551 | 0.00026 | | | **1.0** | 0.0 | 1.0 | 0.0 | | | | |
| tly | tly | - | | | | | | | | | | | | |
| tob | tob | - | 0.98361 | 0.00026 | | | 0.98361 | 0.00025 | **0.99174** | 0.00012 | | | | |
| toi | toi | - | **1.0** | 0.0 | | | 1.0 | 0.0 | 1.0 | 0.0 | | | | |
| toj | toj | - | 0.83688 | 0.00026 | | | 0.83688 | 0.00025 | **0.84892** | 0.0 | | | | |
| ton | ton | - | **1.0** | 0.0 | | | 1.0 | 0.0 | 1.0 | 0.0 | | | | |
| top | top | - | **0.99174** | 0.00013 | | | 0.99174 | 0.00013 | 0.99174 | 0.00012 | | | | |
| tpi | tpi | - | **0.98361** | 0.00013 | | | 0.98361 | 0.00013 | 0.98361 | 0.00012 | | | | |
| tsn | tsn | - | 0.98361 | 0.00026 | | | **0.99174** | 0.00013 | 0.99174 | 0.00012 | | | | |
| tso | tso | - | 0.94158 | 0.0018 | | | 0.94158 | 0.00177 | **0.95139** | 0.00134 | | | | |
| tsz | tsz | - | 0.82517 | 0.00309 | | | 0.86765 | 0.00215 | **0.90625** | 0.00122 | | | | |
| tuk | tuk | tk | 0.96748 | 0.00103 | 0.53659 | 5e-05 | 0.96748 | 0.00101 | **0.98347** | 0.00049 | 0.53988 | 0.0 | 0.47436 | 0.0 |
| tur | tur | tr | 0.49383 | 0.01583 | 0.19704 | 0.02022 | 0.50209 | 0.01503 | **0.51502** | 0.01381 | 0.29777 | 0.013 | 0.35714 | 0.00963 |
| twi | twi | - | 0.95349 | 0.00154 | | | 0.96094 | 0.00126 | **0.9685** | 0.00098 | | | | |
| tyv | tyv | tyv | 0.98361 | 0.00026 | 0.55422 | 0.0 | 0.99174 | 0.00013 | **1.0** | 0.0 | 0.55422 | 0.0 | 0.35616 | 0.0 |
| tzh | tzh | - | 0.98333 | 0.00026 | | | 0.98333 | 0.00025 | **0.9916** | 0.00012 | | | | |
| tzm | tzm | - | 0.01754 | 0.00669 | | | 0.01754 | 0.00657 | **0.0177** | 0.00623 | | | | |
| tzo | tzo | - | 0.97479 | 0.00013 | | | **0.98305** | 0.0 | 0.98305 | 0.0 | | | | |
| udu | udu | - | 0.98361 | 0.00026 | | | **0.99174** | 0.00013 | 0.99174 | 0.00012 | | | | |
| uig | uig | ug | **0.91892** | 0.0 | 0.66667 | 0.0 | 0.91403 | 0.0 | 0.89908 | 0.0 | 0.66667 | 0.0 | 0.66667 | 0.0 |
| ukr | ukr | uk | 0.98361 | 0.00026 | 0.80537 | 0.00137 | 0.98361 | 0.00025 | **0.99174** | 0.00012 | 0.84507 | 0.00101 | 0.86957 | 0.0008 |
| umb | umb | - | **0.87931** | 0.00129 | | | 0.87931 | 0.00126 | 0.87611 | 0.00086 | | | | |
| ura | ura | - | **0.82963** | 0.0 | | | 0.82963 | 0.0 | 0.82963 | 0.0 | | | | |
| urd | urd | ur | **0.96522** | 0.0009 | 0.70701 | 0.00429 | 0.96522 | 0.00088 | 0.96522 | 0.00086 | 0.92116 | 0.00083 | 0.96104 | 0.00036 |
| uzb | uzn | uz | 0.70414 | 0.01274 | 0.40741 | 0.00448 | 0.78033 | 0.00833 | **0.86545** | 0.0044 | 0.57754 | 0.0006 | 0.56977 | 0.00013 |
| vai | vai | - | | | | | | | | | | | | |
| vec | vec | vec | 0.94488 | 0.0009 | 0.62385 | 0.00071 | 0.95238 | 0.00076 | **0.96774** | 0.00049 | 0.58427 | 0.00014 | 0.35616 | 0.0 |
| ven | ven | - | **1.0** | 0.0 | | | 1.0 | 0.0 | 1.0 | 0.0 | | | | |
| vep | vep | vep | | | **0.0625** | 9e-05 | | | | | 0.03279 | 0.0 | 0.03279 | 0.0 |
| vie | vie | vi | **0.66304** | 0.00013 | 0.58095 | 0.00127 | 0.66304 | 0.00013 | 0.66304 | 0.00012 | 0.6455 | 0.00028 | **0.66667** | 0.0 |
| vmw | vmw | - | 0.95798 | 0.00026 | | | **0.97436** | 0.0 | 0.97436 | 0.0 | | | | |
| vol | - | vo | | | | 0.00028 | | | | | | 0.00014 | | 4e-05 |
| war | war | war | **0.9916** | 0.0 | 0.25993 | 0.00853 | 0.9916 | 0.0 | 0.9916 | 0.0 | 0.44872 | 0.0028 | 0.36364 | 0.00094 |
| wln | wln | wa | 0.62105 | 0.00914 | 0.05085 | 0.00259 | 0.62434 | 0.00884 | **0.64481** | 0.00782 | 0.01887 | 0.00207 | 0.0 | 0.00152 |
| wol | wol | - | 0.79747 | 0.00412 | | | 0.86897 | 0.0024 | **0.92537** | 0.0011 | | | | |
| wwa | wwa | - | **0.88636** | 0.0 | | | 0.88636 | 0.0 | 0.87356 | 0.0 | | | | |
| xho | xho | - | 0.94574 | 0.0009 | | | 0.96825 | 0.00051 | **0.976** | 0.00037 | | | | |
| xsm | xsm | - | | | | | | | | | | | | |
| yad | yad | - | **0.8785** | 0.0 | | | 0.85714 | 0.0 | 0.75 | 0.0 | | | | |
| yao | yao | - | 0.97345 | 0.00026 | | | **0.98214** | 0.00013 | 0.98214 | 0.00012 | | | | |
| yap | yap | - | 0.96721 | 0.00026 | | | **0.98333** | 0.0 | 0.98333 | 0.0 | | | | |
| yid | ydd | yi | **1.0** | 0.0 | **1.0** | 0.0 | 1.0 | 0.0 | 1.0 | 0.0 | 1.0 | 0.0 | 1.0 | 0.0 |
| ykg | ykg | - | | | | | | | | | | | | |
| yor | yor | yo | 0.85106 | 0.0027 | 0.86441 | 0.00033 | 0.88889 | 0.00189 | **0.93023** | 0.0011 | 0.91892 | 0.0 | 0.8785 | 0.0 |
| yrk | yrk | - | **0.8381** | 0.00013 | | | 0.83495 | 0.0 | 0.69565 | 0.0 | | | | |
| yua | yua | - | **1.0** | 0.0 | | | 1.0 | 0.0 | 1.0 | 0.0 | | | | |
| zam | zam | - | **0.0** | 0.00013 | | | | | | | | | | |
| zdj | zdj | - | 0.78125 | 0.00245 | | | 0.8496 | 0.00051 | **0.8972** | 0.0 | | | | |
| zgh | zgh | - | **0.17647** | 0.0 | | | 0.17647 | 0.0 | 0.14925 | 0.0 | | | | |
| zho | wuu/gan/yue/hsn/cmn/hak/cjy/nan | wuu/yue/zh | 0.71795 | 0.07248 | 0.74988 | 0.02263 | 0.73346 | 0.06516 | 0.75177 | 0.0566 | 0.88255 | 0.00717 | **0.91715** | 0.00352 |
| zro | zro | - | | | | | | | | | | | | |
| ztu | ztu | - | | | | | | | | | | | | |
| zul | zul/nbl | - | 0.65591 | 0.00077 | | | 0.66304 | 0.00051 | **0.67033** | 0.00024 | | | | |
| zyb | zyb | - | 0.92683 | 0.00051 | | | **0.95** | 0.00013 | 0.94915 | 0.0 | | | | |

Table 45: Comparison of GlotLID vs FT176 on UDHR benchmark (part 4)

| iso639-3 | UDHR Code(s) | OpenLID Code(s) | GlotLID-M F1↑ | FPR↓ | OpenLID F1↑ | FPR↓ | GlotLID-M θ=.3 F1↑ | FPR↓ | GlotLID-M θ=.5 F1↑ | FPR↓ | OpenLID θ=.3 F1↑ | FPR↓ | OpenLID θ=.5 F1↑ | FPR↓ |
|---|---|---|---|---|---|---|---|---|---|---|---|---|---|---|
| ace | ace | ace | 0.90909 | 0.00126 | 0.71856 | 0.00247 | **0.91603** | 0.00107 | 0.91603 | 0.00103 | 0.83916 | 0.00247 | **0.88889** | 0.00075 |
| afr | afr | afr | 0.95238 | 0.00068 | 0.88235 | 0.00086 | 0.96774 | 0.00043 | **0.97561** | 0.00031 | 0.90909 | 0.00086 | 0.96 | 0.00027 |
| als | als | als | **1.0** | 0.00205 | 0.32153 | 0.01331 | 0.7377 | 0.00182 | 0.56863 | 0.00134 | 0.38689 | 0.01331 | 0.472 | 0.00702 |
| amh | amh | amh | **1.0** | 0.0 | **1.0** | 0.0 | 1.0 | 0.0 | 1.0 | 0.0 | 1.0 | 0.0 | 1.0 | 0.0 |
| arb | arb | arb | **0.98333** | 0.00011 | 0.92437 | 0.00021 | 0.92035 | 0.00011 | 0.8381 | 0.0001 | 0.94017 | 0.00021 | **0.94828** | 5e-05 |
| ast | ast | ast | 0.97521 | 0.00011 | 0.66667 | 0.00317 | 0.97521 | 0.00011 | **0.98333** | 0.0 | 0.75472 | 0.00317 | **0.81633** | 0.00139 |
| ayr | ayr | ayr | **0.99174** | 0.00011 | 0.75472 | 0.00209 | 0.91071 | 0.00011 | 0.84615 | 0.0 | 0.81081 | 0.00209 | **0.89552** | 0.00075 |
| azb | azb | azb | | | | | | | | | | | | |
| azj | azj | azj | **0.74306** | 0.00696 | 0.48163 | 0.00354 | 0.15789 | 0.00215 | 0.03008 | 0.00113 | 0.4856 | 0.00354 | **0.4876** | 0.00338 |
| bam | bam | bam | 0.49682 | 0.00662 | 0.19403 | 0.02275 | 0.5098 | 0.0058 | **0.55319** | 0.00433 | 0.20553 | 0.02275 | 0.26 | 0.01543 |
| ban | ban | ban | 0.97561 | 0.00011 | 0.79195 | 0.0015 | **0.98361** | 0.0 | 0.97521 | 0.0 | 0.83453 | 0.0015 | **0.87218** | 0.0007 |
| bel | bel | bel | **0.98333** | 0.0 | 0.34706 | 0.0118 | 0.98333 | 0.0 | 0.98333 | 0.0 | 0.35224 | 0.0118 | **0.39333** | 0.00964 |
| bem | bem | bem | **0.98333** | 0.00023 | 0.40411 | 0.00934 | 0.98333 | 0.00021 | 0.98333 | 0.00021 | 0.41844 | 0.00934 | **0.47581** | 0.00696 |
| ben | ben | ben | **1.0** | 0.0 | 0.99213 | 5e-05 | 1.0 | 0.0 | 1.0 | 0.0 | **1.0** | 5e-05 | 1.0 | 0.0 |
| bho | bho | bho | 0.78519 | 0.00034 | 0.78519 | 0.00016 | 0.78519 | 0.00032 | 0.77273 | 0.00021 | **0.79104** | 0.00016 | 0.78195 | 0.00011 |
| bod | bod | bod | **0.89091** | 0.00011 | 0.7451 | 0.00021 | 0.89091 | 0.00011 | 0.89091 | 0.0001 | 0.7451 | 0.00021 | 0.7451 | 0.00021 |
| bos | bos | bos | 0.18103 | 0.01038 | 0.18841 | 0.00698 | 0.18605 | 0.00805 | 0.14607 | 0.00464 | 0.19403 | 0.00698 | **0.20077** | 0.00605 |
| bug | bug | bug | 0.95312 | 0.00057 | 0.76129 | 0.00182 | 0.96063 | 0.00043 | **0.976** | 0.00021 | 0.80272 | 0.00182 | **0.86765** | 0.0008 |
| bul | bul | bul | 0.96 | 0.00057 | 0.9375 | 0.00043 | 0.96 | 0.00054 | **0.97561** | 0.00031 | 0.95238 | 0.00043 | 0.95238 | 0.00032 |
| cat | cat | cat | 0.93023 | 0.00103 | 0.91603 | 0.00059 | 0.95238 | 0.00064 | **0.96774** | 0.00041 | 0.92308 | 0.00059 | 0.96 | 0.00027 |
| ceb | ceb | ceb | 0.96721 | 0.00046 | 0.58416 | 0.00451 | 0.9916 | 0.00011 | **1.0** | 0.0 | 0.60825 | 0.00451 | **0.64481** | 0.00348 |
| ces | ces | ces | **0.98387** | 0.0 | 0.93846 | 0.00032 | 0.98387 | 0.0 | 0.98387 | 0.0 | 0.96063 | 0.00032 | **0.976** | 5e-05 |
| cjk | cjk | cjk | **0.92641** | 0.00034 | 0.61995 | 0.00724 | 0.92641 | 0.00032 | 0.92035 | 0.0001 | 0.63014 | 0.00724 | **0.66667** | 0.00563 |
| ckb | ckb | ckb | | | | | | | | | | | | |
| crh | crh | crh | 0.97561 | 0.00034 | 0.82759 | 0.00134 | **0.98361** | 0.00021 | 0.98361 | 0.00021 | 0.86331 | 0.00134 | **0.88235** | 0.00086 |
| cym | cym | cym | **1.0** | 0.0 | 0.82667 | 0.0014 | 1.0 | 0.0 | 1.0 | 0.0 | 0.91852 | 0.0014 | **0.96124** | 0.00027 |
| dan | dan | dan | 0.85135 | 0.00251 | 0.84932 | 0.00113 | 0.91304 | 0.00129 | **0.98437** | 0.00021 | 0.88571 | 0.00113 | **0.96124** | 0.00027 |
| deu | deu | deu | **0.98745** | 0.00011 | 0.97119 | 0.00027 | 0.98745 | 0.00011 | 0.98745 | 0.0001 | 0.97521 | 0.00027 | 0.98333 | 0.00011 |
| dyu | dyu | dyu | **0.23188** | 0.00719 | 0.05594 | 0.00429 | 0.22059 | 0.00665 | 0.17323 | 0.00587 | 0.05714 | 0.00429 | **0.06452** | 0.00327 |
| dzo | dzo | dzo | **0.90769** | 0.00126 | 0.81159 | 0.00118 | 0.90769 | 0.00118 | 0.90769 | 0.00113 | 0.81159 | 0.00118 | 0.81159 | 0.00118 |
| ell | ell | ell | 0.97908 | 0.0 | **1.0** | 0.0 | 0.97908 | 0.0 | 0.97908 | 0.0 | 1.0 | 0.0 | 1.0 | 0.0 |
| eng | eng | eng | 0.85294 | 0.00205 | 0.43123 | 0.0081 | 0.87218 | 0.00161 | **0.8855** | 0.00134 | 0.46586 | 0.0081 | **0.50435** | 0.006 |
| epo | epo | epo | 0.96825 | 0.00046 | 0.63492 | 0.00365 | **0.976** | 0.00032 | 0.976 | 0.00031 | 0.69767 | 0.00365 | **0.77419** | 0.00182 |
| est | ekk | est | 0.9375 | 0.00091 | 0.33803 | 0.01261 | **1.0** | 0.0 | 1.0 | 0.0 | 0.41096 | 0.01261 | **0.55556** | 0.00514 |
| eus | eus | eus | 0.91729 | 0.00126 | 0.16901 | 0.0316 | 0.96825 | 0.00043 | **0.98387** | 0.00021 | 0.22901 | 0.0316 | **0.34091** | 0.01238 |
| ewe | ewe | ewe | **0.98361** | 0.00023 | 0.38462 | 0.0103 | 0.98361 | 0.00021 | 0.98361 | 0.00021 | 0.4 | 0.0103 | **0.43165** | 0.00847 |
| fao | fao | fao | **0.98305** | 0.0 | 0.94309 | 0.00027 | 0.98305 | 0.0 | 0.98305 | 0.0 | 0.96667 | 0.00027 | **0.98305** | 0.0 |
| fij | fij | fij | **1.0** | 0.0 | 0.9 | 0.00075 | 1.0 | 0.0 | 1.0 | 0.0 | 0.94737 | 0.00075 | **0.96923** | 0.00021 |
| fin | fin | fin | 0.36311 | 0.02522 | 0.20064 | 0.02694 | 0.38769 | 0.02136 | **0.41311** | 0.01844 | 0.23909 | 0.02694 | **0.29717** | 0.01597 |
| fon | fon | fon | 0.94118 | 0.00046 | 0.34627 | 0.0117 | 0.94915 | 0.00032 | **0.95726** | 0.00021 | 0.35692 | 0.0117 | **0.42804** | 0.00825 |
| fra | fra | fra | 0.95238 | 0.00068 | 0.95161 | 0.00027 | 0.95935 | 0.00043 | 0.9661 | 0.0001 | 0.96721 | 0.00027 | **0.98333** | 5e-05 |
| fur | fur | fur | 0.944 | 0.00068 | 0.45736 | 0.00746 | 0.95935 | 0.00043 | **0.96721** | 0.00031 | 0.46825 | 0.00746 | **0.5514** | 0.00509 |
| fuv | fuv | fuv | 0.77912 | 0.00217 | 0.52133 | 0.0096 | 0.80833 | 0.00107 | **0.81197** | 0.00062 | 0.57743 | 0.0096 | **0.69401** | 0.00396 |
| gaz | gaz | gaz | 0.83221 | 0.00285 | 0.248 | 0.02017 | **0.88235** | 0.0015 | 0.83761 | 0.00062 | 0.27991 | 0.02017 | **0.36364** | 0.01163 |
| gla | gla | gla | 0.94574 | 0.00023 | 0.57416 | 0.00445 | 0.95312 | 0.00011 | **0.96063** | 0.0 | 0.64516 | 0.00445 | **0.71429** | 0.00225 |
| gle | gle | gle | 0.95 | 0.00057 | 0.7037 | 0.00343 | **0.96815** | 0.00054 | 0.96815 | 0.00052 | 0.80423 | 0.00343 | **0.89412** | 0.0096 |
| glg | glg | glg | **0.98305** | 0.00011 | 0.88372 | 0.0007 | 0.98305 | 0.00011 | 0.98305 | 0.0001 | 0.89764 | 0.0007 | **0.91935** | 0.00043 |
| grn | gug | grn | 0.816 | 0.00023 | 0.23742 | 0.01964 | **0.82927** | 0.0 | 0.81967 | 0.0 | 0.26281 | 0.01964 | **0.30287** | 0.01355 |
| guj | guj | guj | **1.0** | 0.0 | **1.0** | 0.0 | 1.0 | 0.0 | 1.0 | 0.0 | 1.0 | 0.0 | 1.0 | 0.0 |
| hat | hat | hat | 0.90706 | 0.00285 | 0.41368 | 0.01835 | 0.9313 | 0.00193 | **0.95312** | 0.00124 | 0.45149 | 0.01835 | **0.51489** | 0.01216 |
| hau | hau | hau | 0.94488 | 0.0024 | 0.57508 | 0.00427 | 0.96 | 0.00161 | **0.97297** | 0.00103 | 0.6679 | 0.00427 | **0.78261** | 0.00536 |
| heb | heb | heb | 0.99145 | 0.00011 | 0.99145 | 5e-05 | **1.0** | 0.0 | 1.0 | 0.0 | 0.99145 | 5e-05 | 0.99145 | 5e-05 |
| hin | hin | hin | 0.62 | 0.00867 | 0.6359 | 0.00381 | 0.62312 | 0.00805 | 0.62944 | 0.00752 | 0.6359 | 0.00381 | **0.63918** | 0.00375 |
| hrv | hrv | hrv | 0.60302 | 0.00902 | 0.52252 | 0.00558 | 0.62176 | 0.00784 | **0.69364** | 0.00546 | 0.57426 | 0.00558 | 0.62032 | 0.0037 |
| hun | hun | hun | 0.82192 | 0.00297 | 0.56459 | 0.00483 | 0.84507 | 0.00236 | **0.89552** | 0.00144 | 0.71084 | 0.00483 | **0.81379** | 0.00139 |
| hye | hye | hye | **1.0** | 0.0 | **1.0** | 0.0 | 1.0 | 0.0 | 1.0 | 0.0 | 1.0 | 0.0 | 1.0 | 0.0 |
| ibo | ibo | ibo | 0.98718 | 0.00023 | 0.63115 | 0.00483 | 0.99355 | 0.00011 | **1.0** | 0.0 | 0.66379 | 0.00483 | **0.74038** | 0.00289 |
| ilo | ilo | ilo | 0.91339 | 0.00126 | 0.71166 | 0.00252 | 0.928 | 0.00097 | **0.97479** | 0.00031 | 0.8227 | 0.00252 | **0.92063** | 0.00054 |
| ind | ind | ind | 0.72483 | 0.00411 | 0.68531 | 0.00188 | 0.76056 | 0.00311 | **0.78261** | 0.00258 | 0.71014 | 0.00188 | **0.72593** | 0.00145 |
| isl | isl | isl | **0.9916** | 0.00011 | **0.9916** | 5e-05 | 0.9916 | 0.00011 | 0.9916 | 0.0001 | 0.9916 | 5e-05 | 0.9916 | 5e-05 |
| ita | ita | ita | 0.7947 | 0.00342 | 0.67416 | 0.00306 | 0.83916 | 0.00236 | **0.86957** | 0.00175 | 0.73171 | 0.00306 | **0.78431** | 0.00171 |
| jav | jav | jav | **0.97581** | 0.00046 | 0.41404 | 0.00553 | 0.97581 | 0.00043 | 0.97561 | 0.00031 | 0.46275 | 0.00553 | **0.57214** | 0.00284 |
| jpn | jpn | jpn | 0.72195 | 0.01301 | 0.64745 | 0.00842 | 0.7861 | 0.00848 | **0.79245** | 0.00783 | 0.71921 | 0.00842 | **0.77249** | 0.0045 |
| kan | kan | kan | **1.0** | 0.0 | 0.99394 | 5e-05 | 1.0 | 0.0 | 1.0 | 0.0 | **1.0** | 5e-05 | 1.0 | 0.0 |
| kat | kat | kat | **1.0** | 0.0 | **1.0** | 0.0 | 1.0 | 0.0 | 1.0 | 0.0 | **1.0** | 5e-05 | 1.0 | 0.0 |
| kaz | kaz | kaz | 0.96721 | 0.00034 | 0.40989 | 0.00885 | **0.97521** | 0.00021 | 0.96667 | 0.00021 | 0.41281 | 0.00885 | **0.42336** | 0.00836 |
| kbp | kbp | kbp | 0.85714 | 0.00228 | 0.24048 | 0.02103 | 0.90909 | 0.00129 | **0.9375** | 0.00082 | 0.24048 | 0.02103 | **0.26786** | 0.01757 |
| kea | kea | kea | **0.72727** | 0.00114 | 0.07627 | 0.00896 | 0.72727 | 0.00107 | 0.69811 | 0.00093 | 0.08333 | 0.00896 | **0.07254** | 0.00675 |
| khk | khk | khk | **0.99225** | 0.0 | 0.5 | 0.00681 | 0.63158 | 0.0 | 0.33333 | 0.0 | 0.50593 | 0.00681 | **0.55172** | 0.00552 |
| khm | khm | khm | **1.0** | 0.0 | **1.0** | 0.0 | 1.0 | 0.0 | 1.0 | 0.0 | 1.0 | 0.0 | 1.0 | 0.0 |
| kin | kin | kin | 0.76336 | 0.0024 | 0.53061 | 0.00258 | 0.81967 | 0.00129 | **0.8547** | 0.00072 | 0.56522 | 0.00258 | **0.64463** | 0.00118 |
| kir | kir | kir | 0.94488 | 0.0008 | 0.35503 | 0.0117 | 0.95238 | 0.00064 | **0.96774** | 0.00041 | 0.35714 | 0.0117 | **0.39216** | 0.00997 |
| kmb | kmb | kmb | **0.99194** | 0.00011 | 0.82034 | 0.00268 | 0.99194 | 0.00011 | 0.99194 | 0.0001 | 0.86738 | 0.00268 | **0.92015** | 0.00096 |
| kmr | kmr | kmr | **0.66667** | 0.00673 | 0.61458 | 0.00397 | 0.66667 | 0.00633 | 0.66667 | 0.00608 | 0.63441 | 0.00397 | **0.64481** | 0.00348 |
| knc | knc | knc | 0.97059 | 0.00034 | 0.5641 | 0.00542 | **0.97778** | 0.00021 | 0.97778 | 0.00021 | 0.63768 | 0.00542 | **0.76301** | 0.00214 |
| kon | kng | kon | 0.40789 | 0.0 | 0.55172 | 0.00816 | 0.40789 | 0.0 | 0.39735 | 0.0 | 0.56522 | 0.00816 | **0.56742** | 0.00718 |
| kor | kor | kor | 0.94488 | 0.0008 | 0.83333 | 0.00129 | 0.96 | 0.00054 | **0.99174** | 0.0001 | 0.90909 | 0.00129 | **0.96774** | 0.00021 |
| lao | lao | lao | **1.0** | 0.0 | **1.0** | 0.0 | 1.0 | 0.0 | 1.0 | 0.0 | 1.0 | 0.0 | 1.0 | 0.0 |
| lij | lij | lij | 0.49785 | 0.01324 | 0.43446 | 0.00805 | 0.53953 | 0.01052 | **0.64804** | 0.00639 | 0.46032 | 0.00805 | **0.53211** | 0.00541 |

Table 46: Comparison of GlotLID vs OpenLID on UDHR benchmark (part 1)

| | | | GlotLID-M | | OpenLID | | with confidence threshold θ | | | | | | | |
| | | | | | | | GlotLID-M θ=.3 | | GlotLID-M θ=.5 | | OpenLID θ=.3 | | OpenLID θ=.5 | |
| iso639-3 | UDHR Code(s) | OpenLID Code(s) | F1↑ | FPR↓ | F1↑ | FPR↓ | F1↑ | FPR↓ | F1↑ | FPR↓ | F1↑ | FPR↓ | F1↑ | FPR↓ |
|---|---|---|---|---|---|---|---|---|---|---|---|---|---|---|
| lin | lin | lin | 0.99145 | 0.00023 | 0.93173 | 0.00091 | **1.0** | 0.0 | 1.0 | 0.0 | 0.95868 | 0.00091 | 0.98712 | 0.00011 |
| lit | lit | lit | 0.9375 | 0.00091 | 0.78146 | 0.00172 | 0.96774 | 0.00043 | **0.98361** | 0.00021 | 0.90769 | 0.00172 | 0.95161 | 0.00027 |
| ltz | ltz | ltz | **0.98305** | 0.0 | 0.87879 | 0.00075 | 0.98305 | 0.0 | 0.98305 | 0.0 | 0.87879 | 0.00075 | 0.90625 | 0.00054 |
| lua | lua | lua | 0.71186 | 0.00183 | 0.68263 | 0.00268 | 0.75 | 0.00107 | **0.78846** | 0.00031 | 0.72152 | 0.00268 | 0.7619 | 0.00166 |
| lug | lug | lug | 0.9589 | 0.00068 | 0.4 | 0.01127 | 0.97222 | 0.00043 | **0.98592** | 0.00021 | 0.4375 | 0.01127 | 0.49822 | 0.00755 |
| lus | lus | lus | 0.93548 | 0.00091 | 0.28087 | 0.01594 | 0.95868 | 0.00054 | **0.97479** | 0.00031 | 0.30208 | 0.01594 | 0.31868 | 0.01329 |
| lvs | lvs | lvs | 0.94118 | 0.00171 | 0.9375 | 0.0008 | 0.92623 | 0.00118 | 0.90213 | 0.00093 | 0.97959 | 0.00086 | **0.98361** | 0.00021 |
| mag | mag | mag | 0.75385 | 0.00046 | 0.76336 | 0.00021 | **0.76562** | 0.00021 | 0.76562 | 0.00021 | 0.76336 | 0.00021 | 0.75969 | 0.00016 |
| mai | mai | mai | **0.83099** | 0.0 | 0.81944 | 0.00011 | 0.83099 | 0.0 | 0.83099 | 0.0 | 0.81944 | 0.00011 | 0.81944 | 0.00011 |
| mal | mal | mal | **1.0** | 0.0 | 0.99595 | 5e-05 | 1.0 | 0.0 | 1.0 | 0.0 | 0.99595 | 5e-05 | **1.0** | 0.0 |
| mar | mar | mar | **0.99174** | 0.00011 | 0.98361 | 0.00011 | 0.99174 | 0.00011 | 0.99174 | 0.0001 | **0.99174** | 0.00011 | 0.99174 | 5e-05 |
| min | min | min | 0.88235 | 0.00171 | 0.63492 | 0.00365 | 0.91603 | 0.00107 | **0.92187** | 0.00082 | 0.69767 | 0.00365 | 0.74534 | 0.00214 |
| mkd | mkd | mkd | **0.99174** | 0.0 | 0.97561 | 0.00011 | 0.99174 | 0.0 | 0.99174 | 0.0 | 0.97561 | 0.00011 | 0.98361 | 5e-05 |
| mlt | mlt | mlt | 0.77419 | 0.00399 | 0.59 | 0.00435 | 0.78947 | 0.00343 | **0.82759** | 0.00258 | 0.68208 | 0.00435 | 0.82517 | 0.00129 |
| mos | mos | mos | 0.97015 | 0.00046 | 0.56769 | 0.00531 | 0.98485 | 0.00021 | **0.99237** | 0.0001 | 0.62201 | 0.00531 | 0.7027 | 0.00295 |
| mri | mri | mri | 0.8227 | 0.00023 | 0.26608 | 0.01663 | **0.82857** | 0.0001 | 0.82857 | 0.0001 | 0.27523 | 0.01663 | 0.2864 | 0.01489 |
| mya | mya | mya | 0.66292 | 0.00673 | 0.66667 | 0.00311 | 0.66292 | 0.00633 | **0.67045** | 0.00587 | 0.66667 | 0.00311 | 0.66667 | 0.00311 |
| nld | nld | nld | 0.70238 | 0.00571 | 0.57843 | 0.00461 | 0.71084 | 0.00515 | **0.71515** | 0.00484 | 0.68208 | 0.00461 | **0.77124** | 0.00188 |
| nno | nno | nno | 0.95868 | 0.00034 | 0.912 | 0.00043 | 0.9661 | 0.0001 | 0.9661 | 0.0001 | 0.92683 | 0.00043 | 0.94215 | 0.00021 |
| nob | nob | nob | 0.98462 | 0.00023 | 0.85906 | 0.00113 | **0.99225** | 0.00011 | 0.98438 | 0.0001 | 0.92754 | 0.00113 | 0.96241 | 0.00027 |
| npi | npi | npi | **0.98214** | 0.0 | 0.97345 | 5e-05 | 0.575 | 0.0 | 0.32353 | 0.0 | 0.97345 | 5e-05 | 0.97297 | 0.0 |
| nso | nso | nso | 0.86957 | 0.00205 | 0.80537 | 0.00156 | 0.87591 | 0.00182 | **0.88235** | 0.00165 | 0.82759 | 0.00156 | 0.83916 | 0.00123 |
| nya | nya | nya | 0.96414 | 0.00103 | 0.74462 | 0.00445 | 0.97581 | 0.00064 | **0.99588** | 0.0001 | 0.7707 | 0.00445 | 0.86121 | 0.00209 |
| oci | oci | oci | 0.41101 | 0.0 | **0.41187** | 0.00118 | 0.40516 | 0.0 | 0.38131 | 0.0 | 0.40773 | 0.00118 | 0.38838 | 0.00059 |
| pan | pan | pan | **1.0** | 0.0 | **1.0** | 0.0 | 1.0 | 0.0 | 1.0 | 0.0 | 1.0 | 0.0 | 1.0 | 0.0 |
| pap | pap | pap | 0.79195 | 0.00354 | 0.35224 | 0.01164 | 0.83099 | 0.00258 | **0.86765** | 0.00185 | 0.41404 | 0.01164 | 0.5514 | 0.00514 |
| pes | pes | pes | **0.65922** | 0.00696 | 0.62745 | 0.00247 | 0.63855 | 0.0058 | 0.54135 | 0.00391 | **0.63158** | 0.00247 | 0.63158 | 0.00241 |
| plt | plt | plt | **0.98182** | 0.0 | 0.92308 | 0.00038 | 0.98182 | 0.0 | 0.98182 | 0.0 | 0.95575 | 0.00038 | **0.97297** | 5e-05 |
| pol | pol | pol | 0.74074 | 0.00479 | 0.64516 | 0.00354 | 0.76923 | 0.00386 | **0.81633** | 0.00278 | 0.68966 | 0.00354 | 0.76433 | 0.00198 |
| por | por | por | 0.86331 | 0.00434 | 0.70588 | 0.00376 | 0.87273 | 0.00376 | **0.89219** | 0.00299 | 0.73171 | 0.00537 | 0.7717 | 0.0038 |
| prs | prs | prs | 0.0 | 0.00274 | 0.33333 | 0.00075 | 0.0 | 0.00258 | 0.0 | 0.00247 | **0.34483** | 0.00075 | 0.34483 | 0.00059 |
| quy | quy | quy | 0.70115 | 0.00593 | 0.10816 | 0.05398 | 0.82993 | 0.00268 | **0.88406** | 0.00165 | 0.11244 | 0.05398 | 0.11949 | 0.04817 |
| ron | ron | ron | 0.80992 | 0.00514 | 0.76078 | 0.00317 | 0.80833 | 0.00472 | **0.82906** | 0.00391 | 0.776 | 0.00317 | 0.79508 | 0.00257 |
| run | run | run | 0.88235 | 0.00183 | 0.44776 | 0.00794 | 0.88889 | 0.00161 | **0.90909** | 0.00124 | 0.50633 | 0.00794 | 0.6 | 0.00429 |
| rus | rus | rus | 0.43321 | 0.01792 | 0.37037 | 0.01095 | 0.47431 | 0.01427 | **0.51064** | 0.01185 | 0.37855 | 0.01095 | 0.41522 | 0.00905 |
| sag | sag | sag | **0.81553** | 0.0 | 0.65806 | 0.00231 | 0.80392 | 0.0 | 0.79208 | 0.0 | 0.74453 | 0.00231 | 0.81356 | 0.00048 |
| san | san | san | **0.66667** | 0.0 | 0.66376 | 5e-05 | 0.66667 | 0.0 | 0.66667 | 0.0 | 0.66376 | 5e-05 | 0.66376 | 5e-05 |
| shn | shn | shn | **0.99145** | 0.0 | 0.99145 | 0.0 | 0.99145 | 0.0 | 0.99145 | 0.0 | 0.99145 | 0.0 | 0.99145 | 0.0 |
| sin | sin | sin | **1.0** | 0.0 | **1.0** | 0.0 | 1.0 | 0.0 | 1.0 | 0.0 | 1.0 | 0.0 | 1.0 | 0.0 |
| slk | slk | slk | 0.87591 | 0.00194 | 0.68571 | 0.00295 | 0.9375 | 0.00086 | **0.94488** | 0.00072 | 0.75472 | 0.00295 | 0.90909 | 0.00064 |
| slv | slv | slv | 0.89552 | 0.0016 | 0.50847 | 0.00622 | 0.95238 | 0.00064 | **0.97561** | 0.00031 | 0.65934 | 0.00622 | 0.78947 | 0.00171 |
| smo | smo | smo | **1.0** | 0.0 | 0.84564 | 0.00123 | 1.0 | 0.0 | 1.0 | 0.0 | 0.85135 | 0.00123 | 0.875 | 0.00096 |
| sna | sna | sna | 0.93846 | 0.00091 | 0.41781 | 0.00912 | 0.96063 | 0.00054 | **0.98387** | 0.00021 | 0.54955 | 0.00912 | 0.7673 | 0.00198 |
| som | som | som | 0.75817 | 0.00422 | 0.44106 | 0.00789 | 0.80556 | 0.00301 | **0.89231** | 0.00144 | 0.48333 | 0.00789 | 0.59184 | 0.00422 |
| sot | sot | sot | 0.98333 | 0.00011 | 0.86667 | 0.00043 | **0.9916** | 0.0 | 0.9916 | 0.0 | 0.87395 | 0.00043 | 0.88889 | 0.00027 |
| spa | spa | spa | 0.72321 | 0.00685 | 0.69333 | 0.00343 | 0.75701 | 0.00537 | **0.78** | 0.00402 | 0.72558 | 0.00343 | 0.75362 | 0.00246 |
| srd | srd | src | **0.9916** | 0.0 | 0.63158 | 0.00376 | 0.9916 | 0.0 | 0.9916 | 0.0 | 0.66667 | 0.00376 | 0.76433 | 0.00198 |
| srp | srp | srp | **0.5124** | 0.00685 | 0.48133 | 0.00338 | 0.4958 | 0.00633 | 0.48945 | 0.00608 | 0.48333 | 0.00338 | 0.48536 | 0.00327 |
| ssw | ssw | ssw | 0.94891 | 0.00068 | 0.72626 | 0.00258 | 0.97015 | 0.00032 | **0.99237** | 0.0 | 0.76923 | 0.00258 | 0.89655 | 0.00075 |
| sun | sun | sun | 0.9697 | 0.00034 | 0.52282 | 0.00606 | **0.9771** | 0.00021 | 0.9771 | 0.00021 | 0.63317 | 0.00606 | 0.77301 | 0.00188 |
| swe | swe | swe | 0.86301 | 0.00228 | 0.89362 | 0.0008 | 0.93333 | 0.00097 | **1.0** | 0.0 | 0.96183 | 0.0008 | 0.98437 | 0.00011 |
| swh | swh | swh | **0.84956** | 0.00046 | 0.1868 | 0.02694 | 0.84685 | 0.00032 | 0.78 | 0.00021 | 0.22351 | 0.02694 | 0.29517 | 0.01468 |
| tam | tam | tam | **1.0** | 0.0 | **1.0** | 0.0 | 1.0 | 0.0 | 1.0 | 0.0 | 1.0 | 0.0 | 1.0 | 0.0 |
| tat | tat | tat | 0.65556 | 0.00696 | 0.40972 | 0.00907 | 0.68208 | 0.0058 | **0.69822** | 0.00515 | 0.42143 | 0.00907 | 0.46457 | 0.00723 |
| tel | tel | tel | **1.0** | 0.0 | **1.0** | 0.0 | 1.0 | 0.0 | 1.0 | 0.0 | 1.0 | 0.0 | 1.0 | 0.0 |
| tgk | tgk | tgk | 0.67429 | 0.0065 | 0.44106 | 0.00783 | 0.92913 | 0.00097 | **0.95082** | 0.00052 | 0.45312 | 0.00783 | 0.53211 | 0.00541 |
| tgl | tgl | tgl | 0.9403 | 0.0008 | 0.58879 | 0.00467 | 0.95455 | 0.00054 | **0.97674** | 0.00021 | 0.61165 | 0.00467 | 0.6738 | 0.00321 |
| tha | tha | tha | **1.0** | 0.0 | **1.0** | 0.0 | 1.0 | 0.0 | 1.0 | 0.0 | 1.0 | 0.0 | 1.0 | 0.0 |
| tir | tir | tir | **1.0** | 0.0 | **1.0** | 0.0 | 1.0 | 0.0 | 1.0 | 0.0 | 1.0 | 0.0 | 1.0 | 0.0 |
| tpi | tpi | tpi | **0.98361** | 0.00011 | 0.47431 | 0.00708 | 0.98361 | 0.00011 | 0.98361 | 0.0001 | 0.51724 | 0.00708 | 0.56338 | 0.00493 |
| tsn | tsn | tsn | 0.98361 | 0.00023 | 0.85106 | 0.00113 | **0.99174** | 0.00011 | 0.99174 | 0.0001 | 0.90909 | 0.00113 | 0.92308 | 0.00054 |
| tso | tso | tso | 0.94158 | 0.0016 | 0.63205 | 0.00875 | 0.94158 | 0.0015 | **0.95139** | 0.00113 | 0.64965 | 0.00875 | 0.68293 | 0.00696 |
| tuk | tuk | tuk | 0.94821 | 0.00148 | 0.62564 | 0.0008 | 0.96748 | 0.00086 | **0.98347** | 0.00041 | 0.65591 | 0.0008 | 0.66667 | 0.00016 |
| tur | tur | tur | 0.4918 | 0.01415 | 0.34384 | 0.01229 | 0.50209 | 0.01277 | **0.51502** | 0.01164 | 0.35928 | 0.01229 | 0.37975 | 0.0105 |
| twi | twi | twi | 0.95349 | 0.00137 | 0.5234 | 0.01202 | 0.96094 | 0.00107 | **0.9685** | 0.00082 | 0.55034 | 0.01202 | 0.60891 | 0.00847 |
| tzm | tzm | tzm | 0.01754 | 0.00593 | **0.01653** | 0.00317 | 0.01754 | 0.00558 | **0.0177** | 0.00525 | 0.01653 | 0.00317 | 0.01653 | 0.00316 |
| uig | uig | uig | 0.90295 | 0.00114 | 0.66667 | 0.0 | **0.93333** | 0.0 | 0.89908 | 0.0 | 0.66667 | 0.0 | 0.66667 | 0.0 |
| ukr | ukr | ukr | 0.98361 | 0.00023 | 0.96 | 0.00027 | 0.98361 | 0.00021 | **0.99174** | 0.0001 | 0.96 | 0.00027 | 0.98361 | 0.00011 |
| umb | umb | umb | 0.87931 | 0.00114 | 0.85409 | 0.0022 | 0.87931 | 0.00107 | 0.87611 | 0.00072 | 0.62011 | 0.0022 | **0.90226** | 0.00139 |
| urd | urd | urd | **0.96522** | 0.0008 | 0.61838 | 0.00075 | 0.96522 | 0.00072 | 0.96522 | 0.00072 | 0.62011 | 0.0073 | 0.62535 | 0.00707 |
| uzn | uzn | uzn | 0.50407 | 0.0073 | 0.18072 | 0.02597 | 0.53881 | 0.00429 | **0.56716** | 0.00247 | 0.21277 | 0.02597 | 0.27523 | 0.01372 |
| vec | vec | vec | 0.92308 | 0.00114 | 0.76923 | 0.00193 | 0.95238 | 0.00064 | **0.96774** | 0.00041 | 0.7947 | 0.00193 | 0.89552 | 0.00075 |
| vie | vie | vie | **0.66304** | 0.00011 | 0.61616 | 0.0008 | 0.66304 | 0.00011 | 0.66304 | 0.0001 | 0.62887 | 0.0008 | 0.63874 | 0.00043 |
| war | war | war | **0.9916** | 0.0 | 0.90909 | 0.00064 | 0.9916 | 0.0 | 0.9916 | 0.0 | 0.9375 | 0.00064 | 0.94488 | 0.00038 |
| wol | wol | wol | 0.79747 | 0.00365 | 0.39117 | 0.0103 | 0.86897 | 0.00204 | **0.92537** | 0.00093 | 0.45756 | 0.0103 | 0.56881 | 0.00498 |
| xho | xho | xho | 0.93846 | 0.00091 | 0.62245 | 0.00397 | 0.96825 | 0.00043 | **0.976** | 0.00031 | 0.69714 | 0.00397 | 0.78205 | 0.00182 |
| ydd | ydd | ydd | **0.99187** | 0.0 | 0.99187 | 0.0 | 0.99187 | 0.0 | 0.99187 | 0.0 | 0.99187 | 0.0 | 0.99187 | 0.0 |
| yor | yor | yor | 0.85106 | 0.0024 | 0.33898 | 0.01256 | 0.88889 | 0.00161 | **0.93023** | 0.00093 | 0.41667 | 0.01256 | 0.57692 | 0.00471 |
| zho | cjy/hak/cmn/hsn/yue/gan/wuu/nan | zho/yue | 0.96345 | 0.00011 | 0.62049 | 0.04561 | **0.96408** | 0.0 | 0.96408 | 0.0 | 0.71366 | 0.04561 | 0.8037 | 0.01645 |
| zul | zul | zul | **0.98305** | 0.0 | 0.58883 | 0.00424 | 0.98305 | 0.0 | 0.98305 | 0.0 | 0.61053 | 0.00424 | 0.65169 | 0.00321 |

Table 47: Comparison of GlotLID vs OpenLID on UDHR benchmark (part 2)

| | | | GlotLID-M | | NLLB | | with confidence threshold $\theta$ | | | | | | | |
| | | | | | | | GlotLID-M $\theta=.3$ | | GlotLID-M $\theta=.5$ | | NLLB $\theta=.3$ | | NLLB $\theta=.5$ | |
| iso639-3 | UDHR Code(s) | NLLB Code(s) | F1↑ | FPR↓ | F1↑ | FPR↓ | F1↑ | FPR↓ | F1↑ | FPR↓ | F1↑ | FPR↓ | F1↑ | FPR↓ |
|---|---|---|---|---|---|---|---|---|---|---|---|---|---|---|
| abk | abk | abk | | | 0.98333 | 6e-05 | | | | | **0.9916** | 6e-05 | 0.9916 | 0.0 |
| ace | ace | ace | 0.90909 | 0.00126 | 0.53153 | 0.0057 | **0.91603** | 0.00107 | 0.91603 | 0.00103 | 0.57005 | 0.0057 | 0.62434 | 0.0038 |
| ady | ady | ady | **0.83495** | 0.0 | 0.63441 | 0.00375 | 0.83495 | 0.0 | 0.83495 | 0.0 | 0.65193 | 0.00375 | 0.67045 | 0.00314 |
| afr | afr | afr | 0.95238 | 0.00068 | 0.90769 | 0.00062 | 0.96774 | 0.00043 | **0.97561** | 0.00031 | 0.91473 | 0.00062 | 0.93651 | 0.00039 |
| aka | fat/twi | aka/twi | **1.0** | 0.0 | 0.60884 | 0.01264 | 1.0 | 0.0 | 1.0 | 0.0 | 0.65209 | 0.01264 | 0.72032 | 0.00743 |
| als | als | als | **0.86131** | 0.00205 | 0.43066 | 0.00867 | 0.7377 | 0.00182 | 0.56863 | 0.00134 | 0.48163 | 0.00867 | 0.58416 | 0.00457 |
| alt | alt | alt | 0.95495 | 0.00034 | 0.76259 | 0.00173 | **0.96364** | 0.00021 | 0.96364 | 0.00021 | 0.76259 | 0.00173 | 0.80303 | 0.00132 |
| amh | amh | amh | **1.0** | 0.0 | 0.88189 | 6e-05 | 1.0 | 0.0 | 1.0 | 0.0 | 0.7156 | 6e-05 | 0.6 | 0.0 |
| arb | arb | arb | **0.98333** | 0.00011 | 0.60914 | 0.00431 | 0.92035 | 0.00011 | 0.8381 | 0.0001 | 0.76433 | 0.00431 | 0.88235 | 0.00088 |
| arn | arn | arn | 0.93913 | 0.00011 | 0.75676 | 0.00179 | **0.94737** | 0.0 | 0.94737 | 0.0 | 0.86154 | 0.00179 | 0.9322 | 0.00017 |
| ast | ast | ast | 0.97521 | 0.00011 | 0.57561 | 0.00475 | 0.97521 | 0.00011 | **0.98333** | 0.0 | 0.76129 | 0.00475 | 0.86765 | 0.00088 |
| ayr | ayr | ayr | **0.99174** | 0.00011 | 0.69461 | 0.00274 | 0.91071 | 0.00011 | 0.84615 | 0.0 | 0.76821 | 0.00274 | 0.87218 | 0.00083 |
| azb | azb | azb | | | | | | | | | | | | |
| azj | azj | azj | **0.74306** | 0.00696 | 0.42754 | 0.00542 | 0.15789 | 0.00215 | 0.03008 | 0.00113 | 0.45736 | 0.00542 | 0.4739 | 0.00385 |
| bam | bam | bam | 0.49682 | 0.00662 | 0.20961 | 0.01957 | 0.5098 | 0.0058 | **0.55319** | 0.00433 | 0.22857 | 0.01957 | 0.27666 | 0.01315 |
| ban | ban | ban | 0.97561 | 0.00011 | 0.7044 | 0.00229 | **0.98361** | 0.0 | 0.97521 | 0.0 | 0.77241 | 0.00229 | 0.82963 | 0.00094 |
| bel | bel | bel | **0.98333** | 0.0 | 0.80272 | 0.00151 | 0.98333 | 0.0 | 0.98333 | 0.0 | 0.81379 | 0.00151 | 0.88722 | 0.00072 |
| bem | bem | bem | **0.98333** | 0.00023 | 0.40714 | 0.00917 | 0.98333 | 0.00021 | 0.98333 | 0.00021 | 0.41455 | 0.00917 | 0.46914 | 0.00699 |
| ben | ben | ben | **1.0** | 0.0 | 0.992 | 0.0 | 1.0 | 0.0 | 1.0 | 0.0 | 0.992 | 0.0 | 0.992 | 0.0 |
| bho | bho | bho | **0.78519** | 0.00034 | 0.66116 | 0.00011 | 0.78519 | 0.00032 | 0.77273 | 0.00021 | 0.66116 | 0.00011 | 0.66116 | 0.00011 |
| bis | bis | bis | **1.0** | 0.0 | 0.71006 | 0.00274 | 1.0 | 0.0 | 1.0 | 0.0 | 0.71856 | 0.00274 | 0.75949 | 0.00209 |
| bod | bod | bod | **0.89091** | 0.00011 | 0.61224 | 0.00425 | 0.89091 | 0.00011 | 0.89091 | 0.0001 | 0.72727 | 0.00425 | 0.77922 | 0.00187 |
| bos | bos | bos | 0.18103 | 0.01039 | 0.14857 | 0.01141 | **0.18605** | 0.00805 | 0.14607 | 0.00464 | 0.16456 | 0.01141 | 0.18056 | 0.00781 |
| bug | bug | bug | 0.95312 | 0.00057 | 0.7439 | 0.00229 | 0.96063 | 0.00043 | **0.976** | 0.00021 | 0.82432 | 0.00229 | 0.91729 | 0.00055 |
| bul | bul | bul | 0.96 | 0.00057 | 0.95868 | 0.00017 | 0.96 | 0.00054 | **0.97561** | 0.00031 | 0.95868 | 0.00017 | 0.97479 | 6e-05 |
| cat | cat | cat | 0.93023 | 0.00103 | 0.49793 | 0.00677 | 0.95238 | 0.00064 | **0.96774** | 0.00041 | 0.63492 | 0.00677 | 0.81633 | 0.00149 |
| ceb | ceb | ceb | 0.96721 | 0.00046 | 0.57143 | 0.00481 | 0.9916 | 0.00011 | **1.0** | 0.0 | 0.59487 | 0.00481 | 0.61702 | 0.00391 |
| ces | ces | ces | **0.98387** | 0.0 | 0.79487 | 0.00173 | 0.98387 | 0.0 | 0.98387 | 0.0 | 0.89855 | 0.00173 | 0.94656 | 0.00033 |
| chv | chv | chv | **0.86154** | 0.0 | 0.8 | 0.00028 | 0.86154 | 0.0 | 0.86154 | 0.0 | 0.82353 | 0.00028 | 0.83582 | 0.00011 |
| cjk | cjk | cjk | **0.92641** | 0.00034 | 0.56296 | 0.00951 | 0.92641 | 0.00032 | 0.92035 | 0.0001 | 0.57431 | 0.00951 | 0.6137 | 0.00726 |
| ckb | ckb | ckb | | | | | | | | | | | | |
| crh | crh | crh | 0.97561 | 0.00034 | 0.9375 | 0.00045 | 0.98361 | 0.00021 | 0.98361 | 0.00021 | 0.96 | 0.00045 | **1.0** | 0.0 |
| cym | cym | cym | **1.0** | 0.0 | 0.77215 | 0.00196 | 1.0 | 0.0 | 1.0 | 0.0 | 0.82432 | 0.00196 | 0.87143 | 0.00094 |
| dan | dan | dan | 0.85135 | 0.00251 | 0.87143 | 0.00089 | 0.91304 | 0.00129 | **0.98437** | 0.00021 | 0.96825 | 0.00089 | 0.96825 | 0.00011 |
| deu | deu | deu | **0.98745** | 0.00011 | 0.64 | 0.00755 | 0.98745 | 0.00011 | 0.98745 | 0.0001 | 0.73846 | 0.00755 | 0.79208 | 0.00347 |
| dyu | dyu | dyu | **0.23188** | 0.00719 | 0.04167 | 0.00721 | 0.22059 | 0.00666 | 0.17323 | 0.00587 | 0.04396 | 0.00721 | 0.02581 | 0.00517 |
| dzo | dzo | dzo | **0.90769** | 0.00126 | 0.68132 | 0.0 | 0.90769 | 0.00118 | 0.90769 | 0.00113 | 0.68132 | 0.0 | 0.58824 | 0.0 |
| ell | ell | ell | 0.97908 | 0.0 | **0.98333** | 0.0 | 0.97908 | 0.0 | 0.97908 | 0.0 | 0.97479 | 0.0 | 0.97479 | 0.0 |
| eng | eng | eng | 0.85294 | 0.00205 | 0.34188 | 0.01292 | 0.87218 | 0.00161 | **0.8855** | 0.00134 | 0.39867 | 0.01292 | 0.46693 | 0.00754 |
| epo | epo | epo | 0.96825 | 0.00046 | 0.31202 | 0.01504 | **0.976** | 0.00032 | 0.976 | 0.00031 | 0.36858 | 0.01504 | 0.44203 | 0.00847 |
| est | ekk | est | 0.9375 | 0.00091 | 0.27778 | 0.01745 | **1.0** | 0.0 | 1.0 | 0.0 | 0.3252 | 0.01745 | 0.43165 | 0.00869 |
| eus | eus | eus | 0.91729 | 0.00126 | 0.67052 | 0.00302 | 0.96825 | 0.00043 | **0.98387** | 0.00021 | 0.81119 | 0.00302 | 0.91339 | 0.00044 |
| ewe | ewe | ewe | **0.98361** | 0.00023 | 0.36137 | 0.01135 | 0.98361 | 0.00021 | 0.98361 | 0.00021 | 0.4 | 0.01135 | 0.44106 | 0.00798 |
| fao | fao | fao | **0.98305** | 0.0 | 0.64444 | 6e-05 | 0.98305 | 0.0 | 0.98305 | 0.0 | 0.64444 | 6e-05 | 0.65169 | 0.0 |
| fij | fij | fij | **1.0** | 0.0 | 0.96825 | 0.00011 | 1.0 | 0.0 | 1.0 | 0.0 | 0.96825 | 0.00011 | 0.976 | 6e-05 |
| fin | fin | fin | 0.36311 | 0.02522 | 0.2344 | 0.02259 | 0.38769 | 0.02136 | **0.41311** | 0.01844 | 0.27434 | 0.02259 | 0.33155 | 0.0137 |
| fon | fon | fon | 0.94118 | 0.00046 | 0.40702 | 0.00939 | 0.94915 | 0.00032 | **0.95726** | 0.00021 | 0.41135 | 0.00939 | 0.48333 | 0.00677 |
| fra | fra | fra | 0.95238 | 0.00068 | 0.58252 | 0.00481 | 0.95935 | 0.00043 | **0.9661** | 0.0001 | 0.6383 | 0.00481 | 0.68966 | 0.00297 |
| fur | fur | fur | 0.944 | 0.00068 | 0.78146 | 0.00179 | 0.95935 | 0.00043 | **0.96721** | 0.00031 | 0.80822 | 0.00179 | 0.92187 | 0.0005 |
| fuv | fuv | fuv | 0.77912 | 0.00217 | 0.4793 | 0.01208 | 0.80833 | 0.00107 | **0.81197** | 0.00062 | 0.53528 | 0.01208 | 0.66066 | 0.00495 |
| gaz | gaz | gaz | 0.83221 | 0.00285 | 0.41892 | 0.00962 | **0.88235** | 0.0015 | 0.83761 | 0.00062 | 0.43357 | 0.00962 | 0.49206 | 0.00704 |
| gla | gla | gla | 0.94574 | 0.00023 | 0.9375 | 0.00011 | 0.95312 | 0.00011 | **0.96063** | 0.0 | 0.9375 | 0.00011 | 0.95238 | 0.0 |
| gle | gle | gle | 0.95 | 0.00091 | 0.90909 | 0.00078 | **0.96815** | 0.00054 | 0.96815 | 0.00052 | 0.98684 | 0.00078 | **0.99338** | 0.0 |
| glg | glg | glg | **0.98305** | 0.00011 | 0.65169 | 0.00341 | 0.98305 | 0.00011 | 0.98305 | 0.0001 | 0.76821 | 0.00341 | 0.90625 | 0.00061 |
| grn | grn | grn | 0.816 | 0.00023 | 0.28409 | 0.01286 | **0.82927** | 0.0 | 0.81967 | 0.0 | 0.36765 | 0.01286 | 0.53191 | 0.00363 |
| guj | guj | guj | **1.0** | 0.0 | **1.0** | 0.0 | 1.0 | 0.0 | 1.0 | 0.0 | 1.0 | 0.0 | 1.0 | 0.0 |
| hat | hat | hat | 0.90706 | 0.00285 | 0.68768 | 0.00598 | 0.9313 | 0.00193 | **0.95312** | 0.00124 | 0.69971 | 0.00598 | 0.71856 | 0.00506 |
| hau | hau | hau | 0.94488 | 0.0024 | 0.83256 | 0.00397 | 0.96 | 0.00161 | **0.97297** | 0.00103 | 0.87745 | 0.00397 | 0.93229 | 0.00138 |
| heb | heb | heb | 0.99145 | 0.00011 | **1.0** | 0.0 | 1.0 | 0.0 | 1.0 | 0.0 | 1.0 | 0.0 | 1.0 | 0.0 |
| hin | hin | hin | 0.62 | 0.00867 | 0.6359 | 0.00397 | 0.62312 | 0.00805 | 0.62944 | 0.00752 | **0.64583** | 0.00397 | 0.64583 | 0.00374 |
| hrv | hrv | hrv | 0.60302 | 0.00902 | 0.42636 | 0.008 | 0.62176 | 0.00784 | **0.69364** | 0.00546 | 0.51643 | 0.008 | 0.58511 | 0.00402 |
| hun | hun | hun | 0.82192 | 0.00297 | 0.22814 | 0.0227 | 0.84507 | 0.00236 | **0.89552** | 0.00144 | 0.3183 | 0.0227 | 0.46332 | 0.00765 |
| hye | hye | hye | **1.0** | 0.0 | 0.97778 | 0.0 | 1.0 | 0.0 | 1.0 | 0.0 | 0.97778 | 0.0 | 0.97778 | 0.0 |
| ibo | ibo | ibo | 0.98718 | 0.00023 | 0.50877 | 0.0052 | 0.99355 | 0.00011 | **1.0** | 0.0 | 0.52968 | 0.0052 | 0.59184 | 0.00336 |
| ilo | ilo | ilo | 0.91339 | 0.00126 | 0.7651 | 0.0019 | 0.928 | 0.00097 | **0.97479** | 0.00031 | 0.83824 | 0.0019 | 0.93443 | 0.00039 |
| ind | ind | ind | 0.72483 | 0.00411 | 0.397 | 0.00867 | 0.76056 | 0.00311 | **0.78261** | 0.00258 | 0.53807 | 0.00867 | 0.69799 | 0.00209 |
| isl | isl | isl | **0.9916** | 0.00011 | 0.76129 | 0.00207 | 0.9916 | 0.00011 | 0.9916 | 0.0001 | 0.80272 | 0.00207 | 0.80272 | 0.0016 |
| ita | ita | ita | 0.7947 | 0.00342 | 0.44195 | 0.00822 | 0.83916 | 0.00236 | **0.86957** | 0.00175 | 0.52444 | 0.00822 | 0.6178 | 0.00391 |
| jav | jav | jav | **0.97581** | 0.00046 | 0.36646 | 0.00783 | 0.97581 | 0.00043 | 0.97561 | 0.00031 | 0.41696 | 0.00783 | 0.47012 | 0.0038 |
| jpn | jpn | jpn | 0.72195 | 0.01301 | 0.73 | 0.00593 | 0.7861 | 0.00848 | 0.79245 | 0.00783 | 0.82486 | 0.00593 | **0.85373** | 0.00242 |
| kal | kal | kal | **0.98305** | 0.0 | 0.76 | 0.00185 | 0.98305 | 0.0 | 0.98305 | 0.0 | 0.83824 | 0.00185 | 0.94215 | 0.00022 |
| kan | kan | kan | **1.0** | 0.0 | **1.0** | 0.0 | 1.0 | 0.0 | 1.0 | 0.0 | 1.0 | 0.0 | 1.0 | 0.0 |
| kat | kat | kat | **1.0** | 0.0 | **1.0** | 0.0 | 1.0 | 0.0 | 1.0 | 0.0 | 1.0 | 0.0 | 1.0 | 0.0 |
| kaz | kaz | kaz | 0.96721 | 0.00034 | 0.42182 | 0.00878 | **0.97521** | 0.00021 | 0.96667 | 0.00021 | 0.42182 | 0.00878 | 0.42491 | 0.00853 |
| kbp | kbp | kbp | 0.85714 | 0.00228 | 0.27685 | 0.01683 | 0.90909 | 0.00129 | **0.9375** | 0.00082 | 0.28019 | 0.01683 | 0.30851 | 0.0142 |
| kea | kea | kea | **0.72727** | 0.00114 | 0.08571 | 0.00789 | 0.72727 | 0.00107 | 0.69811 | 0.00093 | 0.09 | 0.00789 | 0.07527 | 0.00655 |
| khk | khk | khk | **0.99225** | 0.0 | 0.71591 | 0.00268 | 0.63158 | 0.0 | 0.33333 | 0.0 | 0.73256 | 0.00268 | 0.82895 | 0.00132 |
| khm | khm | khm | **1.0** | 0.0 | 0.96825 | 0.00022 | 1.0 | 0.0 | 1.0 | 0.0 | **1.0** | 0.00022 | 1.0 | 0.0 |
| kin | kin | kin | 0.76336 | 0.0024 | 0.82258 | 0.00073 | 0.81967 | 0.00129 | **0.8547** | 0.00072 | 0.82927 | 0.00073 | 0.84298 | 0.00055 |
| kir | kir | kir | 0.94488 | 0.0008 | 0.57416 | 0.00498 | 0.95238 | 0.00064 | **0.96774** | 0.00041 | 0.58537 | 0.00498 | 0.625 | 0.00396 |
| kmb | kmb | kmb | **0.99194** | 0.00011 | 0.82562 | 0.00229 | 0.99194 | 0.00011 | 0.99194 | 0.0001 | 0.85926 | 0.00229 | 0.928 | 0.00055 |
| kmr | kmr | kmr | **0.66667** | 0.00673 | 0.47773 | 0.00721 | 0.66667 | 0.00633 | 0.66667 | 0.00608 | 0.50862 | 0.00721 | 0.54378 | 0.00545 |
| knc | knc | knc | 0.97059 | 0.00034 | 0.50187 | 0.00744 | **0.97778** | 0.00021 | 0.97778 | 0.00021 | 0.57021 | 0.00744 | 0.71277 | 0.00297 |
| kon | kon | kon | 0.40789 | 0.0 | **0.45399** | 0.00733 | 0.40789 | 0.0 | 0.39735 | 0.0 | 0.4486 | 0.00733 | 0.43226 | 0.00671 |
| kor | kor | kor | 0.94488 | 0.0008 | 0.51064 | 0.00643 | 0.96 | 0.00054 | **0.99174** | 0.0001 | 0.64516 | 0.00643 | 0.67416 | 0.00319 |
| lao | lao | lao | **1.0** | 0.0 | 0.992 | 6e-05 | 1.0 | 0.0 | 1.0 | 0.0 | **1.0** | 6e-05 | 1.0 | 0.0 |

Table 48: Comparison of GlotLID vs NLLB on UDHR benchmark (part 1)

Table 49 (part 2): the "with confidence threshold θ" spans the GlotLID-M θ=.3, GlotLID-M θ=.5, NLLB θ=.3, and NLLB θ=.5 column groups.

| iso639-3 | UDHR Code(s) | NLLB Code(s) | GlotLID-M F1↑ | FPR↓ | NLLB F1↑ | FPR↓ | GlotLID-M θ=.3 F1↑ | FPR↓ | GlotLID-M θ=.5 F1↑ | FPR↓ | NLLB θ=.3 F1↑ | FPR↓ | NLLB θ=.5 F1↑ | FPR↓ |
|---|---|---|---|---|---|---|---|---|---|---|---|---|---|---|
| lij | lij | lij | 0.49785 | 0.01324 | 0.56701 | 0.00447 | 0.53953 | 0.01052 | 0.64804 | 0.00639 | 0.57592 | 0.00447 | **0.67485** | 0.0027 |
| lin | lin | lin | 0.99145 | 0.00023 | 0.94561 | 0.00056 | **1.0** | 0.0 | 1.0 | 0.0 | 0.96996 | 0.00056 | 0.98261 | 6e-05 |
| lit | lit | lit | 0.9375 | 0.00091 | 0.68208 | 0.00302 | 0.96774 | 0.00043 | **0.98361** | 0.00021 | 0.74684 | 0.00302 | 0.90076 | 0.00066 |
| ltz | ltz | ltz | **0.98305** | 0.0 | 0.60733 | 0.00408 | 0.98305 | 0.0 | 0.98305 | 0.0 | 0.63388 | 0.00408 | 0.67836 | 0.00292 |
| lua | lua | lua | 0.71186 | 0.00183 | 0.61017 | 0.00352 | 0.75 | 0.00107 | **0.78846** | 0.00031 | 0.63158 | 0.00352 | 0.69231 | 0.00231 |
| lug | lug | lug | 0.9589 | 0.00068 | 0.37398 | 0.01286 | 0.97222 | 0.00043 | **0.98592** | 0.00021 | 0.39205 | 0.01286 | 0.42724 | 0.01013 |
| lus | lus | lus | 0.93548 | 0.00091 | 0.33238 | 0.01303 | 0.95868 | 0.00054 | **0.97479** | 0.00031 | 0.33623 | 0.01303 | 0.34421 | 0.01216 |
| lvs | lvs | lvs | 0.94118 | 0.00171 | 0.95582 | 0.00056 | 0.92623 | 0.00118 | 0.90213 | 0.00093 | 0.97143 | 0.00056 | **0.98347** | 0.00017 |
| mag | mag | mag | 0.75385 | 0.00046 | 0.61635 | 0.00185 | **0.76562** | 0.0021 | 0.76562 | 0.0021 | 0.62821 | 0.00185 | 0.66216 | 0.00121 |
| mai | mai | mai | **0.83099** | 0.0 | 0.80537 | 0.00034 | 0.83099 | 0.0 | 0.83099 | 0.0 | 0.80537 | 0.00034 | 0.81081 | 0.00028 |
| mal | mal | mal | **1.0** | 0.0 | **1.0** | 0.0 | 1.0 | 0.0 | 1.0 | 0.0 | 1.0 | 0.0 | 0.975 | 0.0 |
| mar | mar | mar | 0.99174 | 0.00011 | 0.91603 | 0.00062 | 0.99174 | 0.00011 | 0.99174 | 0.0001 | **1.0** | 0.00062 | 1.0 | 0.0 |
| min | min | min | 0.88235 | 0.00171 | 0.22857 | 6e-05 | 0.91603 | 0.00107 | **0.92187** | 0.00082 | 0.23188 | 6e-05 | 0.20588 | 0.0 |
| mkd | mkd | mkd | **0.99174** | 0.0 | **0.99174** | 0.0 | 0.99174 | 0.0 | 0.99174 | 0.0 | 0.99174 | 0.0 | 0.99174 | 0.0 |
| mlt | mlt | mlt | 0.77419 | 0.00399 | 0.78146 | 0.00179 | 0.78947 | 0.00343 | 0.82759 | 0.00258 | 0.83099 | 0.00179 | **0.88722** | 0.00077 |
| mos | mos | mos | 0.97015 | 0.00046 | 0.75581 | 0.00235 | **0.99237** | 0.0001 | 0.99237 | 0.0001 | 0.78313 | 0.00235 | 0.81761 | 0.0016 |
| mri | mri | mri | 0.8227 | 0.00023 | 0.35913 | 0.01029 | **0.82857** | 0.00011 | 0.82857 | 0.0001 | 0.37061 | 0.01029 | 0.40138 | 0.00825 |
| mya | mya | mya | 0.66292 | 0.00673 | 0.67416 | 0.00324 | 0.66292 | 0.00633 | 0.67045 | 0.00587 | 0.68182 | 0.00324 | **0.7362** | 0.00237 |
| nav | nav | nav | 0.9916 | 0.00011 | 0.57282 | 0.00492 | **1.0** | 0.0 | 1.0 | 0.0 | 0.58706 | 0.00492 | 0.65556 | 0.00341 |
| nld | nld | nld | 0.70238 | 0.00571 | 0.59184 | 0.00442 | 0.71084 | 0.00515 | **0.71515** | 0.00484 | 0.65909 | 0.00442 | 0.6988 | 0.0027 |
| nno | nno | nno | 0.95868 | 0.00034 | 0.8855 | 0.00073 | 0.9661 | 0.0021 | 0.9661 | 0.0001 | 0.95082 | 0.00073 | **0.97479** | 6e-05 |
| nob | nob | nob | 0.98462 | 0.00023 | 0.96183 | 0.00022 | **0.99225** | 0.00011 | 0.98438 | 0.0001 | 0.98438 | 0.00022 | 0.98438 | 6e-05 |
| npi | npi | npi | **0.98214** | 0.0 | 0.88189 | 0.00078 | 0.575 | 0.0 | 0.32353 | 0.0 | 0.96552 | 0.00078 | **0.97391** | 0.00011 |
| nso | nso | nso | 0.86957 | 0.00205 | 0.86131 | 0.00101 | 0.87591 | 0.00182 | 0.88235 | 0.00165 | 0.86765 | 0.00101 | **0.91473** | 0.00055 |
| nya | nya | nya | 0.96414 | 0.00103 | 0.79333 | 0.00336 | 0.97581 | 0.00064 | **0.99588** | 0.0001 | 0.83509 | 0.00336 | 0.91892 | 0.00105 |
| oci | oci | oci | 0.41101 | 0.0 | 0.41688 | 0.0104 | 0.40516 | 0.0 | 0.38131 | 0.0 | **0.45698** | 0.0104 | 0.4357 | 0.00462 |
| oss | oss | oss | 0.50273 | 0.00696 | 0.71875 | 0.00034 | 0.50549 | 0.00644 | 0.50829 | 0.00608 | 0.74797 | 0.00034 | **0.7541** | 0.0 |
| pan | pan | pan | **1.0** | 0.0 | **1.0** | 0.0 | 1.0 | 0.0 | 1.0 | 0.0 | 1.0 | 0.0 | 1.0 | 0.0 |
| pap | pap | pap | 0.79195 | 0.00354 | 0.65537 | 0.00336 | 0.83099 | 0.00258 | **0.86765** | 0.00185 | 0.7205 | 0.00336 | 0.8227 | 0.00132 |
| pcm | pcm | pcm | 0.71739 | 0.0 | 0.84112 | 0.00017 | 0.71739 | 0.0 | 0.71739 | 0.0 | **0.84906** | 0.00017 | 0.84906 | 0.00011 |
| pes | pes | pes | 0.65922 | 0.00696 | 0.64407 | 0.00341 | 0.63855 | 0.0058 | 0.54135 | 0.00392 | 0.65143 | 0.00341 | 0.65143 | 0.00325 |
| plt | plt | plt | **0.98182** | 0.0 | 0.88525 | 0.00067 | 0.98182 | 0.0 | 0.98182 | 0.0 | 0.90756 | 0.00067 | 0.93913 | 0.00028 |
| pol | pol | pol | 0.74074 | 0.00479 | 0.41404 | 0.00928 | 0.76923 | 0.00386 | **0.81633** | 0.00278 | 0.4856 | 0.00928 | 0.60513 | 0.00418 |
| por | por | por | 0.86331 | 0.00434 | 0.73171 | 0.00492 | 0.87273 | 0.00376 | **0.89219** | 0.00299 | 0.81081 | 0.00492 | 0.84806 | 0.00237 |
| prs | prs | prs | 0.0 | 0.00274 | 0.08824 | 0.00022 | 0.0 | 0.00258 | 0.0 | 0.00247 | 0.08824 | 0.00022 | **0.08955** | 0.00017 |
| quy | quy | quy | 0.70115 | 0.00593 | 0.09147 | 0.06543 | 0.82993 | 0.00268 | **0.88406** | 0.00165 | 0.09992 | 0.06543 | 0.11018 | 0.05233 |
| roh | roh | roh | **0.99268** | 0.0 | 0.97515 | 0.00112 | 0.99145 | 0.0 | 0.98775 | 0.0 | 0.98095 | 0.00112 | 0.9892 | 0.00044 |
| ron | ron | ron | 0.80992 | 0.00514 | 0.81667 | 0.0024 | 0.80833 | 0.00472 | 0.82906 | 0.00392 | 0.83051 | 0.0024 | **0.85965** | 0.00171 |
| run | run | run | 0.88235 | 0.00183 | 0.71951 | 0.00252 | 0.88889 | 0.00161 | **0.90909** | 0.00124 | 0.75159 | 0.00252 | 0.81944 | 0.00138 |
| rus | rus | rus | 0.43321 | 0.01792 | 0.43321 | 0.00878 | 0.47431 | 0.01428 | 0.51064 | 0.01185 | 0.45455 | 0.00878 | **0.58824** | 0.00462 |
| sag | sag | sag | **0.81553** | 0.0 | 0.21176 | 0.00084 | 0.80392 | 0.0 | 0.79208 | 0.0 | 0.225 | 0.00084 | 0.23377 | 0.00039 |
| san | san | san | 0.66667 | 0.0 | **0.67521** | 0.00017 | 0.66667 | 0.0 | 0.66667 | 0.0 | 0.67249 | 0.00017 | 0.67249 | 0.0 |
| shn | shn | shn | 0.99145 | 0.0 | **1.0** | 0.0 | 0.99145 | 0.0 | 0.99145 | 0.0 | 1.0 | 0.0 | 1.0 | 0.0 |
| sin | sin | sin | **1.0** | 0.0 | **1.0** | 0.0 | 1.0 | 0.0 | 1.0 | 0.0 | 1.0 | 0.0 | 1.0 | 0.0 |
| slk | slk | slk | 0.87591 | 0.00194 | 0.85926 | 0.00095 | 0.9375 | 0.00086 | **0.94488** | 0.00072 | 0.89231 | 0.00095 | 0.93548 | 0.00033 |
| slv | slv | slv | 0.89552 | 0.0016 | 0.71006 | 0.00274 | 0.95238 | 0.00064 | **0.97561** | 0.00031 | 0.78947 | 0.00274 | 0.85714 | 0.0011 |
| smo | smo | smo | **1.0** | 0.0 | 0.68182 | 0.00296 | 1.0 | 0.0 | 1.0 | 0.0 | 0.70175 | 0.00296 | 0.76433 | 0.00187 |
| sna | sna | sna | 0.93846 | 0.00091 | 0.46964 | 0.00716 | 0.96063 | 0.00054 | **0.98387** | 0.00021 | 0.57711 | 0.00716 | 0.69461 | 0.00264 |
| som | som | som | 0.75817 | 0.00422 | 0.54286 | 0.00531 | 0.80556 | 0.00301 | **0.89231** | 0.00144 | 0.61957 | 0.00531 | 0.7451 | 0.00209 |
| sot | sot | sot | 0.98333 | 0.00011 | 0.69118 | 0.00162 | **0.9916** | 0.0 | 0.9916 | 0.0 | 0.72868 | 0.00162 | 0.78333 | 0.00072 |
| spa | spa | spa | 0.72321 | 0.00685 | 0.39709 | 0.01387 | 0.75701 | 0.00537 | **0.78** | 0.00402 | 0.43968 | 0.01387 | 0.51923 | 0.00814 |
| srd | src | srd | **0.9916** | 0.0 | 0.86567 | 0.00089 | 0.9916 | 0.0 | 0.9916 | 0.0 | 0.95082 | 0.00089 | **0.97479** | 6e-05 |
| srp | srp | srp | **0.5124** | 0.00685 | 0.48739 | 0.00336 | 0.4958 | 0.00633 | 0.48945 | 0.00608 | 0.48945 | 0.00336 | 0.49153 | 0.00319 |
| ssw | ssw | ssw | 0.94891 | 0.00068 | 0.80272 | 0.00123 | **0.99237** | 0.0001 | | | 0.86131 | 0.00123 | 0.90769 | 0.00028 |
| sun | sun | sun | 0.9697 | 0.00034 | 0.75294 | 0.00229 | **0.9771** | 0.00021 | 0.9771 | 0.00021 | 0.84211 | 0.00229 | 0.9078 | 0.00066 |
| swe | swe | swe | 0.86301 | 0.00228 | 0.81579 | 0.00151 | 0.93333 | 0.00097 | **1.0** | 0.0 | 0.95385 | 0.00151 | 0.98413 | 6e-05 |
| swh | swh | swh | **0.84956** | 0.0 | 0.24742 | 0.02036 | 0.84685 | 0.00032 | 0.78 | 0.0 | 0.29268 | 0.02036 | 0.35821 | 0.01178 |
| tah | tah | tah | 0.91892 | 0.00011 | 0.89922 | 0.00067 | 0.91892 | 0.00011 | 0.92727 | 0.0 | 0.92063 | 0.00067 | **0.98305** | 6e-05 |
| tam | tam | tam | **1.0** | 0.0 | **1.0** | 0.0 | 1.0 | 0.0 | 1.0 | 0.0 | 1.0 | 0.0 | 1.0 | 0.0 |
| tat | tat | tat | 0.65556 | 0.00696 | 0.46457 | 0.00755 | 0.68208 | 0.0058 | **0.69822** | 0.00515 | 0.46825 | 0.00755 | 0.48163 | 0.00693 |
| tel | tel | tel | **1.0** | 0.0 | **1.0** | 0.0 | 1.0 | 0.0 | 1.0 | 0.0 | 1.0 | 0.0 | 1.0 | 0.0 |
| tgk | tgk | tgk | 0.67429 | 0.00651 | 0.47581 | 0.00727 | 0.92913 | 0.00097 | **0.95082** | 0.00052 | 0.50213 | 0.00727 | 0.59296 | 0.00446 |
| tgl | tgl | tgl | 0.9403 | 0.0008 | 0.5368 | 0.00587 | 0.95455 | 0.00054 | **0.97674** | 0.00021 | 0.54148 | 0.00587 | 0.56364 | 0.00517 |
| tha | tha | tha | **1.0** | 0.0 | **1.0** | 0.0 | 1.0 | 0.0 | 1.0 | 0.0 | 1.0 | 0.0 | 1.0 | 0.0 |
| tir | tir | tir | **1.0** | 0.0 | 0.97561 | 0.00017 | 1.0 | 0.0 | 1.0 | 0.0 | 0.99174 | 0.00017 | **1.0** | 0.0 |
| ton | ton | ton | **1.0** | 0.0 | | | 1.0 | 0.0 | 1.0 | 0.0 | | | | |
| tpi | tpi | tpi | **0.98361** | 0.00011 | 0.91339 | 0.00045 | 0.98361 | 0.00011 | 0.98361 | 0.0001 | 0.94309 | 0.00045 | 0.95868 | 0.00011 |
| tsn | tsn | tsn | 0.98361 | 0.00023 | 0.79452 | 0.00157 | **0.99174** | 0.00011 | 0.99174 | 0.0001 | 0.85926 | 0.00157 | 0.89922 | 0.00061 |
| tso | tso | tso | 0.94158 | 0.0016 | 0.64073 | 0.00986 | **0.95139** | 0.0015 | 0.95139 | 0.00113 | 0.66986 | 0.00986 | 0.70886 | 0.00633 |
| tuk | tuk | tuk | 0.94821 | 0.00148 | 0.65574 | 0.00022 | 0.96748 | 0.00086 | **0.98347** | 0.00041 | 0.65574 | 0.00022 | 0.65574 | 0.00022 |
| tur | tur | tur | 0.4918 | 0.01415 | 0.32967 | 0.01277 | 0.50209 | 0.01164 | **0.51502** | 0.01164 | 0.37037 | 0.01365 | 0.39604 | 0.01007 |
| tzm | tzm | tzm | 0.01754 | 0.00593 | 0.01653 | 0.0033 | 0.01754 | 0.00558 | **0.0177** | 0.00525 | 0.01653 | 0.0033 | 0.01653 | 0.00325 |
| uig | uig | uig | 0.90295 | 0.00114 | 0.66667 | 0.0 | **0.93333** | 0.0 | 0.89908 | 0.0 | 0.66667 | 0.0 | 0.66667 | 0.0 |
| ukr | ukr | ukr | 0.98361 | 0.00023 | 0.91339 | 0.0005 | 0.98361 | 0.00021 | **0.99174** | 0.0001 | 0.94309 | 0.0005 | 0.97479 | 6e-05 |
| umb | umb | umb | 0.87931 | 0.00114 | 0.89313 | 0.0014 | 0.87931 | 0.00107 | 0.87611 | 0.00072 | 0.91051 | 0.0014 | **0.94737** | 0.00055 |
| urd | urd | urd | **0.96522** | 0.0008 | 0.61838 | 0.00761 | 0.96522 | 0.00075 | 0.96522 | 0.00072 | 0.63068 | 0.00761 | 0.63068 | 0.0071 |
| uzn | uzn | uzn | 0.50407 | 0.0073 | 0.28365 | 0.01325 | 0.53881 | 0.00429 | **0.56716** | 0.00247 | 0.33908 | 0.01325 | 0.41404 | 0.00583 |
| vec | vec | vec | 0.92308 | 0.00114 | 0.85714 | 0.00112 | 0.95238 | 0.00064 | **0.96774** | 0.00041 | 0.87591 | 0.00112 | 0.90909 | 0.00088 |
| vie | vie | vie | **0.66304** | 0.00011 | 0.45675 | 0.00565 | 0.66304 | 0.00011 | 0.66304 | 0.0001 | 0.56621 | 0.00565 | 0.61307 | 0.00088 |
| war | war | war | **0.9916** | 0.0 | 0.75159 | 0.00213 | 0.9916 | 0.0 | 0.9916 | 0.0 | 0.78146 | 0.00213 | 0.81379 | 0.00143 |
| wol | wol | wol | 0.79747 | 0.00365 | 0.23985 | 0.02192 | 0.86897 | 0.00204 | **0.92537** | 0.00093 | 0.2684 | 0.02192 | 0.33973 | 0.01321 |
| xho | xho | xho | 0.93846 | 0.00091 | 0.64835 | 0.00347 | 0.96825 | 0.00043 | **0.976** | 0.00031 | 0.68605 | 0.00347 | 0.74214 | 0.00215 |
| ydd | ydd | ydd | **0.99187** | 0.0 | **1.0** | 0.0 | 1.0 | 0.0 | 1.0 | 0.0 | 1.0 | 0.0 | 1.0 | 0.0 |
| yor | yor | yor | 0.85106 | 0.0024 | 0.44697 | 0.00811 | 0.88889 | 0.00161 | **0.93023** | 0.00093 | 0.50213 | 0.00811 | 0.63784 | 0.00363 |
| zho | cjy/hak/cmn/hsn/yue/gan/wuu/nan | zho | 0.96345 | 0.00011 | 0.54715 | 0.05559 | **0.96408** | 0.0 | 0.96408 | 0.0 | 0.64193 | 0.05559 | 0.56574 | 0.02179 |
| zul | zul | zul | **0.98305** | 0.0 | 0.59487 | 0.00431 | 0.98305 | 0.0 | 0.98305 | 0.0 | 0.61702 | 0.00431 | 0.64088 | 0.00347 |

Table 49: Comparison of GlotLID vs NLLB on UDHR benchmark (part 2)