# OpenReview forum: "GlotLID: Language Identification for Low-Resource Languages"
_EMNLP/2023/Conference — EMNLP 2023 Findings_

### Official Review · Reviewer_9oC8 · 2023-08-01

**Typos Grammar Style And Presentation Improvements:** 1. It would be nice to expand FPR in …
**Soundness:** 4

**Excitement:**

4: Strong: This paper deepens the understanding of some phenomenon or lowers the barriers to an existing research direction.

**Paper Topic And Main Contributions:**

The paper proposes a language id method, specifically focussed towards covering low-resource languages and covers 1665 languages, which is an improvement over previous such works on lang id which are focussed more on the high-to-med resource languages. As part of this effort, they also curate a dataset covering these set of languages for lang id training.

**Questions For The Authors:**

1. In Section 3 Preprocessing, how exactly do you ensure that? Do you train a model on the dataset from vaan Ensh et al?
2. In the L233, its mentioned that fastext is good as it provides prob estimates. but how do we know that they are well-calibrated? It would be good to report the calibration metrics such as  Measuring Calibration in Deep Learning, Nixon et al.

**Reasons To Accept:**

1. Extensive experiments and analysis, I especially liked how for each result/experiment/evaluation measure the authors have justified appropriately.
2. Overall, the paper is well written and easy to understand, I liked how section 2 has clearly outlined the motivation of the work by using the titles in bold before every paragraph. Also, I liked how the authors have clearly explained the choice of their baselines, its a very good practice.
3. The authors plan to release the code/data.

**Reasons To Reject:**

1. It would be good to add statistical deviation of the model across say 3 runs?
2. Apart from results reported in numbers, it would be good to show some examples where the proposed method wins over existing methods and where it fails.
3. Since corpus creation is one major contribution, it would have been nice to get some of the data manually verified as well, I know its not cheap to do that but would further ensure that the corpus is of high quality. Additionally it would be good to know more details on the corpus collected, like distributon by domain etc.

**Reproducibility:**

4: Could mostly reproduce the results, but there may be some variation because of sample variance or minor variations in their interpretation of the protocol or method.

**Reviewer Confidence:**

3: Pretty sure, but there's a chance I missed something. Although I have a good feel for this area in general, I did not carefully check the paper's details, e.g., the math, experimental design, or novelty.

---

> ### Author Rebuttal · Authors · 2023-08-29
>
> We thank the reviewer for their insightful and helpful comments!
>
> **Reasons To Reject:**
>
> > '1_It_would_be_good_to_add_statistical_deviation_of_the_model_across_say_3_runs?'
>
> - In contrast to typical deep learning models, fasttext variations across runs are extremely small, so we deemphasized this investigation for the submitted paper.
>
> - We will add mean and std of 3 runs in the revised version.
>
> > '2_Apart from results reported in numbers, it would be good to show some examples where the proposed method wins over existing methods and where it fails.'
>
> - We will use the extra space to add such examples of wins and fails.
>
> > '3_1_Since_corpus_creation_is_one_major_contribution,_it_would_have_been_nice_to_get_some_of_the_data_manually_verified_as_well,_I_know_its_not_cheap_to_do_that_but_would_further_ensure_that_the_corpus_is_of_high_quality.'
>
> - Manual verification would require native speakers: identifying, wetting and organizing the work of such a large number of experts would need to allocate a substantial effort to this task; instead we performed some automated verifications, for example based on the practice introduced in https://arxiv.org/pdf/2305.12182.pdf (CF2 in section 3.4). We will elaborate on it.
>
> > '3_2_Additionally_it_would_be_good_to_know_more_details_on_the_corpus_collected,_like_distributon_by_domain_etc.'
>
> - Granted extra space we will give more information about this.
>
> **Questions For The Authors:**
>
> > '1_In_Section_3_Preprocessing,_how_exactly_do_you_ensure_that?_Do_you_train_a_model_on_the_dataset_from_vaan_Ensh_et_al?'
>
> - Using the Unicode character database (https://www.unicode.org/Public/15.0.0/ucd/Scripts.txt), we identify the script in which each sentence is written. After that, we verify whether the combination of the script and claimed language for the sentence actually exists.
>
> > '2_In_the_L233,_its_mentioned_that_fastext_is_good_as_it_provides_prob_estimates._but_how_do_we_know_that_they_are_well-calibrated?_It_would_be_good_to_report_the_calibration_metrics_such_as_Measuring_Calibration_in_Deep_Learning,_Nixon_et_al.'
>
> - We will add calibration scores and reliability diagrams - note that for n-way clustering, calibration can be defined in several ways. We intend to use both (aggregated) marginal calibration scores (as in Nixon et al, 2020), but also confidence calibration metrics.
>
> **Typos Grammar Style And Presentation Improvements**
>
> > '1_It_would_be_nice_to_expand_FPR_in_L080_(F1_is_pretty_common).'
> - Sure, we will add this in the revised version.

---

### Official Review · Reviewer_6ooP · 2023-08-04

**Soundness:** 3

**Excitement:**

3: Ambivalent: It has merits (e.g., it reports state-of-the-art results, the idea is nice), but there are key weaknesses (e.g., it describes incremental work), and it can significantly benefit from another round of revision. However, I won't object to accepting it if my co-reviewers champion it.

**Missing References:**

Brown generated a very similar high-coverage training set: https://www.cs.cmu.edu/~ralf/langid.html

**Paper Topic And Main Contributions:**

The paper is concerned with creating a corpus for training language identification models which covers as many languages as possible and presents a model trained on this corpus. This is for the purpose of creating corpora for low-resource languages.

The main contributions of this paper are the following:

- A dataset intended for training language identification models, which includes 1665+ languages
- A list of desiderata for low-resource language identification models
- A fastText language identification model trained on the above dataset
- Comparison of the trained model with four strong baselines
- Analysis of the model's performance with a focus on their intended application of corpus creation

**Questions For The Authors:**

A) Exactly which sources are used to make the training dataset?

B) Can you show that the UDHR test set does not appear in train?

C) How do you handle "undetermined" labels when calculating the metrics?

**Reasons To Accept:**

1) Openly-available dataset including a large number of languages, plus a fastText LID model trained on this data
2) Clear list of desiderata for low-resource LID models and how their choice for comparison LID models fits into this (section 2)
3) Large number of strong baseline models
4) Careful justification for some of their evaluation measures: for example, their argument for their evaluation scenario in section 6.1 and 'cl' in section 7 with relation to their chosen task of corpus creation.

**Reasons To Reject:**

I think that the main reason to reject this paper is that I do not find their UDHR-based test set convincing. Firstly, I suspect that it appears in the training data for the model, specifically in Wikipedia (line 190). A brief search found several examples of Wikipedia pages in different languages containing the UDHR in whole or in part ( https://en.wikipedia.org/wiki/Dignity, https://sco.wikipedia.org/wiki/Universal_Declaration_o_Human_Richts, https://eo.wikipedia.org/wiki/Universala_Deklaracio_de_Homaj_Rajtoj, https://arz.wikipedia.org/wiki/%D8%A7%D9%84%D8%A7%D8%B9%D9%84%D8%A7%D9%86_%D8%A7%D9%84%D8%B9%D8%A7%D9%84%D9%85%D9%89_%D9%84%D8%AD%D9%82%D9%88%D9%82_%D8%A7%D9%84%D8%A7%D9%86%D8%B3%D8%A7%D9%86, https://es.wikipedia.org/wiki/Declaraci%C3%B3n_Universal_de_los_Derechos_Humanos). This contradicts the claim on line 276 that the test data is not part of train. I appreciate that the FLORES-200 test set is also sourced from wiki articles, but most of it consists of new translations. In addition, there are key details missing or unclear about the test set, such as the number of languages covered (section 5.1 mentions 419 translations but table 1 says 374), the number of lines after the filtering described in section 5.1 and the length of these lines given the shortest ones were discarded. Finally, the authors claim on line 454 that the significantly lower F1 scores on UDHR are due to domain shift and an increase in languages, but I feel that more analysis is needed to support this claim (e.g. error analysis). It may be that there is some undetected flaw in the test set.

A secondary reason to reject this paper is that I believe it currently over-claims or is unclear on the number of languages the model supports. The dataset contains at most 1832 languages, but many of these only contain a very small number of sentences. I think the disparity of data should be made clearer (e.g. the authors should say x number of languages have more than 1000 lines of data, with y additional languages with smaller amounts of data - see https://www.cs.cmu.edu/~ralf/langid.html for a similar corpus). It is also not clear at points how many classes are in the model (line 72 claims the model covers 1665, but table 1 tests on 1832 at most). Even for the smaller number of languages (1665), it is not clear how the authors chose which languages to support: line 1204 says that 1667 languages have more than 1000 training sentences, but the tables in the appendix report scores on the GlotLID-C test set for languages with very small amounts of data (e.g.  Dutton World Speedwords, Evenki, Fe' Fe', Kukna, Lezghian). It is doubtful that a LID classifier could form a useful representation on such a small amount of data. In line 621, the paper does mention that there should have been a much higher threshold for the amount of training data: I would agree with this, since it would likely result in a stronger system with less danger of 'representation washing' by over-claiming the number of languages supported.

Finally, whilst I think that comparing with several strong baselines is good, it is currently difficult to make a direct comparison between the different models due to a lack of clarity in the evaluation. I understand point about testing on the full set of languages to make the set-up more realistic (section 6.1), but I think making a direct comparison is misleading since a higher-coverage model will always 'win' in the naive case. I thought the confidence filtering/thresholding aspect was interesting, but it was unclear how 'undetermined' labels affected the results, especially F1. This is particularly important when comparing with models missing languages, as for them a 'successful' result on a language they don't cover is to return no label at all. The paper should make it clearer how the 'underdetermined' label feeds in to the metrics.

**Reproducibility:**

3: Could reproduce the results with some difficulty. The settings of parameters are underspecified or subjectively determined; the training/evaluation data are not widely available.

**Reviewer Confidence:**

4: Quite sure. I tried to check the important points carefully. It's unlikely, though conceivable, that I missed something that should affect my ratings.

**Typos Grammar Style And Presentation Improvements:**

- 'Granularity flexibility' (line 155) - interesting point, but unsure how this has influenced the choice of low-resource LID. Perhaps move to the discussion?
- Tables in general: explain meaning of bold and underline in caption
 - The font is way too small for the tables in appendix section B, even on a screen. Perhaps just have single language per line and use longtable? It would also be good to clarify if the number of sentences refers to train or train and test.
- In line 250, it would be good to expand a bit more on how fastText is explainable (line 250)
- Line 424 – not necessarily the 'cousin' problem, could just be out of model
- Paragraphs on lines 531 and line 548 seem to be implying that the training data has mixed languages. Is there evidence of this?
- Line 578 – what script is the Cherokee data in? Latin?
- Line 714 – expand in limitation section on why corpora from religious sources are a limitation in particular

---

> ### Author Rebuttal · Authors · 2023-08-29
>
> We thank the reviewer for their insightful and helpful comments!
>
> **Summary**
>
> To summarize our response:
>
> (i) UDHR is the best benchmark available for low-resource
> languages. As we show below it has very little contamination
> for low-resource languages. So as an evaluation for
> low-resource languages (as opposed to high-resource
> languages) it serves our goals well.  All prior work has the
> problem of contamination of UDHR for high-resource
> languages. We believe we should not be penalized for
> following established methodology and for focusing our
> evaluation on low-resource languages (which are the point of
> our paper).
>
> (ii) We clearly state that GlotLID's core support
> is for the 1665 languages with low FPR: see L432-437. We do
> not overclaim that we support all 1832 languages (but it's
> important to still train on them to minimize close-couzin
> errors).
>
> (iii) We agree that we should have used formal notation to define our decision rule.
> We will add the following to the paper:
> Let $p' = \max_{l \in M_i} P_{m_i}(l|s)$ and $l' =
> \argmax_{l \in M_i} P_{m_i}(l|s)$.
> Here $s$ is the string to be classified and $M_i$ is the set of classes that the LID $m_i$ predicts.
> We assign $l'$ to $s$ if
> $p' > \theta$ and undetermined otherwise. $\theta$ is the
> classification threshold.  For setting $B$!, we additionally
> set all probabilities of languages not in $B$ to zero before
> applying max/argmax.
>
>
> **Reasons To Reject**:
>
> > '1_I_do_not_find_their_UDHR-based_test_set_convincing. A_brief_search_found_several_examples_of_Wikipedia_pages_in_different_languages_containing_the_UDHR_in_whole_or_in_part.'
>
> - The purpose of adding UDHR baseline besides the flores-200 is because of the many low-resource languages that flores-200 does not support.
>
> - Here, we calculated the contamination as follows. We count a UDHR test sentence as occurring in our training data if all of its word fourgrams occur in the training data.
>
> - For 292 languages: none of the UDHR test sentences appear in our training data.
>
> - For 57 languages: less than 10 percent of UDHR test sentences appear in our training data.
>
> - The 25 languages with a contamination rate over 10 percent are all high resource languages.
>
> (292+57+25=374 -- these are all our UDHR languages)
>
> So UDHR is an excellent benchmark for testing the performance of LID for *low-resource* languages (although not for high-resource languages).
>
> We will discuss and resolve these contamination issues in the revised version. Thanks for pointing them out.
>
> > '2_key_details_missing_or_unclear_about_the_test_set'
>
> > '2_1_such as the number of languages covered (section 5.1 mentions 419 translations but table 1 says 374)'
>
> In all our evaluations, GlotLID as well as the baselines can only make predictions for languages that they are trained on. GlotLID's training set contains 374 of the 419 UDHR languages, so GlotLID can only make predictions for those 374 languages. The baselines are treated the same way.
>
> However, it is very important to point out that we do *not* remove any language from UDHR test. So the 419-374 languages not covered by GlotLID are still in UDHR test and GlotLID generates many false positives based on them. We view this setting as the only realistic model of how LID is used in the real world: there will always be data from languages that the LID was not trained on.
>
> > '2_2_the number of lines after the filtering described in section 5.1 and the length of these lines given the shortest ones were discarded'
>
> - UDHR consist of 30 short articles, and we do the filtering only to remove short sentences such as: "Article 1-30". We will add more details on length distribution in the appendix of the revised version.
>
> > '3_The_authors_claim_on_line_454_that_the_significantly_lower_F1_scores_on_UDHR_are_due_to_domain_shift_and_an_increase_in_languages,_but_I_feel_that_more_analysis_is_needed_to_support_this_claim_(e.g._error_analysis)._It_may_be_that_there_is_some_undetected_flaw_in_the_test_set.'
>
> - Yes, as said in L308-312, there are some flaws in the UDHR data. Some of these issues have now been resolved, we will make them available in the revised version.
>
> > '4_I_think_the_disparity_of_data_should_be_made_clearer_(e.g._the_authors_should_say_x_number_of_languages_have_more_than_1000_lines_of_data,_with_y_additional_languages_with_smaller_amounts_of_data_'
>
> - We explained in L204-205 for 1677 languages we have more than 1000 sentences and additional details for each language are available in Appendix §B. This of course means that for the rest of languages (1832 - 1677) we have less than 1000 sentences. We will make it clearer in the revised version.
>
> > '5_It_is_also_not_clear_at_points_how_many_classes_are_in_the_model_(line_72_claims_the_model_covers_1665,_but_table_1_tests_on_1832_at_most)'
> '
>
> - We explained in L219-224 our model always runs on all of the 1832 languages, however our claim for the number of languages that GlotLID supports is only for the 1665 languages with low FPR (L432-437). The reason we always use all 1832 languages is to mitigate the out-of-model cousin errors.
>
> > '6_It is not clear how the authors chose which languages to support: line 1204 says that 1667 languages have more than 1000 training sentences, but the tables in the appendix report scores on the GlotLID-C test set for languages with very small amounts of data'
>
> - We explained in L219-224 our model always runs on all of the 1832 languages, so we report all of the scores. Choosing 1665 languages as said in L432-437 is based on the low FPR.
>
> > '7_In_line_621,_the_paper_does_mention_that_there_should_have_been_a_much_higher_threshold_for_the_amount_of_training_data:_I_would_agree_with_this,_since_it_would_likely_result_in_a_stronger_system_with_less_danger_of_'representation_washing'_by_over-claiming_the_number_of_languages_supported.'
>
> - Yes, as mentioned after L621 the number of 1665 languages that we use throughout the paper already reflects this insight. Even though we train on 1832 languages, we claim reasonable performance for only 1665 (Table 1). We agree that ideally we would have several tiers of reliability and also provide a system with (say) 1000 languages that has even higher reliability than we have for the system with 1665.
>
> > '8_Finally,_whilst_I_think_that_comparing_with_several_strong_baselines_is_good,_it_is_currently_difficult_to_make_a_direct_comparison_between_the_different_models_due_to_a_lack_of_clarity_in_the_evaluation._I_understand_point_about_testing_on_the_full_set_of_languages_to_make_the_set-up_more_realistic_(section_6.1),_but_I_think_making_a_direct_comparison_is_misleading_since_a_higher-coverage_model_will_always_'win'_in_the_naive_case.'
>
> We agree that we should have used formal notation to define our decision rule.
> We will add the following to the paper:
> Let $p' = \max_{l \in M_i} P_{m_i}(l|s)$ and $l' =
> \argmax_{l \in M_i} P_{m_i}(l|s)$.
> Here $s$ is the string to be classified and $M_i$ is the set of classes that the LID $m_i$ predicts.
> We assign $l'$ to $s$ if
> $p' > \theta$ and undetermined otherwise. $\theta$ is the
> classification threshold.  For setting $B$!, we additionally
> set all probabilities of languages not in $B$ to zero before
> applying max/argmax. In direct comparisons of a model $m_i$ and a model $m_j$ on F1, we
> restrict the set of languages to $M_i \cap M_j$. So we do not take an average of F1 over languages
> that a model does not cover. The average is over the languages that both models support.
>
>
> **Questions For The Authors:**
>
> > 'A)_Exactly_which_sources_are_used_to_make_the_training_dataset?'
>
> - We reviewed sources referenced in L183-185 and ended up with more than 150 sources.
>
> - We cited the ones with the most contributions in the paper in L190-195. We will cite/mention all of the sources even with small contributions in the appendix and in the code repository.
>
> > 'B)_Can_you_show_that_the_UDHR_test_set_does_not_appear_in_train?'
>
> - Answer is given above.
>
> > 'C)_How_do_you_handle_undetermined_labels_when_calculating_the_metrics?'
>
> - Answer is given above.
>
> **Missing References:**
>
> > '1_Brown_generated_a_very_similar_high-coverage_training_set:_https://www.cs.cmu.edu/~ralf/langid.html'
>
> - This reference has been cited in L127, both the paper and the whatlang tool, which the mentioned link is refering to.
>
> **Typos Grammar Style And Presentation Improvements:**
>
> > '1_Granularity flexibility' (line 155) - interesting point, but unsure how this has influenced the choice of low-resource LID. Perhaps move to the discussion?
>
> - GlotLID-M, along with its many attributes, shows some limitations in effectively separating very close languages, such as dialects. To match this challenge, our model is designed with the capability to adjust granularity. For such languages, the model can provide predictions at a macro level by adding the probabilities of its varieties to macrolanguage probability.
>
> > '2_Tables_in_general:_explain_meaning_of_bold_and_underline_in_caption'
> - We will add this.
>
> > '3_1_The_font_is_way_too_small_for_the_tables_in_appendix_section_B,_even_on_a_screen._Perhaps_just_have_single_language_per_line_and_use_longtable?
> - We will enlarge the font.
>
> >'3_2_It_would_also_be_good_to_clarify_if_the_number_of_sentences_refers_to_train_or_train_and_test.'
> - It is referenced before train/test split in L205. It's for both. We will clarify it.
>
> > '4_Line_250,_it_would_be_good_to_expand_a_bit_more_on_how_fastText_is_explainable'
> - We will expand on this using the extra space granted.
>
> > '5_Line_424,_not_necessarily_the_'cousin'_problem,_could_just_be_out_of_model'
> - Out-of model cousin occurs when any language that is not supported by the model interferes with some related languages that are supported. We acknowledge that unrelated out-of-model languages may also interfere. However for the sake of simplicity we called both as out-of model cousin. We will clarify this point.
>
> > '6_Paragraphs_on_lines_531_and_line_548_seem_to_be_implying_that_the_training_data_has_mixed_languages._Is_there_evidence_of_this?'
>
> - We did a number of audits and did find cases like this. For example, the Farsi training data contains a fair bit of Arabic (because Arabic phrases and citations are frequently used).
>
> > '7_Line_578,_what_script_is_the_Cherokee_data_in?_Latin?
> - Our Cherokee training data is in Latin.
>
> > '8_Line_714,_expand_in_limitation_section_on_why_corpora_from_religious_sources_are_a_limitation_in_particular'
>
> - Because there is often a domain mismatch between religious sources and the text we want to apply LID to, e.g., most text found on the web is not religious.

---

### Official Review · Reviewer_TXgM · 2023-08-04

**Soundness:** 4

**Excitement:**

3: Ambivalent: It has merits (e.g., it reports state-of-the-art results, the idea is nice), but there are key weaknesses (e.g., it describes incremental work), and it can significantly benefit from another round of revision. However, I won't object to accepting it if my co-reviewers champion it.

**Paper Topic And Main Contributions:**

The paper's main contribution is the collection of a diverse 1832-language dataset (many low-resource) for Language ID, GlotLID-M. Unlike previous efforts, the data is not scraped from the web - the authors note that scraped corpora such as mc4 have many data quality issues (Kreutzer et al 2022).

The authors train an off-the-shelf FastText classifier on the data, and compare the results on other LangID datasets (like UDHR) with a few baseline LangID systems (like CLD3). Results are comparable or slightly better in terms of F1 measures.



**Reasons To Accept:**

The dataset associated with the paper, GlotLID-M, will likely be useful to Language ID researchers in the future. Their evaluation procedures are also designed not to make LID too easy, explicitly attempting to mark test data that doesn't belong to any of the languages of interest (for which the classifier will output labels). This is possible because their dataset includes 1832 languages, but the set of interest for classification only includes 1665.

**Reasons To Reject:**

The paper uses only off-the-shelf models for evaluation, so the contribution is limited. The proposed FastText system still suffers from issues like poor separation of related languages.

**Reproducibility:**

4: Could mostly reproduce the results, but there may be some variation because of sample variance or minor variations in their interpretation of the protocol or method.

**Reviewer Confidence:**

4: Quite sure. I tried to check the important points carefully. It's unlikely, though conceivable, that I missed something that should affect my ratings.

**Typos Grammar Style And Presentation Improvements:**

FastText is cited, but it would be worthwhile to add a sentence describing what kind of architecture it uses - I believe the classification itself is multinomial logistic regression. This would facilitate comparisons with the other baselines.

---

> ### Author Rebuttal · Authors · 2023-08-29
>
> We thank the reviewer for their insightful and helpful comments!
>
> **Reasons To Reject**:
>
> > 1_The_paper_uses_only_off-the-shelf_models_for_evaluation,_so_the_contribution_is_limited.
>
> - We agree that our contribution in terms of novel deep learning / machine learning methods is limited. However, this is not the goal and focus of the paper.
>
> - Please note that, in section 2, we explain why these models were chosen. Our main use case is  corpus creation and because of the requirements that we explain (broad coverage, open source, ease of use, uncertatiy assessment, efficiency and granular flexibility), we were not able to find viable alternative options. In addition, these models are the only ones that have gained sufficient trust of the community and are used in most corpus development.
>
> - Our work is the first to rigorously evaluate these models in different setups: eg. when the base set is known and when it is not (B?/B!), when confidence thresholds are used or not, etc. None of these setups were systematically considered in previous works.
>
> - We hope the reviewer can acknowledge this work in language identification, not only for its coverage of a broad set of low-resource languages but also for its substantial contribution to the design of evaluation and baseline selection.
>
> > 2_The_proposed_FastText_system_still_suffers_from_issues_like_poor_separation_of_related_languages.
>
> - Yes, the fasttext model, along with its many attributes, shows some limitations in effectively separation of very close languages, such as dialects. To address this challenge, our model is designed with the capability to adjust its level of granularity. For such languages, the model can provide predictions at a macro level by adding the probabilities of its varieties to macrolanguage probability.
>
> **Typos Grammar Style And Presentation Improvements:**:
> > 1_FastText_is_cited,_but_it_would_be_worthwhile_to_add_a_sentence_describing_what_kind_of_architecture_it_uses.
>
> - We will add a more detailed description of the FastText architecture.

---

### Meta-Review · Area_Chair_bz5j · 2023-09-24

**Recommendation:** 3

**Metareview:**

Summary (adapted from Reviewer 9oC8): The paper proposes language identification datasets and models, specifically focused on covering low-resource languages and covering 1665 languages, which is an improvement over previous works.

As reviewers pointed out, there are a number of things missing from the paper to fully understand the experiments and resource construction. For example, the paper does not actually contain a precise list of the sources for dataset construction, and the decision rule was not clearly explained. These are well-addressed in the authors’ response.

There are substantial risks of dataset mislabeling and contamination involved in creating LID datasets. The authors address some of these issues in the paper, and others are addressed in the response period. Particularly, the “reasons to reject” raised by Reviewer 6ooP all received clear responses, and I do not believe that they represent soundness issues with the paper, although they do reflect some information that should have been provided in the original submission.

The authors must ensure that any clarifications and information they provided in the response period are added to the paper in revision, as well as the formatting requests (the appendix results are unreadable).

The paper includes a substantial analysis section which explores the performance of the models and discusses some quirks of the assembled data. This is a very useful contribution on top of the rest of the paper.

---

### Decision · Program_Chairs · 2023-10-07

**Decision:**

Accept-Findings

**Comment:**

Summary (adapted from Reviewer 9oC8): The paper proposes language identification datasets and models, specifically focused on covering low-resource languages and covering 1665 languages, which is an improvement over previous works.

As reviewers pointed out, there are a number of things missing from the paper to fully understand the experiments and resource construction. For example, the paper does not actually contain a precise list of the sources for dataset construction, and the decision rule was not clearly explained. These are well-addressed in the authors’ response.

There are substantial risks of dataset mislabeling and contamination involved in creating LID datasets. The authors address some of these issues in the paper, and others are addressed in the response period. Particularly, the “reasons to reject” raised by Reviewer 6ooP all received clear responses, and I do not believe that they represent soundness issues with the paper, although they do reflect some information that should have been provided in the original submission.

The authors must ensure that any clarifications and information they provided in the response period are added to the paper in revision, as well as the formatting requests (the appendix results are unreadable).

The paper includes a substantial analysis section which explores the performance of the models and discusses some quirks of the assembled data. This is a very useful contribution on top of the rest of the paper.